# A Hyperparameter Benchmark of VAE-Based Methods for scRNA-seq Batch Integration

## Abstract

We present the first systematic model architecture hyperparameter benchmark of variational-autoencoder (VAE)–based for single-cell RNA sequencing batch integration. We focused on models available under the scvi-tools framework, and compared the scVI, MrVI, and LDVAE models across four datasets with heterogeneous designs under two feature regimes: training with all, and utilizing only highly variable genes (HVGs). Our study executes 960 training runs spanning 120 configurations for the three models that vary latent size capacity, network depth/width, and evaluates with a comprehensive, standardized metric suite from the scib package capturing both batch removal and biological conservation (Batch ASW, PCR-batch, iLISI, graph connectivity, NMI, ARI, label ASW, isolated-label F1/ASW, cLISI, and trajectory conservation), qualitative analysis with UMAP and t-SNE, alongside PCA, random projection, and unintegrated baselines. We find trade-offs across datasets: scVI delivers the strongest overall integration, driven by superior batch correction; LDVAE shows dataset-specific gains in biological structure preservation; MrVI shows stability and batch correction superiority under multi-protocol datasets, however, it is more resource-intensive. Selecting for HVG features generally outperforms full-gene training for all models. Model architecture hyperparameter analysis indicates that moderate to high latent dimensionality (more than 30 dimensions) often yields the best balance, while sensitivity to latent size appears to be related to dataset heterogeneity (diverse tissues, laboratories, chemistries, and gene-coverage profiles), and larger latent spaces tend to improve batch mixing but can reduce biological conservation. We provide model and dataset-specific guidelines that translate our analysis into practical defaults and tuning rules for the practical deployment of VAE-based integration in single-cell studies.

## 1 Introduction

In recent years, single-cell gene expression resources have expanded at an unprecedented pace. Advances in high-throughput technologies now enable profiling from thousands to millions of cells in a single experiment Macosko et al. (2015); Klein et al. (2015); Datlinger et al. (2021); Cao et al. (2017); Han et al. (2018); Satpathy et al. (2019); Picelli et al. (2013). Public portals reflect this scale: as of October 2024, CELLxGENE reported 169.3 million cells, including 93.6 million unique cells spanning more than 2,000 cell types Program et al. (2025), and the Human Cell Atlas portal listed 64.4 million cells by September 2025 hca (2025). The recent Tahoe-100M resource further illustrates this trend, providing one hundred million perturbation-resolved single-cell profiles Zhang et al. (2025). Alongside this growth, a diverse ecosystem of scRNA-seq chemistries and platforms, with differing capture efficiencies, read coverage, throughput, and laboratory workflows, can introduce batch effects that obscure biological signal Haghverdi et al. (2018); Tung et al. (2017); Mereu et al. (2020). Without careful experimental design and appropriate computational integration, these effects can dominate downstream analysis, causing cells to cluster by run, chemistry, or laboratory rather than by true biological state Tran et al. (2020).

A substantial literature evaluates batch-effect correction (integration) methods. Some studies propose new algorithms and compare them to existing approaches Danino et al. (2024); Li et al. (2022); Grønbech et al. (2018); Haghverdi et al. (2018); Butler et al. (2018); Zhang et al. (2024); Polański et al. (2020); Hrovatin et al. (2024); Zhang et al. (2023); others focus on broad, method-agnostic benchmarking Tran et al. (2020); Arevalo et al. (2024); Nguyen et al. (2023); Chen et al. (2021); Luecken et al. (2022); Antonsson & Melsted (2025); Chazarra-Gil et al. (2021) or on principled evaluation metrics Büttner et al. (2017; 2019). However, despite this progress, a gap remains: to our knowledge, no prior work has systematically benchmarked VAEs architecture hyperparameter configurations of variational-autoencoder (VAE)– based methods specifically for batch correction.

Figure 1 shows the graphical models for scVI, LDVAE, and MrVI. We adapt the original figures from Lopez et al. (2018); Boyeau et al. (2022); Svensson et al. (2020) and aggregate them to illustrate the three VAEs benchmarked

in this study. All of these models are considered as deep-generative models for scRNA-seq data, but differ in focus and interpretability. scVI (single cell variational inference) is a hierarchical Bayesian VAE that models gene counts with a zero inflated negative binomial distribution, accounts for batch effects, and learns a low-dimensional latent space for tasks such as normalization, clustering, and differential expression Lopez et al. (2018). LDVAE (Linearly Decoded VAE) is built directly on scVI, but replaces the non-linear decoder with a linear factor model, sacrificing reconstruction accuracy for interpretability where each latent dimension corresponds to a gene program that can be biologically interpreted as co-expression axes Svensson et al. (2020). MrVI (Multi-resolution Variational Inference) generalizes scVI to cohort-level studies by introducing hierarchical latent variables where one latent space captures cell states independent of samples, and another that incorporates sample-level covariates Boyeau et al. (2022).

VAE models in the scvi-tools remain the most widely used probabilistic approach for the integration and analysis of scRNA-seq, offering strong performance with modest computing and mature tooling Feng et al. (2025); Reed et al. (2024); He et al. (2024); Ergen et al. (2025); Long et al. (2023); Mayr et al. (2024); Kandasamy et al. (2025); Ogden et al. (2025); Cillo et al. (2024). In parallel with the rise of foundation models, several transformer-based single-cell models have appeared, e.g., scFoundation Hao et al. (2023) and AIDO.Cell Ho et al. (2024). In particular, AIDO.Cell was pre-trained on 50M cells and required a large training run (256 H100 GPUs over three days for the 100M parameters FM, and eight days for the 650M parameters FM) Ho et al. (2024). However, independent zero-shot evaluations report that popular single-cell FMs can underperform simpler methods, including scVI, on PBMC68k cell identification tasks when measured by NMI/ARI (Ho et al., 2024). These results make scVI and its variants (MrVI for cohort-level effects; LDVAE for interpretability with a linear decoder) a competitive and practical choice for scRNA-seq, even in the foundation model era.

Furthermore, many studies that utilize scVI in their analysis tend to use the default settings (latent dimensionality of 10, hidden layer of 1, and 128 nodes per layer) Long et al. (2023); Long et al.; Mayr et al. (2024); Kandasamy et al. (2025); Ogden et al. (2025); Cillo et al. (2024). Some studies report using different parameters Feng et al. (2025); He et al. (2024); Ergen et al. (2025); however, they tend to use a single combination and do not provide a clear reasoning for the choice of parameters.

Motivated by this gap, we benchmarked three VAE models: scVI, MrVI, and LDVAE, to integrate scRNA-seq data while preserving meaningful biological variation. Our selection targets complementary use cases in scRNA-seq analysis: scVI serves as a general-purpose VAE enabling robust batch integration and downstream tasks; MrVI extends this framework by incorporating cohort structure, explicitly modeling sample and technology covariates for multi-protocol or multi-site designs; and LDVAE employs a linear decoder, trading modest accuracy for interpretability through gene-program loadings. We systematically explored 120 hyperparameter configurations (30 for scVI, 60 for MrVI, 30 for LDVAE) and evaluated each under two gene selection regimes: the entire gene set and the top 5,000 highly variable genes (HVGs) across three datasets, yielding 960 training runs in total. The grid varied four core hyperparameters as provided in Table 1. This design allows us to identify robust operating regimes, quantify sensitivity to key hyperparameters, and enable fair, fully quantitative comparisons among the models. Our study centers on architecture capacity, including latent size, depth, width, and (for MrVI) the sample-aware latent, because these knobs change what the model can represent and therefore directly govern the batch versus biology trade-off. Our study does not imply that parameters like learning rate, batch size, and dropout are of a less important, however, they affect how fast and stable VAEs reach good optima, and they are already supported by a mature literature and standard recipes that users can adopt to their use case in VAEs Goyal et al. (2017); Kingma et al. (2015); Srivastava et al. (2014); Mandt et al. (2017); Smith et al. (2021); Goyal et al. (2017); Adam et al. (2014). Unlike the scope of our study, to our knowledge, no previous study has systematically benchmarked VAEs architecture choices in the domain of scRNA, and our study provides guidelines for users to avoid extensive computations. We evaluated integration quality using the following quantitative metrics: Batch ASW score, PCR batch score, iLISI score, graph connectivity score, NMI score, ARI score, label ASW score, isolated label F1 score, isolated label ASW score, cLISI score and trajectory conservation score. Note that isolated label F1, isolated label ASW, and trajectory conservation did not apply to all datasets due to label/trajectory availability. We additionally performed qualitative assessments using t-SNE and UMAP. All methods were compared against three baselines: unintegrated, random projection, and PCA. The datasets used include Human Immune dataset from OpenProblems Luecken et al. (2022), Zenodo 8020792 (24 PBMC Samples) Brown et al. (2023), Zenodo 11100300 (18 PBMC Samples) Brown et al. (2024), and Tabula Muris tab (2020).

**Contributions.** We (i) defined a unified, reproducible protocol for VAE-based batch integration (standardized preprocessing, feature regimes, training, and metrics); (ii) benchmarked scVI, LDVAE, and MrVI on a coherent hyperparameter grid (latent size, depth, width, cohort-aware where applicable); (iii) evaluated our benchmark on a balanced metrics suite for batch removal and biological conservation with standardized visual summaries; and (iv) provided concise configuration templates to guide model and capacity selection.

**Table 1:** Summary of hyperparameter configurations explored for each model. ✓ indicates applicability.

| Hyperparameter | Values | scVI | MrVI | LDVAE |
|---|---|---|---|---|
| Dimensionality of the the sample-aware latent space | 10, 20, 30, 40, 50 | ✓ | ✓ | ✓ |
| Dimensionality of the sample-unaware latent space | 10, 20 | | ✓ | |
| Number of hidden layers | 1, 2, 3 | ✓ | ✓ | ✓ |
| Number of nodes per hidden layer | 128, 256 | ✓ | ✓ | ✓ |

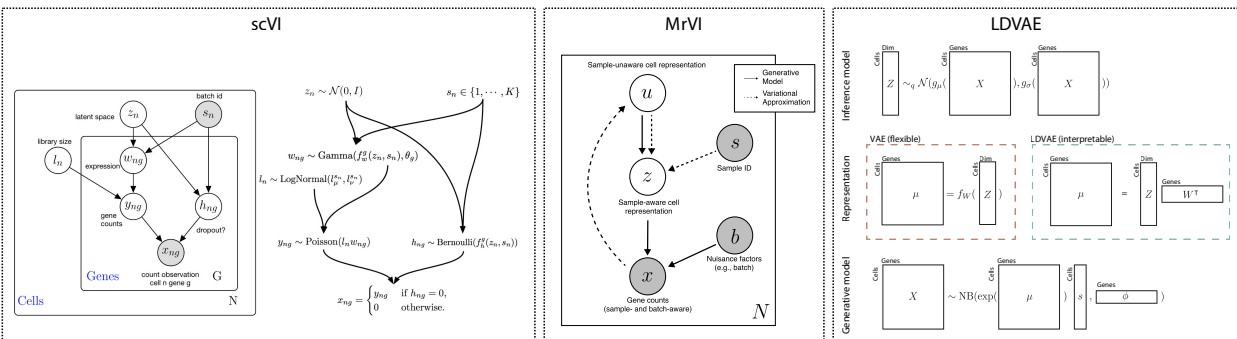

**Figure 1:** *Graphical representations of probabilistic models scVI, MrVI, and LDVAE Lopez et al. (2018); Boyeau et al. (2022); Svensson et al. (2020). This Figure is adopted from the authors in Lopez et al. (2018); Boyeau et al. (2022); Svensson et al. (2020) and it illustrates the underlying graphical models for three variational inference frameworks used in single-cell.*

## 2 DATASETS AND BENCHMARKING PIPELINE

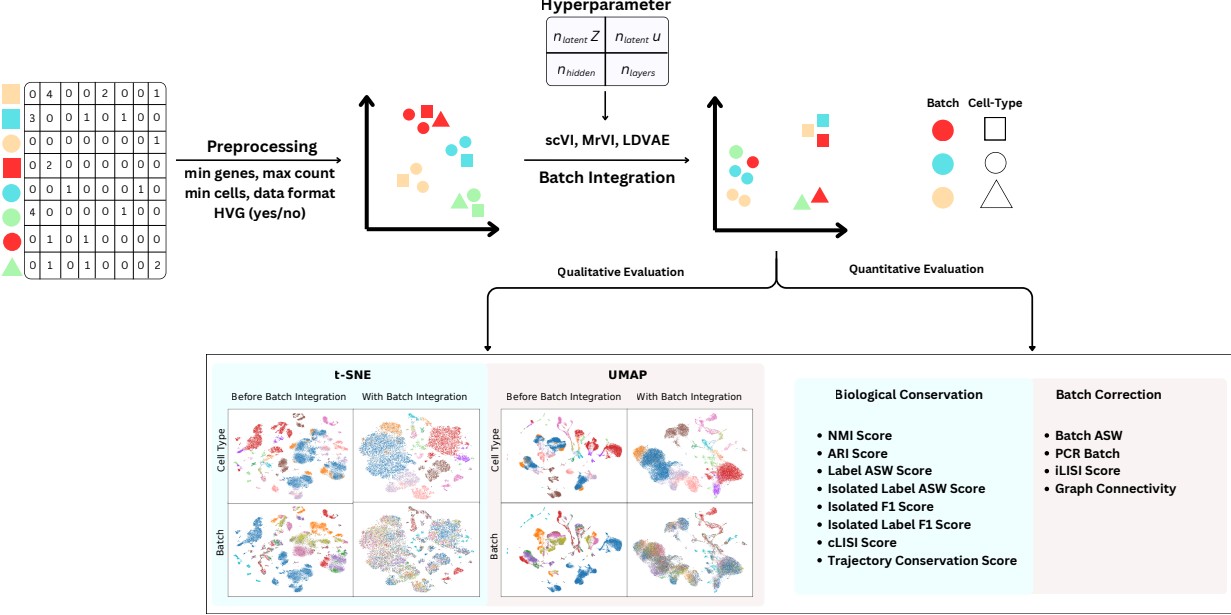

**Figure 2:** *Our proposed workflow for hyperparameter benchmarking of VAE-based methods in scRNA-seq data integration.*

The choice of datasets in our study provides a rigorous benchmark for batch integration because they span complementary sources of variation. Together, they capture cross-study heterogeneity, concentrate many batches within a

unified protocol, and incorporate repeated sampling over time. This combination challenges methods to handle differences driven by study design, run-and-handling effects, and temporal shifts. Each dataset includes explicit batch identifiers and harmonized annotations, allowing a consistent application of batch mixing and structure preservation metrics. The dataset sizes balance realism with tractability, enabling exhaustive hyperparameter sweeps and repeated trials, and the shared file format with standardized metadata supports reproducible pipelines and fair comparisons. All used datasets underwent standardized preprocessing steps, including removal of low-quality cells (bottom 1% gene count, top 1% total count), high mitochondrial content cells, and duplicates. Genes expressed in fewer than three cells are also filtered. Filtering thresholds were defined per batch or library based on each dataset's distribution.

**Human Immune dataset**. The OpenProblems-scIB benchmark serves as a widely adopted reference for evaluating computational methods in the integration of data scRNA-seq. It aggregates immune cells from peripheral blood and bone marrow in five datasets and ten batches, generated using various versions of 10X Genomics (v2 and v3 chemistries) and Smart-seq2 technologies, as illustrated in Figure 7 in the Appendix. This dataset includes 33,506 cells and 12,303 genes, provided in AnnData format. After preprocessing, gene and total count thresholds varied across batches, generally ranging from 400–2700 genes and 8,000–890,000 total counts.

**Zenodo 11100300 (18 PBMC Samples)**. A longitudinal scRNA-sequence dataset of peripheral blood mononuclear cells from 2 donors with myalgic encephalomyelitis/chronic fatigue syndrome, collected before, during, and after an antibiotic-induced remission event; it contains 55,260 cells and 36,601 genes across 4 batches, generated using 10x Genomics, and is provided in AnnData format as illustrated in Figure 8. After preprocessing, gene and total count thresholds ranged between 380–490 genes and 15,000–18,000 total counts. This dataset contains 18 PBMC samples.

**Zenodo 8020792 (24 PBMC Samples)**. A capillary-blood PBMC scRNA-seq resource aggregating 28 samples from 3 donors across 14 experimental batches, generated using 10x Genomics v3.1 chemistry with MULTI-seq barcodes; it includes 76,535 cells and 36,601 genes and is provided in AnnData format as illustrated in Figure 9. This dataset includes a four-level hierarchical cell type annotation, we have used the cell type level 3 label to ensure a fine but stable granularity for biological conservation assessment. After preprocessing, thresholds ranged from 1,000–1,700 genes and 14,000–31,000 total counts, and were additionally filtered to retain only PBMCs based on cell type annotations. The final number of samples after preprocessing is 24 PBMC samples.

**Tabula Muris**. The Tabula Muris atlas, a cross-tissue mouse single-cell RNA-seq resource profiling on the order of $10^5$ cells across roughly 20 organs using two complementary modalities, droplet-based 3′ UMI (10x) for breadth and FACS/Smart-seq2 for depth, yielding rich coverage of immune and non-immune compartments tab (2020). The dataset includes 356,213 cells and 20,116 genes, as illustrated in Figure. 10. The atlas includes 51 immune cell-type labels (e.g., T/B/NK, myeloid and dendritic subsets, tissue-resident macrophages) and 103 non-immune labels (epithelial, endothelial/vascular, stromal/mesenchymal, neural, muscle, hepatic, renal, pancreatic, etc.), for a total of about 154 distinct labels.

**Benchmarking Pipeline**. We benchmark the pipeline in Figure 2 by harmonizing three datasets into a unified AnnData schema with standardized metadata (batch, Participant ID, cell type) and preserved raw counts in layers; after standard QC, each model is run with either all genes or HVGs. We evaluate three VAE-based methods—scVI, MrVI, and LDVAE—sweeping model capacity and latent dimensionality as in Table 1. Performance is quantified with batch-mixing metrics (Batch ASW, PCR batch, iLISI, graph connectivity) and biological-conservation metrics (NMI, ARI, Label ASW and isolated variants, cLISI, trajectory conservation), complemented by UMAP/t-SNE visualizations colored by batch and cell-type. All models were trained on Ubuntu 22.04 LTS with an NVIDIA RTX 4090 (24 GB VRAM) and 128 GB RAM.

To assess seed sensitivity directly, we randomly sampled models and configurations and retrained them with two independent seeds using training of 200 epochs. The variability across seeds for the primary aggregate metric was approximately 0.001, indicating numerical stability at convergence. Taken together, the cross-dataset consistency and the very small seed variance show that stochasticity does not drive our results, and that the reported rankings and recommendations are generalizable.

## 3 RESULTS

The following section reports results for each dataset across the three models, condensed for readers seeking practical tuning guidelines for the studied VAEs. All metrics follow the definitions in Luecken et al. (2022); see that work for details. We compute an overall score as the mean of two group-wise averages: (i) biological conservation and (ii) batch correction. To reflect the primacy of preserving biological signal, we aggregate seven biological metrics and four batch-correction metrics, since removing batch noise without maintaining biology is not a desirable outcome.

## 3.1 IMMUNE DATASETS

Figure 3 compares one–parameter sweeps in which two hyperparameters are fixed while the third is varied; a detailed paired, cross–hyperparameter analysis follows in the next section. Visually, a moderate to high latent dimensionality $n_{\text{latent}}$ usually strikes the best balance for scVI, MrVI, and LDVAE. Sensitivity to $n_{\text{latent}}$ is strongest in the Human Immune dataset: pushing $n_{\text{latent}}$ to the high end boosts the batch composite (BC) but tends to soften the biology composite (Bio) for MrVI and LDVAE. Increasing the hidden width $n_{\text{hidden}}$ yields at most small, marginal gains in a few settings and is otherwise flat or mildly negative. Increasing depth $n_{\text{layers}}$ often lifts BC while generally reducing overall (Bio) at higher $n_{\text{layers}}$, producing a clear trade–off; among the models, LDVAE appears least sensitive to this depth change.

Figure 4 shows the best-performing configuration of each model across the immune datasets, alongside Unintegrated, PCA, and Random Projection baselines. Overall, scVI consistently leads: it achieves the strongest batch correction on Human Immune and Zenodo 8020792 (24 PBMC Samples) (with HVG), and on Zenodo 11100300 (18 PBMC Samples) it delivers top biological conservation with a batch score close to the best. MrVI is notably steady across datasets for both Full and HVG features, with batch and biology scores that are closely matched. LDVAE exhibits data set-specific gains, most prominently in Human Immune, where it attains the highest biology score, and benefits highly from the HVG features in the remaining datasets.

The t-SNE in figure 11, and the UMAP provided in Figure 15 in the supplementary material, compare scVI, MrVI, and LDVAE in Human Immune, Zenodo 8020792 (24 PBMC Samples), and Zenodo 11100300 (18 PBMC Samples) datasets, with each method colored by cell type and batch. Across datasets, scVI consistently produces the most compact within–type clusters with clear boundaries, while maintaining strong cross-batch overlap. MrVI clusters cells correctly by type most of the time, but we can still see traces of batch effect in the embedding, as shown in Zenodo 8020792 (24 PBMC Samples). LDVAE performs well on Human Immune, where clusters are recognizable and batches mix reasonably. However, in Zenodo 8020792 (24 PBMC Samples) it shows poor performance in clustering.

Figure 12 presents t-SNE visualizations of the Zenodo 8020792 (24 PBMC samples) using HVGs (for the other two immune datasets, we have included the t-SNE plots in the Appendix Section F and G). Embeddings show the separation of cell populations under different VAE model configurations (scVI, MrVI, LDVAE) across varying numbers of latent dimensions, hidden units, and layers. The first panel in this Figure shows scVI. We can observe that increasing latent dimensionality generally tightens the same cell-type clusters and increases the separation between phenotypes. The clearest improvements appear at higher latent sizes, with the best embeddings at $n_{\text{latent}} = 50$ for shallow networks (e.g., $n_{\text{layers}} = 1$ with $n_{\text{hidden}} \in \{128, 256\}$). Moderate gains from increasing hidden width (128→256) are most visible in smaller latent sizes (around 20), while benefits are marginal in larger latent spaces. In contrast, deeper encoders ($n_{\text{layers}} = 3$) tend to fragment the embedding and reduce the gains of larger $n_{\text{latent}}$, suggesting detrimental interactions between depth and latent capacity for this dataset.

In the second panel of Figure 12 we see the embeddings of MrVI. For MrVI, compact clustering typically emerges in the mid–latent regime ($n_{\text{latent}} \approx 20$–40). Increasing from $u = 10$ to $u = 20$ yields improvements in select settings (e.g., $n_{\text{hidden}} \in \{128, 256\}$, $n_{\text{layers}} = 3$, $n_{\text{latent}} \in \{40, 50\}$), but this effect is not uniform across the grid. At $u = 10$, pushing $n_{\text{latent}} > 40$ can overspread clusters (e.g., $n_{\text{hidden}} = 128$, $n_{\text{layers}} = 2$, $n_{\text{latent}} = 40 \to 50$)), whereas $u = 20$ better preserves compactness at comparable latent sizes. Depth gives a little advantage and can modestly degrade separation, indicating MrVI is more sensitive to balancing $n_{\text{latentU}}$ with $n_{\text{latentZ}}$ than to adding layers.

In the third panel of Figure 12, we see the embeddings of LDVAE. In the case of LDVAE, it benefits most from increased hidden width. Raising $n_{\text{latent}}$ from 10 to 50 improves separation, while adding the depth of the encoder ($n_{\text{layers}} = 2$ or 3) rarely changes the qualitative picture.

### 3.1.1 HUMAN IMMUNE

Here we analyze per-metric performance on the Human Immune dataset; full analysis is provided in the supplementary material I. For scVI, the best Overall is 0.78105 at $(256, 40, 2, \text{HVG})$ and the worst is 0.73765 at $(256, 10, 1, \text{FULL})$.

Batch correction peaks at $(256, 50, 1, \text{HVG})$, whereas biological conservation peaks at $(256, 10, 1, \text{HVG})$. Moving $128 \to 256$ gives small gains in batch-oriented metrics but often decreases biology/label metrics; increasing layers $1 \to 3$ tilts toward batch mixing with declines in clustering agreement and biological overall. HVG outperforms Full on average. Increasing $n_{\text{latent}}$ stepwise improves batch metrics and agreement with mixed effects on label compactness and biological overall; endpoints $10 \to 50$ improve Overall in 11/12.

For MrVI, the best Overall is 0.76041 at $(256, 40, 1, u=20, \text{Full})$ and the worst is 0.66557 at $(128, 10, 3, u=20, \text{HVG})$. Changing $128 \to 256$ generally depresses composites (iLISI is the counter-trend), and increasing depth $1 \to 3$ pushes

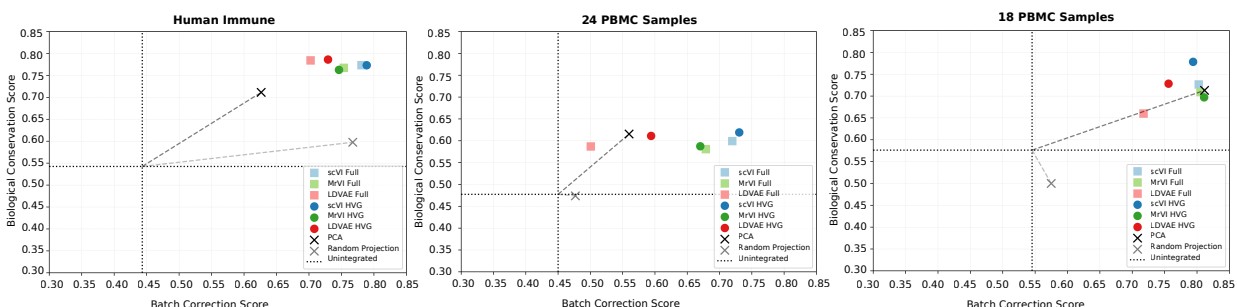

**Figure 3:** *This figure presents a comparison where two hyperparameters are fixed while the third is varied, evaluated on both the full gene set and the HVG features; the first panel summarizes results on the Human Immune dataset, the second on Zenodo 8020792 (24 PBMC Samples), and the third on Zenodo 11100300 (18 PBMC Samples), with performance assessed using the overall batch score and overall biological conservation score.*

**Figure 4:** *This figure compares the overall batch correction (BC) and biological conservation (Bio) scores of the three methods across the three datasets, alongside unintegrated, PCA, and random projection baselines. The points shown represent the best-performing configurations for each model.*

composites down while inflating iLISI. Latent $z$ is the strongest favorable driver: endpoints 50 vs 10 improve composites and agreement broadly with trajectory often down. Moving $u=10 \rightarrow 20$ aids Batch ASW and label compactness with modest composite movement.

For LDVAE, the best Overall is $0.75774$ at $(128, 30, 2, \text{HVG})$ and the worst is $0.61916$ at $(256, 50, 2, \text{Full})$. Increasing $128 \rightarrow 256$ generally reduces composites; increasing depth $1 \rightarrow 3$ is net favorable for composites but softens Batch ASW and agreement. Large $n_{\text{latent}}$ is not advantageous; moderate latent ($\sim 30$) with shallow-to-moderate depth peaks, and HVG yields consistent, modest gains across composites and agreement.

### 3.1.2 ZENODO 8020792 (24 PBMC SAMPLES)

Here we summarize per-metric results on Zenodo 8020792 (24 PBMC samples); full details and tables are provided in the supplementary material J.

For scVI, the best Overall is $0.67480$ at $(256, 50, 1, \text{HVG})$ and the worst is $0.58505$ at $(128, 10, 2, \text{Full})$. Larger hidden size generally helps (Overall up in $20/30$ pairs; batch Overall $26/30$), deeper networks hurt composites ($1 \rightarrow 3$:

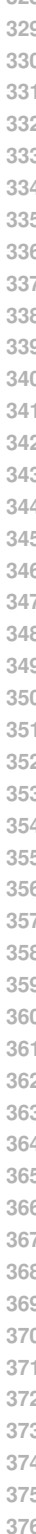
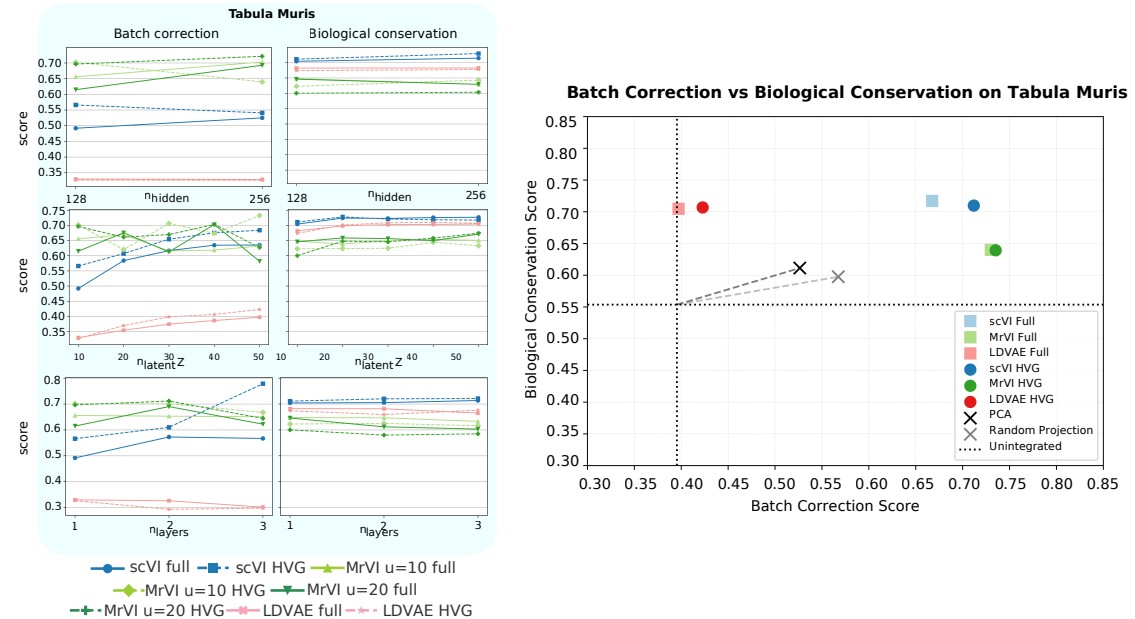

**Figure 5:** *The first panel presents a comparison where two hyperparameters are fixed while the third is varied, evaluated on both the full gene set and the HVG features for Tabula Muris dataset. The second panel compares the overall batch correction (BC) and biological conservation (Bio) scores of the three methods on Tabula Muris, alongside unintegrated, PCA, and random projection baselines. The points shown represent the best-performing configurations for each model.*

Overall $4/20$ up), and increasing $n_{\text{latent}}$ most strongly boosts batch metrics and agreement (endpoint $50-10$: Overall $12/12$ up; batch Overall $12/12$ up). HVG is favorable on as they increase overall, batch, and bio with additional gains often in NMI/ARI and connectivity.

For MrVI, the best Overall is $0.62984$ at $(256, 40, 1, u=20, \text{Full})$ and the worst is $0.54528$ at $(256, 10, 3, u=20, \text{HVG})$. Increasing hidden size and depth typically lowers the composites on this dataset, while $u$ latent modestly improves batch-oriented metrics (PCR, iLISI) and label compactness with limited movement in the composite scores. The optimal Overall appears at moderate–high $z$ latent with shallow depth.

For LDVAE, the best Overall is $0.60260$ at $(256, 50, 1, \text{HVG})$ and the worst is $0.49913$ at $(256, 10, 2, \text{Full})$. Larger hidden size is broadly beneficial (Overall $19/30$ up), depth 3 versus 1 is generally unfavorable for composites, and increasing $n_{\text{latent}}$ boosts batch composites with endpoint $50-10$ improvements in $11/12$ triplets. HVG strongly dominates Full for LDVAE ($30/30$ up for Overall and batch, $29/30$ up for bio).

### 3.1.3 ZENODO 11100300 (18 PBMC SAMPLES)

Here we summarize per-metric results on Zenodo 11100300 (18 PBMC samples); full details and tables are provided in the supplementary material K.

For scVI, the best Overall is $0.78615$ at $(256, 30, 1, \text{HVG})$ and the worst is $0.69409$ at $(256, 40, 1, \text{HVG})$. Increasing hidden size $128 \rightarrow 256$ improves batch removal but tends to reduce biology and the combined Overall; added depth trades batch for biology with little net effect on the primary aggregate; and larger latent dimensionality most clearly benefits batch composites and modestly lifts Overall, with mixed-to-slightly positive effects on biology.

For MrVI, the best Overall is $0.75718$ at $(256, 50, 1, u=20, \text{Full})$ and the worst is $0.70797$ at $(256, 20, 3, u=10, \text{Full})$. Larger hidden size gives small, consistent gains across composites; added depth is broadly unfavorable for the Overall and biology and does not improve the batch composite on average; higher $z$-latent capacity tends to align with the strongest configurations (shallow depth). Increasing $u$ from 10 to 20 reliably improves batch metrics and usually lifts the Overall, with modest gains in clustering agreement and biology and little change in neighborhood mixing; these benefits hold for both Full and HVG.

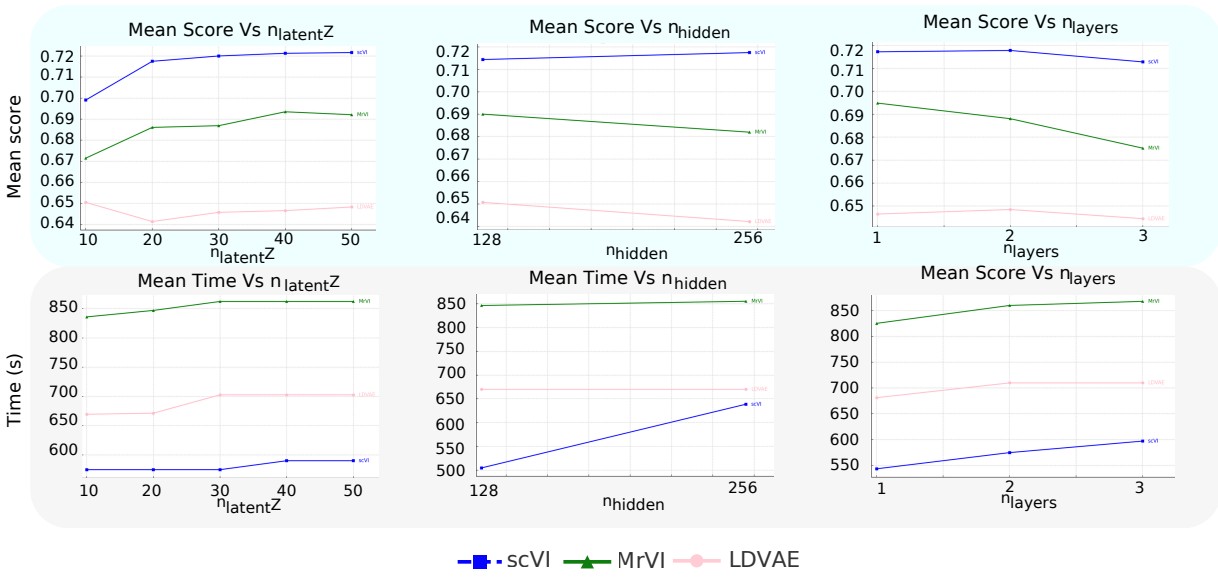

**Figure 6:** *Each plot varies a single hyperparameter while holding the other two fixed. The first panel shows the mean score, and the second panel shows the mean training time, both aggregated across datasets. Curves for scVI, MrVI, and LDVAE summarize how performance and runtime change as the varied hyperparameter increases.*

For LDVAE, the best Overall is $0.74219$ at $(256, 50, 1, \text{HVG})$ and the worst is $0.60572$ at $(256, 20, 3, \text{Full})$. Moving $128 \rightarrow 256$ yields a small net gain in the primary aggregate and a clearer lift in biology with a slight reduction in batch composite; deeper networks generally depress the Overall and biology while leaving batch flat to modestly positive only at three layers; increasing $n_{\text{latent}}$ most strongly helps batch composites and modestly raises the Overall, while biology often softens stepwise but is roughly balanced end-to-end. HVG is modestly better than Full on the primary, batch, and biology composites in this grid.

### 3.2 TABULA MURIS DATASET

We use this dataset to validate our findings and assess whether the same hyperparameter effects hold. As shown in Figure 5, the first panel demonstrates that sweeping one hyperparameter while holding the other two fixed leads to the same conclusion as before: increasing the latent dimension $z$ beyond 30 improves both batch correction and biological conservation. The second panel further indicates that scVI attains the highest overall score. Notably, because Tabula Muris is highly heterogeneous and involves two protocols and 154 cell types (51 immune and 103 non-immune), the sample-aware covariate is influential; accordingly, MrVI excels in batch correction, which is consistent with its design.

Quantitatively, the best scVI configuration is $(256, 50, 2, \text{HVG})$ with an overall score of $0.7108245$, whereas the weakest is $(128, 10, 1, \text{Full})$ with a score of $0.598120875$. For MrVI, the top configuration is $(256, 30, 1, u = 20, \text{HVG})$ scoring $0.68723225$, while the poorest is $(256, 20, 3, u = 10, \text{Full})$ scoring $0.5392775$. For LDVAE, the best setting is $(128, 50, 1, \text{HVG})$ with a score of $0.564824338$, and the worst is $(128, 10, 2, \text{Full})$ with a score of $0.476394$. These results mirror our earlier datasets, supporting the generalizability of our study to unseen, highly heterogeneous, complex non-immune data.

## 4 DISCUSSION

### 4.1 COMPUTATIONAL COMPLEXITY

Each fully connected (dense) layer is implemented as a general matrix–matrix multiplication (GEMM). For a mini-batch of size $B$, input width $in$, and output width $out$, the forward/backward work scales as $\mathcal{O}(B.in.out)$NVIDIA Corporation (2023). Consequently, a layer that maps latent $z$ to hidden $H$ (or vice versa) with $B$ samples costs $\mathcal{O}(BHz)$. Stacking hidden layers adds additional GEMMs: an $H \rightarrow H$ block contributes $\mathcal{O}(BH^2)$ per layer, while

gene-facing heads $H \to G$ cost $\mathcal{O}(BHG)$. Hence, increasing $n_{\text{latent}}$ affects only the $H \leftrightarrow z$ interfaces (linear in $z$), whereas the dominant terms are the gene heads $H \to G$ and any extra hidden layers $H \to H$.

**Practical compute implications of tuning $z$, $H$, $G$, and $B$.**    Each fully connected (dense) layer is implemented as a general matrix–matrix multiplication (GEMM). For a mini-batch of size $B$, input width $in$, and output width $out$, the forward/backward work scales as $\mathcal{O}(B.in.out)$ NVIDIA Corporation (2023). We use this rule to reason about how training cost changes when we adjust the latent size $z$, hidden width $H$, the number of genes $G$, and the batch size $B$. Throughout, we keep in mind the typical magnitudes in single-cell integration: $G$ in the *thousands*, $H$ in the *hundreds*, and $z$ in the *tens*. In practice, both $H$ and $G$ tend to move in large steps (e.g., $H\!: 128 \to 256$; $G\!:$ all-genes vs. $\sim 5{,}000$ HVGs), and $B$ is usually chosen to saturate memory, so the most nimble knob is $z$.

**scVI (MLP decoder, $z \to H \to G$).**    With at least one hidden layer in the decoder, the terms that depend on $z$ are confined to the $H \leftrightarrow z$ interfaces: the encoder heads $H \to z$ and the decoder entrance $z \to H$, giving an incremental cost.

$$\Delta C_{\text{scVI}}(z) \;=\; \mathcal{O}(BH\,\Delta z).$$

The dominant gene-facing multiplies $G \to H$ (encoder front) and $H \to G$ (decoder head) remain $\mathcal{O}(BHG)$ and do not grow with $z$. Hence the fractional overhead of raising $z$ by $\Delta z$ is approximately

$$\frac{\Delta C_{\text{scVI}}(z)}{C_{\text{base}}} \;\approx\; \frac{\Delta z}{G}.$$

Since $G$ is in the thousands and $\Delta z$ is in the tens, this overhead is sub-percent in typical settings. Conclusion: for scVI, increasing $z$ is the cheapest way to add capacity and increase performance as proved by our extensive experiments; it barely perturbs the big $\mathcal{O}(BHG)$ terms. This makes scVI particularly attractive when one wants to improve integration quality with minimal training-time impact.

**LDVAE and MrVI (linear gene projection; decoder cost includes $BGz$).**    LDVAE replaces the non-linear decoder with a single linear map $z \to G$, and MrVI's generative path also produces $G$-dimensional outputs via a linear projection of a $z$-width representation. In both cases, the decoder contributes a term $\mathcal{O}(BGz)$. The encoder front $G \to H$ remains $\mathcal{O}(BGH)$. Consequently, the leading-order training cost can be summarized as

$$C_{\text{LDVAE/MrVI}} \;\approx\; \mathcal{O}\big(BG(H + z)\big) \;+\; (BH^2,\, BHz \text{ terms}).$$

A small increase in $z$ yields

$$\Delta C_{\text{LDVAE/MrVI}}(z) \;\approx\; \mathcal{O}(BG\,\Delta z),$$

while a small increase in $H$ yields

$$\Delta C_{\text{LDVAE/MrVI}}(H) \;\approx\; \mathcal{O}(BG\,\Delta H).$$

*Per unit of width*, $\Delta z$ and $\Delta H$ cost the same order: both scale like $BG$. What makes $z$ cheaper *in practice* is the size of the step: we usually take $\Delta z$ in the tens, whereas $\Delta H$ is commonly a large jump (e.g., $+128$). Formally, the fractional overheads satisfy

$$\frac{\Delta C}{C}\Big|_{z} \;\approx\; \frac{\Delta z}{H + z}, \qquad \frac{\Delta C}{C}\Big|_{H} \;\approx\; \frac{\Delta H}{H + z},$$

so for typical magnitudes ($H$ hundreds, $z$ tens) a $+10$–$+40$ change in $z$ is usually less expensive than a $+128$ change in $H$, even though $z$ appears in a "big" $BGz$ term. By contrast, changing $G$ or $B$ directly scales every large multiply ($\propto BG$), so those knobs are inherently costly; therefore, restricting to HVGs (smaller $G$) with tuning $z$ is such a powerful lever for LDVAE/MrVI.

**Memory footprint.**    In scVI, raising $z$ only widens the $H \leftrightarrow z$ blocks, adding $\mathcal{O}(H\,\Delta z)$ parameters. In LDVAE/MrVI, the gene projection widens with $z$, adding $\mathcal{O}(G\,\Delta z)$ parameters, which are manageable with HVGs, but much larger if using all genes.

**Guidance synthesizing magnitudes and step sizes.**    Because $H$ and $G$ typically move in large chunks and $B$ multiplies all costs, the small and targeted adjustments available through $z$ make it the most economical capacity knob across models for different reasons: in scVI it is intrinsically cheap ($\Delta T \propto BH\,\Delta z$ while $\mathcal{O}(BHG)$ is unchanged), and in LDVAE/MrVI it is cheaper in practice because $\Delta z$ (tens) is much smaller than common $\Delta H$ (hundreds) or changes to $G$ (thousands). Accordingly, our practical recipe is: use HVGs to control $G$, keep networks shallow, and treat $z$ as the first tuning knob. In particular, scVI remains the go-to model when one wants strong integration with minimal incremental training cost from increasing latent dimensionality.

## 4.2 SCORE–COMPLEXITY TRADE-OFFS

Figure 6 summarizes the effect of varying a single hyperparameter while holding the others fixed. The first panel reports the mean score, and the second reports the mean training time, both averaged across datasets. Across models, increasing the Z-latent dimensionality $n_{\text{latent}}Z$ yields the most favorable score–cost trade-off, especially for scVI, typically improving performance with the smallest increase in memory and only modest growth in time. Pairwise hyperparameter comparisons in the Appendix further show in detail that raising $n_{\text{latent}}Z$ benefits most individual metrics and outperforms alternatives such as increasing depth or hidden width in terms of score–time/memory efficiency.

## 4.3 MODEL RECOMMENDATIONS

**scVI**. Among the evaluated models, scVI is the least expensive to tune in terms of the latent dimension z. Across four datasets spanning a wide range of cell types and technologies, scVI consistently outperformed the alternatives, making it a strong default choice for scRNA-seq analysis. In practice, we recommend tuning the latent dimension $z$ first—using highly variable genes (HVGs)—as this offers the best accuracy–efficiency trade-off and minimizes computational cost. Across all datasets we observe that increasing the latent dimensionality $z$ consistently improves both batch integration and biological conservation; in practice, we recommend moderate–to–high $z$ (30 -50). In contrast, increasing the hidden dimension from $H = 128$ to $H = 256$ yields only marginal gains, especially once $z > 30$ (see Fig. 12) and can even reduce biological conservation, while adding depth typically provides small benefits at best and more often degrades performance at higher layer counts. It is important to note that scVI comprises an MLP encoder with two output heads and one decoder, so the operations that scale with $z$ are confined to $H \to z$ (two heads) and $z \to H$, giving train-time complexity $\mathcal{O}(BHz)$, which is linear in $z$. In contrast, the genes facing layers $G \to H$ and $H \to G$ scale as $\mathcal{O}(BGH)$, and each additional hidden layer contributes $\mathcal{O}(BH^2)$. Since typical regimes have $z$ in the tens, $H$ in the hundreds, and $G$ in the thousands, the marginal compute of expanding $z$ is negligible, making $n_{\text{latent}}$ a compute efficient lever that often improves accuracy without meaningful run-time or memory penalties

**MrVI**. We observe that setting the latent $z$ dimensionality in the moderate–high range yields the best results; increasing $n_{\text{layers}}$ offers little benefit and, in the Human Immune dataset, setting $n_{\text{layers}} = 3$ steeply decreases overall biological conservation. The t-SNE maps show improvement when increasing the auxiliary latent $u$ from 10 to 20, and in certain settings at higher $n_{\text{latent}}z$, $u = 20$ preserves the compactness of clustering. We therefore recommend, for MrVI with HVG features, using moderate–high $n_{\text{latent}} \in [20, 40]$ with $u = 20$, a shallow depth $n_{\text{layers}} = 1$, and $n_{\text{hidden}} = 128$. Furthermore, from the Tabula Muris dataset, we see that MrVI exhibited strong stability and excelled at batch correction on highly heterogeneous datasets (154 cell types), largely due to its architecture, which includes a sample-aware covariate capable of accommodating multiple protocols. We therefore recommend MrVI for complex, multi-protocol datasets. However, it is important to note that MrVI is the most computationally expensive model.

**LDVAE**. Excels at interpretability and scaling because it allows to visualize how latent dimensions link to genes, but has less capacity to model complex, nonlinear manifolds and to integrate datasets with complex batch effects. It performs best when heterogeneity is dominated by broad lineages (as in Human Immune), where gene programs are nearly axis-aligned and a linear decoder cleanly preserves biology across batches. By contrast, on single-tissue PBMC cohorts rich in donor/time effects and fine subtype boundaries and on Tabula Muris, which is a multi-tissue, cross-protocol atlas, nonlinearity buys accuracy, and LDVAE's linearity becomes a bottleneck.

Our results indicate that HVG is a preferable gene-selection strategy, consistent with prior studies Zappia et al. (2025); Luecken et al. (2022), thereby serving as a sanity check for our analysis. In general, we recommend scVI as the go-to model across datasets; although many studies utilize scVI in their analysis, they tend to use the default settings ($n_{\text{latent}}$=10, $n_{\text{layers}}$=1, $n_{\text{hidden}}$=128), as cited in the introduction, which is not the best performing choice, and this reliance on defaults overlooks the fact that parameter tuning can yield improved results, as our analysis demonstrates. A strong, compute-efficient starting point is to use HVG features and a moderate to high latent size $z$ in the range 30–50, a shallow encoder with $n_{\text{layers}}$=1, 2, and $n_{\text{hidden}}$=128.

## 5 CONCLUSION

In this study, we systematically benchmarked VAE-based batch-integration methods (scVI, MrVI, LDVAE) across three datasets by sweeping feature sets (HVG vs. Full), latent size, network depth, and width. We found that widely used defaults are not optimal: HVG features and a moderate to high latent dimension (30 to 50) improve the overall quality of integration. scVI emerges as the most reliable general choice, LDVAE best preserves biology in the most heterogeneous dataset, and MrVI benefits from $u$=20 and larger $z$ but adds computational overhead. Practically, we recommend adopting HVG by default, first tuning $z$, keeping depth shallow and only widening when necessary, yielding a robust and compute-efficient recipe for single-cell integration.

ACKNOWLEDGMENTS

We acknowledge using OpenAI's generative AI (ChatGPT; `https://openai.com`) to improve the clarity of the writing and to assist with formatting tables and equations.

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

## A  DATA AVAILABILITY AND REPRODUCIBILITY

Benchmarking was done with Snakemake v9.11.5 to ensure a reproducible and scalable analysis pipeline. Upon acceptance, we will release the complete workflow including environment and code on GitHub, and publish all 720 trained models on Hugging Face to enable reuse and replication. To facilitate exploration, we provide an interactive Jupyter notebook for hyperparameter sweeps in which users can vary $n_{\text{latent}}(z)$, $n_{\text{latent}}(u)$, $n_{\text{hidden}}$, $n_{\text{layers}}$, and the feature set; plots update in real time to streamline comparison and guide selection of configurations for their study.

We conducted all experiments on Ubuntu 22.04 LTS with an NVIDIA RTX 4090 (24 GB VRAM) and 128 GB RAM, Python v3.12.2 (`https://www.python.org`); Scanpy v1.11.0 for preprocessing and visualization `https://scanpy.readthedocs.io`); scvi-tools v1.3.0 for probabilistic modeling with scVI, MrVI, and LDVAE `https://docs.scvi-tools.org`); PyTorch v2.6.0+cu124 as a primary deep-learning backend `https://pytorch.org`); JAX v0.4.35 (Google AI, `https://github.com/google/jax`); scIB v1.1.7 for integration and conservation metrics `https://scib.readthedocs.io`, and snakemake v9.11.5 `https://snakemake.readthedocs.io/en/stable/`.

This study uses three single-cell RNA sequencing datasets that span various experimental conditions and batch structures. The first dataset is the Human Immune Cells dataset from the Open Problems in Single-Cell Analysis initiative Luecken et al. (2022), accessible at `https://openproblems.bio/datasets/openproblems_v1/immune_cells`. It includes 33,506 immune cells and 12,303 genes collected from peripheral blood and bone marrow, distributed across ten batches and generated using both 10X Genomics and Smart-seq2 technologies. In addition, two datasets were obtained from Zenodo upon request. The 24 PBMC dataset contains 76,535 cells and 36,601 genes collected from capillary blood samples across 14 batches using 10X Genomics Barrow & Inc. (2023a), accessible upon request at `https://zenodo.org/records/8020792`. The 18 PBMC dataset includes 55,260 cells and 36,601 genes from four batches, corresponding to samples collected during a remission-inducing antibiotic intervention Barrow & Inc. (2023b)accessible upon request at `https://zenodo.org/records/11100300`.

## B    DATASETS' STATISTICS

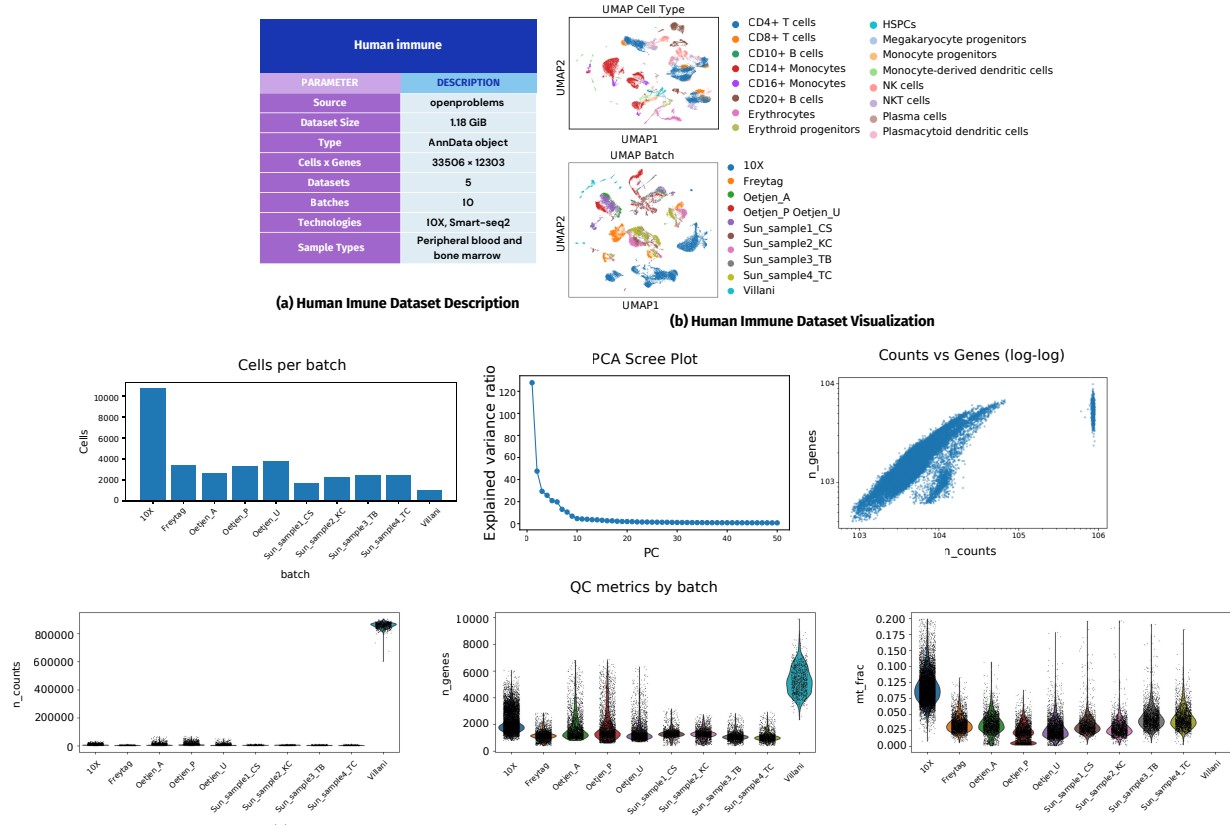

(a) Human Imune Dataset Description

(b) Human Immune Dataset Visualization

(c) cells-per-batch, PCA scree, counts–genes, and batch-wise QC  across all cells/batches of the Human Immune dataset.

**Figure 7:** *Summary of dataset description and visualization used in Human Immune Dataset. (a) OpenProblems immune dataset: 33,506 cells and 12,303 genes across 5 datasets and 10 batches, generated using 10X and Smart-seq2 technologies from peripheral blood and bone marrow samples. (b) UMAP projections of the OpenProblems dataset colored by cell type (top) and batch (bottom), highlighting biological diversity and batch heterogeneity. (c) PCA scree plot, counts–genes, and mitochondrial fraction (top), dataset-level QC summary (bottom).*

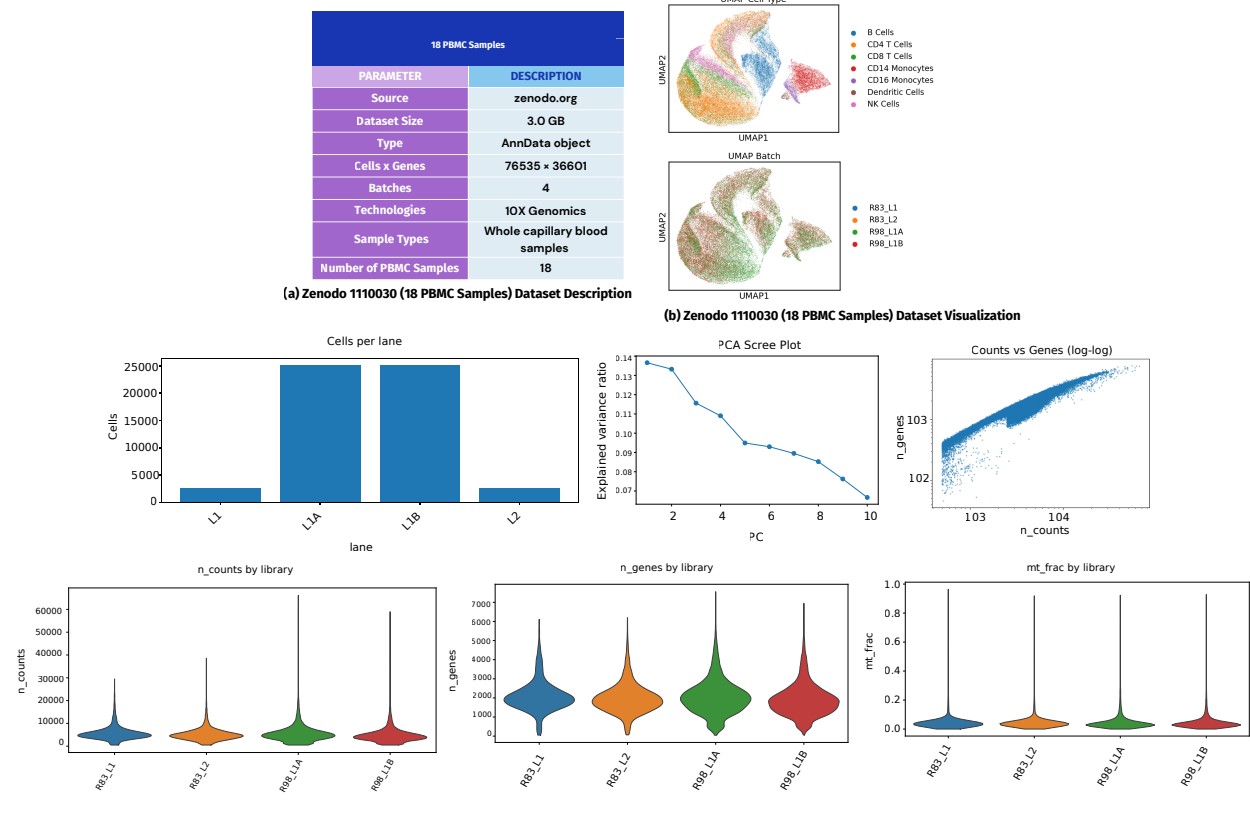

(a) Zenodo 1110030 (18 PBMC Samples) Dataset Description

(b) Zenodo 1110030 (18 PBMC Samples) Dataset Visualization

(c) cells-per-batch, PCA scree, counts–genes, and batch-wise QC across all cells/batches of the 18 PBMC Samples dataset.

**Figure 8:** *Summary of dataset description and visualization used in Zenodo 11100300 (18 PBMC Samples). (a) Zenodo 11100300 (18 PBMC Samples): 55,260 cells and 36,601 genes across 4 batches from whole capillary blood samples generated using 10X Genomics. (b) UMAP projections of Zenodo 8020792 (24 PBMC Samples) colored by cell type (top) and batch (bottom), showing pronounced batch effects and distinct cell type clustering. (c) PCA scree plot, counts–genes, and mitochondrial fraction (top), dataset-level QC summary (bottom).*

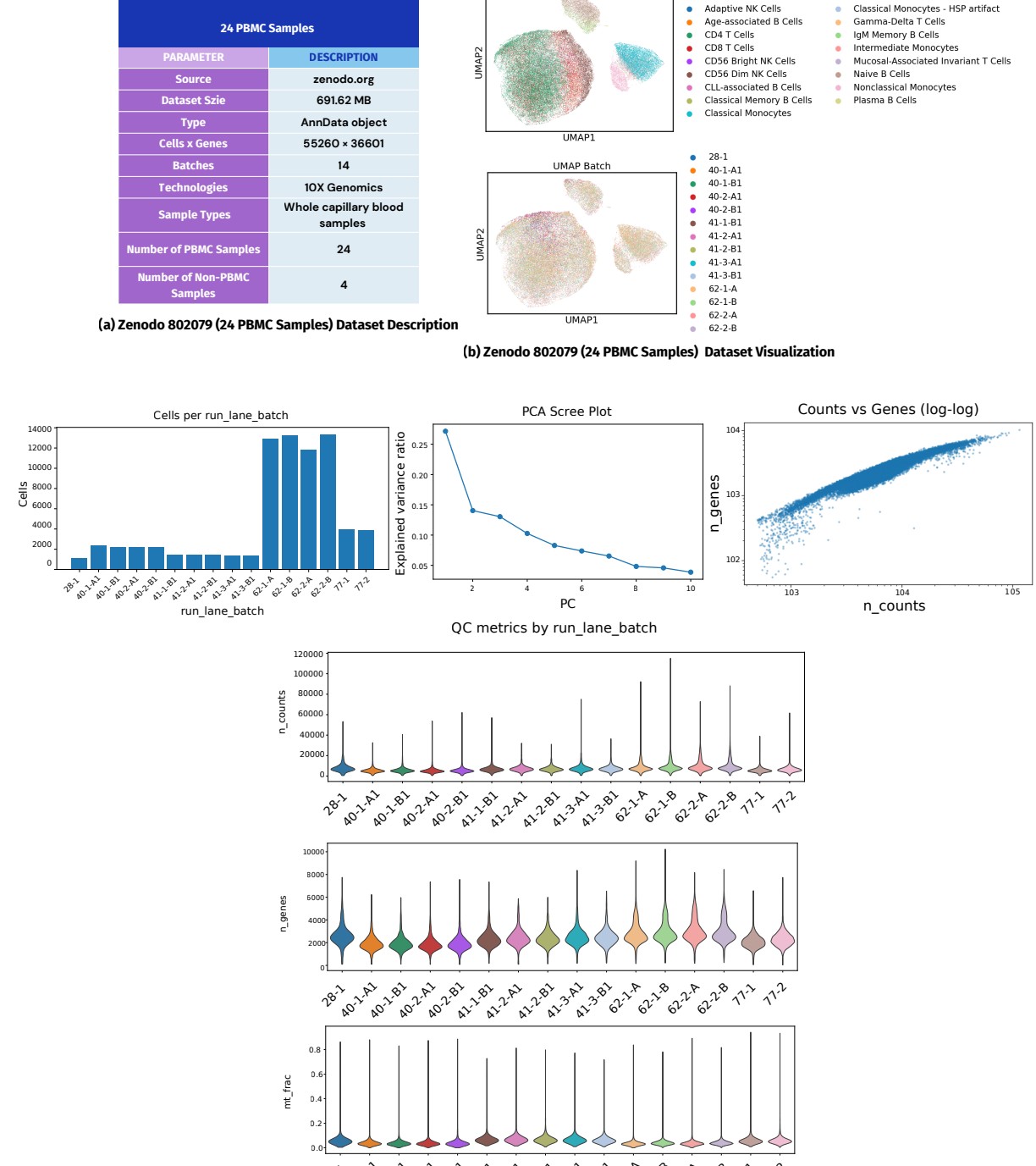

(a) Zenodo 802079 (24 PBMC Samples) Dataset Description

(b) Zenodo 802079 (24 PBMC Samples) Dataset Visualization

(c) cells-per-batch, PCA scree, counts–genes, and batch-wise QC across all cells/batches of the 24 PBMC Samples dataset.

**Figure 9:** *Summary of dataset description and visualization used in Zenodo 8020792 (24 PBMC Samples). (a) Zenodo 8020792 (24 PBMC Samples): 76,535 cells and 36,601 genes across 14 batches from whole capillary blood samples processed using 10X Genomics. (b) UMAP projections of Zenodo 8020792 (24 PBMC Samples) colored by cell type (top) and batch (bottom), showing pronounced batch effects and distinct cell type clustering. (c) PCA scree plot, counts–genes, and mitochondrial fraction (top), dataset-level QC summary (bottom).*

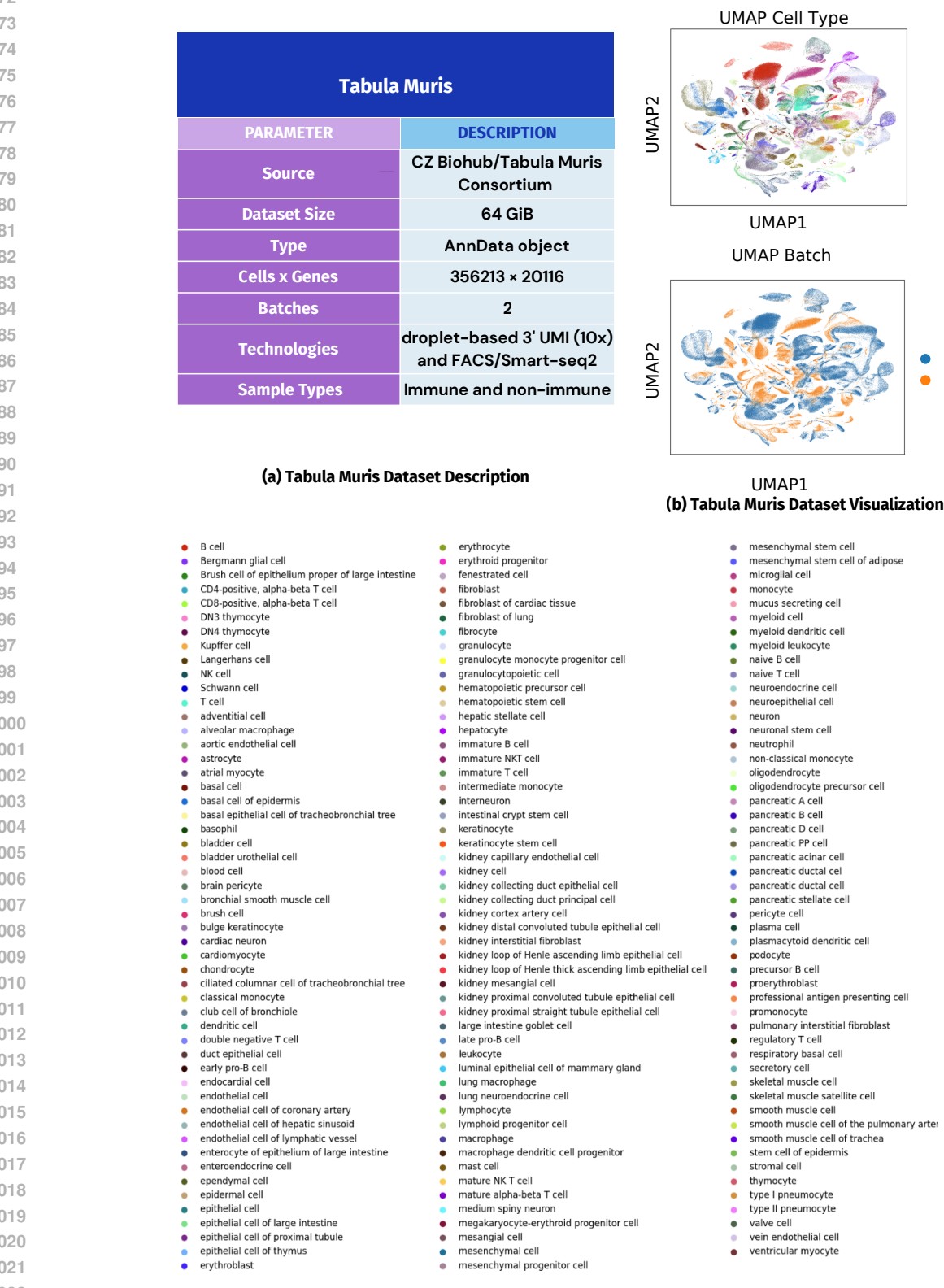

**Figure 10:** *Summary of dataset description and visualization used in Tabula Muris. (a) Tabula Muris: 356,123 cells and 20,116 genes across 154 distinct cell types (b) UMAP projections of Tabula Muris colored by cell type (top) and batch (bottom).*

## C METHODS

### C.1 SCVI FROM LOPEZ ET AL. (2018)

**Observed data and annotations.** For cells $n = 1, \ldots, N$ and genes $g = 1, \ldots, G$, let $x_{ng} \in \mathbb{N}$ be observed counts. Each cell has a (known) batch annotation $s_n \in \{1, \ldots, B\}$. The model conditions on $s_n$.

**Latent variables.** scVI introduces a low-dimensional biological latent $z_n \in \mathbb{R}^d$ and a cell-specific library-size factor $\ell_n > 0$ (on the log scale). Priors:

$$z_n \sim \mathcal{N}(0, I), \qquad \ell_n \sim \mathrm{LogNormal}(\ell_\mu, \ell_\sigma^2),$$

with batch-specific hyperparameters $(\ell_\mu, \ell_\sigma) \in \mathbb{R}_+^B \times \mathbb{R}_+^B$.

**Decoder parameterization (batch-conditioned).** Given $(z_n, s_n)$, scVI decodes gene-wise parameters via neural networks:

$$\rho_n = f_w(z_n, s_n) \in \Delta^{G-1}, \qquad \pi_{ng} = f_g^h(z_n, s_n) \in (0, 1),$$

where $\rho_n$ is a batch-corrected, normalized vector of transcript proportions (summing to one) and $\pi_{ng}$ is the zero-inflation probability. The mean of the NB component is $\mu_{ng} = \ell_n \rho_{ng}$ with gene-specific inverse-dispersion $\theta_g > 0$.

**Generative process.** For each pair $(n, g)$, the model draws

$$z_n \sim \mathcal{N}(0, I), \quad \ell_n \sim \mathrm{LogNormal}(\ell_\mu, \ell_\sigma^2), \quad \rho_n = f_w(z_n, s_n),$$
$$w_{ng} \sim \mathrm{Gamma}(\rho_{ng}, \theta_g), \quad y_{ng} \sim \mathrm{Poisson}(\ell_n w_{ng}),$$
$$h_{ng} \sim \mathrm{Bernoulli}\big(f_g^h(z_n, s_n)\big), \quad x_{ng} = \begin{cases} y_{ng}, & h_{ng} = 0 \\ 0, & h_{ng} = 1. \end{cases}$$

Marginalizing $(w_{ng}, y_{ng})$ yields a zero-inflated negative binomial (ZINB) likelihood for $x_{ng}$ with mean $\mu_{ng} = \ell_n \rho_{ng}$, dispersion $\theta_g$, and zero-inflation $\pi_{ng}$.

**Approximate posterior (amortized VI).** Encoders parameterize a mean-field variational family

$$q(z_n, \ell_n \,|\, x_n, s_n) = q(z_n \,|\, x_n, s_n) \, q(\ell_n \,|\, x_n, s_n),$$

with $q(z_n|\cdot)$ Gaussian and $q(\ell_n|\cdot)$ log-normal. The evidence lower bound is

$$\log p(x \,|\, s) \; \geq \; \mathbb{E}_{q(z,\ell \,|\, x,s)}\big[\log p(x \,|\, z, \ell, s)\big] - \sum_n \mathrm{KL}\big(q(z_n|x_n, s_n) \,\|\, p(z_n)\big) - \sum_n \mathrm{KL}\big(q(\ell_n|x_n, s_n) \,\|\, p(\ell_n)\big),$$

with $\{\theta_g\}$ optimized as global variables. Training uses reparameterization and mini-batching.

**How batch integration is realized in the model.** Batch enters the generative path only through the decoder conditionals $f_w(\cdot, s_n)$ and $f^h(\cdot, s_n)$. This induces a latent $z_n$ that emphasizes biology while the decoder accounts for batch-dependent shifts in the observation model.

### C.2 LDVAE FROM SVENSSON ET AL. (2020)

LDVAE replaces scVI's non-linear decoder with a *linear* reconstruction map, yielding logits $\eta_{ng} = v_g^\top z_n + b_g$ and proportions $h_n = \mathrm{softmax}(\eta_n)$, so rows of $V$ act as interpretable gene loadings directly linking latent axes to co-expressed "gene programs." This design deliberately trades a modest increase in reconstruction error for interpretability in very large datasets. In the paper's formulation, LDVAE is demonstrated in a single-batch setting with zero-inflation deactivated while retaining the scVI NB count model with a library-size factor.

### C.3 MRVI FROM BOYEAU ET AL. (2022)

**Observed variables and covariates.** For cell $n \in \{1, \ldots, N\}$ and genes $g \in \{1, \ldots, G\}$, let counts be $x_{ng} \in \mathbb{N}$. Each cell has a *target* sample ID $s_n \in \{1, \ldots, S\}$ and a *nuisance* covariate $b_n \in \{1, \ldots, B\}$ (e.g. batch/chemistry).

**Latent structure (two–level representation).** MrVI introduces (i) a *sample–unaware* latent $u_n \in \mathbb{R}^{L_u}$ that captures cell state and is independent of $(s_n, b_n)$, and (ii) a *sample–aware* latent $z_n \in \mathbb{R}^{L_z}$ that augments $u_n$ with sample effects while remaining independent of $b_n$. Priors:

$$u_n \sim \sum_{k=1}^{K} \pi_k \, \mathcal{N}(\mu_k, \Sigma_k) \quad \text{(MoG; } K{=}1 \text{ recovers a Gaussian prior),}$$

$$z_n \mid u_n \sim \mathcal{N}(u_n, I) \quad \text{or} \quad z_n \mid u_n \sim \mathcal{N}(A_{uz} u_n + \gamma_{uz}, \, I) \text{ if } L_z \neq L_u.$$

This hierarchy makes $u_n$ the harmonized cell-state space and $z_n$ the sample-conditioned refinement.

**Decoder with nuisance conditioning and NB likelihood.** Let $g_\theta(z_n, b_n)$ denote a multi-head attention module that injects nuisance effects. Define logits for gene-wise normalized expression

$$h_n = \text{softmax}\big(A_{zh}\,[\,z_n + g_\theta(z_n, b_n)\,] + \gamma_{zh}\big) \in \Delta^{G-1},$$

with learned $A_{zh} \in \mathbb{R}^{G \times L_z}$, $\gamma_{zh} \in \mathbb{R}^G$. Given a size factor $l_n > 0$ (taken as the library size in the paper) and gene-wise inverse dispersion $r_{ng} \geq 0$, the observation model is Negative Binomial

$$x_{ng} \sim \text{NB}\big(l_n h_{ng}, \, r_{ng}\big).$$

Thus nuisance $b_n$ affects only the observation path via $g_\theta$, while sample effects enter through $z_n$.

**Joint distribution.** Let $\phi = \{\pi_k, \mu_k, \Sigma_k\}_{k=1}^{K}$ and $\psi = \{A_{uz}, \gamma_{uz}, A_{zh}, \gamma_{zh}, \theta, r\}$ denote parameters. The generative model factorizes as

$$p_\phi(u_n)\, p_\psi(z_n \mid u_n)\, p_\psi(x_n \mid z_n, b_n) \quad \text{and} \quad p(x, u, z \mid s, b) = \prod_{n=1}^{N} p_\phi(u_n)\, p_\psi(z_n \mid u_n)\, p_\psi(x_n \mid z_n, b_n).$$

**Variational inference and ELBO.** With amortized posteriors $q_\lambda(u_n \mid x_n)$ and $q_\lambda(z_n \mid x_n, s_n)$, the evidence lower bound is

$$\mathcal{L} = \sum_{n=1}^{N} \mathbb{E}_{q_\lambda(u_n, z_n \mid x_n, s_n)}\big[\log p_\psi(x_n \mid z_n, b_n)\big] - \text{KL}\big(q_\lambda(z_n \mid x_n, s_n)\,\|\,p_\psi(z_n \mid u_n)\big) - \text{KL}\big(q_\lambda(u_n \mid x_n)\,\|\,p_\phi(u_n)\big),$$

optimized over $(\lambda, \phi, \psi)$ with reparameterization gradients and mini-batching.

# D    T-SNE EMBEDDINGS OF SCVI, MRVI, AND LDVAE ON IMMUNE BENCHMARK DATASET

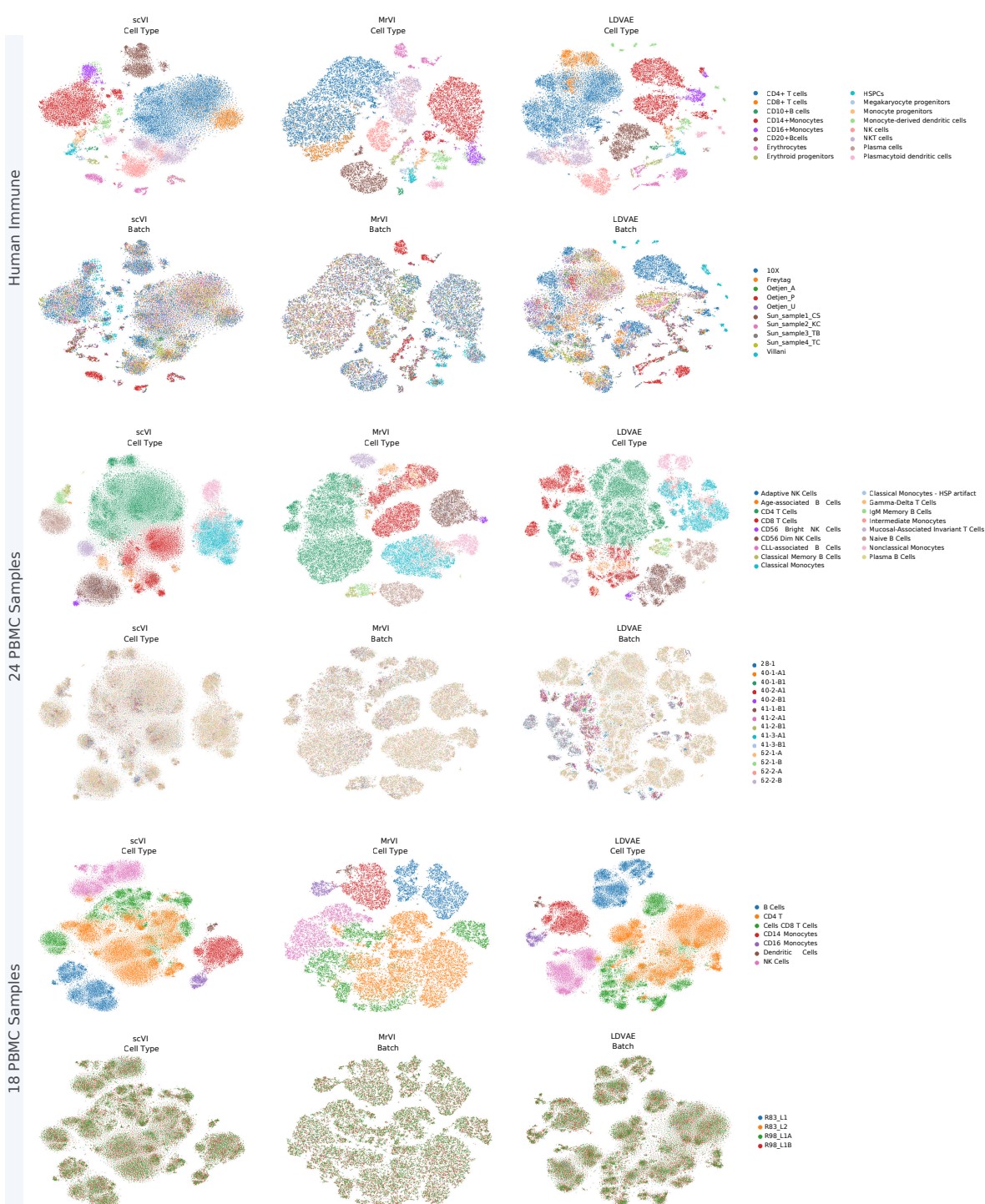

**Figure 11:** *t-SNE embeddings of scVI, MrVI, and LDVAE on immune benchmark datasets; for each panel, top = cell types, bottom = batches; models shown are the top-performers per dataset.*

# E    ZENODO 8020792 (24 PBMC SAMPLES) T-SNE MAPS

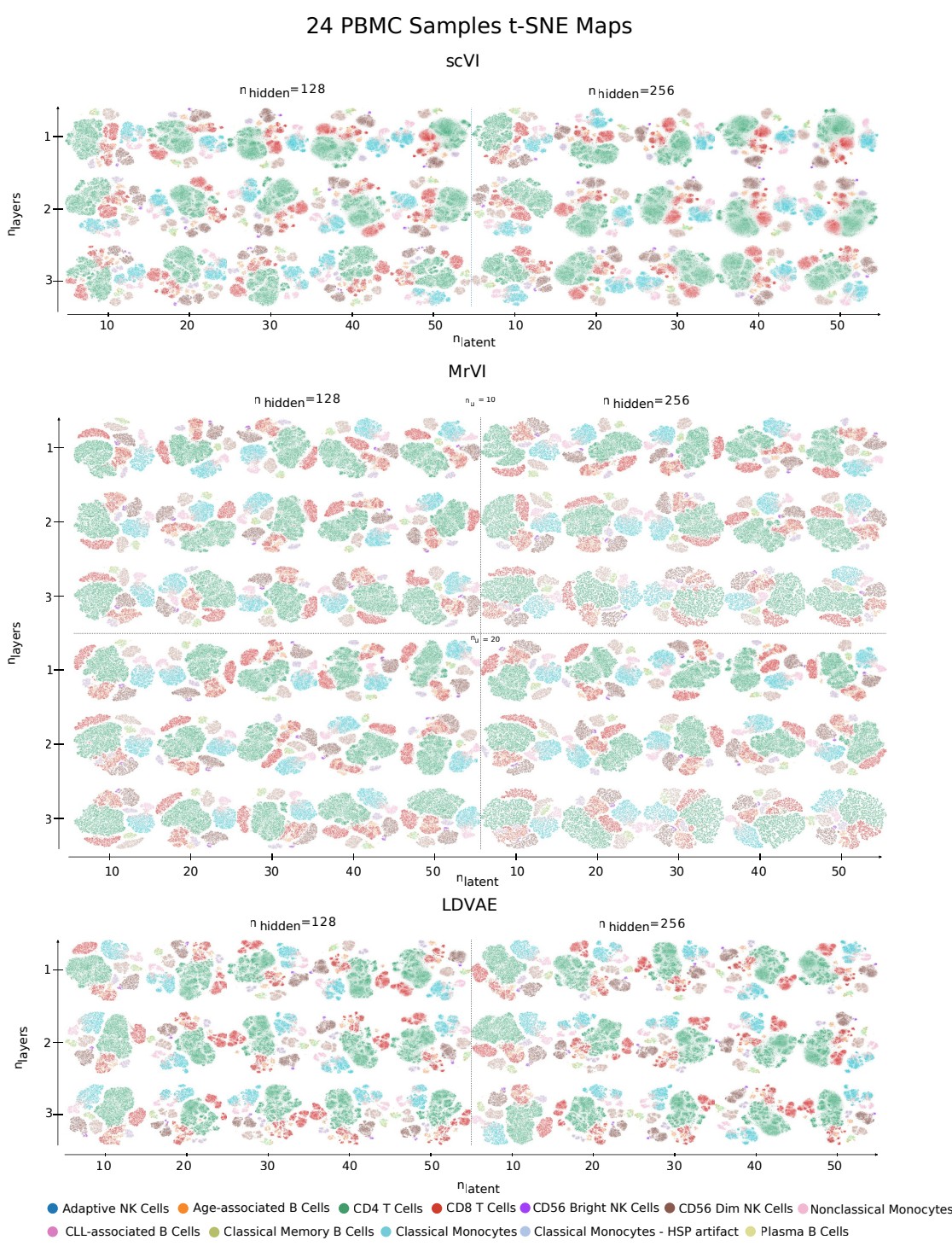

**Figure 12:** *t-SNE visualizations of the Zenodo 8020792 (24 PBMC samples) using HVGs. Embeddings show the separation of cell populations under different VAE model configurations (scVI, MrVI, LDVAE) across varying numbers of latent dimensions, hidden units, and layers.*

## F HUMAN IMMUNE T-SNE MAPS

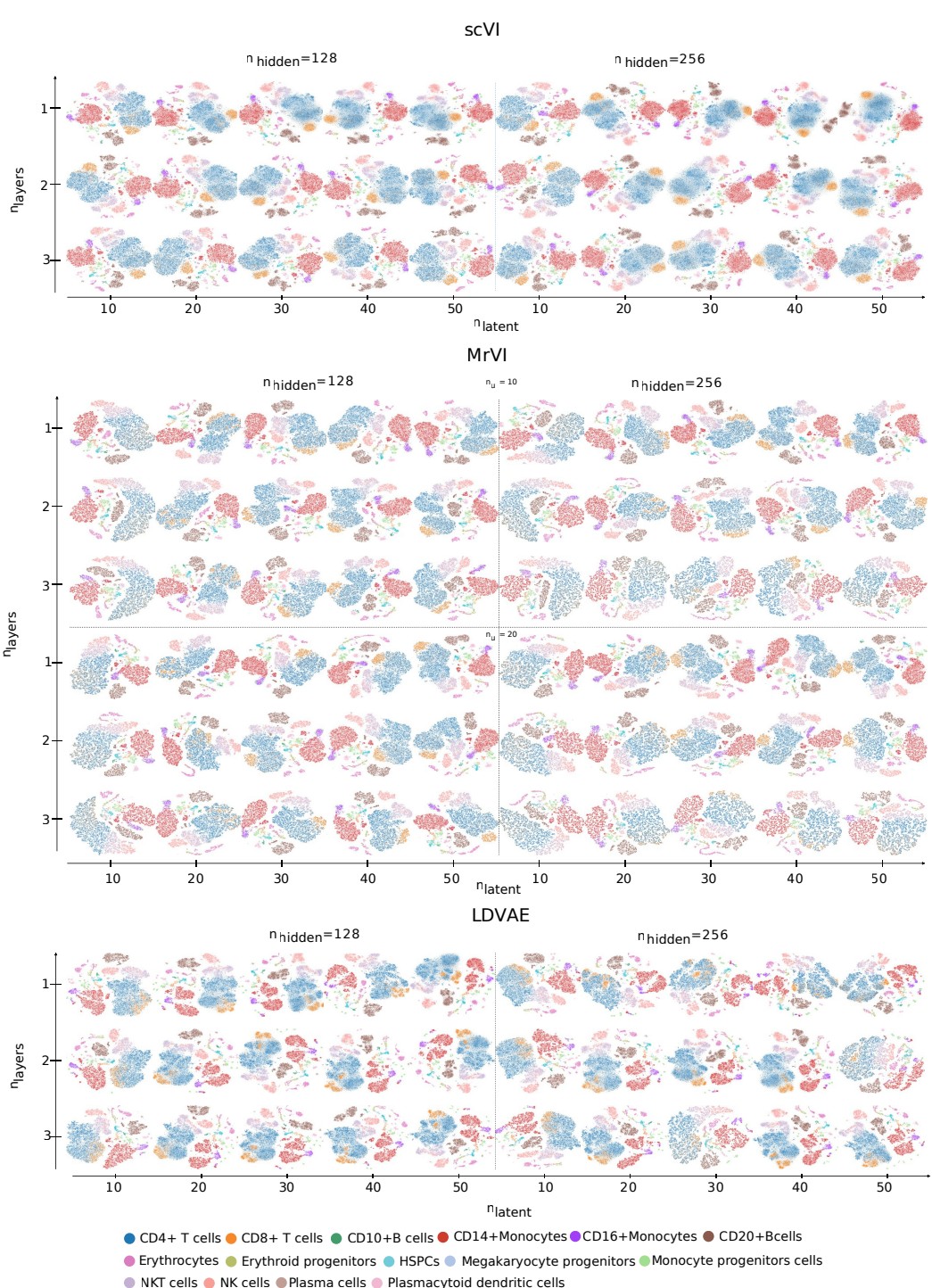

**Figure 13:** *t-SNE visualizations of the Human Immune dataset using HVGs. Embeddings show the separation of cell populations under different VAE model configurations (scVI, MrVI, LDVAE) across varying numbers of latent dimensions, hidden units, and layers.*

## G    18 PBMC Samples t-SNE Maps

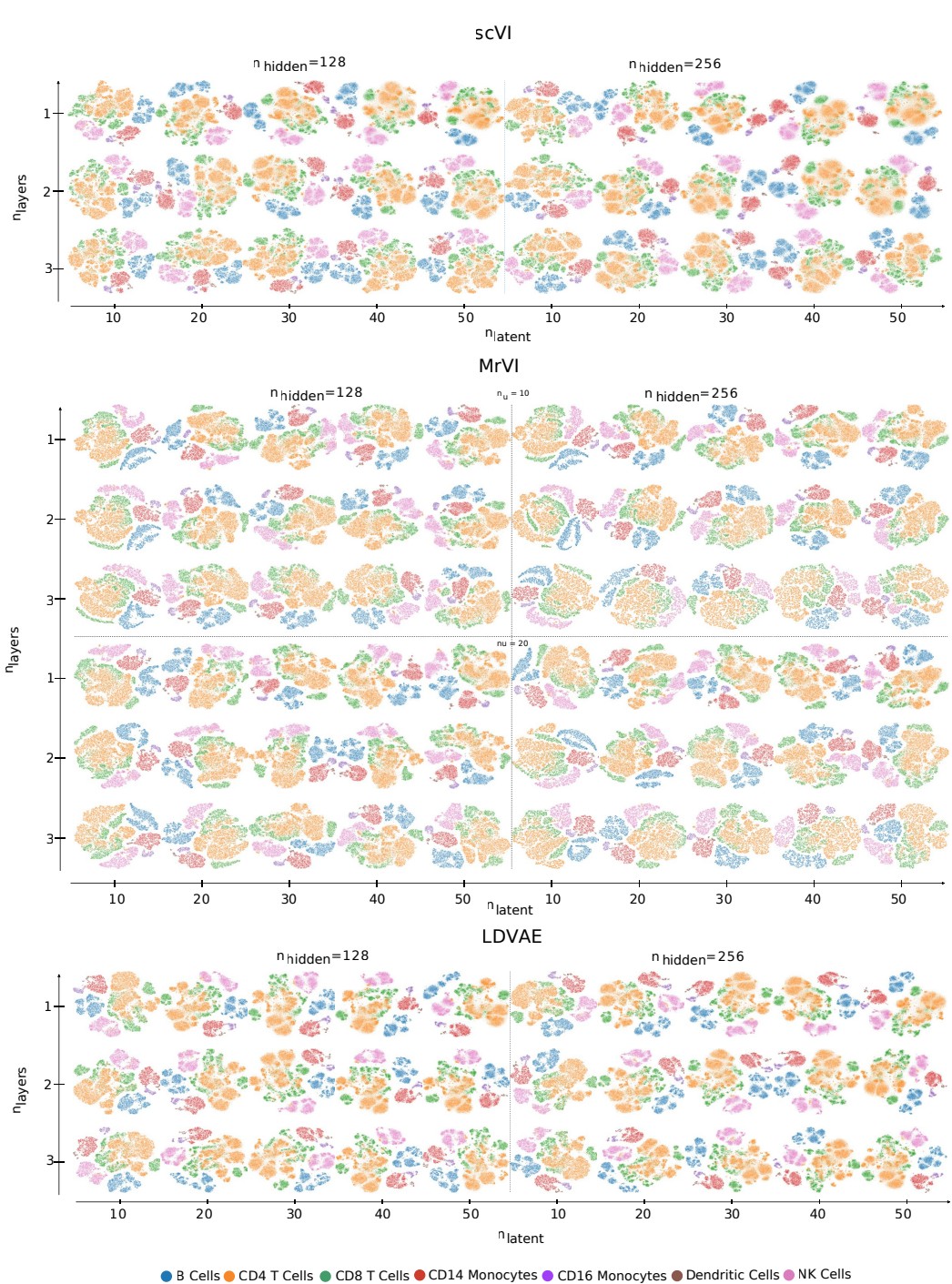

**Figure 14:** *t-SNE visualizations of the Zenodo 11100300 (18 PBMC samples) using HVGs. Embeddings show the separation of cell populations under different VAE model configurations (scVI, MrVI, LDVAE) across varying numbers of latent dimensions, hidden units, and layers.*

# H  UMAP VISUALIZATION FOR THE BEST PERFORMING MODELS

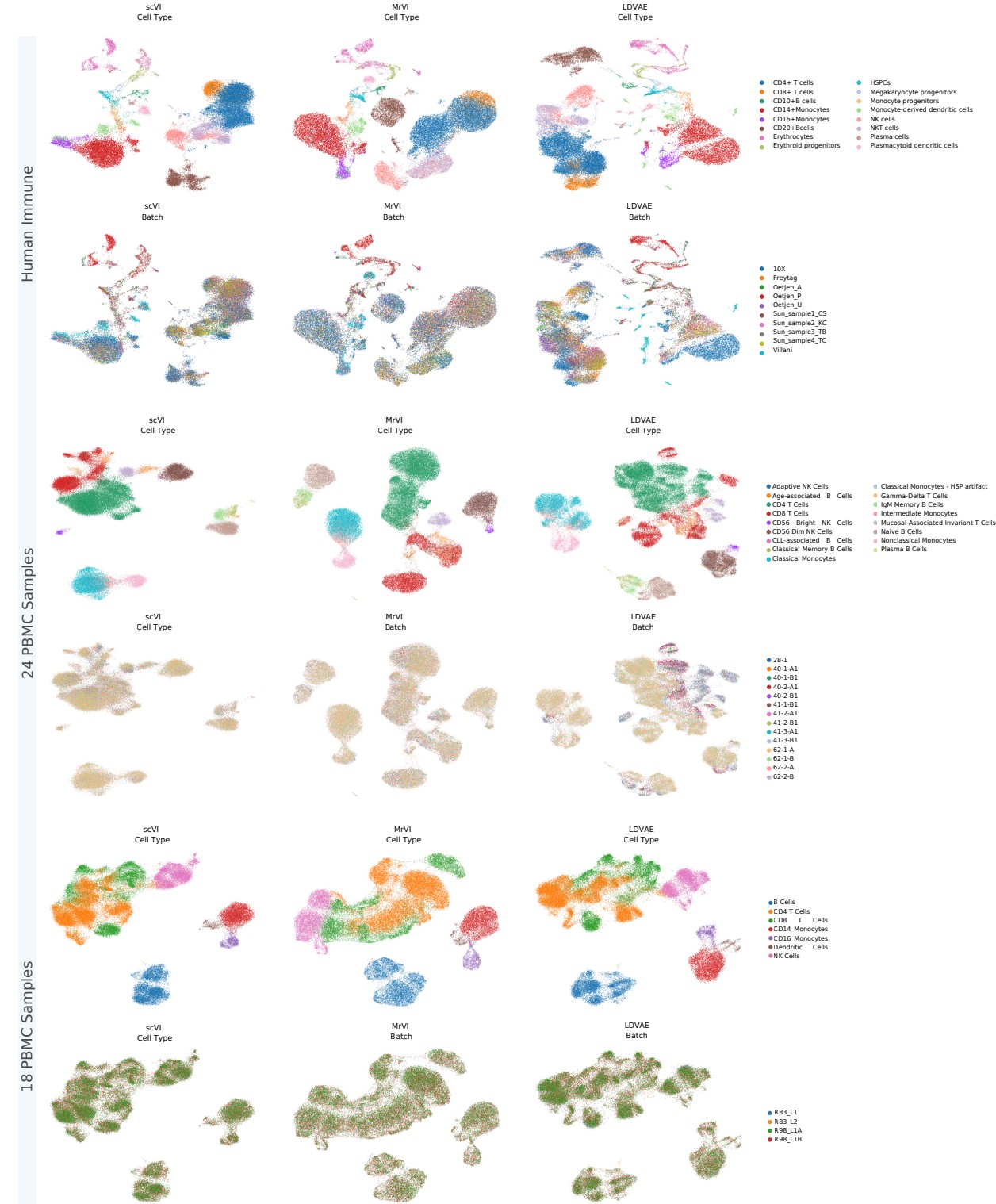

**Figure 15:** *UMAP of best performing models across datasets.*

# I  PER METRIC PERFORMANCE ON HUMAN IMMUNE

| Hyperparameters | Output | Features | Overall | Overall(BC) | Overall(Bio) | Batch correction | | | | Bio Conservation | | | | | | |
|---|---|---|---|---|---|---|---|---|---|---|---|---|---|---|---|---|
| | | | Overall | Overall(BC) | Overall(Bio) | ASW | PCR | iLISI | GC | NMI | ARI | ASW | IL F1 | IL ASW | cLISI | TC |
| n_hidden: 128, n_latent: 10, n_layers: 1 | Embedding | FULL | 0.74109 | 0.73658 | 0.74560 | 0.85750 | 0.90338 | 0.21506 | 0.97039 | 0.76544 | 0.60452 | **0.57565** | 0.80544 | **0.63365** | 0.99625 | 0.83826 |
| n_hidden: 128, n_latent: 10, n_layers: 1 | Embedding | HVG | 0.74541 | 0.73601 | 0.75481 | 0.86407 | 0.89363 | 0.21283 | 0.97349 | 0.77100 | 0.62854 | 0.57336 | 0.82949 | 0.62676 | 0.99571 | 0.85883 |
| n_hidden: 128, n_latent: 10, n_layers: 2 | Embedding | FULL | 0.75193 | 0.75452 | 0.74933 | 0.86168 | 0.92285 | 0.26481 | 0.96875 | 0.76669 | 0.62821 | 0.56394 | 0.82780 | 0.60720 | 0.99563 | 0.85587 |
| n_hidden: 128, n_latent: 10, n_layers: 2 | Embedding | HVG | 0.74780 | 0.75296 | 0.74263 | 0.87045 | 0.93187 | 0.24452 | 0.96501 | 0.77877 | 0.64201 | 0.57415 | 0.74930 | 0.61149 | 0.99605 | 0.84668 |
| n_hidden: 128, n_latent: 10, n_layers: 3 | Embedding | FULL | 0.74664 | 0.76424 | 0.72903 | 0.86513 | 0.94162 | 0.27824 | 0.97198 | 0.75048 | 0.61805 | 0.56166 | 0.72105 | 0.59635 | 0.99476 | 0.86089 |
| n_hidden: 128, n_latent: 10, n_layers: 3 | Embedding | HVG | 0.76483 | 0.75927 | 0.77039 | 0.86861 | 0.94354 | 0.27630 | 0.94862 | 0.79237 | 0.80098 | 0.56579 | 0.77012 | 0.60047 | 0.99465 | 0.86837 |
| n_hidden: 128, n_latent: 20, n_layers: 1 | Embedding | FULL | 0.75133 | 0.75740 | 0.74526 | 0.88847 | 0.91482 | 0.23726 | 0.98907 | 0.78594 | 0.64529 | 0.55151 | 0.79935 | 0.59170 | 0.99529 | 0.84771 |
| n_hidden: 128, n_latent: 20, n_layers: 1 | Embedding | HVG | 0.75209 | 0.75831 | 0.74587 | 0.89419 | 0.91633 | 0.24556 | 0.97715 | 0.79248 | 0.63690 | 0.55009 | 0.80334 | 0.59874 | 0.99527 | 0.84430 |
| n_hidden: 128, n_latent: 20, n_layers: 2 | Embedding | FULL | 0.74937 | 0.76638 | 0.73236 | 0.88598 | 0.93223 | 0.27149 | 0.97581 | 0.76364 | 0.62376 | 0.55434 | 0.73270 | 0.59764 | 0.99419 | 0.86027 |
| n_hidden: 128, n_latent: 20, n_layers: 2 | Embedding | HVG | 0.76433 | 0.76797 | 0.76069 | 0.88331 | 0.94011 | 0.26562 | 0.98284 | 0.78972 | 0.75033 | 0.55814 | 0.79299 | 0.58758 | 0.99452 | 0.85159 |
| n_hidden: 128, n_latent: 20, n_layers: 3 | Embedding | FULL | 0.75410 | 0.76979 | 0.73841 | 0.87126 | 0.95042 | 0.27956 | 0.97791 | 0.75889 | 0.62168 | 0.56206 | 0.75722 | 0.61547 | 0.99492 | 0.85860 |
| n_hidden: 128, n_latent: 20, n_layers: 3 | Embedding | HVG | 0.76768 | 0.76714 | 0.76822 | 0.87014 | 0.94381 | 0.27300 | 0.98159 | 0.80338 | 0.74817 | 0.55904 | 0.74226 | 0.60023 | 0.99455 | 0.86840 |
| n_hidden: 128, n_latent: 30, n_layers: 1 | Embedding | FULL | 0.76460 | 0.76814 | 0.76107 | 0.90181 | 0.93117 | 0.26108 | 0.97849 | 0.80338 | 0.74817 | 0.53801 | 0.80425 | 0.58747 | 0.99403 | 0.85218 |
| n_hidden: 128, n_latent: 30, n_layers: 1 | Embedding | HVG | 0.77531 | 0.77572 | 0.77490 | 0.90442 | 0.93728 | 0.27249 | 0.98871 | 0.82492 | 0.83917 | 0.53631 | 0.80515 | 0.57811 | 0.99335 | 0.84729 |
| n_hidden: 128, n_latent: 30, n_layers: 2 | Embedding | FULL | 0.77106 | 0.76808 | 0.77404 | 0.88450 | 0.93077 | 0.27044 | 0.98661 | 0.80807 | 0.80970 | 0.55478 | 0.80421 | 0.60358 | 0.99526 | 0.84268 |
| n_hidden: 128, n_latent: 30, n_layers: 2 | Embedding | HVG | 0.77259 | 0.76856 | 0.77662 | 0.88751 | 0.94036 | 0.26528 | 0.98107 | 0.82291 | 0.83212 | 0.55580 | 0.80803 | 0.59502 | 0.99461 | 0.82786 |
| n_hidden: 128, n_latent: 30, n_layers: 3 | Embedding | FULL | 0.76426 | 0.76632 | 0.76220 | 0.87014 | 0.94565 | 0.27398 | 0.97550 | 0.78975 | 0.81158 | 0.55412 | 0.72123 | 0.60202 | 0.99393 | 0.86277 |
| n_hidden: 128, n_latent: 30, n_layers: 3 | Embedding | HVG | 0.76530 | 0.76678 | 0.76382 | 0.87597 | 0.94754 | 0.27493 | 0.96866 | 0.78897 | 0.74628 | 0.55757 | 0.77200 | 0.61464 | 0.99470 | **0.87256** |
| n_hidden: 128, n_latent: 40, n_layers: 1 | Embedding | FULL | 0.77359 | 0.77675 | 0.77043 | 0.90675 | 0.93790 | 0.27453 | 0.98783 | 0.82471 | 0.83377 | 0.53614 | 0.78481 | 0.57646 | 0.99356 | 0.84356 |
| n_hidden: 128, n_latent: 40, n_layers: 1 | Embedding | HVG | 0.77735 | 0.78046 | 0.77424 | 0.90859 | 0.94579 | 0.28045 | 0.98699 | 0.83020 | 0.84034 | 0.53363 | 0.79981 | 0.57557 | 0.99348 | 0.84668 |
| n_hidden: 128, n_latent: 40, n_layers: 2 | Embedding | FULL | 0.77246 | 0.77081 | 0.77411 | 0.88343 | 0.93825 | 0.27558 | 0.98598 | 0.79546 | 0.81210 | 0.55521 | 0.80049 | 0.59897 | 0.99458 | 0.86196 |
| n_hidden: 128, n_latent: 40, n_layers: 2 | Embedding | HVG | 0.77213 | 0.76996 | 0.77429 | 0.89031 | 0.94810 | 0.26454 | 0.97688 | 0.81900 | 0.82970 | 0.55924 | 0.75020 | 0.60516 | 0.99524 | 0.86152 |
| n_hidden: 128, n_latent: 40, n_layers: 3 | Embedding | FULL | 0.75363 | 0.76973 | 0.73753 | 0.86818 | 0.94334 | 0.28063 | 0.98676 | 0.76367 | 0.62209 | 0.55904 | 0.75700 | 0.60466 | 0.99493 | 0.86131 |
| n_hidden: 128, n_latent: 40, n_layers: 3 | Embedding | HVG | 0.76391 | 0.76522 | 0.76259 | 0.87387 | 0.94480 | 0.27175 | 0.97049 | 0.78926 | 0.74938 | 0.56738 | 0.75262 | 0.62286 | 0.99564 | 0.86101 |
| n_hidden: 128, n_latent: 50, n_layers: 1 | Embedding | FULL | 0.76726 | 0.77670 | 0.75782 | 0.90743 | 0.93746 | 0.27333 | 0.98856 | 0.81804 | 0.77807 | 0.53295 | 0.77336 | 0.56556 | 0.99421 | 0.84258 |
| n_hidden: 128, n_latent: 50, n_layers: 1 | Embedding | HVG | 0.77909 | 0.77979 | 0.77838 | 0.91114 | 0.93990 | 0.27617 | 0.99197 | 0.82597 | 0.83647 | 0.53600 | 0.83615 | 0.57412 | 0.99394 | 0.84598 |
| n_hidden: 128, n_latent: 50, n_layers: 2 | Embedding | FULL | 0.76338 | 0.77136 | 0.75541 | 0.88362 | 0.93284 | 0.28353 | 0.98542 | 0.78824 | 0.73800 | 0.55307 | 0.75113 | 0.60387 | 0.99464 | 0.85888 |
| n_hidden: 128, n_latent: 50, n_layers: 2 | Embedding | HVG | 0.76694 | 0.77297 | 0.76091 | 0.89069 | 0.94386 | 0.27053 | 0.98678 | 0.80237 | 0.76385 | 0.55779 | 0.75211 | 0.59792 | 0.99443 | 0.85790 |
| n_hidden: 128, n_latent: 50, n_layers: 3 | Embedding | FULL | 0.76906 | 0.76541 | 0.77271 | 0.87201 | 0.94214 | 0.27809 | 0.96940 | 0.79578 | 0.80192 | 0.56064 | 0.77108 | 0.62462 | 0.99532 | 0.85960 |
| n_hidden: 128, n_latent: 50, n_layers: 3 | Embedding | HVG | 0.76010 | 0.76447 | 0.75572 | 0.87064 | 0.94108 | 0.27674 | 0.96942 | 0.79188 | 0.81125 | 0.56613 | 0.74262 | 0.60495 | 0.99551 | 0.77770 |
| n_hidden: 256, n_latent: 10, n_layers: 1 | Embedding | FULL | 0.73765 | 0.72590 | 0.74941 | 0.86791 | 0.85906 | 0.19985 | 0.97676 | 0.76443 | 0.61377 | 0.57531 | 0.80675 | 0.63292 | **0.99649** | 0.85619 |
| n_hidden: 256, n_latent: 10, n_layers: 1 | Embedding | HVG | 0.75881 | 0.73413 | **0.78349** | 0.86473 | 0.88178 | 0.21515 | 0.97485 | 0.82291 | 0.77604 | 0.57476 | **0.84222** | 0.62813 | 0.99600 | 0.84435 |
| n_hidden: 256, n_latent: 10, n_layers: 2 | Embedding | FULL | 0.75235 | 0.75769 | 0.74701 | 0.87514 | 0.90851 | 0.25952 | 0.98762 | 0.76655 | 0.63290 | 0.57115 | 0.78928 | 0.61806 | 0.99463 | 0.85650 |
| n_hidden: 256, n_latent: 10, n_layers: 2 | Embedding | HVG | 0.74879 | 0.75771 | 0.73988 | 0.87278 | 0.92695 | 0.25330 | 0.97779 | 0.77807 | 0.64158 | 0.56197 | 0.73953 | 0.61381 | 0.99455 | 0.84961 |
| n_hidden: 256, n_latent: 10, n_layers: 3 | Embedding | FULL | 0.74675 | 0.76976 | 0.72374 | 0.87549 | 0.94121 | 0.27909 | 0.98326 | 0.75576 | 0.62569 | 0.55608 | 0.67845 | 0.59681 | 0.99357 | 0.85980 |
| n_hidden: 256, n_latent: 10, n_layers: 3 | Embedding | HVG | 0.74782 | 0.76739 | 0.72825 | 0.87689 | 0.94852 | 0.26989 | 0.97425 | 0.76176 | 0.63656 | 0.55978 | 0.69199 | 0.60547 | 0.99267 | 0.84950 |
| n_hidden: 256, n_latent: 20, n_layers: 1 | Embedding | FULL | 0.75949 | 0.74367 | 0.77532 | 0.89341 | 0.87086 | 0.22113 | 0.98925 | 0.80739 | 0.77166 | 0.54988 | 0.81006 | 0.61006 | 0.99494 | 0.85517 |
| n_hidden: 256, n_latent: 20, n_layers: 1 | Embedding | HVG | 0.76087 | 0.75704 | 0.76471 | 0.89502 | 0.90197 | 0.24212 | 0.98904 | 0.80536 | 0.76248 | 0.54771 | 0.80470 | 0.59659 | 0.99479 | 0.84133 |
| n_hidden: 256, n_latent: 20, n_layers: 2 | Embedding | FULL | 0.77076 | 0.77606 | 0.76546 | 0.89876 | 0.93096 | 0.28652 | 0.98799 | 0.80484 | 0.76385 | 0.54715 | 0.80492 | 0.58677 | 0.99262 | 0.85808 |
| n_hidden: 256, n_latent: 20, n_layers: 2 | Embedding | HVG | 0.77897 | 0.78078 | 0.77717 | 0.90156 | 0.94879 | 0.28732 | 0.98543 | 0.82575 | 0.84478 | 0.53921 | 0.79517 | 0.58889 | 0.99176 | 0.85466 |
| n_hidden: 256, n_latent: 20, n_layers: 3 | Embedding | FULL | 0.77049 | 0.77646 | 0.76453 | 0.88676 | 0.95075 | 0.28659 | 0.98175 | 0.80885 | 0.78114 | 0.54437 | 0.78194 | 0.57785 | 0.99333 | 0.86421 |
| n_hidden: 256, n_latent: 20, n_layers: 3 | Embedding | HVG | 0.76355 | 0.78337 | 0.74372 | 0.89753 | 0.95638 | 0.29679 | 0.98281 | 0.77692 | 0.72984 | 0.54532 | 0.71473 | 0.58694 | 0.99143 | 0.86085 |
| n_hidden: 256, n_latent: 30, n_layers: 1 | Embedding | FULL | 0.75988 | 0.76035 | 0.75940 | 0.90602 | 0.88936 | 0.25675 | 0.98928 | 0.79706 | 0.74818 | 0.53376 | 0.80350 | 0.59512 | 0.99273 | 0.84543 |
| n_hidden: 256, n_latent: 30, n_layers: 1 | Embedding | HVG | 0.76621 | 0.77211 | 0.76030 | 0.90919 | 0.92024 | 0.26802 | 0.99099 | 0.80106 | 0.75157 | 0.53450 | 0.82047 | 0.57773 | 0.99362 | 0.84315 |
| n_hidden: 256, n_latent: 30, n_layers: 2, | Embedding | FULL | 0.77128 | 0.78135 | 0.76122 | 0.90486 | 0.94321 | 0.28848 | 0.98884 | 0.79420 | 0.74372 | 0.54638 | 0.80574 | 0.59372 | 0.99397 | 0.85081 |
| n_hidden: 256, n_latent: 30, n_layers: 2 | Embedding | HVG | 0.77771 | 0.78743 | 0.76799 | 0.91004 | 0.95392 | 0.29644 | 0.98932 | 0.82030 | 0.78462 | 0.53897 | 0.79992 | 0.58871 | 0.99211 | 0.85128 |
| n_hidden: 256, n_latent: 30, n_layers: 3 | Embedding | FULL | 0.75711 | 0.77726 | 0.73695 | 0.88628 | 0.94834 | 0.29082 | 0.98361 | 0.76429 | 0.62610 | 0.54607 | 0.78855 | 0.58835 | 0.99157 | 0.86133 |
| n_hidden: 256, n_latent: 30, n_layers: 3 | Embedding | HVG | 0.76645 | 0.78455 | 0.74835 | 0.89828 | **0.95909** | 0.29843 | 0.98241 | 0.77431 | 0.72206 | 0.54681 | 0.75846 | 0.58416 | 0.99158 | 0.86104 |
| n_hidden: 256, n_latent: 40, n_layers: 1 | Embedding | FULL | 0.75612 | 0.77057 | 0.74167 | 0.91668 | 0.90994 | 0.26322 | **0.99243** | 0.82291 | 0.77604 | 0.52619 | 0.78735 | 0.57285 | 0.99333 | 0.84196 |
| n_hidden: 256, n_latent: 40, n_layers: 1 | Embedding | HVG | 0.77540 | 0.78555 | 0.76526 | 0.91844 | 0.93806 | 0.29794 | 0.98774 | 0.81880 | 0.81456 | 0.52321 | 0.79726 | 0.56817 | 0.99113 | 0.84366 |
| n_hidden: 256, n_latent: 40, n_layers: 2 | Embedding | FULL | 0.75586 | 0.78025 | 0.73148 | 0.90287 | 0.93639 | 0.29254 | 0.98920 | 0.77502 | 0.61290 | 0.54468 | 0.74137 | 0.59004 | 0.99285 | 0.86347 |
| n_hidden: 256, n_latent: 40, n_layers: 2 | Embedding | HVG | **0.78105** | 0.78891 | 0.77318 | 0.90819 | 0.95547 | 0.30407 | 0.98791 | 0.81559 | 0.82496 | 0.54112 | 0.79797 | 0.58526 | 0.99149 | 0.85590 |
| n_hidden: 256, n_latent: 40, n_layers: 3 | Embedding | FULL | 0.76592 | 0.77928 | 0.75256 | 0.89098 | 0.95210 | 0.29472 | 0.97932 | 0.79260 | 0.76096 | 0.54845 | 0.71558 | 0.59043 | 0.99275 | 0.86715 |
| n_hidden: 256, n_latent: 40, n_layers: 3 | Embedding | HVG | 0.76612 | 0.78101 | 0.75124 | 0.89555 | 0.95302 | 0.29158 | 0.98388 | 0.78258 | 0.73259 | 0.54422 | 0.77145 | 0.58417 | 0.99131 | 0.85236 |
| n_hidden: 256, n_latent: 50, n_layers: 1 | Embedding | FULL | 0.77714 | 0.78089 | 0.77339 | 0.91940 | 0.92131 | 0.29318 | 0.98964 | **0.83233** | **0.85148** | 0.52172 | 0.79707 | 0.57136 | 0.99106 | 0.84869 |
| n_hidden: 256, n_latent: 50, n_layers: 1 | Embedding | HVG | 0.77438 | **0.79110** | 0.75766 | **0.92495** | 0.94135 | **0.30929** | 0.98883 | 0.81599 | 0.77335 | 0.51907 | 0.80080 | 0.56765 | 0.98967 | 0.83710 |
| n_hidden: 256, n_latent: 50, n_layers: 2 | Embedding | FULL | 0.76639 | 0.77931 | 0.75346 | 0.90551 | 0.93329 | 0.28826 | 0.99018 | 0.80601 | 0.76048 | 0.54120 | 0.72193 | 0.59521 | 0.99260 | 0.85080 |
| n_hidden: 256, n_latent: 50, n_layers: 2 | Embedding | HVG | 0.77577 | 0.78911 | 0.76243 | 0.91032 | 0.95376 | 0.30403 | 0.98831 | 0.79598 | 0.74972 | 0.53843 | 0.81603 | 0.58419 | 0.99173 | 0.86096 |
| n_hidden: 256, n_latent: 50, n_layers: 3 | Embedding | FULL | 0.76248 | 0.77734 | 0.74762 | 0.89070 | 0.95350 | 0.29188 | 0.97330 | 0.77377 | 0.72751 | 0.55226 | 0.72748 | 0.59964 | 0.99248 | 0.86023 |
| n_hidden: 256, n_latent: 50, n_layers: 3 | Embedding | HVG | 0.77459 | 0.78344 | 0.76574 | 0.89671 | 0.95468 | 0.29950 | 0.98289 | 0.80320 | 0.82349 | 0.54563 | 0.73872 | 0.59602 | 0.99206 | 0.86107 |

**Table 2:** *Performance of scVI on the Human Immune Dataset*

Table 2 reports a full hyperparameter sweep on the human immune dataset using scVI. There is a clear trade-off across $n_{\text{hidden}} \in \{128, 256\}$, $n_{\text{latent}} \in \{10, 20, 30, 40, 50\}$, $n_{\text{layers}} \in \{1, 2, 3\}$, and Features $\in \{$Full, HVG$\}$. Using the Overall score, the best configuration is 0.78105 at $(256, 40, 2, \text{HVG})$, while the worst is 0.73765 at $(256, 10, 1, \text{FULL})$. The batch-correction overall peaks under HVG with larger latent dimensionality $(256, 50, 1, \text{HVG})$, whereas the biological-conservation overall prefers HVG with smaller latent $(256, 10, 1, \text{HVG})$. These extremes indicate that model capacity and feature selection jointly set a spectrum between batch mixing and biological separability.

Changing $n_{\text{hidden}}$ from 128 to 256 showed in general small gains in batch-oriented metrics, Batch ASW (30/30), iLISI (22/30), Overall (batch) (23/30), and connectivity (28/30) improved in most paired settings, while PCR is mixed (16/30 up). Biological/label metrics often decrease (e.g., Overall(bio) 11/30 up, NMI 11/30 up, Label ASW 2/30 up). Increasing $n_{\text{layers}}$ from 1 to 3 further tilts toward batch mixing, with PCR (19/20) and iLISI (16/20) improving in most paired settings and overall batch success in 11/20 cases. The trajectory also improves in 19/20 pairs. In contrast, clustering agreement declines (NMI 3/20 up, 17/20 down; ARI 9/20 up, 11/20 down), and the biological overall decreases in 14/20 pairs. Other label metrics are mixed: The ASW label and the isolated ASW label more often increase (14/20 and 13/20, respectively), but the isolated label F1 worsens in 20/20 cases. The Overall score shifts only slightly (11/20 up), consistent with a small net effect at the aggregate level. HVG outperforms FULL on average for overall, batch-correction metrics, and clustering agreement (notably ARI), with only small penalties in connectivity and trajectory and near-neutral effects on label ASW; HVG is therefore a strong default.

To quantify the latent effect, we performed paired comparisons while holding $(n_{\text{hidden}}, n_{\text{layers}}, \text{Features})$ fixed and stepping $n_{\text{latent}}$ through $10 \to 20 \to 30 \to 40 \to 50$ (4 steps $\times$ 12 fixed triplets = 48 stepwise comparisons per metric). Batch-correction metrics improve in the large majority of steps: Overall(batch) 37/48 up, Batch ASW 37/48

up, PCR 33/48 up, and iLISI 35/48 up. Clustering agreement also trends upward (NMI 27/48 up; ARI 28/48 up). Label-compactness metrics tend to soften (Label ASW 14/48 up, 34/48 down; Isolated Label ASW 16/48 up, 32/48 down), and trajectory shows a near balance (23/48 up, 25/48 down). The biological overall is mixed on a stepwise basis (24/48 up, 24/48 down), yet the endpoint comparison ($n_{latent}$=50 versus 10) improves in 10/12 fixed triplets. The Overall rises in 31/48 steps, and the endpoint $10{\rightarrow}50$ improves in 11/12 cases.

**scVI Paired Up/Down/Flat Counts Per Metric by Hyperparameter comparison on Human Immune Dataset**

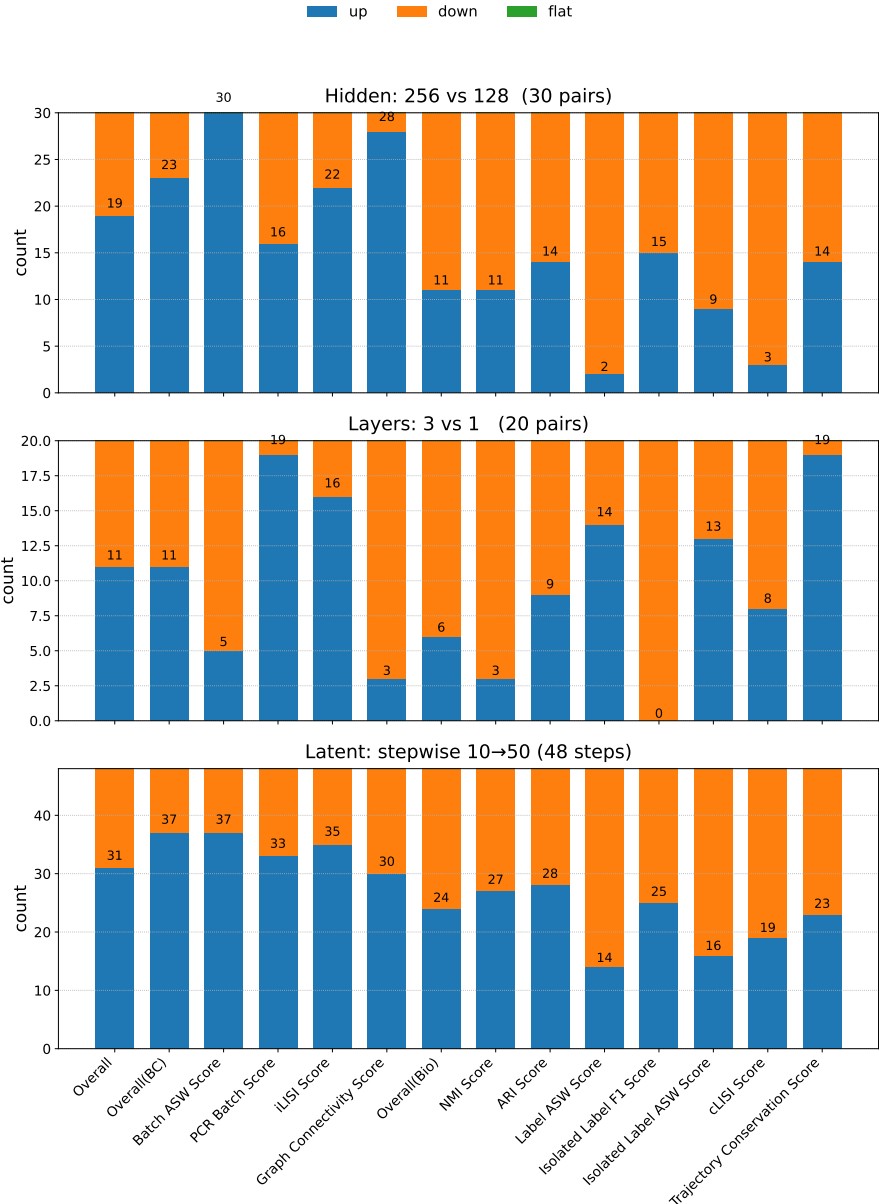

**Figure 16:** *Paired comparison of scVI hyperparameters on the Human Immune dataset, showing the number of metrics that improve ("up"), decline ("down"), or remain unchanged ("flat") when varying hidden units (256 vs. 128), network depth (3 vs. 1 layers), and latent dimensions (stepwise $10{\rightarrow}50$), evaluated on both full and HVG feature sets.*

| Hyperparameters | Output | Features | Overall | | | Batch correction | | | | Bio Conservation | | | | | | |
|---|---|---|---|---|---|---|---|---|---|---|---|---|---|---|---|---|
| | | | Overall | Overall(BC) | Overall(Bio) | ASW | PCR | iLISI | GC | NMI | ARI | ASW | IL F1 | IL ASW | cLISI | TC |
| n_hidden: 128, n_latent: 10, n_layers: 1, n_latent_u: 10 | Embedding | Full | 0.71903 | 0.70139 | 0.73667 | 0.81009 | 0.77938 | 0.30265 | 0.91344 | 0.73165 | 0.57299 | 0.57554 | 0.77221 | 0.64952 | 0.98818 | 0.86663 |
| n_hidden: 128, n_latent: 10, n_layers: 1, n_latent_u: 10 | Embedding | HVG | 0.73438 | 0.73819 | 0.73058 | 0.82166 | 0.84307 | 0.31766 | 0.97038 | 0.74922 | 0.58447 | 0.58702 | 0.69529 | 0.63232 | 0.98760 | 0.87813 |
| n_hidden: 128, n_latent: 10, n_layers: 2, n_latent_u: 10 | Embedding | Full | 0.72315 | 0.72636 | 0.71994 | 0.82906 | 0.80800 | 0.31094 | 0.95745 | 0.71205 | 0.52066 | 0.57417 | 0.71842 | 0.62798 | 0.98788 | **0.89845** |
| n_hidden: 128, n_latent: 10, n_layers: 2, n_latent_u: 10 | Embedding | HVG | 0.73363 | 0.74300 | 0.72426 | 0.82474 | 0.85769 | 0.32233 | 0.96723 | 0.71054 | 0.54515 | 0.58809 | 0.75203 | 0.62022 | 0.98721 | 0.86658 |
| n_hidden: 128, n_latent: 10, n_layers: 3, n_latent_u: 10 | Embedding | Full | 0.72104 | 0.73522 | 0.70686 | 0.82221 | 0.83983 | 0.32549 | 0.95337 | 0.68955 | 0.49511 | 0.58287 | 0.70116 | 0.62380 | 0.98607 | 0.86946 |
| n_hidden: 128, n_latent: 10, n_layers: 3, n_latent_u: 10 | Embedding | HVG | 0.70987 | 0.73345 | 0.68630 | 0.81148 | 0.84061 | **0.33148** | 0.95023 | 0.68028 | 0.46918 | 0.56880 | 0.63456 | 0.59832 | 0.98357 | 0.86936 |
| n_hidden: 128, n_latent: 20, n_layers: 1, n_latent_u: 10 | Embedding | Full | 0.74400 | 0.73475 | 0.75325 | 0.83839 | 0.84275 | 0.30423 | 0.95362 | 0.77613 | 0.61283 | 0.61245 | 0.77802 | 0.62759 | 0.99594 | 0.86981 |
| n_hidden: 128, n_latent: 20, n_layers: 1, n_latent_u: 10 | Embedding | HVG | 0.75015 | 0.73455 | 0.76576 | 0.83837 | 0.84162 | 0.31110 | 0.94709 | 0.77658 | 0.61760 | 0.61297 | 0.81031 | 0.66196 | 0.99571 | 0.88518 |
| n_hidden: 128, n_latent: 20, n_layers: 2, n_latent_u: 10 | Embedding | Full | 0.73733 | 0.73821 | 0.73645 | 0.83015 | 0.85181 | 0.30987 | 0.96102 | 0.75416 | 0.59025 | 0.59272 | 0.71926 | 0.62696 | 0.99347 | 0.87831 |
| n_hidden: 128, n_latent: 20, n_layers: 2, n_latent_u: 10 | Embedding | HVG | 0.74557 | 0.73622 | 0.75492 | 0.83070 | 0.85464 | 0.30839 | 0.95117 | 0.76816 | 0.61183 | 0.59363 | 0.78872 | 0.64544 | 0.99409 | 0.88258 |
| n_hidden: 128, n_latent: 20, n_layers: 3, n_latent_u: 10 | Embedding | Full | 0.72673 | 0.72879 | 0.72468 | 0.82155 | 0.83409 | 0.31015 | 0.94934 | 0.76275 | 0.60838 | 0.60271 | 0.60055 | 0.61891 | 0.99476 | 0.88473 |
| n_hidden: 128, n_latent: 20, n_layers: 3, n_latent_u: 10 | Embedding | HVG | 0.74040 | 0.73881 | 0.74199 | 0.82778 | 0.85413 | 0.30039 | 0.97294 | 0.76274 | 0.60602 | 0.60899 | 0.60747 | 0.63563 | 0.99395 | 0.87913 |
| n_hidden: 128, n_latent: 30, n_layers: 1, n_latent_u: 10 | Embedding | Full | 0.72625 | 0.73839 | 0.71412 | 0.84541 | 0.84501 | 0.30758 | 0.95556 | 0.76142 | 0.60624 | 0.59677 | 0.53924 | 0.61480 | 0.99486 | 0.88548 |
| n_hidden: 128, n_latent: 30, n_layers: 1, n_latent_u: 10 | Embedding | HVG | 0.74698 | 0.74581 | 0.74815 | 0.84994 | 0.85271 | 0.31033 | 0.97026 | 0.78064 | 0.61501 | 0.60333 | 0.70971 | 0.65619 | 0.99447 | 0.87771 |
| n_hidden: 128, n_latent: 30, n_layers: 2, n_latent_u: 10 | Embedding | Full | 0.74482 | 0.74200 | 0.74764 | 0.83833 | 0.85330 | 0.30674 | 0.96963 | 0.76260 | 0.60336 | 0.61195 | 0.75782 | 0.63253 | 0.99564 | 0.86957 |
| n_hidden: 128, n_latent: 30, n_layers: 2, n_latent_u: 10 | Embedding | HVG | 0.74753 | 0.74432 | 0.75074 | 0.83382 | 0.87481 | 0.30485 | 0.96380 | 0.77366 | 0.61033 | 0.60817 | 0.75667 | 0.64330 | 0.99587 | 0.86717 |
| n_hidden: 128, n_latent: 30, n_layers: 3, n_latent_u: 10 | Embedding | Full | 0.73728 | 0.73680 | 0.73776 | 0.81815 | 0.86921 | 0.30520 | 0.95463 | 0.75040 | 0.58746 | 0.60196 | 0.72321 | 0.64402 | 0.99538 | 0.86192 |
| n_hidden: 128, n_latent: 30, n_layers: 3, n_latent_u: 10 | Embedding | HVG | 0.73409 | 0.73352 | 0.73465 | 0.82182 | 0.84566 | 0.30753 | 0.95908 | 0.75428 | 0.59788 | 0.60421 | 0.67427 | 0.64511 | 0.99530 | 0.87151 |
| n_hidden: 128, n_latent: 40, n_layers: 1, n_latent_u: 10 | Embedding | Full | 0.74275 | 0.74681 | 0.73869 | 0.83794 | 0.88296 | 0.30086 | 0.96550 | 0.77406 | 0.60774 | 0.60774 | 0.70586 | 0.60936 | 0.99567 | 0.86559 |
| n_hidden: 128, n_latent: 40, n_layers: 1, n_latent_u: 10 | Embedding | HVG | 0.75441 | 0.74620 | 0.76263 | 0.84336 | 0.87888 | 0.30031 | 0.96224 | 0.77908 | 0.61951 | 0.60485 | **0.83260** | 0.64060 | 0.99598 | 0.86580 |
| n_hidden: 128, n_latent: 40, n_layers: 2, n_latent_u: 10 | Embedding | Full | 0.74870 | 0.74563 | 0.75177 | 0.84357 | 0.87129 | 0.30304 | 0.96463 | 0.77487 | 0.61489 | 0.61167 | 0.75269 | 0.64060 | 0.99625 | 0.87142 |
| n_hidden: 128, n_latent: 40, n_layers: 2, n_latent_u: 10 | Embedding | HVG | 0.74356 | 0.74298 | 0.74413 | 0.84176 | 0.86899 | 0.30351 | 0.95768 | 0.76814 | 0.62955 | 0.60279 | 0.71414 | 0.62957 | 0.99601 | 0.86869 |
| n_hidden: 128, n_latent: 40, n_layers: 3, n_latent_u: 10 | Embedding | Full | 0.73442 | 0.74458 | 0.72425 | 0.84005 | 0.86403 | 0.30318 | 0.97106 | 0.74988 | 0.60541 | 0.59353 | 0.64816 | 0.61075 | 0.99520 | 0.86684 |
| n_hidden: 128, n_latent: 40, n_layers: 3, n_latent_u: 10 | Embedding | HVG | 0.73235 | 0.74334 | 0.72136 | 0.83553 | 0.86250 | 0.30824 | 0.96709 | 0.76286 | 0.60851 | 0.60187 | 0.59332 | 0.61956 | 0.99566 | 0.86777 |
| n_hidden: 128, n_latent: 50, n_layers: 1, n_latent_u: 10 | Embedding | Full | 0.74597 | 0.74528 | 0.74667 | 0.83811 | 0.86223 | 0.29678 | 0.98398 | 0.77586 | 0.61666 | 0.60068 | 0.72480 | 0.64603 | 0.99608 | 0.86661 |
| n_hidden: 128, n_latent: 50, n_layers: 1, n_latent_u: 10 | Embedding | HVG | 0.73212 | 0.74848 | 0.71577 | 0.84643 | 0.87587 | 0.29789 | 0.97372 | 0.76604 | 0.61393 | 0.60237 | 0.54451 | 0.61572 | 0.99515 | 0.87265 |
| n_hidden: 128, n_latent: 50, n_layers: 2, n_latent_u: 10 | Embedding | Full | 0.74872 | 0.74366 | 0.75378 | 0.84590 | 0.87017 | 0.30059 | 0.95797 | 0.77371 | 0.62860 | 0.61267 | 0.73754 | 0.64992 | 0.99608 | 0.87833 |
| n_hidden: 128, n_latent: 50, n_layers: 2, n_latent_u: 10 | Embedding | HVG | 0.73666 | 0.74509 | 0.72824 | 0.83945 | 0.87244 | 0.29754 | 0.97091 | 0.76841 | 0.61148 | 0.60932 | 0.60290 | 0.64328 | 0.99584 | 0.86645 |
| n_hidden: 128, n_latent: 50, n_layers: 3, n_latent_u: 10 | Embedding | Full | 0.73982 | 0.74369 | 0.73594 | 0.82346 | 0.87925 | 0.30505 | 0.96701 | 0.75700 | 0.61822 | 0.62404 | 0.64435 | 0.63473 | 0.99559 | 0.87766 |
| n_hidden: 128, n_latent: 50, n_layers: 3, n_latent_u: 10 | Embedding | HVG | 0.71811 | 0.74065 | 0.69557 | 0.83270 | 0.85444 | 0.30441 | 0.97105 | 0.74777 | 0.59273 | 0.59713 | 0.67861 | 0.61745 | 0.99530 | 0.63999 |
| n_hidden: 256, n_latent: 10, n_layers: 1, n_latent_u: 10 | Embedding | Full | 0.72792 | 0.73638 | 0.71946 | 0.81602 | 0.84708 | 0.32515 | 0.95727 | 0.71338 | 0.52971 | 0.58716 | 0.71009 | 0.63258 | 0.98644 | 0.87688 |
| n_hidden: 256, n_latent: 10, n_layers: 1, n_latent_u: 10 | Embedding | HVG | 0.72727 | 0.73237 | 0.72217 | 0.81712 | 0.83738 | 0.32692 | 0.94808 | 0.73296 | 0.57714 | 0.58156 | 0.69290 | 0.61447 | 0.98687 | 0.86928 |
| n_hidden: 256, n_latent: 10, n_layers: 2, n_latent_u: 10 | Embedding | Full | 0.72376 | 0.73628 | 0.71123 | 0.81394 | 0.83892 | 0.32979 | 0.96249 | 0.68661 | 0.46285 | 0.58493 | 0.76593 | 0.62277 | 0.98488 | 0.87062 |
| n_hidden: 256, n_latent: 10, n_layers: 2, n_latent_u: 10 | Embedding | HVG | 0.71624 | 0.72906 | 0.70341 | 0.80787 | 0.83734 | 0.32030 | 0.95073 | 0.70461 | 0.53322 | 0.58324 | 0.60906 | 0.63544 | 0.98529 | 0.87304 |
| n_hidden: 256, n_latent: 10, n_layers: 3, n_latent_u: 10 | Embedding | Full | 0.66639 | 0.70555 | 0.62722 | 0.79206 | 0.80285 | 0.32246 | 0.90481 | 0.64525 | 0.35197 | 0.57156 | 0.62241 | 0.61856 | 0.98445 | 0.59598 |
| n_hidden: 256, n_latent: 10, n_layers: 3, n_latent_u: 10 | Embedding | HVG | 0.69400 | 0.71589 | 0.67211 | 0.80067 | 0.81445 | 0.32189 | 0.92655 | 0.68175 | 0.43023 | 0.57069 | 0.70353 | 0.61444 | 0.98444 | 0.71967 |
| n_hidden: 256, n_latent: 20, n_layers: 1, n_latent_u: 10 | Embedding | Full | 0.74063 | 0.73693 | 0.74433 | 0.83643 | 0.84772 | 0.31169 | 0.95188 | 0.77524 | 0.61226 | 0.60022 | 0.70903 | 0.63656 | 0.99509 | 0.88189 |
| n_hidden: 256, n_latent: 20, n_layers: 1, n_latent_u: 10 | Embedding | HVG | 0.74200 | 0.73364 | 0.75035 | 0.83014 | 0.84493 | 0.30993 | 0.94955 | 0.77636 | 0.61274 | 0.60893 | 0.74191 | 0.63378 | 0.99434 | 0.88441 |
| n_hidden: 256, n_latent: 20, n_layers: 2, n_latent_u: 10 | Embedding | Full | 0.71311 | 0.72967 | 0.69655 | 0.82600 | 0.83724 | 0.30907 | 0.94638 | 0.74347 | 0.59558 | 0.59183 | 0.63860 | 0.61843 | 0.99317 | 0.69474 |
| n_hidden: 256, n_latent: 20, n_layers: 2, n_latent_u: 10 | Embedding | HVG | 0.72344 | 0.73190 | 0.71497 | 0.83325 | 0.82809 | 0.31281 | 0.95346 | 0.75405 | 0.59384 | 0.61531 | 0.55753 | 0.61161 | 0.99418 | 0.87825 |
| n_hidden: 256, n_latent: 20, n_layers: 3, n_latent_u: 10 | Embedding | Full | 0.69893 | 0.72409 | 0.67376 | 0.80308 | 0.84626 | 0.31551 | 0.93152 | 0.70520 | 0.53075 | 0.58591 | 0.64857 | 0.58794 | 0.99012 | 0.66783 |
| n_hidden: 256, n_latent: 20, n_layers: 3, n_latent_u: 10 | Embedding | HVG | 0.68382 | 0.72993 | 0.63771 | 0.81942 | 0.85847 | 0.32078 | 0.92105 | 0.68475 | 0.49656 | 0.59316 | 0.66019 | 0.61251 | 0.98777 | 0.42902 |
| n_hidden: 256, n_latent: 30, n_layers: 1, n_latent_u: 10 | Embedding | Full | 0.73646 | 0.74022 | 0.73269 | 0.83874 | 0.86409 | 0.30680 | 0.95127 | 0.77212 | 0.63026 | 0.60839 | 0.64910 | 0.61373 | 0.99572 | 0.85949 |
| n_hidden: 256, n_latent: 30, n_layers: 1, n_latent_u: 10 | Embedding | HVG | 0.72793 | 0.74320 | 0.71266 | 0.84409 | 0.85086 | 0.31524 | 0.96261 | 0.76008 | 0.59577 | 0.59676 | 0.55928 | 0.61869 | 0.99487 | 0.86313 |
| n_hidden: 256, n_latent: 30, n_layers: 2, n_latent_u: 10 | Embedding | Full | 0.70286 | 0.73754 | 0.66818 | 0.82069 | 0.84702 | 0.30908 | 0.97337 | 0.76405 | 0.61595 | 0.61516 | 0.61067 | 0.62150 | 0.99516 | 0.45481 |
| n_hidden: 256, n_latent: 30, n_layers: 2, n_latent_u: 10 | Embedding | HVG | 0.73999 | 0.73724 | 0.74274 | 0.82800 | 0.84738 | 0.30918 | 0.96441 | 0.77503 | 0.63215 | 0.61712 | 0.67942 | 0.62474 | 0.99513 | 0.87558 |
| n_hidden: 256, n_latent: 30, n_layers: 3, n_latent_u: 10 | Embedding | Full | 0.68641 | 0.73029 | 0.64252 | 0.81930 | 0.84215 | 0.31476 | 0.94894 | 0.70857 | 0.47595 | 0.61679 | 0.71982 | 0.62899 | 0.99300 | 0.35456 |
| n_hidden: 256, n_latent: 30, n_layers: 3, n_latent_u: 10 | Embedding | HVG | 0.71416 | 0.72164 | 0.70667 | 0.80765 | 0.84652 | 0.31547 | 0.91693 | 0.74868 | 0.58825 | 0.58991 | 0.70108 | 0.59917 | 0.99147 | 0.72810 |
| n_hidden: 256, n_latent: 40, n_layers: 1, n_latent_u: 10 | Embedding | Full | 0.74723 | 0.74396 | 0.75051 | 0.83830 | 0.87779 | 0.30542 | 0.95434 | 0.77314 | 0.62654 | 0.60328 | 0.75751 | 0.61966 | 0.99617 | 0.87723 |
| n_hidden: 256, n_latent: 40, n_layers: 1, n_latent_u: 10 | Embedding | HVG | 0.75340 | 0.74889 | 0.75791 | 0.84027 | 0.88326 | 0.31003 | 0.96202 | 0.78387 | 0.64167 | 0.60823 | 0.74401 | 0.64887 | 0.99600 | 0.88269 |
| n_hidden: 256, n_latent: 40, n_layers: 2, n_latent_u: 10 | Embedding | Full | 0.73316 | 0.73880 | 0.72753 | 0.81698 | 0.86379 | 0.30779 | 0.94333 | 0.75691 | 0.60458 | 0.60548 | 0.64234 | 0.61303 | 0.99531 | 0.87505 |
| n_hidden: 256, n_latent: 40, n_layers: 2, n_latent_u: 10 | Embedding | HVG | 0.73245 | 0.73960 | 0.72530 | 0.82889 | 0.86369 | 0.30779 | 0.95802 | 0.74485 | 0.59713 | 0.59293 | 0.65075 | 0.62031 | 0.99410 | 0.87663 |
| n_hidden: 256, n_latent: 40, n_layers: 3, n_latent_u: 10 | Embedding | Full | 0.70042 | 0.72781 | 0.67302 | 0.80755 | 0.86261 | 0.30914 | 0.93194 | 0.70174 | 0.45081 | 0.60729 | 0.78126 | 0.59293 | 0.99410 | 0.66617 |
| n_hidden: 256, n_latent: 40, n_layers: 3, n_latent_u: 10 | Embedding | HVG | 0.70153 | 0.72144 | 0.68161 | 0.81452 | 0.85127 | 0.31166 | 0.90832 | 0.76437 | 0.58161 | 0.58161 | 0.71623 | 0.64395 | 0.99450 | 0.66691 |
| n_hidden: 256, n_latent: 50, n_layers: 1, n_latent_u: 10 | Embedding | Full | 0.75104 | 0.74770 | 0.75438 | 0.84616 | 0.87082 | 0.30541 | 0.96843 | 0.77699 | 0.63379 | 0.61401 | 0.76946 | 0.62736 | 0.99577 | 0.86326 |
| n_hidden: 256, n_latent: 50, n_layers: 1, n_latent_u: 10 | Embedding | HVG | 0.72944 | 0.74355 | 0.71532 | 0.84189 | 0.86761 | 0.30148 | 0.96322 | 0.76127 | 0.61015 | 0.59927 | 0.54564 | 0.63066 | 0.99523 | 0.86502 |
| n_hidden: 256, n_latent: 50, n_layers: 2, n_latent_u: 10 | Embedding | Full | 0.74379 | 0.74092 | 0.74665 | 0.84298 | 0.85122 | 0.30421 | 0.96529 | 0.74517 | 0.59281 | 0.61340 | 0.77667 | 0.64025 | 0.99536 | 0.86286 |
| n_hidden: 256, n_latent: 50, n_layers: 2, n_latent_u: 10 | Embedding | HVG | 0.72877 | 0.74059 | 0.71694 | 0.83368 | 0.87396 | 0.30290 | 0.95183 | 0.75153 | 0.60861 | 0.62619 | 0.53865 | 0.62969 | 0.99560 | 0.86835 |
| n_hidden: 256, n_latent: 50, n_layers: 3, n_latent_u: 10 | Embedding | Full | 0.67476 | 0.71062 | 0.63891 | 0.78653 | 0.84725 | 0.31739 | 0.89129 | 0.69162 | 0.46488 | 0.59223 | 0.69640 | 0.61175 | 0.99215 | 0.42337 |
| n_hidden: 256, n_latent: 50, n_layers: 3, n_latent_u: 10 | Embedding | HVG | 0.72038 | 0.72777 | 0.71300 | 0.81466 | 0.84134 | 0.31210 | 0.94298 | 0.72538 | 0.53764 | 0.58791 | 0.78116 | 0.63643 | 0.99137 | 0.73109 |
| n_hidden: 128, n_latent: 10, n_layers: 1, n_latent_u: 20 | Embedding | Full | 0.72037 | 0.73030 | 0.71045 | 0.82396 | 0.81350 | 0.31665 | 0.96708 | 0.69483 | 0.48237 | 0.58293 | 0.70104 | 0.64196 | 0.98675 | 0.88324 |
| n_hidden: 128, n_latent: 10, n_layers: 1, n_latent_u: 20 | Embedding | HVG | 0.72251 | 0.72376 | 0.72126 | 0.81961 | 0.80612 | 0.32389 | 0.94511 | 0.70930 | 0.49602 | 0.58902 | 0.73534 | 0.64397 | 0.98748 | 0.88769 |
| n_hidden: 128, n_latent: 10, n_layers: 2, n_latent_u: 20 | Embedding | Full | 0.72682 | 0.72780 | 0.72584 | 0.83979 | 0.80149 | 0.31666 | 0.95325 | 0.71186 | 0.55436 | 0.57195 | 0.75803 | 0.63186 | 0.98597 | 0.88685 |
| n_hidden: 128, n_latent: 10, n_layers: 2, n_latent_u: 20 | Embedding | HVG | 0.73044 | 0.72529 | 0.73560 | 0.82251 | 0.80806 | 0.31672 | 0.95385 | 0.71557 | 0.53209 | 0.58600 | 0.78925 | 0.63769 | 0.98597 | 0.88262 |
| n_hidden: 128, n_latent: 10, n_layers: 3, n_latent_u: 20 | Embedding | Full | 0.68149 | 0.72050 | 0.64247 | 0.82327 | 0.79689 | 0.31579 | 0.94604 | 0.66619 | 0.42438 | 0.57357 | 0.65931 | 0.61663 | 0.98352 | 0.57373 |
| n_hidden: 128, n_latent: 10, n_layers: 3, n_latent_u: 20 | Embedding | HVG | 0.66557 | 0.72127 | 0.60988 | 0.82118 | 0.79178 | 0.32507 | 0.94702 | 0.67366 | 0.48109 | 0.57961 | 0.59798 | 0.59799 | 0.98555 | 0.36929 |
| n_hidden: 128, n_latent: 20, n_layers: 1, n_latent_u: 20 | Embedding | Full | 0.74994 | 0.73794 | 0.76195 | 0.84532 | 0.85098 | 0.30711 | 0.94833 | 0.77629 | 0.61424 | 0.61332 | 0.78354 | 0.66806 | 0.99570 | 0.88253 |
| n_hidden: 128, n_latent: 20, n_layers: 1, n_latent_u: 20 | Embedding | HVG | 0.73916 | 0.73547 | 0.74284 | 0.83678 | 0.84143 | 0.30545 | 0.95823 | 0.77402 | 0.61726 | 0.60296 | 0.67161 | 0.66055 | 0.99504 | 0.87845 |
| n_hidden: 128, n_latent: 20, n_layers: 2, n_latent_u: 20 | Embedding | Full | 0.74473 | 0.73239 | 0.75706 | 0.83153 | 0.83163 | 0.30347 | 0.96294 | 0.77429 | 0.61313 | 0.60712 | 0.77777 | 0.65174 | 0.99503 | 0.88035 |
| n_hidden: 128, n_latent: 20, n_layers: 2, n_latent_u: 20 | Embedding | HVG | 0.74386 | 0.73978 | 0.74795 | 0.84221 | 0.85083 | 0.31450 | 0.95157 | 0.76891 | 0.61811 | 0.61452 | 0.70689 | 0.65421 | 0.99495 | 0.87805 |
| n_hidden: 128, n_latent: 20, n_layers: 3, n_latent_u: 20 | Embedding | Full | 0.74239 | 0.74094 | 0.74385 | 0.83174 | 0.86134 | 0.31633 | 0.95434 | 0.76877 | 0.60776 | 0.58852 | 0.72602 | 0.64448 | 0.99206 | 0.87932 |
| n_hidden: 128, n_latent: 20, n_layers: 3, n_latent_u: 20 | Embedding | HVG | 0.73986 | 0.73551 | 0.74420 | 0.83964 | 0.83498 | 0.30645 | 0.96139 | 0.75024 | 0.60249 | 0.60249 | 0.73883 | 0.65565 | 0.99402 | 0.88319 |
| n_hidden: 128, n_latent: 30, n_layers: 1, n_latent_u: 20 | Embedding | Full | 0.73967 | 0.74076 | 0.73859 | 0.85039 | 0.84570 | 0.29832 | 0.96862 | 0.77013 | 0.61030 | 0.60579 | 0.67266 | 0.65487 | 0.99549 | 0.86087 |
| n_hidden: 128, n_latent: 30, n_layers: 1, n_latent_u: 20 | Embedding | HVG | 0.73548 | 0.74392 | 0.72704 | 0.84816 | 0.86440 | 0.29882 | 0.96432 | 0.76722 | 0.61398 | 0.60879 | 0.70244 | 0.66576 | 0.99588 | 0.73518 |
| n_hidden: 128, n_latent: 30, n_layers: 2, n_latent_u: 20 | Embedding | Full | 0.74633 | 0.74109 | 0.75157 | 0.83861 | 0.84557 | 0.30601 | 0.97419 | 0.76774 | 0.61051 | 0.60858 | 0.76324 | 0.64524 | 0.99584 | 0.86982 |
| n_hidden: 128, n_latent: 30, n_layers: 2, n_latent_u: 20 | Embedding | HVG | 0.73360 | 0.73714 | 0.73005 | 0.83266 | 0.85481 | 0.30507 | 0.95602 | 0.75912 | 0.60313 | 0.59293 | 0.66375 | 0.63527 | 0.99422 | 0.86194 |
| n_hidden: 128, n_latent: 30, n_layers: 3, n_latent_u: 20 | Embedding | Full | 0.72496 | 0.73861 | 0.71130 | 0.82470 | 0.86736 | 0.31217 | 0.95022 | 0.74723 | 0.60562 | 0.60511 | 0.55083 | 0.61411 | 0.99423 | 0.86198 |
| n_hidden: 128, n_latent: 30, n_layers: 3, n_latent_u: 20 | Embedding | HVG | 0.72847 | 0.73971 | 0.71723 | 0.83971 | 0.85342 | 0.30369 | 0.97013 | 0.73758 | 0.58444 | 0.60073 | 0.72896 | 0.64784 | 0.99489 | 0.72620 |
| n_hidden: 128, n_latent: 40, n_layers: 1, n_latent_u: 20 | Embedding | Full | 0.73941 | 0.74914 | 0.72969 | 0.84967 | 0.88080 | 0.29777 | 0.96830 | 0.77320 | 0.63229 | 0.61127 | 0.59231 | 0.62515 | 0.99594 | 0.87769 |
| n_hidden: 128, n_latent: 40, n_layers: 1, n_latent_u: 20 | Embedding | HVG | 0.74787 | 0.74407 | 0.75167 | 0.85139 | 0.85339 | 0.30405 | 0.96666 | 0.76758 | 0.61124 | 0.61089 | 0.78808 | 0.62195 | 0.99586 | 0.86607 |
| n_hidden: 128, n_latent: 40, n_layers: 2, n_latent_u: 20 | Embedding | Full | 0.73957 | 0.74012 | 0.73902 | 0.83714 | 0.85961 | 0.30446 | 0.95927 | 0.75011 | 0.58597 | 0.60110 | 0.70364 | 0.66125 | 0.99540 | 0.87568 |
| n_hidden: 128, n_latent: 40, n_layers: 2, n_latent_u: 20 | Embedding | HVG | 0.71500 | 0.74366 | 0.68633 | 0.83748 | 0.87334 | 0.30045 | 0.96335 | 0.76697 | 0.61408 | 0.60110 | 0.68515 | 0.66133 | 0.99665 | 0.47535 |
| n_hidden: 128, n_latent: 40, n_layers: 3, n_latent_u: 20 | Embedding | Full | 0.74254 | 0.74592 | 0.73917 | 0.83029 | **0.88841** | 0.29876 | 0.96623 | 0.75564 | 0.58708 | 0.61108 | 0.71769 | 0.63953 | 0.99566 | 0.86748 |
| n_hidden: 128, n_latent: 40, n_layers: 3, n_latent_u: 20 | Embedding | HVG | 0.73315 | 0.73364 | 0.73266 | 0.81765 | 0.88009 | 0.30000 | 0.93000 | 0.76921 | 0.63235 | 0.61520 | 0.61106 | 0.63358 | 0.99572 | 0.87108 |
| n_hidden: 128, n_latent: 50, n_layers: 1, n_latent_u: 20 | Embedding | Full | 0.74496 | 0.74685 | 0.74307 | 0.85125 | 0.86521 | 0.30012 | 0.97081 | 0.77677 | 0.61045 | 0.61447 | 0.67171 | 0.65019 | 0.99594 | 0.88199 |
| n_hidden: 128, n_latent: 50, n_layers: 1, n_latent_u: 20 | Embedding | HVG | 0.74877 | 0.74562 | 0.75192 | 0.84615 | 0.87317 | 0.30415 | 0.95902 | 0.77641 | 0.61879 | 0.62027 | 0.73430 | 0.66125 | 0.99627 | 0.85616 |
| n_hidden: 128, n_latent: 50, n_layers: 2, n_latent_u: 20 | Embedding | Full | 0.73894 | 0.74057 | 0.73730 | 0.83685 | 0.86351 | 0.30193 | 0.95998 | 0.76496 | 0.59596 | 0.61367 | 0.65948 | 0.65010 | 0.99587 | 0.88109 |
| n_hidden: 128, n_latent: 50, n_layers: 2, n_latent_u: 20 | Embedding | HVG | 0.74537 | 0.73767 | 0.75306 | 0.83898 | 0.85144 | 0.29464 | 0.96564 | 0.77762 | 0.61793 | 0.61074 | 0.72186 | **0.67074** | 0.99665 | 0.87636 |
| n_hidden: 128, n_latent: 50, n_layers: 3, n_latent_u: 20 | Embedding | Full | 0.71272 | 0.73786 | 0.68757 | 0.83633 | 0.86638 | 0.29644 | 0.95230 | 0.75538 | 0.60778 | 0.61277 | 0.72283 | 0.64538 | 0.99547 | 0.47340 |
| n_hidden: 128, n_latent: 50, n_layers: 3, n_latent_u: 20 | Embedding | HVG | 0.73575 | 0.73857 | 0.73293 | 0.83795 | 0.86313 | 0.30389 | 0.94930 | 0.75328 | 0.58944 | 0.61774 | 0.66835 | 0.63340 | 0.99521 | 0.87833 |
| n_hidden: 256, n_latent: 10, n_layers: 1, n_latent_u: 20 | Embedding | Full | 0.69203 | 0.72220 | 0.66186 | 0.82766 | 0.79757 | 0.31689 | 0.94667 | 0.72501 | 0.56866 | 0.59077 | 0.71130 | 0.63858 | 0.98699 | 0.41268 |
| n_hidden: 256, n_latent: 10, n_layers: 1, n_latent_u: 20 | Embedding | HVG | 0.72339 | 0.73183 | 0.71494 | 0.82426 | 0.82821 | 0.31656 | 0.95837 | 0.70778 | 0.51129 | 0.58711 | 0.70740 | 0.62164 | 0.98670 | 0.88267 |
| n_hidden: 256, n_latent: 10, n_layers: 2, n_latent_u: 20 | Embedding | Full | 0.70589 | 0.71783 | 0.69394 | 0.81371 | 0.79086 | 0.32164 | 0.94512 | 0.66871 | 0.46404 | 0.58425 | 0.65034 | 0.61870 | 0.98475 | 0.88679 |
| n_hidden: 256, n_latent: 10, n_layers: 2, n_latent_u: 20 | Embedding | HVG | 0.70489 | 0.70844 | 0.70133 | 0.82458 | 0.75451 | 0.32272 | 0.93195 | 0.68765 | 0.50125 | 0.56506 | 0.68465 | 0.60586 | 0.98298 | 0.88388 |
| n_hidden: 256, n_latent: 10, n_layers: 3, n_latent_u: 20 | Embedding | Full | 0.68155 | 0.70626 | 0.65683 | 0.80873 | 0.77138 | 0.32214 | 0.92280 | 0.64217 | 0.39676 | 0.57340 | 0.63980 | 0.60595 | 0.98283 | 0.75792 |
| n_hidden: 256, n_latent: 10, n_layers: 3, n_latent_u: 20 | Embedding | HVG | 0.67543 | 0.70675 | 0.64410 | 0.81531 | 0.77743 | 0.32359 | 0.91069 | 0.64089 | 0.42592 | 0.56190 | 0.64478 | 0.59396 | 0.97985 | 0.66142 |
| n_hidden: 256, n_latent: 20, n_layers: 1, n_latent_u: 20 | Embedding | Full | 0.73626 | 0.73693 | 0.73559 | 0.84496 | 0.84619 | 0.30344 | 0.94882 | 0.76710 | 0.60344 | 0.60293 | 0.62991 | 0.65718 | 0.99520 | 0.89333 |
| n_hidden: 256, n_latent: 20, n_layers: 1, n_latent_u: 20 | Embedding | HVG | 0.74488 | 0.73516 | 0.75459 | 0.84203 | 0.83005 | 0.31268 | 0.95589 | 0.76686 | 0.59502 | 0.61351 | 0.74449 | 0.65887 | 0.99478 | 0.87864 |
| n_hidden: 256, n_latent: 20, n_layers: 2, n_latent_u: 20 | Embedding | Full | 0.72908 | 0.72887 | 0.72929 | 0.82815 | 0.83268 | 0.30933 | 0.95176 | 0.74791 | 0.59548 | 0.58642 | 0.64843 | 0.64765 | 0.99490 | 0.88507 |
| n_hidden: 256, n_latent: 20, n_layers: 2, n_latent_u: 20 | Embedding | HVG | 0.73728 | 0.73268 | 0.74188 | 0.83137 | 0.83155 | 0.30846 | 0.95933 | 0.75253 | 0.60047 | 0.60762 | 0.70863 | 0.64669 | 0.99399 | 0.88326 |
| n_hidden: 256, n_latent: 20, n_layers: 3, n_latent_u: 20 | Embedding | Full | 0.67695 | 0.72320 | 0.63070 | 0.82424 | 0.83660 | 0.31714 | 0.91760 | 0.65767 | 0.39839 | 0.58889 | 0.61438 | 0.63228 | 0.99179 | 0.53146 |
| n_hidden: 256, n_latent: 20, n_layers: 3, n_latent_u: 20 | Embedding | HVG | 0.69816 | 0.73082 | 0.66550 | 0.81193 | 0.86841 | 0.31466 | 0.92830 | 0.69297 | 0.43678 | 0.60342 | 0.74054 | 0.64174 | 0.99219 | 0.55083 |
| n_hidden: 256, n_latent: 30, n_layers: 1, n_latent_u: 20 | Embedding | Full | 0.74443 | 0.74661 | 0.74225 | 0.84513 | 0.87757 | 0.31068 | 0.95308 | 0.77809 | 0.61545 | 0.60854 | 0.67419 | 0.64273 | 0.99555 | 0.88119 |
| n_hidden: 256, n_latent: 30, n_layers: 1, n_latent_u: 20 | Embedding | HVG | 0.74847 | 0.74297 | 0.75397 | 0.84522 | 0.86067 | 0.30550 | 0.96049 | 0.76505 | 0.59655 | 0.61025 | 0.77881 | 0.66270 | 0.99627 | 0.86816 |
| n_hidden: 256, n_latent: 30, n_layers: 2, n_latent_u: 20 | Embedding | Full | 0.74730 | 0.73677 | 0.75783 | 0.83340 | 0.85754 | 0.30452 | 0.95160 | 0.76206 | 0.61036 | 0.60204 | 0.82264 | 0.64738 | 0.99487 | 0.87480 |
| n_hidden: 256, n_latent: 30, n_layers: 2, n_latent_u: 20 | Embedding | HVG | 0.72221 | 0.74140 | 0.70301 | 0.83904 | 0.85262 | 0.31063 | 0.96333 | 0.76285 | 0.61649 | **0.62891** | 0.60431 | 0.65227 | 0.99537 | 0.66038 |
| n_hidden: 256, n_latent: 30, n_layers: 3, n_latent_u: 20 | Embedding | Full | 0.71217 | 0.72926 | 0.69509 | 0.82791 | 0.85104 | 0.31677 | 0.94132 | 0.72307 | 0.53304 | 0.58305 | 0.71068 | 0.62748 | 0.99464 | 0.69816 |
| n_hidden: 256, n_latent: 30, n_layers: 3, n_latent_u: 20 | Embedding | HVG | 0.69524 | 0.72490 | 0.66557 | 0.82166 | 0.84102 | 0.30613 | 0.93078 | 0.72564 | 0.53695 | 0.59749 | 0.68945 | 0.63362 | 0.99233 | 0.48355 |
| n_hidden: 256, n_latent: 40, n_layers: 1, n_latent_u: 20 | Embedding | Full | **0.76041** | **0.75361** | **0.76721** | 0.85406 | 0.88199 | 0.30612 | 0.97226 | **0.79773** | 0.74674 | 0.60810 | 0.70918 | 0.65246 | 0.99559 | 0.86273 |
| n_hidden: 256, n_latent: 40, n_layers: 1, n_latent_u: 20 | Embedding | HVG | 0.74116 | 0.74813 | 0.73419 | 0.84561 | 0.87436 | 0.30467 | 0.96788 | 0.77039 | 0.61014 | 0.61045 | 0.65662 | 0.62010 | 0.99577 | 0.87588 |
| n_hidden: 256, n_latent: 40, n_layers: 2, n_latent_u: 20 | Embedding | Full | 0.72417 | 0.73434 | 0.71399 | 0.82776 | 0.85146 | 0.30533 | 0.95281 | 0.73751 | 0.57516 | 0.58323 | 0.69677 | 0.63720 | 0.99555 | 0.86559 |
| n_hidden: 256, n_latent: 40, n_layers: 2, n_latent_u: 20 | Embedding | HVG | 0.74605 | 0.73783 | 0.75426 | 0.83469 | 0.86038 | 0.30240 | 0.95384 | 0.76128 | 0.61879 | 0.61182 | 0.76996 | 0.65659 | 0.99581 | 0.86559 |
| n_hidden: 256, n_latent: 40, n_layers: 3, n_latent_u: 20 | Embedding | Full | 0.70465 | 0.72288 | 0.68642 | 0.81083 | 0.84555 | 0.30836 | 0.92678 | 0.71804 | 0.51324 | 0.59766 | 0.68800 | 0.61525 | 0.99550 | 0.67771 |
| n_hidden: 256, n_latent: 40, n_layers: 3, n_latent_u: 20 | Embedding | HVG | 0.67793 | 0.72531 | 0.63055 | 0.79027 | 0.86242 | 0.31867 | 0.92986 | 0.69909 | 0.46139 | 0.59091 | 0.70937 | 0.63945 | 0.99091 | 0.32271 |
| n_hidden: 256, n_latent: 50, n_layers: 1, n_latent_u: 20 | Embedding | Full | 0.74583 | 0.74394 | 0.74772 | 0.85105 | 0.85909 | 0.29753 | 0.96810 | 0.78384 | 0.64493 | 0.61183 | 0.61355 | 0.64493 | **0.99667** | 0.85837 |
| n_hidden: 256, n_latent: 50, n_layers: 1, n_latent_u: 20 | Embedding | HVG | 0.75146 | 0.75236 | 0.75057 | 0.84438 | 0.87860 | 0.30210 | **0.98436** | 0.77598 | 0.62929 | 0.61387 | 0.70598 | 0.66509 | 0.99634 | 0.86842 |
| n_hidden: 256, n_latent: 50, n_layers: 2, n_latent_u: 20 | Embedding | Full | 0.71792 | 0.74460 | 0.69123 | 0.82404 | 0.88302 | 0.30613 | 0.96522 | 0.75865 | 0.60100 | 0.60116 | 0.53344 | 0.62776 | 0.99594 | 0.71108 |
| n_hidden: 256, n_latent: 50, n_layers: 2, n_latent_u: 20 | Embedding | HVG | 0.73209 | 0.73362 | 0.73056 | 0.81146 | 0.86753 | 0.29925 | 0.95622 | 0.74587 | 0.58430 | 0.61557 | 0.64395 | 0.62040 | 0.99599 | 0.88786 |
| n_hidden: 256, n_latent: 50, n_layers: 3, n_latent_u: 20 | Embedding | Full | 0.68395 | 0.72922 | 0.63868 | 0.80848 | 0.86952 | 0.31630 | 0.92259 | 0.72221 | 0.54119 | 0.59239 | 0.73221 | 0.62141 | 0.99420 | 0.26717 |
| n_hidden: 256, n_latent: 50, n_layers: 3, n_latent_u: 20 | Embedding | HVG | 0.68953 | 0.72365 | 0.65541 | 0.79688 | 0.86839 | 0.31898 | 0.91037 | 0.68238 | 0.44004 | 0.55590 | 0.68621 | 0.62109 | 0.98874 | 0.61350 |

**Table 3:** *Performance of MrVI on the Human Immune Dataset*

Table 3 reports the MrVI sweep on the human immune dataset across $n_{\text{hidden}} \in \{128, 256\}$, $n_{\text{latent}} \in \{10, 20, 30, 40, 50\}$, $n_{\text{layers}} \in \{1, 2, 3\}$, $n_{\text{latent\_u}} \in \{10, 20\}$, and Features $\in \{\text{Full, HVG}\}$. Using the Overall score, the best configuration is 0.76041 at $(256, 40, 1, u{=}20, \text{Full})$, while the worst is 0.66557 at $(128, 10, 3, u{=}20, \text{HVG})$. The same $(256, 40, 1, u{=}20, \text{Full})$ setting also sits at or near the top for the batch and biology Overalls alongside NMI and ARI, indicating a pronounced peak. In contrast, label fidelity and temporal continuity reach their best values at more conservative capacity: Label ASW near $(256, 30, 2, u{=}20, \text{HVG})$ and Trajectory near $(128, 10, 2, u{=}10, \text{Full})$.

Changing hidden size from 128 to 256 generally depresses the aggregate metrics in this MRVI grid. Across matched pairs that fix $(n_{\text{latent}}, n_{\text{layers}}, u, \text{Features})$, Overall rises in $17/60$ and falls in $43/60$; the batch Overall rises in $13/60$ and falls in $47/60$; the biology Overall rises in $13/60$ and falls in $47/60$. Batch ASW drops in $48/60$, NMI in $48/60$, ARI in $44/60$, and Graph Connectivity in $44/60$ pairs. The main counter-trend is iLISI, which improves in $48/60$ when moving to 256. Trajectory is mixed ($26/60$ up, $34/60$ down). Overall, $n_{\text{hidden}}{=}128$ is the safer default unless iLISI is prioritized.

Increasing depth from $n_{\text{layers}}{=}1$ to 3 also pushes downward on the composite metrics. Over matched pairs holding $(n_{\text{hidden}}, n_{\text{latent}}, u, \text{Features})$ fixed, Overall rises in $4/40$ and falls in $36/40$; the batch Overall rises in $4/40$ and falls in $36/40$; the biology Overall rises in $3/40$ and falls in $37/40$. Batch ASW decreases in $37/40$ and Graph Connectivity in $32/40$; clustering agreement weakens (NMI up $1/40$, down $39/40$; ARI up $12/40$, down $28/40$). iLISI improves in $32/40$ and PCR in $12/40$. In short, deeper models tend to inflate iLISI but hurt agreement, connectivity, and the composite Overalls; shallow depth (1) is preferred for balance.

Latent dimensionality in $z$ is the strongest and most favorable driver. Endpoint comparisons (50 vs. 10) at fixed $(n_{\text{hidden}}, n_{\text{layers}}, u, \text{Features})$ show Overall improving in $23/24$ cases, the batch Overall in $24/24$, and the biology Overall in $20/24$. NMI and ARI each improve in $24/24$, Graph Connectivity in $21/24$, Label ASW in $23/24$, and Isolated-Label ASW in $19/24$. The main downside is Trajectory, which falls in $18/24$ endpoints. Stepwise behavior is not strictly monotone for every metric, but the end state at high latent (40–50) is decisively better on aggregate and agreement metrics. Practically, pushing $n_{\text{latent}}$ high with shallow depth is the winning recipe for the composite scores without sacrificing label compactness in this MrVI setting.

The unshared $u$ latent shows nuanced trade-offs. Moving from $u{=}10$ to $u{=}20$ at fixed $(n_{\text{hidden}}, n_{\text{latent}}, n_{\text{layers}}, \text{Features})$, Overall is up in $27/60$ and down in $33/60$; the batch Overall is up in $25/60$ and down in $34/60$; the biology Overall is up in $27/60$ and down in $33/60$. Several structure and label metrics tilt positive: Batch ASW improves in $40/60$, Label ASW in $36/60$, Isolated-Label ASW in $43/60$, and Trajectory in $33/60$ pairs; PCR tends to fall ($24/60$ up, $36/60$ down), and Connectivity is roughly balanced ($27/60$ up, $33/60$ down). Thus, $u{=}20$ aids label compactness and Batch ASW, while composite Overalls move only modestly.

Feature selection has a small but broad positive tilt for HVG. Pairing HVG against Full at fixed $(n_{\text{hidden}}, n_{\text{latent}}, n_{\text{layers}}, u)$, Overall is higher in $32/60$ pairs (lower in $28/60$), ARI in $36/60$, NMI in $31/60$, iLISI in $33/60$, and Trajectory tilts higher on average. Effects are modest in magnitude but consistent enough that HVG is a reasonable default; the single global optimum here, however, happens to be Full at $(256, 40, 1, u{=}20)$.

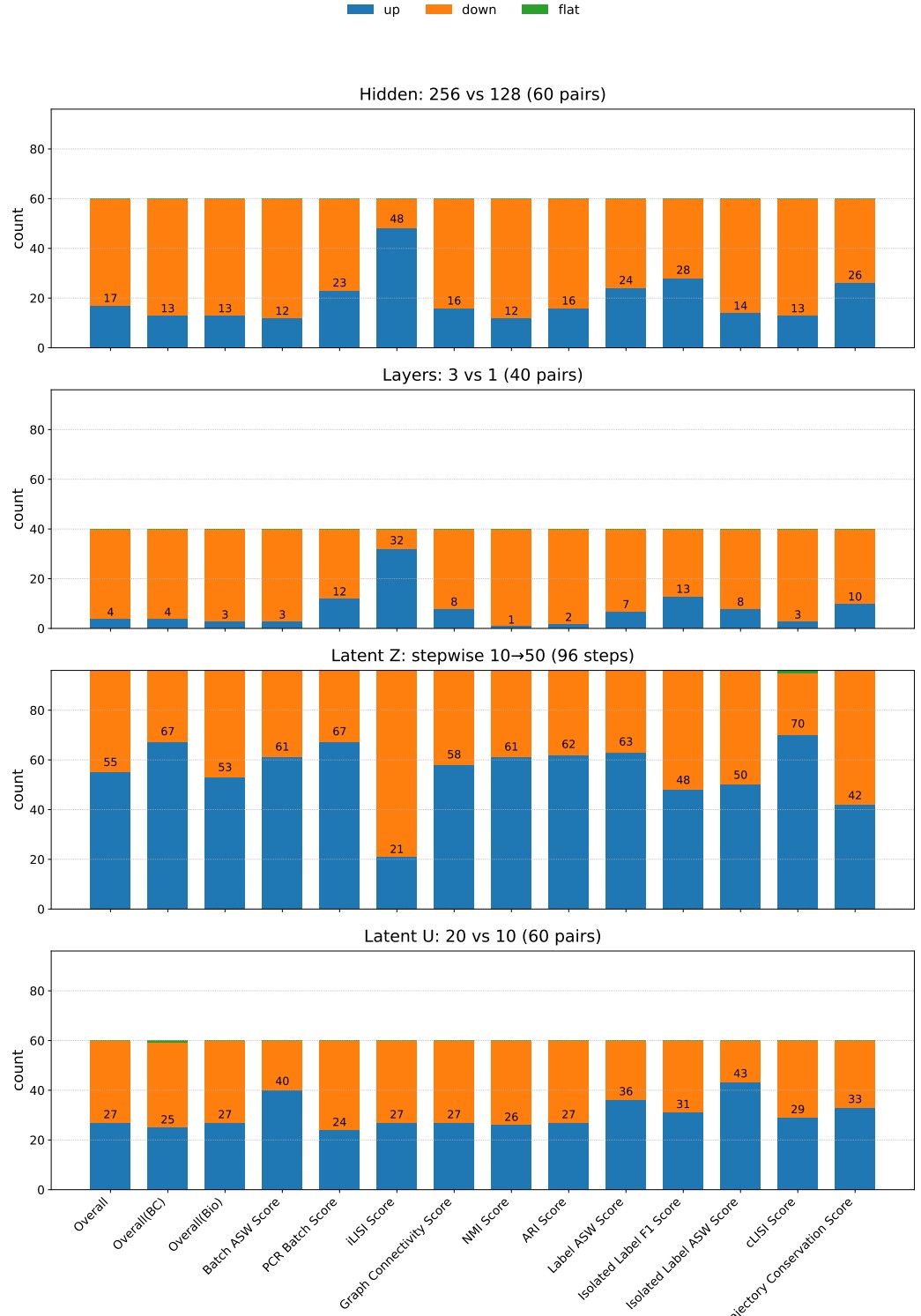

**Figure 17:** *Paired comparison of MrVI hyperparameters on the Human Immune dataset, showing the number of metrics that improve ("up"), decline ("down"), or remain unchanged ("flat") when varying hidden units (256 vs. 128), network depth (3 vs. 1 layers), latent dimension z (stepwise 10→50), and latent dimension u evaluated on both full and HVG feature sets.*

| Hyperparameters | Output | Features | Overall | Overall(BC) | Overall(Bio) | Batch correction | | | | Bio Conservation | | | | | | |
| | | | Overall | Overall(BC) | Overall(Bio) | ASW | PCR | iLISI | GC | NMI | ARI | ASW | IL F1 | IL ASW | cLISI | TC |
|---|---|---|---|---|---|---|---|---|---|---|---|---|---|---|---|---|
| n_hidden: 128, n_latent: 10, n_layers: 1 | Embedding | Full | 0.73381 | 0.71332 | 0.75430 | 0.83631 | 0.89503 | 0.16170 | 0.96023 | 0.75779 | 0.59330 | 0.59702 | 0.82455 | 0.64340 | 0.99719 | 0.86686 |
| n_hidden: 128, n_latent: 10, n_layers: 1 | Embedding | HVG | 0.74264 | 0.72320 | 0.76208 | 0.83792 | 0.89207 | 0.18267 | **0.98014** | 0.79011 | 0.62167 | 0.60762 | 0.81868 | 0.64505 | 0.99747 | 0.85394 |
| n_hidden: 128, n_latent: 10, n_layers: 2 | Embedding | Full | 0.73684 | 0.71939 | 0.75430 | 0.83094 | 0.89305 | 0.18919 | 0.96436 | 0.76519 | 0.59664 | 0.60823 | 0.81969 | 0.64829 | 0.99759 | 0.84451 |
| n_hidden: 128, n_latent: 10, n_layers: 2 | Embedding | HVG | 0.74108 | 0.71996 | 0.76220 | 0.83529 | 0.88794 | 0.19363 | 0.96297 | 0.77771 | 0.60582 | 0.61089 | 0.82151 | 0.66191 | 0.99777 | 0.85978 |
| n_hidden: 128, n_latent: 10, n_layers: 3 | Embedding | Full | 0.73438 | 0.71987 | 0.74889 | 0.82313 | 0.88769 | 0.19865 | 0.97002 | 0.75885 | 0.59023 | 0.61115 | 0.78323 | 0.64336 | 0.99730 | 0.85813 |
| n_hidden: 128, n_latent: 10, n_layers: 3 | Embedding | HVG | 0.74279 | 0.72045 | 0.76513 | 0.82925 | 0.89020 | 0.19376 | 0.96860 | 0.77742 | 0.61265 | 0.61301 | 0.82746 | 0.66900 | 0.99778 | 0.85861 |
| n_hidden: 128, n_latent: 20, n_layers: 1 | Embedding | Full | 0.72279 | 0.70283 | 0.74275 | 0.85634 | 0.88027 | 0.14224 | 0.93247 | 0.75703 | 0.55266 | 0.58828 | 0.79766 | 0.65663 | 0.99840 | 0.84857 |
| n_hidden: 128, n_latent: 20, n_layers: 1 | Embedding | HVG | 0.73706 | 0.72337 | 0.75085 | 0.86746 | 0.89175 | 0.16590 | 0.96796 | 0.76509 | 0.55135 | 0.58856 | 0.85181 | 0.65279 | 0.99798 | 0.84839 |
| n_hidden: 128, n_latent: 20, n_layers: 2 | Embedding | Full | 0.72519 | 0.71381 | 0.73656 | 0.84986 | 0.89132 | 0.15966 | 0.95442 | 0.74718 | 0.53067 | 0.59319 | 0.76759 | 0.67215 | 0.99820 | 0.84696 |
| n_hidden: 128, n_latent: 20, n_layers: 2 | Embedding | HVG | 0.72475 | 0.72480 | 0.72470 | 0.85751 | 0.90228 | 0.17679 | 0.96263 | 0.76506 | 0.55565 | 0.59320 | 0.78727 | 0.67086 | 0.99807 | 0.70278 |
| n_hidden: 128, n_latent: 20, n_layers: 3 | Embedding | Full | 0.72669 | 0.70914 | 0.74424 | 0.83818 | 0.89231 | 0.16882 | 0.93727 | 0.75540 | 0.53959 | 0.60058 | 0.78429 | 0.67134 | **0.99897** | 0.85951 |
| n_hidden: 128, n_latent: 20, n_layers: 3 | Embedding | HVG | 0.73887 | 0.72269 | 0.75504 | 0.84223 | 0.90306 | 0.18334 | 0.96215 | 0.76911 | 0.55521 | 0.59615 | 0.83494 | 0.67869 | 0.99826 | 0.85295 |
| n_hidden: 128, n_latent: 30, n_layers: 1 | Embedding | Full | 0.74336 | 0.70246 | 0.78426 | 0.86654 | 0.89138 | 0.14686 | 0.90506 | 0.79722 | 0.77324 | 0.57978 | 0.85348 | 0.64503 | 0.99845 | 0.84261 |
| n_hidden: 128, n_latent: 30, n_layers: 1 | Embedding | HVG | 0.75239 | 0.73251 | 0.77227 | 0.88617 | 0.89530 | 0.17475 | 0.97381 | 0.78822 | 0.70556 | 0.57398 | 0.85140 | 0.64249 | 0.99794 | 0.84633 |
| n_hidden: 128, n_latent: 30, n_layers: 2 | Embedding | Full | 0.72948 | 0.70828 | 0.75067 | 0.85552 | 0.89590 | 0.15235 | 0.92933 | 0.76480 | 0.53518 | 0.58699 | 0.81406 | 0.66328 | 0.99847 | 0.87197 |
| n_hidden: 128, n_latent: 30, n_layers: 2 | Embedding | HVG | **0.75774** | 0.72933 | **0.78615** | 0.86832 | **0.91037** | 0.18433 | 0.95428 | 0.80267 | 0.77611 | 0.58501 | 0.83030 | 0.65838 | 0.99818 | 0.85240 |
| n_hidden: 128, n_latent: 30, n_layers: 3 | Embedding | Full | 0.73391 | 0.71309 | 0.75473 | 0.84077 | 0.90346 | 0.17063 | 0.93751 | 0.75633 | 0.55693 | 0.59044 | 0.85499 | 0.66831 | 0.99856 | 0.85758 |
| n_hidden: 128, n_latent: 30, n_layers: 3 | Embedding | HVG | 0.73731 | 0.72200 | 0.75262 | 0.85104 | 0.89997 | 0.17624 | 0.96074 | 0.75766 | 0.54301 | 0.58844 | 0.85236 | 0.67232 | 0.99849 | 0.85607 |
| n_hidden: 128, n_latent: 40, n_layers: 1 | Embedding | Full | 0.72838 | 0.71077 | 0.74599 | 0.88467 | 0.88332 | 0.15083 | 0.92427 | 0.75875 | 0.54776 | 0.57483 | 0.84599 | 0.64305 | 0.99843 | 0.85314 |
| n_hidden: 128, n_latent: 40, n_layers: 1 | Embedding | HVG | 0.70982 | 0.67170 | 0.74794 | 0.85647 | 0.86292 | 0.15101 | 0.91639 | 0.75765 | 0.54086 | 0.59712 | 0.85123 | 0.64143 | 0.99873 | 0.84856 |
| n_hidden: 128, n_latent: 40, n_layers: 2 | Embedding | Full | 0.72012 | 0.70870 | 0.73155 | 0.86189 | 0.90074 | 0.14070 | 0.93147 | 0.74349 | 0.51207 | 0.57824 | 0.80028 | 0.65458 | 0.99848 | 0.85371 |
| n_hidden: 128, n_latent: 40, n_layers: 2 | Embedding | HVG | 0.75578 | 0.72695 | 0.78460 | 0.87634 | 0.89494 | 0.17362 | 0.96291 | **0.80390** | **0.77754** | 0.57359 | 0.85408 | 0.64987 | 0.99811 | 0.83511 |
| n_hidden: 128, n_latent: 40, n_layers: 3 | Embedding | Full | 0.72443 | 0.69980 | 0.74906 | 0.84990 | 0.89495 | 0.16198 | 0.89237 | 0.75116 | 0.54048 | 0.58877 | 0.82622 | 0.67566 | 0.99891 | 0.86220 |
| n_hidden: 128, n_latent: 40, n_layers: 3 | Embedding | HVG | 0.72641 | 0.71994 | 0.73288 | 0.85352 | 0.90626 | 0.17789 | 0.94210 | 0.75757 | 0.55531 | 0.58565 | 0.82441 | 0.67474 | 0.99869 | 0.73376 |
| n_hidden: 128, n_latent: 50, n_layers: 1 | Embedding | Full | 0.72849 | 0.71483 | 0.74216 | 0.89185 | 0.88545 | 0.13660 | 0.94541 | 0.74837 | 0.57048 | 0.56424 | 0.83640 | 0.62868 | 0.99805 | 0.84886 |
| n_hidden: 128, n_latent: 50, n_layers: 1 | Embedding | HVG | 0.73745 | 0.73225 | 0.74264 | **0.90301** | 0.90150 | 0.16919 | 0.95530 | 0.76259 | 0.56673 | 0.56078 | 0.84789 | 0.62348 | 0.99760 | 0.83941 |
| n_hidden: 128, n_latent: 50, n_layers: 2 | Embedding | Full | 0.72628 | 0.71038 | 0.74217 | 0.86848 | 0.88698 | 0.14652 | 0.93954 | 0.74433 | 0.52612 | 0.57617 | 0.83708 | 0.65730 | 0.99848 | 0.86803 |
| n_hidden: 128, n_latent: 50, n_layers: 2 | Embedding | HVG | 0.75295 | 0.73063 | 0.77528 | 0.87851 | 0.90079 | 0.17424 | 0.96897 | 0.79170 | 0.72029 | 0.57459 | 0.84087 | 0.65033 | 0.99870 | 0.85043 |
| n_hidden: 128, n_latent: 50, n_layers: 3 | Embedding | Full | 0.73629 | 0.69684 | 0.77573 | 0.84999 | 0.88700 | 0.15682 | 0.89357 | 0.78188 | 0.75267 | 0.58468 | 0.77974 | 0.65501 | 0.99884 | 0.87728 |
| n_hidden: 128, n_latent: 50, n_layers: 3 | Embedding | HVG | 0.73404 | 0.71461 | 0.75348 | 0.85563 | 0.89127 | 0.15811 | 0.95342 | 0.76601 | 0.55410 | 0.58411 | 0.84711 | 0.67286 | 0.99877 | 0.85137 |
| n_hidden: 256, n_latent: 10, n_layers: 1 | Embedding | Full | 0.72664 | 0.71657 | 0.73672 | 0.84004 | 0.88043 | 0.18251 | 0.96329 | 0.75221 | 0.53871 | 0.60246 | 0.76693 | 0.63296 | 0.99785 | 0.86591 |
| n_hidden: 256, n_latent: 10, n_layers: 1 | Embedding | HVG | 0.71124 | 0.68715 | 0.73532 | 0.83712 | 0.81504 | 0.16273 | 0.93373 | 0.74775 | 0.53626 | 0.59646 | 0.79101 | 0.60967 | 0.99706 | 0.86902 |
| n_hidden: 256, n_latent: 10, n_layers: 2 | Embedding | Full | 0.73108 | 0.71806 | 0.74411 | 0.83288 | 0.88673 | 0.18781 | 0.96481 | 0.76339 | 0.54397 | 0.60560 | 0.75972 | 0.64081 | 0.99714 | 0.84775 |
| n_hidden: 256, n_latent: 10, n_layers: 2 | Embedding | HVG | 0.73809 | 0.71798 | 0.75820 | 0.83475 | 0.88700 | 0.18716 | 0.96301 | 0.78182 | 0.62588 | 0.60556 | 0.76912 | 0.66226 | 0.99780 | 0.86497 |
| n_hidden: 256, n_latent: 10, n_layers: 3 | Embedding | Full | 0.73461 | 0.71348 | 0.75575 | 0.82925 | 0.87863 | 0.18330 | 0.96271 | 0.75959 | 0.59017 | 0.60329 | 0.84051 | 0.64650 | 0.99759 | 0.85263 |
| n_hidden: 256, n_latent: 10, n_layers: 3 | Embedding | HVG | 0.72991 | 0.71075 | 0.74907 | 0.81771 | 0.89703 | **0.20366** | 0.92459 | 0.78561 | 0.63241 | **0.61460** | 0.83734 | 0.66531 | 0.99786 | 0.71037 |
| n_hidden: 256, n_latent: 20, n_layers: 1 | Embedding | Full | 0.73328 | 0.71685 | 0.74971 | 0.86319 | 0.87865 | 0.16023 | 0.96534 | 0.76420 | 0.55388 | 0.58905 | 0.83775 | 0.65526 | 0.99795 | 0.84988 |
| n_hidden: 256, n_latent: 20, n_layers: 1 | Embedding | HVG | 0.70496 | 0.66499 | 0.74493 | 0.84248 | 0.76137 | 0.13921 | 0.91687 | 0.76224 | 0.54483 | 0.59581 | 0.83071 | 0.63509 | 0.99762 | 0.84824 |
| n_hidden: 256, n_latent: 20, n_layers: 2 | Embedding | Full | 0.72442 | 0.70112 | 0.74772 | 0.82510 | 0.85819 | 0.17203 | 0.94916 | 0.75155 | 0.53763 | 0.59376 | 0.84267 | 0.67155 | 0.99799 | 0.83890 |
| n_hidden: 256, n_latent: 20, n_layers: 2 | Embedding | HVG | 0.73726 | 0.72220 | 0.75232 | 0.86726 | 0.89023 | 0.16835 | 0.96294 | 0.77060 | 0.54903 | 0.58795 | 0.84928 | 0.65982 | 0.99779 | 0.85179 |
| n_hidden: 256, n_latent: 20, n_layers: 3 | Embedding | Full | 0.72874 | 0.70597 | 0.75152 | 0.84849 | 0.88335 | 0.15960 | 0.93244 | 0.75061 | 0.53653 | 0.59301 | 0.84386 | 0.67266 | 0.99852 | 0.86542 |
| n_hidden: 256, n_latent: 20, n_layers: 3 | Embedding | HVG | 0.73228 | 0.71142 | 0.75315 | 0.83798 | 0.90399 | 0.15991 | 0.94378 | 0.76132 | 0.55384 | 0.59831 | 0.83727 | 0.66835 | 0.99887 | 0.85408 |
| n_hidden: 256, n_latent: 30, n_layers: 1 | Embedding | Full | 0.63886 | 0.60076 | 0.67696 | 0.81601 | 0.66585 | 0.12066 | 0.90051 | 0.64869 | 0.43483 | 0.56623 | 0.62421 | 0.58116 | 0.99196 | **0.89165** |
| n_hidden: 256, n_latent: 30, n_layers: 1 | Embedding | HVG | 0.68086 | 0.62526 | 0.73645 | 0.82752 | 0.62169 | 0.14217 | 0.90966 | 0.74917 | 0.53338 | 0.58830 | 0.82586 | 0.61845 | 0.99614 | 0.84383 |
| n_hidden: 256, n_latent: 30, n_layers: 2, | Embedding | Full | 0.72743 | 0.70310 | 0.75177 | 0.86112 | 0.89795 | 0.15663 | 0.89668 | 0.76490 | 0.55827 | 0.58389 | 0.84389 | 0.66613 | 0.99856 | 0.84676 |
| n_hidden: 256, n_latent: 30, n_layers: 2, | Embedding | HVG | 0.73254 | 0.72681 | 0.73826 | 0.87118 | 0.90196 | 0.17890 | 0.95521 | 0.76774 | 0.57451 | 0.57613 | 0.85594 | 0.66260 | 0.99749 | 0.73340 |
| n_hidden: 256, n_latent: 30, n_layers: 3 | Embedding | Full | 0.68871 | 0.70076 | 0.67666 | 0.85427 | 0.88273 | 0.14649 | 0.91954 | 0.75770 | 0.55737 | 0.57974 | 0.83466 | 0.66706 | 0.99859 | 0.34152 |
| n_hidden: 256, n_latent: 30, n_layers: 3 | Embedding | HVG | 0.72278 | 0.70322 | 0.74235 | 0.81548 | 0.87882 | 0.18377 | 0.93481 | 0.75770 | 0.54768 | 0.59882 | 0.84781 | **0.68282** | 0.99886 | 0.76063 |
| n_hidden: 256, n_latent: 40, n_layers: 1 | Embedding | Full | 0.72405 | 0.70221 | 0.74590 | 0.88862 | 0.86385 | 0.14217 | 0.91418 | 0.75713 | 0.56078 | 0.56727 | 0.83764 | 0.63877 | 0.99817 | 0.86151 |
| n_hidden: 256, n_latent: 40, n_layers: 1 | Embedding | HVG | 0.65281 | 0.58473 | 0.72088 | 0.81660 | 0.51584 | 0.10769 | 0.89878 | 0.71700 | 0.46892 | 0.58009 | 0.79834 | 0.61851 | 0.99638 | 0.87594 |
| n_hidden: 256, n_latent: 40, n_layers: 2 | Embedding | Full | 0.69193 | 0.66108 | 0.72278 | 0.81968 | 0.77632 | 0.15276 | 0.89555 | 0.71246 | 0.48338 | 0.57676 | 0.82907 | 0.60752 | 0.99533 | 0.85491 |
| n_hidden: 256, n_latent: 40, n_layers: 2 | Embedding | HVG | 0.74191 | **0.73827** | 0.74555 | 0.89102 | 0.90658 | 0.18661 | 0.96887 | 0.75759 | 0.54508 | 0.56302 | **0.85656** | 0.64609 | 0.99706 | 0.85343 |
| n_hidden: 256, n_latent: 40, n_layers: 3 | Embedding | Full | 0.71968 | 0.70704 | 0.73233 | 0.85015 | 0.89938 | 0.16134 | 0.91727 | 0.74022 | 0.50777 | 0.57751 | 0.80006 | 0.67012 | 0.99865 | 0.83200 |
| n_hidden: 256, n_latent: 40, n_layers: 3 | Embedding | HVG | 0.69524 | 0.68238 | 0.70811 | 0.80894 | 0.83306 | 0.18048 | 0.90703 | 0.68548 | 0.44195 | 0.57560 | 0.76266 | 0.62695 | 0.98953 | 0.87459 |
| n_hidden: 256, n_latent: 50, n_layers: 1 | Embedding | Full | 0.68179 | 0.71562 | 0.64796 | 0.89672 | 0.87688 | 0.13527 | 0.95360 | 0.73825 | 0.48340 | 0.56534 | 0.84565 | 0.63806 | 0.99814 | 0.26690 |
| n_hidden: 256, n_latent: 50, n_layers: 1 | Embedding | HVG | 0.67454 | 0.62398 | 0.72511 | 0.83336 | 0.65086 | 0.10413 | 0.90758 | 0.71508 | 0.57361 | 0.58216 | 0.73327 | 0.60968 | 0.99477 | 0.86717 |
| n_hidden: 256, n_latent: 50, n_layers: 2 | Embedding | Full | 0.61916 | 0.60313 | 0.63518 | 0.77893 | 0.60514 | 0.16969 | 0.85877 | 0.59428 | 0.43809 | 0.53596 | 0.55690 | 0.50047 | 0.97644 | 0.84412 |
| n_hidden: 256, n_latent: 50, n_layers: 2 | Embedding | HVG | 0.64042 | 0.61186 | 0.66899 | 0.80649 | 0.58881 | 0.16123 | 0.89091 | 0.66962 | 0.51259 | 0.55721 | 0.55268 | 0.53228 | 0.98356 | 0.87496 |
| n_hidden: 256, n_latent: 50, n_layers: 3 | Embedding | Full | 0.68041 | 0.67444 | 0.68638 | 0.80551 | 0.81978 | 0.17217 | 0.90029 | 0.65847 | 0.40744 | 0.56176 | 0.73311 | 0.59501 | 0.98537 | 0.86350 |
| n_hidden: 256, n_latent: 50, n_layers: 3 | Embedding | HVG | 0.69122 | 0.67836 | 0.70408 | 0.78833 | 0.85527 | 0.17167 | 0.89817 | 0.70411 | 0.45698 | 0.58783 | 0.72106 | 0.59343 | 0.99432 | 0.87086 |

**Table 4:** *Performance of LDVAE on the Human Immune Dataset*

Table 4 summarizes the sweep over $n_{\text{hidden}} \in \{128, 256\}$, $n_{\text{latent}} \in \{10, 20, 30, 40, 50\}$, $n_{\text{layers}} \in \{1, 2, 3\}$, and Features $\in \{\text{Full}, \text{HVG}\}$ for LDVAE. Using the Overall score, the best configuration is $0.75774$ at $(n_{\text{hidden}}{=}128, n_{\text{latent}}{=}30, n_{\text{layers}}{=}2, \text{HVG})$, while the worst is $0.61916$ at $(256, 50, 2, \text{Full})$. The batch-correction overall peaks at $(256, 40, 2, \text{HVG})$ and the biological overall coincides with the Overall winner at $(128, 30, 2, \text{HVG})$. Individual metric optima are spread: Batch ASW is highest at $(128, 50, 1, \text{HVG})$; PCR at $(128, 30, 2, \text{HVG})$; iLISI at $(256, 10, 3, \text{HVG})$; graph connectivity at $(128, 10, 1, \text{HVG})$; NMI and ARI at $(128, 40, 2, \text{HVG})$; Label ASW at $(256, 10, 3, \text{HVG})$; trajectory at $(256, 30, 1, \text{Full})$.

Changing hidden size from 128 to 256 generally reduces the composites and many structure metrics in this grid. Across paired settings that fix $(n_{\text{latent}}, n_{\text{layers}}, \text{Features})$, Overall improves in $4/30$ and worsens in $26/30$ pairs; the batch overall in $5/30$ up and $25/30$ down; the biology overall in $6/30$ up and $24/30$ down. By metric, Batch ASW is up in $13/30$, PCR in $5/30$, iLISI in $12/30$, connectivity in $10/30$, NMI in $9/30$, ARI in $9/30$, Label ASW in $11/30$, and trajectory in $16/30$ pairs. The net effect is that moving to 256 tends to hurt the composites and clustering agreement, with trajectory the only metric showing a near-balanced tilt.

Increasing depth from $n_{\text{layers}}{=}1$ to $3$ is net favorable here for the composite scores, while shifting the detailed balance of batch versus biology. Over paired settings fixing $(n_{\text{hidden}}, n_{\text{latent}}, \text{Features})$, Overall rises in $13/20$ and falls in $7/20$; the batch overall $11/20$ up, $9/20$ down; the biology overall $12/20$ up, $8/20$ down. PCR and iLISI improve strongly ($15/20$ and $18/20$ up, respectively), Label ASW improves in $17/20$, and trajectory in $12/20$. In contrast, Batch ASW drops in $19/20$, while NMI and ARI are more often down ($8/12$ down for both). Thus depth promotes certain batch harmonization statistics (PCR, iLISI) and label compactness, with mild trajectory gains, but it softens Batch ASW and clustering agreement.

Latent dimensionality in $z$ does not help in this dataset when pushed to the high end. Endpoint comparisons between $n_{\text{latent}}{=}50$ and $10$ at fixed $(n_{\text{hidden}}, n_{\text{layers}}, \text{Features})$ show Overall up in only $2/12$ and down in $10/12$; the batch overall $3/12$ up, $9/12$ down; the biology overall $2/12$ up, $10/12$ down. Endpoints for individual metrics show Batch

ASW 7/12 up, but PCR 3/12 up, iLISI 0/12 up, connectivity 1/12 up, NMI 2/12 up, ARI 3/12 up, Label ASW 0/12 up, trajectory 5/12 up. Stepwise trends ($10{\to}20{\to}30{\to}40{\to}50$) corroborate this: Batch ASW rises in 35/48 steps, but many other metrics tilt downward by the time we reach 50. In short, large $n_{\text{latent}}$ is not advantageous in this sweep; moderate latent (around 30) with shallow-to-moderate depth is where the composite and biology overalls peak.

Comparing HVG to Full at matched $(n_{\text{hidden}}, n_{\text{latent}}, n_{\text{layers}})$ pairs, HVG gives consistent, modest gains in the composites and agreement metrics: up in 21/30 pairs, batch overall (22/30), biology overall (19/30). For individual metrics, HVG improves iLISI (21/30 up), connectivity (19/30), NMI (22/30), ARI (19/30), and Label ASW (15/30 up with many near-ties). PCR averages slightly down despite more ups than downs, indicating a few large negative shifts. Overall, HVG remains a sensible default here given its small but broad gains.

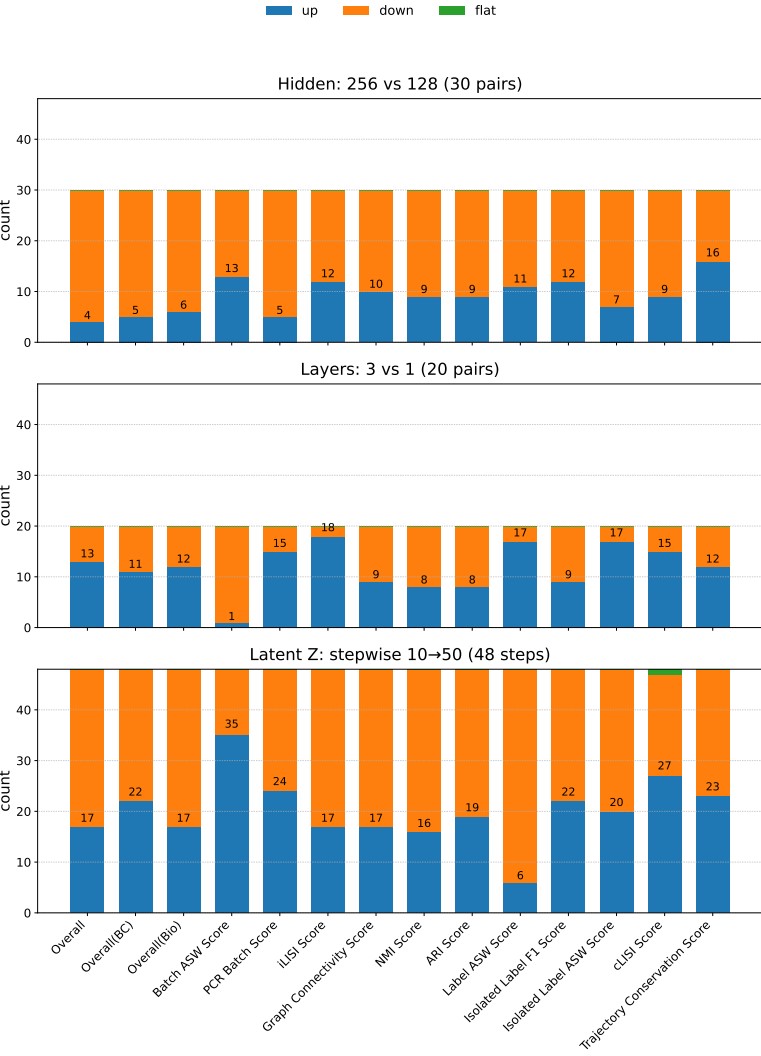

**Figure 18:** *Paired comparison of LDVAE hyperparameters on the Human Immune dataset, showing the number of metrics that improve ("up"), decline ("down"), or remain unchanged ("flat") when varying hidden units (256 vs. 128), network depth (3 vs. 1 layers), latent dimension z (stepwise 10→50), and latent dimension u evaluated on both full and HVG feature sets.*

## J  PER METRIC PERFORMANCE ON ZENODO 8020792 (24 PBMC SAMPLES)

| Hyperparameters | Output | Features | Overall | Overall(BC) | Overall(Bio) | Batch correction | | | | Bio Conservation | | | | | | |
|---|---|---|---|---|---|---|---|---|---|---|---|---|---|---|---|---|
| | | | Overall | Overall(BC) | Overall(Bio) | ASW | PCR | iLISI | GC | NMI | ARI | ASW | IL F1 | IL ASW | cLISI | TC |
| n_hidden: 128, n_latent: 10, n_layers: 1 | Embedding | Full | 0.61346 | 0.65012 | 0.57679 | 0.86403 | 0.58491 | 0.28245 | 0.86910 | 0.78357 | 0.59925 | 0.53016 | 0.01371 | 0.53646 | 0.99761 | N/A |
| n_hidden: 128, n_latent: 10, n_layers: 1 | Embedding | HVG | 0.60135 | 0.62847 | 0.57422 | 0.87143 | 0.49717 | 0.27860 | 0.86668 | 0.79006 | 0.60409 | **0.54473** | 0.01053 | 0.49741 | **0.99852** | N/A |
| n_hidden: 128, n_latent: 10, n_layers: 2 | Embedding | Full | 0.58505 | 0.62665 | 0.54345 | 0.86732 | 0.49275 | 0.28238 | 0.86417 | 0.71426 | 0.46442 | 0.52038 | 0.01511 | 0.55249 | 0.99405 | N/A |
| n_hidden: 128, n_latent: 10, n_layers: 2 | Embedding | HVG | 0.61626 | 0.63040 | 0.60213 | 0.87185 | 0.50683 | 0.27930 | 0.86360 | 0.76940 | 0.77118 | 0.53812 | 0.00745 | 0.52829 | 0.99831 | N/A |
| n_hidden: 128, n_latent: 10, n_layers: 3 | Embedding | Full | 0.61454 | 0.65706 | 0.57202 | 0.85860 | 0.62826 | 0.28536 | 0.85603 | 0.78354 | 0.60540 | 0.52148 | 0.01053 | 0.51406 | 0.99710 | N/A |
| n_hidden: 128, n_latent: 10, n_layers: 3 | Embedding | HVG | 0.58970 | 0.63475 | 0.54465 | 0.86996 | 0.52102 | 0.28100 | 0.86700 | 0.68477 | 0.50440 | 0.51646 | 0.00728 | 0.55878 | 0.99619 | N/A |
| n_hidden: 128, n_latent: 20, n_layers: 1 | Embedding | Full | 0.64391 | 0.67209 | 0.61572 | 0.89105 | 0.64401 | 0.28265 | 0.87067 | 0.82148 | 0.79734 | 0.52559 | 0.01200 | 0.54175 | 0.99618 | N/A |
| n_hidden: 128, n_latent: 20, n_layers: 1 | Embedding | HVG | 0.64638 | 0.67924 | 0.61352 | 0.90464 | 0.64947 | 0.28615 | 0.87669 | 0.81754 | 0.79657 | 0.52760 | 0.00778 | 0.53442 | 0.99722 | N/A |
| n_hidden: 128, n_latent: 20, n_layers: 2 | Embedding | Full | 0.65036 | 0.68775 | 0.61296 | 0.89718 | 0.68883 | 0.29089 | 0.87412 | 0.81900 | 0.79700 | 0.52336 | 0.01241 | 0.53042 | 0.99555 | N/A |
| n_hidden: 128, n_latent: 20, n_layers: 2 | Embedding | HVG | 0.66154 | 0.69674 | 0.62635 | 0.90612 | 0.71226 | 0.29450 | 0.87408 | 0.82917 | 0.85743 | 0.52534 | 0.00723 | 0.54175 | 0.99717 | N/A |
| n_hidden: 128, n_latent: 20, n_layers: 3 | Embedding | Full | 0.62080 | 0.67905 | 0.56254 | 0.88586 | 0.67334 | 0.29212 | 0.86487 | 0.73429 | 0.56232 | 0.51587 | 0.01261 | 0.55524 | 0.99494 | N/A |
| n_hidden: 128, n_latent: 20, n_layers: 3 | Embedding | HVG | 0.63915 | 0.67712 | 0.60119 | 0.89105 | 0.65703 | 0.28839 | 0.87200 | 0.76835 | 0.76421 | 0.52251 | 0.00742 | 0.54692 | 0.99774 | N/A |
| n_hidden: 128, n_latent: 30, n_layers: 1 | Embedding | Full | 0.62669 | 0.68775 | 0.56562 | 0.90969 | 0.68152 | 0.28638 | 0.87341 | 0.76557 | 0.56164 | 0.52001 | 0.01197 | 0.53969 | 0.99488 | N/A |
| n_hidden: 128, n_latent: 30, n_layers: 1 | Embedding | HVG | 0.65485 | 0.70596 | 0.60373 | 0.91943 | 0.73416 | 0.29551 | 0.87474 | 0.79945 | 0.77962 | 0.52560 | 0.00707 | 0.51367 | 0.99698 | N/A |
| n_hidden: 128, n_latent: 30, n_layers: 2 | Embedding | Full | 0.65411 | 0.68831 | 0.61991 | 0.90143 | 0.68173 | 0.29107 | 0.87902 | 0.82291 | 0.79715 | 0.52423 | 0.01176 | 0.53532 | 0.99655 | N/A |
| n_hidden: 128, n_latent: 30, n_layers: 2 | Embedding | HVG | 0.65491 | 0.69831 | 0.61150 | 0.90804 | 0.72090 | 0.29613 | 0.86817 | 0.81180 | 0.79596 | 0.52286 | 0.00706 | 0.53405 | 0.99725 | N/A |
| n_hidden: 128, n_latent: 30, n_layers: 3 | Embedding | Full | 0.61240 | 0.67017 | 0.55463 | 0.88543 | 0.64710 | 0.28562 | 0.86254 | 0.71332 | 0.52559 | 0.52119 | 0.01251 | 0.55896 | 0.99621 | N/A |
| n_hidden: 128, n_latent: 30, n_layers: 3 | Embedding | HVG | 0.61250 | 0.66072 | 0.56429 | 0.89433 | 0.58847 | 0.28710 | 0.87298 | 0.75343 | 0.55882 | 0.52468 | 0.00740 | 0.54389 | 0.99753 | N/A |
| n_hidden: 128, n_latent: 40, n_layers: 1 | Embedding | Full | 0.65381 | 0.69677 | 0.61085 | 0.91594 | 0.70810 | 0.29292 | 0.87011 | 0.82084 | 0.79715 | 0.52184 | 0.01572 | 0.53329 | 0.99628 | N/A |
| n_hidden: 128, n_latent: 40, n_layers: 1 | Embedding | HVG | 0.66276 | 0.71716 | 0.60836 | 0.92330 | 0.76254 | 0.30696 | 0.87584 | 0.81755 | 0.79439 | 0.52166 | 0.00718 | 0.51360 | 0.99579 | N/A |
| n_hidden: 128, n_latent: 40, n_layers: 2 | Embedding | Full | 0.65941 | 0.69193 | 0.62689 | 0.89705 | 0.70583 | 0.29239 | 0.87244 | 0.84152 | 0.85921 | 0.52135 | 0.00764 | 0.53565 | 0.99598 | N/A |
| n_hidden: 128, n_latent: 40, n_layers: 2 | Embedding | HVG | 0.65741 | 0.70388 | 0.61095 | 0.91590 | 0.72494 | 0.30114 | 0.87354 | 0.81200 | 0.79644 | 0.52122 | 0.00782 | 0.53115 | 0.99704 | N/A |
| n_hidden: 128, n_latent: 40, n_layers: 3 | Embedding | Full | 0.61577 | 0.67736 | 0.55418 | 0.88635 | 0.66688 | 0.28892 | 0.86730 | 0.70957 | 0.53747 | 0.52096 | 0.00823 | 0.55229 | 0.99654 | N/A |
| n_hidden: 128, n_latent: 40, n_layers: 3 | Embedding | HVG | 0.61196 | 0.66879 | 0.55514 | 0.88989 | 0.62773 | 0.28807 | 0.86946 | 0.72958 | 0.52774 | 0.52525 | 0.00787 | 0.54275 | 0.99764 | N/A |
| n_hidden: 128, n_latent: 50, n_layers: 1 | Embedding | Full | 0.65774 | 0.71099 | 0.60448 | 0.91629 | 0.75336 | 0.29756 | 0.87676 | 0.79881 | 0.77870 | 0.51969 | 0.00766 | 0.52587 | 0.99614 | N/A |
| n_hidden: 128, n_latent: 50, n_layers: 1 | Embedding | HVG | 0.67235 | 0.72380 | 0.62089 | 0.92597 | 0.78646 | 0.31657 | 0.86621 | 0.83144 | 0.85896 | 0.52105 | 0.00773 | 0.51044 | 0.99572 | N/A |
| n_hidden: 128, n_latent: 50, n_layers: 2 | Embedding | Full | 0.64947 | 0.68897 | 0.60997 | 0.90121 | 0.68864 | 0.29061 | 0.87542 | 0.80356 | 0.78357 | 0.52448 | 0.00765 | 0.54368 | 0.99686 | N/A |
| n_hidden: 128, n_latent: 50, n_layers: 2 | Embedding | HVG | 0.66283 | 0.69805 | 0.62760 | 0.91298 | 0.71603 | 0.29789 | 0.86529 | 0.83308 | **0.86089** | 0.52192 | 0.00728 | 0.54562 | 0.99684 | N/A |
| n_hidden: 128, n_latent: 50, n_layers: 3 | Embedding | Full | 0.61986 | 0.67129 | 0.56843 | 0.88639 | 0.64887 | 0.28735 | 0.86254 | 0.75602 | 0.58338 | 0.52295 | 0.01201 | 0.53945 | 0.99675 | N/A |
| n_hidden: 128, n_latent: 50, n_layers: 3 | Embedding | HVG | 0.60842 | 0.66253 | 0.55431 | 0.88893 | 0.60805 | 0.28637 | 0.86677 | 0.73119 | 0.53076 | 0.52582 | 0.00748 | 0.53298 | 0.99766 | N/A |
| n_hidden: 256, n_latent: 10, n_layers: 1 | Embedding | Full | 0.61165 | 0.65235 | 0.57096 | 0.86527 | 0.58898 | 0.28120 | 0.87395 | 0.78899 | 0.57198 | 0.52930 | 0.01445 | 0.52381 | 0.99721 | N/A |
| n_hidden: 256, n_latent: 10, n_layers: 1 | Embedding | HVG | 0.59689 | 0.61649 | 0.57730 | 0.87229 | 0.44091 | 0.27623 | 0.87653 | 0.79053 | 0.61015 | 0.53843 | 0.00975 | 0.51667 | 0.99488 | N/A |
| n_hidden: 256, n_latent: 10, n_layers: 2 | Embedding | Full | 0.60731 | 0.64689 | 0.56773 | 0.86859 | 0.55718 | 0.28770 | 0.87408 | 0.76193 | 0.55817 | 0.52133 | 0.00775 | 0.56112 | 0.99607 | N/A |
| n_hidden: 256, n_latent: 10, n_layers: 2 | Embedding | HVG | 0.62346 | 0.63475 | 0.61217 | 0.87203 | 0.50980 | 0.28510 | 0.87208 | 0.81245 | 0.79757 | 0.53035 | 0.00765 | 0.52508 | 0.99795 | N/A |
| n_hidden: 256, n_latent: 10, n_layers: 3 | Embedding | Full | 0.64551 | 0.66430 | 0.62671 | 0.86928 | 0.61883 | 0.29025 | 0.87883 | 0.82078 | 0.83384 | 0.52091 | **0.02469** | **0.56293** | 0.99712 | N/A |
| n_hidden: 256, n_latent: 10, n_layers: 3 | Embedding | HVG | 0.63100 | 0.64358 | 0.61842 | 0.87346 | 0.53723 | 0.28346 | 0.88016 | 0.81032 | 0.84141 | 0.52585 | 0.00765 | 0.52760 | 0.99771 | N/A |
| n_hidden: 256, n_latent: 20, n_layers: 1 | Embedding | Full | 0.65787 | 0.68625 | **0.62948** | 0.89727 | 0.68423 | 0.28518 | 0.87831 | **0.84489** | 0.85813 | 0.52553 | 0.00703 | 0.54564 | 0.99569 | N/A |
| n_hidden: 256, n_latent: 20, n_layers: 1 | Embedding | HVG | 0.64887 | 0.68409 | 0.61366 | 0.90912 | 0.66156 | 0.28806 | 0.87763 | 0.82131 | 0.79669 | 0.53006 | 0.00755 | 0.52820 | 0.99745 | N/A |
| n_hidden: 256, n_latent: 20, n_layers: 2 | Embedding | Full | 0.62584 | 0.68605 | 0.56562 | 0.89914 | 0.67630 | 0.29063 | 0.87812 | 0.77730 | 0.56199 | 0.52068 | 0.00948 | 0.52995 | 0.99433 | N/A |
| n_hidden: 256, n_latent: 20, n_layers: 2 | Embedding | HVG | 0.65104 | 0.69236 | 0.60973 | 0.90907 | 0.69069 | 0.29085 | 0.87881 | 0.80883 | 0.79299 | 0.52427 | 0.00767 | 0.52804 | 0.99656 | N/A |
| n_hidden: 256, n_latent: 20, n_layers: 3 | Embedding | Full | 0.65280 | 0.69464 | 0.61095 | 0.89656 | 0.70456 | 0.29539 | 0.88207 | 0.79875 | 0.81649 | 0.51971 | 0.00830 | 0.52771 | 0.99476 | N/A |
| n_hidden: 256, n_latent: 20, n_layers: 3 | Embedding | HVG | 0.64875 | 0.69778 | 0.59972 | 0.90803 | 0.71211 | 0.29468 | 0.87629 | 0.78376 | 0.76247 | 0.51841 | 0.00738 | 0.53057 | 0.99570 | N/A |
| n_hidden: 256, n_latent: 30, n_layers: 1 | Embedding | Full | 0.65585 | 0.69047 | 0.62122 | 0.91145 | 0.68617 | 0.28503 | 0.87923 | 0.83823 | 0.85529 | 0.52191 | 0.00713 | 0.50929 | 0.99549 | N/A |
| n_hidden: 256, n_latent: 30, n_layers: 1 | Embedding | HVG | 0.65739 | 0.70490 | 0.60988 | 0.92313 | 0.72900 | 0.29730 | 0.87016 | 0.81574 | 0.79457 | 0.52190 | 0.00842 | 0.52237 | 0.99628 | N/A |
| n_hidden: 256, n_latent: 30, n_layers: 2, | Embedding | Full | 0.63341 | 0.70886 | 0.55795 | 0.91287 | 0.74254 | 0.29701 | 0.88302 | 0.76289 | 0.55634 | 0.51750 | 0.00826 | 0.51077 | 0.99196 | N/A |
| n_hidden: 256, n_latent: 30, n_layers: 2, | Embedding | HVG | 0.63883 | 0.71430 | 0.56336 | 0.92713 | 0.76269 | 0.29780 | 0.86959 | 0.76609 | 0.56553 | 0.51872 | 0.00725 | 0.52809 | 0.99446 | N/A |
| n_hidden: 256, n_latent: 30, n_layers: 3 | Embedding | Full | 0.65506 | 0.69900 | 0.61113 | 0.89922 | 0.72466 | 0.29775 | 0.87435 | 0.81574 | 0.78873 | 0.51940 | 0.01163 | 0.53679 | 0.99446 | N/A |
| n_hidden: 256, n_latent: 30, n_layers: 3 | Embedding | HVG | 0.65351 | 0.69839 | 0.60864 | 0.90780 | 0.70716 | 0.29730 | 0.88151 | 0.80819 | 0.79138 | 0.51766 | 0.00744 | 0.53231 | 0.99483 | N/A |
| n_hidden: 256, n_latent: 40, n_layers: 1 | Embedding | Full | 0.63427 | 0.70476 | 0.56378 | 0.91711 | 0.73381 | 0.29717 | 0.87096 | 0.78046 | 0.57091 | 0.51976 | 0.00979 | 0.50667 | 0.99512 | N/A |
| n_hidden: 256, n_latent: 40, n_layers: 1 | Embedding | HVG | 0.66624 | 0.71792 | 0.61455 | 0.93110 | 0.76741 | 0.30342 | 0.86976 | 0.82043 | 0.83867 | 0.52025 | 0.00777 | 0.50461 | 0.99558 | N/A |
| n_hidden: 256, n_latent: 40, n_layers: 2 | Embedding | Full | 0.63582 | 0.71045 | 0.56118 | 0.91565 | 0.73793 | 0.30126 | 0.88696 | 0.77508 | 0.56091 | 0.51775 | 0.00825 | 0.51161 | 0.99349 | N/A |
| n_hidden: 256, n_latent: 40, n_layers: 2 | Embedding | HVG | 0.66723 | 0.72784 | 0.60662 | 0.93021 | 0.80163 | 0.30715 | 0.87238 | 0.79704 | 0.81523 | 0.51772 | 0.00760 | 0.50818 | 0.99393 | N/A |
| n_hidden: 256, n_latent: 40, n_layers: 3 | Embedding | Full | 0.65885 | 0.69757 | 0.62014 | 0.90109 | 0.72015 | 0.29695 | 0.87209 | 0.82693 | 0.84340 | 0.51842 | 0.00714 | 0.52890 | 0.99603 | N/A |
| n_hidden: 256, n_latent: 40, n_layers: 3 | Embedding | HVG | 0.64965 | 0.69332 | 0.60597 | 0.90804 | 0.70039 | 0.29500 | 0.86987 | 0.80586 | 0.79057 | 0.51916 | 0.00774 | 0.51675 | 0.99572 | N/A |
| n_hidden: 256, n_latent: 50, n_layers: 1 | Embedding | Full | 0.65949 | 0.71973 | 0.59924 | 0.92461 | 0.76177 | 0.30445 | **0.88811** | 0.79775 | 0.77670 | 0.51784 | 0.01057 | 0.49969 | 0.99288 | N/A |
| n_hidden: 256, n_latent: 50, n_layers: 1 | Embedding | HVG | **0.67480** | 0.73053 | 0.61907 | **0.93546** | **0.80196** | **0.31896** | 0.86575 | 0.83045 | 0.85819 | 0.51812 | 0.00864 | 0.50444 | 0.99455 | N/A |
| n_hidden: 256, n_latent: 50, n_layers: 2 | Embedding | Full | 0.63327 | 0.70822 | 0.55832 | 0.91811 | 0.73961 | 0.29659 | 0.87856 | 0.76670 | 0.55811 | 0.51816 | 0.00762 | 0.50682 | 0.99252 | N/A |
| n_hidden: 256, n_latent: 50, n_layers: 2 | Embedding | HVG | 0.64456 | **0.73114** | 0.55798 | 0.93452 | 0.79943 | 0.31010 | 0.88051 | 0.75948 | 0.55768 | 0.51616 | 0.00959 | 0.51219 | 0.99275 | N/A |
| n_hidden: 256, n_latent: 50, n_layers: 3 | Embedding | Full | 0.65760 | 0.70041 | 0.61478 | 0.89938 | 0.72618 | 0.29835 | 0.87773 | 0.80835 | 0.83159 | 0.51804 | 0.00747 | 0.52737 | 0.99586 | N/A |
| n_hidden: 256, n_latent: 50, n_layers: 3 | Embedding | HVG | 0.65395 | 0.69848 | 0.60942 | 0.91234 | 0.70881 | 0.29537 | 0.87741 | 0.80531 | 0.78837 | 0.51797 | 0.00744 | 0.54100 | 0.99641 | N/A |

**Table 5:** *Performance of scVI on the Zenodo 8020792 (24 PBMC Samples) Dataset*

Table 5 reports the scVI sweep on Zenodo 8020792 over $n_{\text{hidden}} \in \{128, 256\}$, $n_{\text{latent}} \in \{10, 20, 30, 40, 50\}$, $n_{\text{layers}} \in \{1, 2, 3\}$, and Features $\in \{\text{Full, HVG}\}$. The best Overall is $0.67480$ at $(256, 50, 1, \text{HVG})$, while the worst is $0.58505$ at $(128, 10, 2, \text{Full})$. Batch-correction overall peaks at $(256, 50, 2, \text{HVG})$, whereas the biology overall reaches its maximum at $(256, 20, 1, \text{Full})$. By individual metrics, the strongest batch mixing occurs at $(256, 50, 1, \text{HVG})$ for Batch ASW ($0.93546$), PCR ($0.80196$), and iLISI ($0.31896$), graph connectivity peaks at $(256, 50, 1, \text{Full})$ ($0.88811$), clustering agreement peaks at $(256, 20, 1, \text{Full})$ for NMI ($0.84489$) and at $(128, 50, 2, \text{HVG})$ for ARI ($0.86089$), while label compactness (Label ASW) is highest at $(128, 10, 1, \text{HVG})$ ($0.54473$). The trajectory score is not applicable here (N/A) because the dataset lacks the necessary trajectory information.

Paired comparisons clarify the effect of each hyperparameter. Increasing hidden size from $128$ to $256$ improves Overall in $20/30$ matched pairs, the batch overall in $26/30$, and the biology overall in $17/30$. Increasing depth from $n_{\text{layers}}{=}1$ to $3$ is generally unfavorable for the composites: Overall rises in $4/20$ and falls in $16/20$ pairs; the batch overall is $8/20$ up and $12/20$ down; the biology overall is $4/20$ up and $16/20$ down. In contrast, increasing latent dimensionality has the most systematic positive effect on batch correction: stepwise $10{\to}20{\to}30{\to}40{\to}50$ comparisons show Overall up in $34/48$ steps and batch overall up in $39/48$; Batch ASW and iLISI rise in $41/48$ and $38/48$ steps, respectively. The biology overall is mixed stepwise ($23/48$ up, $25/48$ down), and the endpoint comparison $50{-}10$ is strongly positive for Overall ($12/12$ up) and batch overall ($12/12$ up) while only modestly positive for the biology overall ($7/12$ up). Finally, HVG outperforms Full: HVG$-$Full results in Overall ($18/30$ up), batch overall ($17/30$ up), and biology overall ($15/30$ up), with additional gains often seen in NMI/ARI and connectivity.

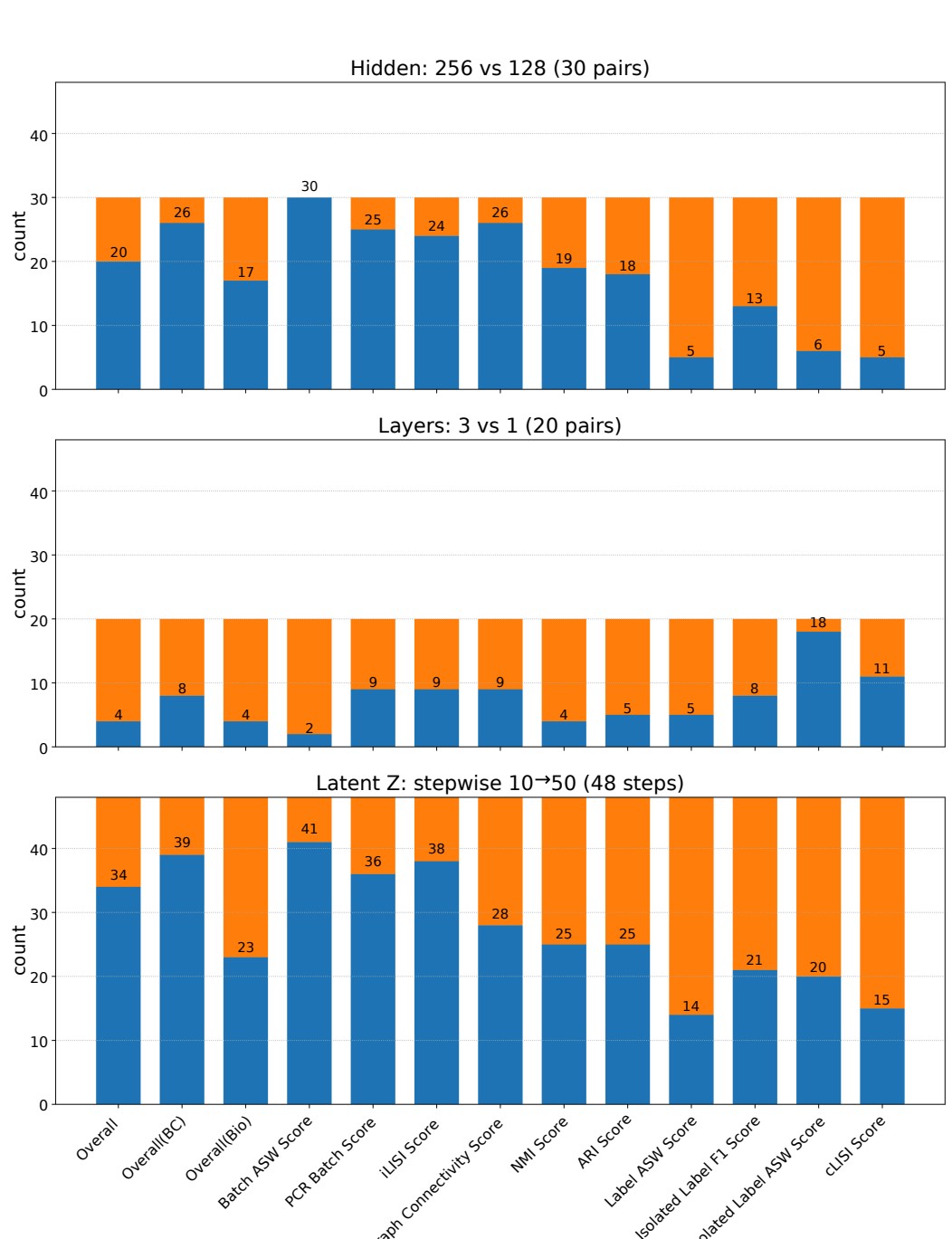

**Figure 19:** *Paired comparison of scVI hyperparameters on the Zenodo 8020792 (24 PBMC Samples) dataset, showing the number of metrics that improve ("up"), decline ("down"), or remain unchanged ("flat") when varying hidden units (256 vs. 128), network depth (3 vs. 1 layers), latent dimension z (stepwise 10→50), and latent dimension u evaluated on both full and HVG feature sets.*

| Hyperparameters | Output | Features | Overall | Overall(BC) | Overall(Bio) | Batch correction | | | | Bio Conservation | | | | | | |
|---|---|---|---|---|---|---|---|---|---|---|---|---|---|---|---|---|
| | | | Overall | Overall(BC) | Overall(Bio) | ASW | PCR | iLISI | GC | NMI | ARI | ASW | IL F1 | IL ASW | cLISI | TC |
| n_hidden: 128, n_latent: 10, n_layers: 1, n_latent_u: 10 | Embedding | Full | 0.59013 | 0.60702 | 0.57324 | 0.81813 | 0.45701 | 0.30326 | 0.84967 | 0.78521 | 0.59622 | 0.54838 | 0.01545 | 0.49550 | 0.99866 | N/A |
| n_hidden: 128, n_latent: 10, n_layers: 1, n_latent_u: 10 | Embedding | HVG | 0.59203 | 0.61041 | 0.57364 | 0.81743 | 0.46514 | 0.30407 | 0.85500 | 0.79153 | 0.60175 | 0.56112 | 0.01422 | 0.47420 | 0.99904 | N/A |
| n_hidden: 128, n_latent: 10, n_layers: 2, n_latent_u: 10 | Embedding | Full | 0.59641 | 0.62303 | 0.56979 | 0.83054 | 0.52853 | 0.29887 | 0.83419 | 0.78071 | 0.59562 | 0.54134 | 0.01258 | 0.49011 | 0.99838 | N/A |
| n_hidden: 128, n_latent: 10, n_layers: 2, n_latent_u: 10 | Embedding | HVG | 0.57880 | 0.58568 | 0.57192 | 0.82467 | 0.38221 | 0.30005 | 0.83579 | 0.78540 | 0.59639 | 0.55647 | 0.01197 | 0.48290 | 0.99843 | N/A |
| n_hidden: 128, n_latent: 10, n_layers: 3, n_latent_u: 10 | Embedding | Full | 0.58070 | 0.58401 | 0.57740 | 0.81260 | 0.38456 | 0.29624 | 0.84262 | 0.79868 | 0.60701 | 0.56400 | 0.01337 | 0.48240 | 0.99897 | N/A |
| n_hidden: 128, n_latent: 10, n_layers: 3, n_latent_u: 10 | Embedding | HVG | 0.58231 | 0.60521 | 0.55942 | 0.82383 | 0.45715 | 0.29495 | 0.84490 | 0.76014 | 0.57599 | 0.55951 | 0.01098 | 0.45190 | 0.99797 | N/A |
| n_hidden: 128, n_latent: 20, n_layers: 1, n_latent_u: 10 | Embedding | Full | 0.62857 | 0.67040 | 0.58674 | 0.85367 | 0.66353 | 0.29653 | 0.86787 | 0.80647 | 0.60946 | 0.56967 | 0.01241 | 0.52313 | 0.99927 | N/A |
| n_hidden: 128, n_latent: 20, n_layers: 1, n_latent_u: 10 | Embedding | HVG | 0.61620 | 0.64940 | 0.58300 | 0.85515 | 0.58008 | 0.29480 | 0.86758 | 0.80156 | 0.60742 | 0.54550 | 0.00874 | 0.53672 | 0.99807 | N/A |
| n_hidden: 128, n_latent: 20, n_layers: 2, n_latent_u: 10 | Embedding | Full | 0.60225 | 0.62362 | 0.58088 | 0.85528 | 0.47405 | 0.29749 | 0.86768 | 0.78830 | 0.58982 | 0.56681 | 0.01018 | 0.53115 | 0.99905 | N/A |
| n_hidden: 128, n_latent: 20, n_layers: 2, n_latent_u: 10 | Embedding | HVG | 0.62096 | 0.65567 | 0.58626 | 0.84905 | 0.61230 | 0.29321 | 0.86812 | 0.80242 | 0.60980 | 0.56551 | 0.01237 | 0.52832 | 0.99914 | N/A |
| n_hidden: 128, n_latent: 20, n_layers: 3, n_latent_u: 10 | Embedding | Full | 0.59705 | 0.61514 | 0.57896 | 0.83567 | 0.47761 | 0.29195 | 0.85534 | 0.78118 | 0.59338 | 0.54454 | 0.01275 | 0.54397 | 0.99796 | N/A |
| n_hidden: 128, n_latent: 20, n_layers: 3, n_latent_u: 10 | Embedding | HVG | 0.59377 | 0.63203 | 0.55551 | 0.83491 | 0.53864 | 0.29586 | 0.85873 | 0.76999 | 0.52186 | 0.54733 | 0.01086 | 0.48486 | 0.99817 | N/A |
| n_hidden: 128, n_latent: 30, n_layers: 1, n_latent_u: 10 | Embedding | Full | 0.60152 | 0.61529 | 0.58774 | 0.86221 | 0.44063 | 0.28793 | 0.87039 | 0.80509 | 0.60843 | 0.56963 | 0.01132 | 0.53286 | 0.99912 | N/A |
| n_hidden: 128, n_latent: 30, n_layers: 1, n_latent_u: 10 | Embedding | HVG | 0.60394 | 0.62885 | 0.57902 | 0.86044 | 0.49541 | 0.29269 | 0.86688 | 0.79888 | 0.60766 | 0.54196 | 0.01247 | 0.51527 | 0.99791 | N/A |
| n_hidden: 128, n_latent: 30, n_layers: 2, n_latent_u: 10 | Embedding | Full | 0.59500 | 0.61432 | 0.57568 | 0.85602 | 0.44839 | 0.29167 | 0.86119 | 0.79628 | 0.60382 | 0.54055 | 0.01104 | 0.50484 | 0.99758 | N/A |
| n_hidden: 128, n_latent: 30, n_layers: 2, n_latent_u: 10 | Embedding | HVG | 0.60041 | 0.62894 | 0.57188 | 0.86073 | 0.49241 | 0.29052 | 0.87212 | 0.78779 | 0.56820 | 0.55362 | 0.00887 | 0.51430 | 0.99849 | N/A |
| n_hidden: 128, n_latent: 30, n_layers: 3, n_latent_u: 10 | Embedding | Full | 0.58625 | 0.60204 | 0.57046 | 0.84875 | 0.40955 | 0.29019 | 0.85969 | 0.76298 | 0.57452 | 0.54528 | 0.01408 | 0.52918 | 0.99673 | N/A |
| n_hidden: 128, n_latent: 30, n_layers: 3, n_latent_u: 10 | Embedding | HVG | 0.59971 | 0.61919 | 0.58023 | 0.84358 | 0.48414 | 0.29440 | 0.85465 | 0.78152 | 0.59503 | 0.54806 | 0.01748 | 0.54112 | 0.99818 | N/A |
| n_hidden: 128, n_latent: 40, n_layers: 1, n_latent_u: 10 | Embedding | Full | 0.61446 | 0.65060 | 0.57831 | 0.86013 | 0.57664 | 0.29424 | 0.87140 | 0.79020 | 0.60027 | 0.56063 | 0.01212 | 0.50754 | 0.99909 | N/A |
| n_hidden: 128, n_latent: 40, n_layers: 1, n_latent_u: 10 | Embedding | HVG | 0.61665 | 0.64728 | 0.58601 | 0.86661 | 0.55707 | 0.29226 | 0.87318 | 0.80251 | 0.60849 | 0.56958 | 0.00941 | 0.52671 | 0.99937 | N/A |
| n_hidden: 128, n_latent: 40, n_layers: 2, n_latent_u: 10 | Embedding | Full | 0.59099 | 0.61241 | 0.56958 | 0.85798 | 0.43722 | 0.29021 | 0.86422 | 0.76561 | 0.57878 | 0.53479 | 0.01026 | 0.53036 | 0.99770 | N/A |
| n_hidden: 128, n_latent: 40, n_layers: 2, n_latent_u: 10 | Embedding | HVG | 0.61538 | 0.64152 | 0.58924 | 0.86105 | 0.54392 | 0.29370 | 0.86739 | 0.80668 | 0.60894 | 0.56947 | 0.01029 | 0.54071 | 0.99933 | N/A |
| n_hidden: 128, n_latent: 40, n_layers: 3, n_latent_u: 10 | Embedding | Full | 0.60671 | 0.62997 | 0.58344 | 0.85498 | 0.51578 | 0.29314 | 0.85598 | 0.80037 | 0.60767 | 0.56079 | 0.01190 | 0.52079 | 0.99912 | N/A |
| n_hidden: 128, n_latent: 40, n_layers: 3, n_latent_u: 10 | Embedding | HVG | 0.58066 | 0.60063 | 0.56069 | 0.84539 | 0.41052 | 0.29243 | 0.85418 | 0.76237 | 0.53922 | 0.54309 | 0.01106 | 0.51069 | 0.99770 | N/A |
| n_hidden: 128, n_latent: 50, n_layers: 1, n_latent_u: 10 | Embedding | Full | 0.60774 | 0.63526 | 0.58022 | 0.86071 | 0.52417 | 0.29042 | 0.86572 | 0.79193 | 0.59969 | 0.55085 | 0.01045 | 0.52943 | 0.99897 | N/A |
| n_hidden: 128, n_latent: 50, n_layers: 1, n_latent_u: 10 | Embedding | HVG | 0.61505 | 0.64749 | 0.58260 | 0.86062 | 0.56669 | 0.29072 | 0.87194 | 0.80170 | 0.60706 | 0.54514 | 0.01186 | 0.53126 | 0.99860 | N/A |
| n_hidden: 128, n_latent: 50, n_layers: 2, n_latent_u: 10 | Embedding | Full | 0.62537 | 0.63479 | **0.61595** | 0.87185 | 0.50124 | 0.29118 | 0.87489 | **0.81015** | **0.80776** | 0.54509 | 0.00980 | 0.52528 | 0.99763 | N/A |
| n_hidden: 128, n_latent: 50, n_layers: 2, n_latent_u: 10 | Embedding | HVG | 0.61293 | 0.64728 | 0.57859 | 0.86235 | 0.56536 | 0.29204 | 0.86936 | 0.79393 | 0.60306 | 0.54867 | 0.01143 | 0.51639 | 0.99806 | N/A |
| n_hidden: 128, n_latent: 50, n_layers: 3, n_latent_u: 10 | Embedding | Full | 0.61281 | 0.64410 | 0.58152 | 0.84847 | 0.57305 | 0.29164 | 0.86321 | 0.77748 | 0.61921 | 0.55068 | 0.01542 | 0.52902 | 0.99730 | N/A |
| n_hidden: 128, n_latent: 50, n_layers: 3, n_latent_u: 10 | Embedding | HVG | 0.59231 | 0.61020 | 0.57442 | 0.84264 | 0.44550 | 0.29281 | 0.85984 | 0.78658 | 0.59879 | 0.54870 | 0.01147 | 0.50243 | 0.99852 | N/A |
| n_hidden: 256, n_latent: 10, n_layers: 1, n_latent_u: 10 | Embedding | Full | 0.59218 | 0.61020 | 0.57416 | 0.82067 | 0.47137 | 0.30083 | 0.84791 | 0.78909 | 0.60025 | 0.55708 | 0.01244 | 0.48717 | 0.99892 | N/A |
| n_hidden: 256, n_latent: 10, n_layers: 1, n_latent_u: 10 | Embedding | HVG | 0.58516 | 0.60199 | 0.56834 | 0.82725 | 0.43749 | 0.30144 | 0.84176 | 0.77967 | 0.59857 | 0.55390 | 0.01172 | 0.46751 | 0.99869 | N/A |
| n_hidden: 256, n_latent: 10, n_layers: 2, n_latent_u: 10 | Embedding | Full | 0.57619 | 0.58158 | 0.57079 | 0.81296 | 0.37973 | 0.29911 | 0.83454 | 0.78007 | 0.59222 | 0.57097 | 0.01100 | 0.47154 | 0.99894 | N/A |
| n_hidden: 256, n_latent: 10, n_layers: 2, n_latent_u: 10 | Embedding | HVG | 0.57687 | 0.58434 | 0.56941 | 0.81655 | 0.38017 | 0.29993 | 0.84470 | 0.77580 | 0.59942 | 0.55161 | 0.01048 | 0.48143 | 0.99772 | N/A |
| n_hidden: 256, n_latent: 10, n_layers: 3, n_latent_u: 10 | Embedding | Full | 0.54718 | 0.57444 | 0.51991 | 0.78311 | 0.42086 | 0.29842 | 0.79538 | 0.68952 | 0.43996 | 0.52513 | 0.01368 | 0.45420 | 0.99697 | N/A |
| n_hidden: 256, n_latent: 10, n_layers: 3, n_latent_u: 10 | Embedding | HVG | 0.56522 | 0.59149 | 0.53895 | 0.79510 | 0.48630 | 0.29568 | 0.78890 | 0.71649 | 0.51965 | 0.54651 | 0.01361 | 0.44022 | 0.99725 | N/A |
| n_hidden: 256, n_latent: 20, n_layers: 1, n_latent_u: 10 | Embedding | Full | 0.61786 | 0.65153 | 0.58420 | 0.85603 | 0.58371 | 0.29674 | 0.86964 | 0.80848 | 0.61257 | 0.56260 | 0.00753 | 0.51518 | 0.99884 | N/A |
| n_hidden: 256, n_latent: 20, n_layers: 1, n_latent_u: 10 | Embedding | HVG | 0.61268 | 0.64424 | 0.58113 | 0.85260 | 0.56295 | 0.29942 | 0.86200 | 0.80826 | 0.61077 | 0.56455 | 0.01021 | 0.49394 | 0.99903 | N/A |
| n_hidden: 256, n_latent: 20, n_layers: 2, n_latent_u: 10 | Embedding | Full | 0.60335 | 0.62207 | 0.58462 | 0.83744 | 0.50178 | 0.29344 | 0.85561 | 0.80647 | 0.60996 | 0.56368 | 0.01011 | 0.51843 | 0.99911 | N/A |
| n_hidden: 256, n_latent: 20, n_layers: 2, n_latent_u: 10 | Embedding | HVG | 0.59519 | 0.61619 | 0.57419 | 0.83063 | 0.47935 | 0.29358 | 0.86119 | 0.78734 | 0.59689 | 0.54673 | 0.01706 | 0.49949 | 0.99762 | N/A |
| n_hidden: 256, n_latent: 20, n_layers: 3, n_latent_u: 10 | Embedding | Full | 0.59002 | 0.60920 | 0.57083 | 0.82470 | 0.48407 | 0.29333 | 0.83470 | 0.77503 | 0.59661 | 0.56937 | 0.01256 | 0.47221 | 0.99922 | N/A |
| n_hidden: 256, n_latent: 20, n_layers: 3, n_latent_u: 10 | Embedding | HVG | 0.55302 | 0.57829 | 0.52775 | 0.80220 | 0.45188 | 0.29723 | 0.76183 | 0.71309 | 0.42156 | 0.53444 | 0.01519 | 0.48363 | 0.99857 | N/A |
| n_hidden: 256, n_latent: 30, n_layers: 1, n_latent_u: 10 | Embedding | Full | 0.60355 | 0.63399 | 0.57311 | 0.85903 | 0.50854 | 0.29572 | 0.87268 | 0.78786 | 0.56942 | 0.54735 | 0.00903 | 0.52674 | 0.99824 | N/A |
| n_hidden: 256, n_latent: 30, n_layers: 1, n_latent_u: 10 | Embedding | HVG | 0.60806 | 0.63165 | 0.58446 | 0.85803 | 0.50322 | 0.29309 | 0.87225 | 0.80241 | 0.61340 | 0.54944 | 0.01072 | 0.53207 | 0.99874 | N/A |
| n_hidden: 256, n_latent: 30, n_layers: 2, n_latent_u: 10 | Embedding | Full | 0.59265 | 0.60820 | 0.57709 | 0.83312 | 0.45616 | 0.29249 | 0.85102 | 0.78946 | 0.60172 | 0.54905 | 0.01375 | 0.51040 | 0.99819 | N/A |
| n_hidden: 256, n_latent: 30, n_layers: 2, n_latent_u: 10 | Embedding | HVG | 0.58636 | 0.62730 | 0.54542 | 0.83407 | 0.52648 | 0.29493 | 0.85371 | 0.73396 | 0.52166 | 0.54458 | 0.01002 | 0.46569 | 0.99664 | N/A |
| n_hidden: 256, n_latent: 30, n_layers: 3, n_latent_u: 10 | Embedding | Full | 0.57285 | 0.59052 | 0.55518 | 0.83071 | 0.40138 | 0.29497 | 0.83502 | 0.74628 | 0.54341 | 0.54868 | 0.01446 | 0.48093 | 0.99735 | N/A |
| n_hidden: 256, n_latent: 30, n_layers: 3, n_latent_u: 10 | Embedding | HVG | 0.54843 | 0.59064 | 0.50621 | 0.80231 | 0.45665 | 0.29532 | 0.80829 | 0.67034 | 0.39078 | 0.52556 | 0.01383 | 0.43951 | 0.99726 | N/A |
| n_hidden: 256, n_latent: 40, n_layers: 1, n_latent_u: 10 | Embedding | Full | 0.61305 | 0.64536 | 0.58073 | 0.86413 | 0.55124 | 0.29319 | 0.87290 | 0.80015 | 0.60800 | 0.55329 | 0.01039 | 0.51369 | 0.99888 | N/A |
| n_hidden: 256, n_latent: 40, n_layers: 1, n_latent_u: 10 | Embedding | HVG | 0.62126 | 0.66464 | 0.57787 | 0.86784 | 0.62335 | 0.29705 | 0.87034 | 0.80221 | 0.60605 | 0.55171 | 0.00908 | 0.49939 | 0.99879 | N/A |
| n_hidden: 256, n_latent: 40, n_layers: 2, n_latent_u: 10 | Embedding | Full | 0.60360 | 0.62764 | 0.57956 | 0.85401 | 0.49613 | 0.29323 | 0.86720 | 0.80285 | 0.61129 | 0.55466 | 0.01348 | 0.49635 | 0.99871 | N/A |
| n_hidden: 256, n_latent: 40, n_layers: 2, n_latent_u: 10 | Embedding | HVG | 0.60628 | 0.64884 | 0.56372 | 0.84842 | 0.59534 | 0.29654 | 0.85505 | 0.76533 | 0.55390 | 0.55356 | 0.00903 | 0.50203 | 0.99849 | N/A |
| n_hidden: 256, n_latent: 40, n_layers: 3, n_latent_u: 10 | Embedding | Full | 0.59956 | 0.63911 | 0.56002 | 0.86002 | 0.60382 | 0.29386 | 0.84139 | 0.74303 | 0.52980 | 0.55632 | 0.01477 | 0.51815 | 0.99803 | N/A |
| n_hidden: 256, n_latent: 40, n_layers: 3, n_latent_u: 10 | Embedding | HVG | 0.59143 | 0.61349 | 0.56936 | 0.82524 | 0.49434 | 0.29576 | 0.83862 | 0.78415 | 0.59467 | 0.56391 | 0.01871 | 0.45607 | 0.99867 | N/A |
| n_hidden: 256, n_latent: 50, n_layers: 1, n_latent_u: 10 | Embedding | Full | 0.62042 | 0.66264 | 0.57820 | 0.86919 | 0.61570 | 0.29041 | 0.87527 | 0.79453 | 0.60424 | 0.54768 | 0.00946 | 0.51500 | 0.99830 | N/A |
| n_hidden: 256, n_latent: 50, n_layers: 1, n_latent_u: 10 | Embedding | HVG | 0.61281 | 0.63865 | 0.58697 | 0.86729 | 0.52899 | 0.28961 | 0.86872 | 0.80251 | 0.60827 | 0.55128 | 0.01126 | 0.55005 | 0.99845 | N/A |
| n_hidden: 256, n_latent: 50, n_layers: 2, n_latent_u: 10 | Embedding | Full | 0.60155 | 0.64772 | 0.55537 | 0.83803 | 0.60494 | 0.29365 | 0.85426 | 0.72827 | 0.47986 | 0.53801 | 0.01445 | **0.57528** | 0.99636 | N/A |
| n_hidden: 256, n_latent: 50, n_layers: 2, n_latent_u: 10 | Embedding | HVG | 0.60676 | 0.64160 | 0.57193 | 0.84618 | 0.56691 | 0.29307 | 0.86023 | 0.77895 | 0.59342 | 0.54650 | 0.01664 | 0.49833 | 0.99773 | N/A |
| n_hidden: 256, n_latent: 50, n_layers: 3, n_latent_u: 10 | Embedding | Full | 0.59673 | 0.64697 | 0.54650 | 0.81925 | 0.62848 | 0.29541 | 0.84474 | 0.73691 | 0.50024 | 0.55338 | 0.01865 | 0.47053 | 0.99930 | N/A |
| n_hidden: 256, n_latent: 50, n_layers: 3, n_latent_u: 10 | Embedding | HVG | 0.60831 | 0.65029 | 0.56633 | 0.81212 | **0.68825** | 0.29416 | 0.82661 | 0.76398 | 0.57294 | 0.57688 | 0.01600 | 0.46896 | 0.99922 | N/A |
| n_hidden: 128, n_latent: 10, n_layers: 1, n_latent_u: 20 | Embedding | Full | 0.59499 | 0.63481 | 0.55516 | 0.82472 | 0.56013 | 0.29444 | 0.85497 | 0.76177 | 0.51165 | 0.55822 | 0.01130 | 0.48954 | 0.99849 | N/A |
| n_hidden: 128, n_latent: 10, n_layers: 1, n_latent_u: 20 | Embedding | HVG | 0.58780 | 0.61329 | 0.56230 | 0.82831 | 0.47584 | 0.29904 | 0.84999 | 0.76679 | 0.58016 | 0.55355 | 0.01587 | 0.45957 | 0.99785 | N/A |
| n_hidden: 128, n_latent: 10, n_layers: 2, n_latent_u: 20 | Embedding | Full | 0.58574 | 0.61216 | 0.55932 | 0.81606 | 0.49700 | 0.29813 | 0.83743 | 0.74994 | 0.57361 | 0.54011 | 0.01307 | 0.48147 | 0.99772 | N/A |
| n_hidden: 128, n_latent: 10, n_layers: 2, n_latent_u: 20 | Embedding | HVG | 0.57706 | 0.61244 | 0.54168 | 0.81146 | 0.48509 | 0.30090 | 0.85232 | 0.72910 | 0.49740 | 0.53841 | 0.01323 | 0.47456 | 0.99736 | N/A |
| n_hidden: 128, n_latent: 10, n_layers: 3, n_latent_u: 20 | Embedding | Full | 0.58113 | 0.59659 | 0.56566 | 0.81353 | 0.43318 | 0.29732 | 0.84234 | 0.76719 | 0.58297 | 0.54086 | 0.01524 | 0.48946 | 0.99823 | N/A |
| n_hidden: 128, n_latent: 10, n_layers: 3, n_latent_u: 20 | Embedding | HVG | 0.58604 | 0.60088 | 0.57119 | 0.80762 | 0.46030 | 0.29504 | 0.84055 | 0.77966 | 0.59651 | 0.56070 | 0.01547 | 0.47625 | 0.99857 | N/A |
| n_hidden: 128, n_latent: 20, n_layers: 1, n_latent_u: 20 | Embedding | Full | 0.60429 | 0.62316 | 0.58543 | 0.85097 | 0.47760 | 0.29734 | 0.86671 | 0.80545 | 0.60909 | 0.57571 | 0.00865 | 0.51462 | 0.99907 | N/A |
| n_hidden: 128, n_latent: 20, n_layers: 1, n_latent_u: 20 | Embedding | HVG | 0.60536 | 0.62754 | 0.58319 | 0.84894 | 0.49986 | 0.29966 | 0.86170 | 0.80564 | 0.60926 | 0.56841 | 0.00964 | 0.50692 | 0.99926 | N/A |
| n_hidden: 128, n_latent: 20, n_layers: 2, n_latent_u: 20 | Embedding | Full | 0.59916 | 0.61349 | 0.58483 | 0.85239 | 0.43059 | 0.29838 | 0.87232 | 0.80626 | 0.60950 | 0.57317 | 0.01022 | 0.51062 | 0.99922 | N/A |
| n_hidden: 128, n_latent: 20, n_layers: 2, n_latent_u: 20 | Embedding | HVG | 0.60321 | 0.62336 | 0.58306 | 0.84361 | 0.50152 | 0.29742 | 0.85086 | 0.80505 | 0.60867 | 0.55804 | 0.00989 | 0.51772 | 0.99898 | N/A |
| n_hidden: 128, n_latent: 20, n_layers: 3, n_latent_u: 20 | Embedding | Full | 0.59545 | 0.61166 | 0.57924 | 0.83960 | 0.46310 | 0.29285 | 0.85111 | 0.80263 | 0.60728 | 0.56477 | 0.01060 | 0.49163 | 0.99850 | N/A |
| n_hidden: 128, n_latent: 20, n_layers: 3, n_latent_u: 20 | Embedding | HVG | 0.58865 | 0.60789 | 0.56940 | 0.83452 | 0.45138 | 0.29346 | 0.85221 | 0.76744 | 0.58478 | 0.55311 | 0.01029 | 0.50264 | 0.99816 | N/A |
| n_hidden: 128, n_latent: 30, n_layers: 1, n_latent_u: 20 | Embedding | Full | 0.60858 | 0.64239 | 0.57477 | 0.87632 | 0.51845 | 0.29869 | 0.87609 | 0.79075 | 0.56905 | 0.56883 | 0.00810 | 0.51253 | 0.99936 | N/A |
| n_hidden: 128, n_latent: 30, n_layers: 1, n_latent_u: 20 | Embedding | HVG | 0.61909 | 0.65440 | 0.58378 | 0.88046 | 0.56916 | 0.29291 | 0.87507 | 0.78787 | 0.60504 | **0.58799** | 0.01400 | 0.50847 | 0.99932 | N/A |
| n_hidden: 128, n_latent: 30, n_layers: 2, n_latent_u: 20 | Embedding | Full | 0.60287 | 0.62942 | 0.57632 | 0.85694 | 0.50490 | 0.29167 | 0.86416 | 0.78460 | 0.60622 | 0.54815 | 0.01000 | 0.51145 | 0.99752 | N/A |
| n_hidden: 128, n_latent: 30, n_layers: 2, n_latent_u: 20 | Embedding | HVG | 0.61313 | 0.64384 | 0.58243 | 0.85983 | 0.55844 | 0.29742 | 0.85967 | 0.80668 | 0.60893 | 0.55996 | 0.01005 | 0.51004 | 0.99892 | N/A |
| n_hidden: 128, n_latent: 30, n_layers: 3, n_latent_u: 20 | Embedding | Full | 0.59936 | 0.63012 | 0.56860 | 0.84201 | 0.52724 | 0.29513 | 0.85611 | 0.78366 | 0.59865 | 0.54829 | 0.01192 | 0.47095 | 0.99810 | N/A |
| n_hidden: 128, n_latent: 30, n_layers: 3, n_latent_u: 20 | Embedding | HVG | 0.59819 | 0.62795 | 0.56843 | 0.83585 | 0.53136 | 0.29291 | 0.85169 | 0.78100 | 0.56740 | 0.54943 | 0.01018 | 0.50400 | 0.99853 | N/A |
| n_hidden: 128, n_latent: 40, n_layers: 1, n_latent_u: 20 | Embedding | Full | 0.62263 | 0.65599 | 0.58927 | 0.88492 | 0.57129 | 0.29730 | 0.87044 | 0.80728 | 0.60928 | 0.55337 | 0.00836 | 0.53797 | 0.99939 | N/A |
| n_hidden: 128, n_latent: 40, n_layers: 1, n_latent_u: 20 | Embedding | HVG | 0.61821 | 0.65592 | 0.58049 | 0.88211 | 0.57252 | 0.29629 | 0.87277 | 0.79669 | 0.60449 | 0.55397 | 0.01034 | 0.51979 | 0.99766 | N/A |
| n_hidden: 128, n_latent: 40, n_layers: 2, n_latent_u: 20 | Embedding | Full | 0.62002 | 0.65710 | 0.58293 | 0.86682 | 0.59115 | 0.29650 | 0.87394 | 0.79634 | 0.60150 | 0.57321 | 0.01055 | 0.51655 | 0.99944 | N/A |
| n_hidden: 128, n_latent: 40, n_layers: 2, n_latent_u: 20 | Embedding | HVG | 0.60917 | 0.63307 | 0.58526 | 0.86903 | 0.50547 | 0.29000 | 0.86780 | 0.80476 | 0.61163 | 0.56213 | 0.00968 | 0.52462 | 0.99877 | N/A |
| n_hidden: 128, n_latent: 40, n_layers: 3, n_latent_u: 20 | Embedding | Full | 0.61384 | 0.65236 | 0.57531 | 0.85553 | 0.60171 | 0.29399 | 0.85822 | 0.79413 | 0.58794 | 0.56850 | 0.00916 | 0.49303 | 0.99912 | N/A |
| n_hidden: 128, n_latent: 40, n_layers: 3, n_latent_u: 20 | Embedding | HVG | 0.62263 | 0.66503 | 0.58023 | 0.86035 | 0.64656 | 0.29271 | 0.86050 | 0.79005 | 0.60957 | 0.58133 | 0.01126 | 0.48995 | 0.99919 | N/A |
| n_hidden: 128, n_latent: 50, n_layers: 1, n_latent_u: 20 | Embedding | Full | 0.61735 | 0.65461 | 0.58009 | 0.88343 | 0.55803 | 0.30139 | 0.87558 | 0.79517 | 0.57273 | 0.57286 | 0.01009 | 0.53022 | 0.99948 | N/A |
| n_hidden: 128, n_latent: 50, n_layers: 1, n_latent_u: 20 | Embedding | HVG | 0.61803 | 0.66332 | 0.57275 | **0.88575** | 0.59936 | 0.30057 | 0.86758 | 0.78699 | 0.57040 | 0.54128 | 0.00860 | 0.53056 | 0.99867 | N/A |
| n_hidden: 128, n_latent: 50, n_layers: 2, n_latent_u: 20 | Embedding | Full | 0.61490 | 0.65648 | 0.57331 | 0.87055 | 0.58469 | 0.29601 | 0.87469 | 0.78816 | 0.60509 | 0.55238 | 0.00881 | 0.48637 | 0.99904 | N/A |
| n_hidden: 128, n_latent: 50, n_layers: 2, n_latent_u: 20 | Embedding | HVG | 0.60166 | 0.63560 | 0.56773 | 0.87278 | 0.51292 | 0.29008 | 0.86660 | 0.76316 | 0.55855 | 0.54778 | 0.00990 | 0.52921 | 0.99777 | N/A |
| n_hidden: 128, n_latent: 50, n_layers: 3, n_latent_u: 20 | Embedding | Full | 0.61364 | 0.64711 | 0.58017 | 0.85297 | 0.58090 | 0.29237 | 0.86220 | 0.80517 | 0.60868 | 0.56213 | 0.01013 | 0.49620 | 0.99873 | N/A |
| n_hidden: 128, n_latent: 50, n_layers: 3, n_latent_u: 20 | Embedding | HVG | 0.60614 | 0.63424 | 0.57803 | 0.84200 | 0.55029 | 0.29057 | 0.85411 | 0.79505 | 0.60882 | 0.55502 | 0.01088 | 0.49993 | 0.99850 | N/A |
| n_hidden: 256, n_latent: 10, n_layers: 1, n_latent_u: 20 | Embedding | Full | 0.57716 | 0.61300 | 0.54132 | 0.82572 | 0.48398 | 0.29927 | 0.84303 | 0.74396 | 0.50586 | 0.54759 | 0.00983 | 0.44308 | 0.99758 | N/A |
| n_hidden: 256, n_latent: 10, n_layers: 1, n_latent_u: 20 | Embedding | HVG | 0.59947 | 0.62226 | 0.57668 | 0.82450 | 0.51661 | 0.30065 | 0.84726 | 0.79374 | 0.60657 | 0.56076 | 0.01224 | 0.48790 | 0.99888 | N/A |
| n_hidden: 256, n_latent: 10, n_layers: 2, n_latent_u: 20 | Embedding | Full | 0.59069 | 0.61903 | 0.56234 | 0.81562 | 0.52733 | 0.29687 | 0.83631 | 0.76081 | 0.57504 | 0.54555 | 0.01279 | 0.48163 | 0.99823 | N/A |
| n_hidden: 256, n_latent: 10, n_layers: 2, n_latent_u: 20 | Embedding | HVG | 0.58774 | 0.60757 | 0.56792 | 0.79933 | 0.49995 | 0.29636 | 0.83462 | 0.77841 | 0.59242 | 0.54639 | 0.01279 | 0.47903 | 0.99845 | N/A |
| n_hidden: 256, n_latent: 10, n_layers: 3, n_latent_u: 20 | Embedding | Full | 0.55390 | 0.58751 | 0.52029 | 0.77521 | 0.51535 | 0.29348 | 0.76601 | 0.69694 | 0.41358 | 0.52811 | 0.01915 | 0.46658 | 0.99740 | N/A |
| n_hidden: 256, n_latent: 10, n_layers: 3, n_latent_u: 20 | Embedding | HVG | 0.54528 | 0.58024 | 0.51031 | 0.77411 | 0.47629 | 0.29480 | 0.77577 | 0.68059 | 0.38835 | 0.54643 | 0.02114 | 0.42723 | 0.99813 | N/A |
| n_hidden: 256, n_latent: 20, n_layers: 1, n_latent_u: 20 | Embedding | Full | 0.60733 | 0.63468 | 0.57998 | 0.85474 | 0.51841 | 0.29938 | 0.86620 | 0.78821 | 0.59204 | 0.58115 | 0.01114 | 0.50813 | 0.99920 | N/A |
| n_hidden: 256, n_latent: 20, n_layers: 1, n_latent_u: 20 | Embedding | HVG | 0.60349 | 0.62615 | 0.58083 | 0.84779 | 0.49911 | 0.29653 | 0.86119 | 0.79503 | 0.60123 | 0.57349 | 0.00993 | 0.50605 | 0.99923 | N/A |
| n_hidden: 256, n_latent: 20, n_layers: 2, n_latent_u: 20 | Embedding | Full | 0.60903 | 0.63668 | 0.58139 | 0.83983 | 0.55634 | 0.29780 | 0.85274 | 0.80276 | 0.61323 | 0.56794 | 0.01157 | 0.49358 | 0.99926 | N/A |
| n_hidden: 256, n_latent: 20, n_layers: 2, n_latent_u: 20 | Embedding | HVG | 0.60556 | 0.63053 | 0.58059 | 0.83687 | 0.53400 | 0.29798 | 0.85327 | 0.80357 | 0.60887 | 0.58279 | 0.00945 | 0.47955 | 0.99928 | N/A |
| n_hidden: 256, n_latent: 20, n_layers: 3, n_latent_u: 20 | Embedding | Full | 0.57724 | 0.59882 | 0.55567 | 0.81453 | 0.44497 | 0.29463 | 0.84115 | 0.74443 | 0.53802 | 0.55055 | 0.01988 | 0.48329 | 0.99783 | N/A |
| n_hidden: 256, n_latent: 20, n_layers: 3, n_latent_u: 20 | Embedding | HVG | 0.57046 | 0.58565 | 0.55527 | 0.81559 | 0.40691 | 0.29695 | 0.82315 | 0.74136 | 0.56064 | 0.55423 | 0.01278 | 0.46442 | 0.99818 | N/A |
| n_hidden: 256, n_latent: 30, n_layers: 1, n_latent_u: 20 | Embedding | Full | 0.60763 | 0.63893 | 0.57633 | 0.87619 | 0.50844 | 0.29628 | 0.87480 | 0.78510 | 0.59783 | 0.54885 | 0.00823 | 0.52020 | 0.99779 | N/A |
| n_hidden: 256, n_latent: 30, n_layers: 1, n_latent_u: 20 | Embedding | HVG | 0.62285 | 0.66232 | 0.58337 | 0.87100 | 0.60460 | 0.30242 | 0.87126 | 0.80079 | 0.60872 | 0.56606 | 0.00679 | 0.51877 | 0.99909 | N/A |
| n_hidden: 256, n_latent: 30, n_layers: 2, n_latent_u: 20 | Embedding | Full | 0.59797 | 0.61908 | 0.57686 | 0.84434 | 0.47724 | 0.29455 | 0.86019 | 0.78636 | 0.64654 | 0.55749 | 0.00945 | 0.46317 | 0.99814 | N/A |
| n_hidden: 256, n_latent: 30, n_layers: 2, n_latent_u: 20 | Embedding | HVG | 0.59729 | 0.61352 | 0.58107 | 0.84807 | 0.44811 | 0.29347 | 0.86441 | 0.78006 | 0.64351 | 0.54813 | 0.00883 | 0.50726 | 0.99865 | N/A |
| n_hidden: 256, n_latent: 30, n_layers: 3, n_latent_u: 20 | Embedding | Full | 0.57741 | 0.60938 | 0.54543 | 0.81993 | 0.49063 | 0.29379 | 0.83318 | 0.73130 | 0.51983 | 0.56289 | 0.01473 | 0.44520 | 0.99861 | N/A |
| n_hidden: 256, n_latent: 30, n_layers: 3, n_latent_u: 20 | Embedding | HVG | 0.56552 | 0.60797 | 0.52308 | 0.80285 | 0.53134 | 0.29794 | 0.79974 | 0.69314 | 0.42871 | 0.51600 | 0.01603 | 0.48733 | 0.99726 | N/A |
| n_hidden: 256, n_latent: 40, n_layers: 1, n_latent_u: 20 | Embedding | Full | **0.62984** | **0.67882** | 0.58080 | 0.86298 | 0.66298 | 0.30316 | 0.87221 | 0.79097 | 0.57217 | 0.57839 | 0.00854 | 0.53562 | 0.99946 | N/A |
| n_hidden: 256, n_latent: 40, n_layers: 1, n_latent_u: 20 | Embedding | HVG | 0.62879 | 0.67019 | 0.58738 | 0.87630 | 0.62490 | 0.30232 | 0.87723 | 0.79684 | 0.60195 | 0.57823 | 0.01140 | 0.53640 | **0.99948** | N/A |
| n_hidden: 256, n_latent: 40, n_layers: 2, n_latent_u: 20 | Embedding | Full | 0.60797 | 0.64855 | 0.56740 | 0.85010 | 0.59676 | 0.29299 | 0.85434 | 0.74223 | 0.56070 | 0.54849 | 0.01532 | 0.53871 | 0.99851 | N/A |
| n_hidden: 256, n_latent: 40, n_layers: 2, n_latent_u: 20 | Embedding | HVG | 0.61228 | 0.64597 | 0.57858 | 0.84132 | 0.59432 | 0.29342 | 0.85482 | 0.78951 | 0.60172 | 0.54810 | 0.01426 | 0.52004 | 0.99785 | N/A |
| n_hidden: 256, n_latent: 40, n_layers: 3, n_latent_u: 20 | Embedding | Full | 0.59987 | 0.64093 | 0.55882 | 0.81955 | 0.63396 | 0.29555 | 0.81426 | 0.75830 | 0.54101 | 0.58722 | 0.00998 | 0.45706 | 0.99933 | N/A |
| n_hidden: 256, n_latent: 40, n_layers: 3, n_latent_u: 20 | Embedding | HVG | 0.57922 | 0.63943 | 0.51901 | 0.80809 | 0.62005 | 0.29756 | 0.83201 | 0.70288 | 0.41998 | 0.55261 | 0.01317 | 0.42671 | 0.99874 | N/A |
| n_hidden: 256, n_latent: 50, n_layers: 1, n_latent_u: 20 | Embedding | Full | 0.61174 | 0.65856 | 0.56492 | 0.87889 | 0.57983 | 0.28819 | **0.87733** | 0.76140 | 0.54598 | 0.54713 | 0.00887 | 0.52997 | 0.99618 | N/A |
| n_hidden: 256, n_latent: 50, n_layers: 1, n_latent_u: 20 | Embedding | HVG | 0.61918 | 0.65888 | 0.57947 | 0.88199 | 0.58412 | 0.29592 | 0.87350 | 0.79395 | 0.60565 | 0.54191 | 0.00853 | 0.52847 | 0.99831 | N/A |
| n_hidden: 256, n_latent: 50, n_layers: 2, n_latent_u: 20 | Embedding | Full | 0.62288 | 0.65974 | 0.58602 | 0.88578 | 0.58570 | 0.29577 | 0.86659 | 0.80578 | 0.61535 | 0.57405 | 0.00798 | 0.51368 | 0.99927 | N/A |
| n_hidden: 256, n_latent: 50, n_layers: 2, n_latent_u: 20 | Embedding | HVG | 0.59956 | 0.64010 | 0.55903 | 0.85118 | 0.55740 | 0.29329 | 0.85853 | 0.75003 | 0.53974 | 0.54425 | 0.01252 | 0.51081 | 0.99680 | N/A |
| n_hidden: 256, n_latent: 50, n_layers: 3, n_latent_u: 20 | Embedding | Full | 0.59025 | 0.63906 | 0.54144 | 0.80727 | 0.62901 | 0.29769 | 0.82227 | 0.72911 | 0.47169 | 0.57904 | 0.01183 | 0.45787 | 0.99908 | N/A |
| n_hidden: 256, n_latent: 50, n_layers: 3, n_latent_u: 20 | Embedding | HVG | 0.57848 | 0.62720 | 0.52976 | 0.80175 | 0.58353 | 0.29812 | 0.82541 | 0.71767 | 0.42656 | 0.57477 | 0.01780 | 0.44301 | 0.99874 | N/A |

**Table 6:** *Performance of MrVI on the Zenodo 8020792 (24 PBMC Samples)*

Table 6 reports the MrVI sweep on Zenodo 8020792 over $n_{\text{hidden}} \in \{128, 256\}$, $n_{\text{latent}} \in \{10, 20, 30, 40, 50\}$, $n_{\text{layers}} \in \{1, 2, 3\}$, $n_{\text{latent\_u}} \in \{10, 20\}$ and Features $\in \{\text{Full}, \text{HVG}\}$. The best Overall is $0.62984$ at $(256, 40, 1, \text{Full})$, while the worst is $0.54528$ at $(256, 10, 3, \text{HVG})$.

Changing hidden size from 128 to 256 tends to lower the composite scores in this grid. Across matched pairs that fix $(n_{\text{latent}}, n_{\text{layers}}, \text{Features})$, Overall improves in $9/30$ and worsens in $21/30$ pairs; the batch overall in $9/30$ up and $21/30$ down; the biology overall in $8/30$ up and $22/30$ down. Metric-wise, moving to 256 often improves PCR and iLISI (many ups) and frequently raises isolated-label F1, but it more often reduces Batch ASW, graph connectivity, and clustering agreement (NMI/ARI), with label-compactness metrics showing mixed behavior. In short, larger hidden size favors certain batch harmonization statistics but usually hurts the aggregate objectives on this dataset.

Increasing depth from $n_{\text{layers}}=1$ to 3 is broadly unfavorable for the composites. In matched pairs, Overall rises in $1/20$ and falls in $19/20$; the batch overall $1/20$ up and $19/20$ down; the biology overall $3/20$ up and $17/20$ down. Batch ASW and graph connectivity decrease in nearly all pairs, iLISI and PCR improve only occasionally, and clustering agreement (NMI/ARI) more often declines. The consistent exception is isolated-label F1, which increases in all pairs. Practically, depth erodes batch mixing and graph cohesion here and does not pay off in the aggregate.

The latent dimension shows interactions with capacity and features, but in this table the exact stepwise trends ($10 \to 50$) cannot be paired systematically across fixed $(n_{\text{hidden}}, n_{\text{layers}}, \text{Features})$, so we avoid over-generalization. The best Overall occurs at moderate-to-high latent (40) with $(256, 1, \text{Full})$, while several batch and structure metrics peak at higher latent values in specific settings, illustrating that latent interacts with both hidden size and feature selection.

Sweeping $n_{\text{latent\_u}}$ causes the primary Overall increases in $37/60$ pairs, the batch composite Overall(BC) in $40/60$, while the biology composite Overall(Bio) decreases in aggregate (up $23/60$, down $37/60$). Batch metrics are the main beneficiaries: PCR rises in $39/60$, iLISI in $40/60$, and Batch ASW in $34/60$. Graph connectivity slightly declines on average (up $23/60$). For clustering agreement, NMI and ARI tilt down overall (NMI up $23/60$; ARI up $24/60$). Label compactness is mixed: Label ASW improves (up $37/60$), whereas Isolated Label ASW and Isolated Label F1 tend to fall (up $20/60$; up $26/60$). cLISI is essentially flat (up $32/60$).

Comparing feature selection at matched $(n_{\text{hidden}}, n_{\text{latent}}, n_{\text{layers}})$, Full has a slight edge on the composite Overalls for this dataset: HVG minus Full shows small negative mean differences for Overall and the batch overall, whereas the biology overall is roughly balanced to slightly positive for HVG. However, we consider these differences negligible and still recommend HVG as a more favarobale option.

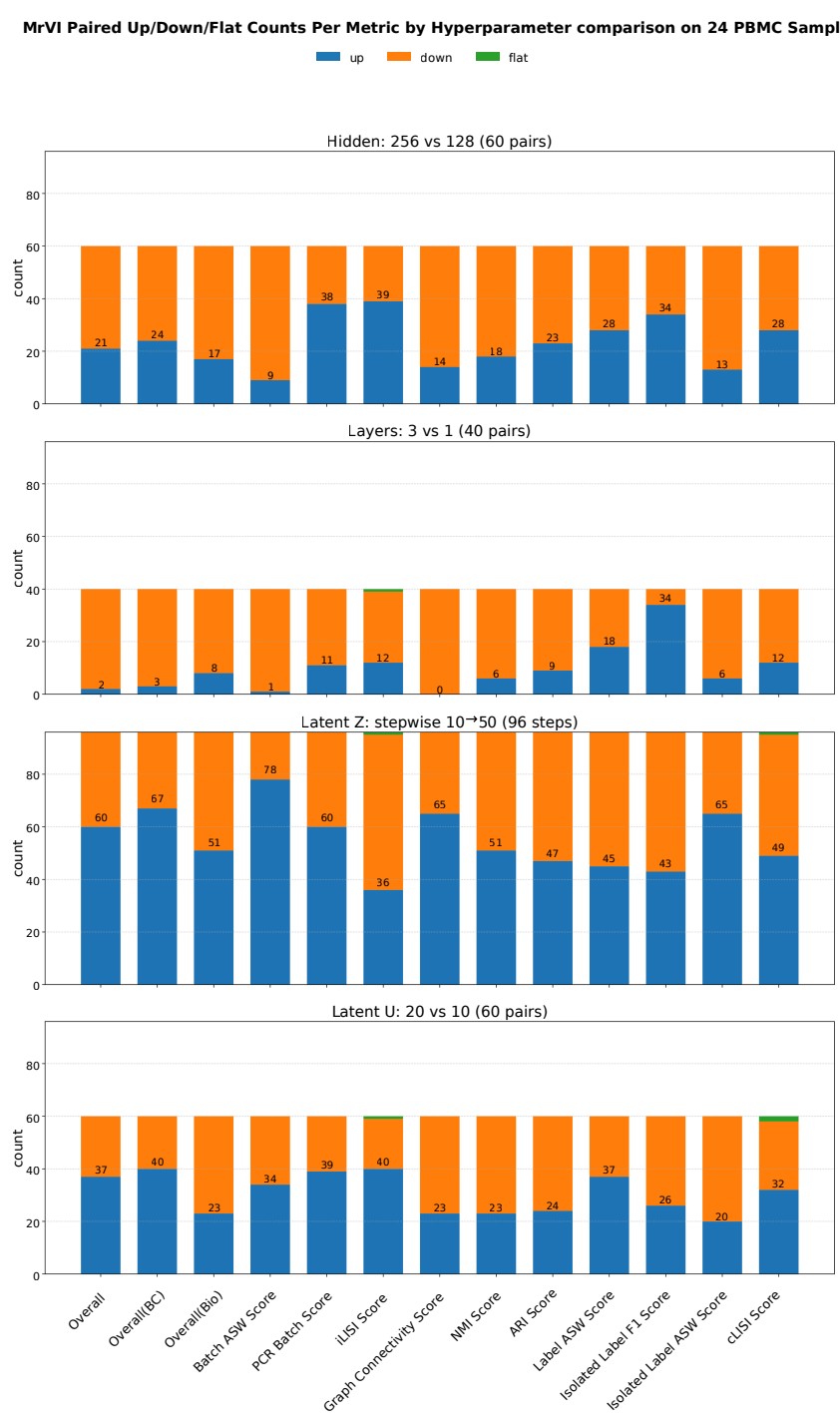

**Figure 20:** *Paired comparison of MrVI hyperparameters on the Zenodo 8020792 (24 PBMC Samples) dataset, showing the number of metrics that improve ("up"), decline ("down"), or remain unchanged ("flat") when varying hidden units (256 vs. 128), network depth (3 vs. 1 layers), latent dimension z (stepwise 10→50), and latent dimension u evaluated on both full and HVG feature sets.*

| Hyperparameters | Output | Features | Overall | Overall(BC) | Overall(Bio) | Batch correction | | | | Bio Conservation | | | | | | |
|---|---|---|---|---|---|---|---|---|---|---|---|---|---|---|---|---|
| | | | Overall | Overall(BC) | Overall(Bio) | ASW | PCR | iLISI | GC | NMI | ARI | ASW | IL F1 | IL ASW | cLISI | TC |
| n_hidden: 128, n_latent: 10, n_layers: 1 | Embedding | Full | 0.51549 | 0.48411 | 0.54686 | 0.88813 | 0.00000 | 0.19367 | 0.85466 | 0.74521 | 0.51185 | 0.52900 | 0.01184 | 0.48594 | 0.99735 | N/A |
| n_hidden: 128, n_latent: 10, n_layers: 1 | Embedding | HVG | 0.55120 | 0.52689 | 0.57550 | 0.86717 | 0.13432 | **0.24613** | 0.85993 | 0.79920 | 0.60907 | 0.55167 | 0.00964 | 0.48464 | 0.99880 | N/A |
| n_hidden: 128, n_latent: 10, n_layers: 2 | Embedding | Full | 0.51041 | 0.48118 | 0.53965 | 0.88043 | 0.00000 | 0.19450 | 0.84980 | 0.74076 | 0.48000 | 0.52395 | 0.01077 | 0.48471 | 0.99770 | N/A |
| n_hidden: 128, n_latent: 10, n_layers: 2 | Embedding | HVG | 0.54472 | 0.52298 | 0.56646 | 0.86198 | 0.13740 | 0.24010 | 0.85246 | 0.78505 | 0.57112 | 0.54935 | 0.01203 | 0.48235 | 0.99886 | N/A |
| n_hidden: 128, n_latent: 10, n_layers: 3 | Embedding | Full | 0.50921 | 0.48070 | 0.53772 | 0.88309 | 0.00000 | 0.19557 | 0.84415 | 0.72170 | 0.47779 | 0.52928 | 0.01210 | 0.48811 | 0.99736 | N/A |
| n_hidden: 128, n_latent: 10, n_layers: 3 | Embedding | HVG | 0.55734 | 0.54111 | 0.57356 | 0.86361 | 0.21277 | 0.24388 | 0.84425 | 0.79678 | 0.60688 | 0.55104 | 0.01178 | 0.47626 | 0.99866 | N/A |
| n_hidden: 128, n_latent: 20, n_layers: 1 | Embedding | Full | 0.51273 | 0.49885 | 0.52660 | 0.93720 | 0 | 0.19062 | 0.86759 | 0.69067 | 0.40265 | 0.52923 | 0.00764 | 0.53124 | 0.99819 | N/A |
| n_hidden: 128, n_latent: 20, n_layers: 1 | Embedding | HVG | 0.54876 | 0.52437 | 0.57315 | 0.92383 | 0.08196 | 0.21913 | 0.87258 | 0.77654 | 0.59528 | 0.54117 | 0.00865 | 0.51838 | 0.99886 | N/A |
| n_hidden: 128, n_latent: 20, n_layers: 2 | Embedding | Full | 0.51583 | 0.49352 | 0.53815 | 0.92086 | 0.00000 | 0.19163 | 0.86157 | 0.72863 | 0.45521 | 0.52969 | 0.00834 | 0.50868 | 0.99834 | N/A |
| n_hidden: 128, n_latent: 20, n_layers: 2 | Embedding | HVG | 0.54682 | 0.52379 | 0.56984 | 0.91814 | 0.08562 | 0.21899 | 0.87241 | 0.77373 | 0.58014 | 0.53987 | 0.00913 | 0.51782 | 0.99836 | N/A |
| n_hidden: 128, n_latent: 20, n_layers: 3 | Embedding | Full | 0.50840 | 0.48891 | 0.52790 | 0.91136 | 0.00000 | 0.19155 | 0.85272 | 0.68801 | 0.43344 | 0.53151 | 0.01242 | 0.50373 | 0.99831 | N/A |
| n_hidden: 128, n_latent: 20, n_layers: 3 | Embedding | HVG | 0.53866 | 0.50211 | 0.57521 | 0.90559 | 0.01943 | 0.21554 | 0.86786 | 0.78726 | 0.60724 | 0.53937 | 0.01404 | 0.50455 | 0.99879 | N/A |
| n_hidden: 128, n_latent: 30, n_layers: 1 | Embedding | Full | 0.51273 | 0.49885 | 0.52660 | 0.93720 | 0.00000 | 0.19062 | 0.86759 | 0.69067 | 0.40265 | 0.52922 | 0.00764 | 0.53124 | 0.99819 | N/A |
| n_hidden: 128, n_latent: 30, n_layers: 1 | Embedding | HVG | 0.57523 | 0.53752 | 0.61293 | 0.93584 | 0.11433 | 0.21885 | 0.88108 | 0.82037 | 0.79473 | 0.53291 | 0.00866 | 0.52285 | 0.99805 | N/A |
| n_hidden: 128, n_latent: 30, n_layers: 2 | Embedding | Full | 0.51535 | 0.49682 | 0.53389 | 0.92970 | 0.00000 | 0.19141 | 0.86616 | 0.70637 | 0.42632 | 0.53066 | 0.00773 | 0.53381 | 0.99842 | N/A |
| n_hidden: 128, n_latent: 30, n_layers: 2 | Embedding | HVG | 0.57928 | 0.54182 | 0.61675 | 0.92683 | 0.13715 | 0.22184 | 0.88144 | 0.82317 | 0.79763 | 0.53684 | 0.01240 | 0.53168 | 0.99879 | N/A |
| n_hidden: 128, n_latent: 30, n_layers: 3 | Embedding | Full | 0.51507 | 0.49594 | 0.53420 | 0.92641 | 0.00000 | 0.18920 | 0.86814 | 0.69894 | 0.43236 | 0.52729 | 0.00913 | 0.53880 | 0.99842 | N/A |
| n_hidden: 128, n_latent: 30, n_layers: 3 | Embedding | HVG | 0.54050 | 0.51687 | 0.56413 | 0.91983 | 0.05721 | 0.21379 | 0.87664 | 0.75531 | 0.54305 | 0.53874 | 0.00884 | 0.53981 | **0.99902** | N/A |
| n_hidden: 128, n_latent: 40, n_layers: 1 | Embedding | Full | 0.51784 | 0.50298 | 0.53269 | 0.94561 | 0.00000 | 0.18812 | 0.87820 | 0.68877 | 0.43152 | 0.52629 | 0.00783 | 0.54352 | 0.99823 | N/A |
| n_hidden: 128, n_latent: 40, n_layers: 1 | Embedding | HVG | 0.59014 | 0.56627 | 0.61402 | 0.93920 | 0.21880 | 0.22223 | 0.88484 | 0.81606 | 0.80252 | 0.53054 | 0.01254 | 0.52399 | 0.99845 | N/A |
| n_hidden: 128, n_latent: 40, n_layers: 2 | Embedding | Full | 0.53146 | 0.49891 | 0.56401 | 0.93759 | 0.00000 | 0.18941 | 0.86863 | 0.71649 | 0.58238 | 0.52608 | 0.01274 | 0.54789 | 0.99847 | N/A |
| n_hidden: 128, n_latent: 40, n_layers: 2 | Embedding | HVG | 0.57452 | 0.53447 | 0.61456 | 0.93492 | 0.10944 | 0.21649 | 0.87703 | 0.81629 | 0.80038 | 0.53328 | 0.00817 | 0.53054 | 0.99870 | N/A |
| n_hidden: 128, n_latent: 40, n_layers: 3 | Embedding | Full | 0.51104 | 0.49317 | 0.52890 | 0.92956 | 0.00000 | 0.18566 | 0.85747 | 0.67512 | 0.40898 | 0.53148 | 0.01394 | 0.54531 | 0.99858 | N/A |
| n_hidden: 128, n_latent: 40, n_layers: 3 | Embedding | HVG | 0.55509 | 0.54320 | 0.56699 | 0.91711 | 0.16817 | 0.21576 | 0.87176 | 0.76721 | 0.56395 | 0.53647 | 0.00796 | 0.52746 | 0.99888 | N/A |
| n_hidden: 128, n_latent: 50, n_layers: 1 | Embedding | Full | 0.52835 | 0.50435 | 0.55234 | 0.94982 | 0.00000 | 0.18901 | 0.87857 | 0.69812 | 0.53790 | 0.52606 | 0.01113 | 0.54286 | 0.99799 | N/A |
| n_hidden: 128, n_latent: 50, n_layers: 1 | Embedding | HVG | 0.60066 | 0.59141 | 0.60991 | 0.94234 | 0.30567 | 0.22689 | **0.89075** | 0.81293 | 0.79819 | 0.52873 | 0.00772 | 0.51393 | 0.99797 | N/A |
| n_hidden: 128, n_latent: 50, n_layers: 2 | Embedding | Full | 0.54379 | 0.50065 | 0.58693 | 0.93706 | 0.00000 | 0.19340 | 0.87215 | 0.75799 | 0.68500 | 0.52597 | 0.01363 | 0.54074 | 0.99827 | N/A |
| n_hidden: 128, n_latent: 50, n_layers: 2 | Embedding | HVG | 0.58113 | 0.55571 | 0.60656 | 0.93699 | 0.19159 | 0.21733 | 0.87692 | 0.79881 | 0.77401 | 0.53232 | 0.00765 | 0.52797 | 0.99858 | N/A |
| n_hidden: 128, n_latent: 50, n_layers: 3 | Embedding | Full | 0.51113 | 0.49919 | 0.52306 | 0.93440 | 0.00000 | 0.19005 | 0.87231 | 0.67265 | 0.38670 | 0.52782 | 0.01211 | 0.54038 | 0.99870 | N/A |
| n_hidden: 128, n_latent: 50, n_layers: 3 | Embedding | HVG | 0.55031 | 0.53456 | 0.56605 | 0.92026 | 0.12686 | 0.21516 | 0.87597 | 0.75612 | 0.55914 | 0.53880 | 0.00746 | 0.53587 | 0.99890 | N/A |
| n_hidden: 256, n_latent: 10, n_layers: 1 | Embedding | Full | 0.51072 | 0.48619 | 0.53525 | 0.89236 | 0.00000 | 0.19487 | 0.85755 | 0.72506 | 0.47273 | 0.51889 | 0.01139 | 0.48594 | 0.99750 | N/A |
| n_hidden: 256, n_latent: 10, n_layers: 1 | Embedding | HVG | 0.54479 | 0.51305 | 0.57653 | 0.87855 | 0.09456 | 0.23165 | 0.84747 | 0.79532 | 0.60684 | 0.55135 | 0.01210 | 0.49475 | 0.99884 | N/A |
| n_hidden: 256, n_latent: 10, n_layers: 2 | Embedding | Full | 0.49913 | 0.48442 | 0.51385 | 0.89182 | 0.00000 | 0.19218 | 0.85366 | 0.66681 | 0.41222 | 0.51792 | 0.01103 | 0.47943 | 0.99567 | N/A |
| n_hidden: 256, n_latent: 10, n_layers: 2 | Embedding | HVG | 0.54687 | 0.51688 | 0.57687 | 0.87524 | 0.10033 | 0.23703 | 0.85492 | 0.78916 | 0.61363 | 0.54798 | **0.01871** | 0.49290 | 0.99883 | N/A |
| n_hidden: 256, n_latent: 10, n_layers: 3 | Embedding | Full | 0.51227 | 0.48289 | 0.54166 | 0.88822 | 0.00000 | 0.19761 | 0.84573 | 0.74189 | 0.48051 | 0.52873 | 0.01183 | 0.48871 | 0.99827 | N/A |
| n_hidden: 256, n_latent: 10, n_layers: 3 | Embedding | HVG | 0.54915 | 0.52319 | 0.57511 | 0.87930 | 0.11750 | 0.24076 | 0.85521 | 0.79588 | 0.60181 | **0.55489** | 0.01457 | 0.48463 | 0.99892 | N/A |
| n_hidden: 256, n_latent: 20, n_layers: 1 | Embedding | Full | 0.51629 | 0.49456 | 0.53801 | 0.92246 | 0.00000 | 0.19237 | 0.86342 | 0.71824 | 0.45804 | 0.53059 | 0.00999 | 0.51320 | 0.99799 | N/A |
| n_hidden: 256, n_latent: 20, n_layers: 1 | Embedding | HVG | 0.56250 | 0.50208 | **0.62292** | 0.92420 | 0.00000 | 0.21611 | 0.86802 | **0.83027** | **0.85160** | 0.53646 | 0.00830 | 0.51231 | 0.99857 | N/A |
| n_hidden: 256, n_latent: 20, n_layers: 2 | Embedding | Full | 0.51282 | 0.49331 | 0.53233 | 0.92101 | 0.00000 | 0.18900 | 0.86322 | 0.70897 | 0.43137 | 0.53243 | 0.01003 | 0.51379 | 0.99741 | N/A |
| n_hidden: 256, n_latent: 20, n_layers: 2 | Embedding | HVG | 0.53244 | 0.49874 | 0.56613 | 0.92241 | 0.00000 | 0.21629 | 0.85627 | 0.75994 | 0.58423 | 0.53256 | 0.01013 | 0.51591 | 0.99846 | N/A |
| n_hidden: 256, n_latent: 20, n_layers: 3 | Embedding | Full | 0.51795 | 0.49323 | 0.54266 | 0.91942 | 0.00000 | 0.19159 | 0.86192 | 0.73711 | 0.47139 | 0.53033 | 0.00962 | 0.50935 | 0.99818 | N/A |
| n_hidden: 256, n_latent: 20, n_layers: 3 | Embedding | HVG | 0.53919 | 0.49853 | 0.57986 | 0.91986 | 0.00000 | 0.21455 | 0.85971 | 0.78164 | 0.63522 | 0.53612 | 0.01108 | 0.51649 | 0.99861 | N/A |
| n_hidden: 256, n_latent: 30, n_layers: 1 | Embedding | Full | 0.53118 | 0.49962 | 0.56274 | 0.93657 | 0.00000 | 0.19236 | 0.86956 | 0.72094 | 0.59635 | 0.52527 | 0.01010 | 0.52587 | 0.99789 | N/A |
| n_hidden: 256, n_latent: 30, n_layers: 1 | Embedding | HVG | 0.57281 | 0.53069 | 0.61492 | 0.94221 | 0.08748 | 0.21845 | 0.87463 | 0.82088 | 0.79471 | 0.53051 | 0.01102 | 0.53384 | 0.99856 | N/A |
| n_hidden: 256, n_latent: 30, n_layers: 2, | Embedding | Full | 0.53282 | 0.49878 | 0.56687 | 0.93559 | 0.00000 | 0.18843 | 0.87110 | 0.73199 | 0.59668 | 0.52739 | 0.00767 | 0.53902 | 0.99848 | N/A |
| n_hidden: 256, n_latent: 30, n_layers: 2, | Embedding | HVG | 0.55224 | 0.52756 | 0.57692 | 0.93774 | 0.07999 | 0.21856 | 0.87397 | 0.79435 | 0.60002 | 0.52893 | 0.00758 | 0.53241 | 0.99822 | N/A |
| n_hidden: 256, n_latent: 30, n_layers: 3 | Embedding | Full | 0.51660 | 0.49891 | 0.53428 | 0.93900 | 0.00000 | 0.18691 | 0.86975 | 0.70018 | 0.43221 | 0.52635 | 0.00770 | 0.54072 | 0.99850 | N/A |
| n_hidden: 256, n_latent: 30, n_layers: 3 | Embedding | HVG | 0.56166 | 0.51011 | 0.61321 | 0.93323 | 0.01694 | 0.21476 | 0.87552 | 0.82534 | 0.79267 | 0.53350 | 0.00759 | 0.52171 | 0.99847 | N/A |
| n_hidden: 256, n_latent: 40, n_layers: 1 | Embedding | Full | 0.53223 | 0.50200 | 0.56247 | 0.94627 | 0.00000 | 0.18792 | 0.87379 | 0.71182 | 0.59224 | 0.52392 | 0.00776 | 0.54097 | 0.99810 | N/A |
| n_hidden: 256, n_latent: 40, n_layers: 1 | Embedding | HVG | 0.57098 | 0.56753 | 0.57444 | 0.94793 | 0.21896 | 0.22045 | 0.88278 | 0.78151 | 0.60593 | 0.52441 | 0.00767 | 0.52934 | 0.99777 | N/A |
| n_hidden: 256, n_latent: 40, n_layers: 2 | Embedding | Full | 0.53191 | 0.50145 | 0.56238 | 0.94215 | 0.00000 | 0.18921 | 0.87442 | 0.71506 | 0.57860 | 0.52796 | 0.01247 | 0.54185 | 0.99836 | N/A |
| n_hidden: 256, n_latent: 40, n_layers: 2 | Embedding | HVG | 0.57055 | 0.57238 | 0.56873 | 0.94346 | 0.23474 | 0.22313 | 0.88819 | 0.78294 | 0.57279 | 0.52748 | 0.00808 | 0.52305 | 0.99804 | N/A |
| n_hidden: 256, n_latent: 40, n_layers: 3 | Embedding | Full | 0.53530 | 0.50218 | 0.56842 | 0.93922 | 0.00000 | 0.19081 | 0.87871 | 0.72709 | 0.59332 | 0.52672 | 0.01240 | **0.55266** | 0.99833 | N/A |
| n_hidden: 256, n_latent: 40, n_layers: 3 | Embedding | HVG | 0.56762 | 0.57191 | 0.56333 | 0.94124 | 0.23910 | 0.22425 | 0.88307 | 0.76647 | 0.55386 | 0.52849 | 0.00741 | 0.52540 | 0.99835 | N/A |
| n_hidden: 256, n_latent: 50, n_layers: 1 | Embedding | Full | 0.53060 | 0.50570 | 0.55549 | **0.95092** | 0.00000 | 0.19255 | 0.87934 | 0.71753 | 0.54795 | 0.52302 | 0.00884 | 0.53762 | 0.99801 | N/A |
| n_hidden: 256, n_latent: 50, n_layers: 1 | Embedding | HVG | **0.60260** | **0.59410** | 0.61110 | 0.94816 | **0.31321** | 0.23013 | 0.88488 | 0.81683 | 0.79643 | 0.52234 | 0.00813 | 0.52584 | 0.99706 | N/A |
| n_hidden: 256, n_latent: 50, n_layers: 2 | Embedding | Full | 0.52861 | 0.50139 | 0.55582 | 0.94951 | 0.00000 | 0.18629 | 0.86927 | 0.69463 | 0.56546 | 0.52482 | 0.00773 | 0.54404 | 0.99824 | N/A |
| n_hidden: 256, n_latent: 50, n_layers: 2 | Embedding | HVG | 0.59579 | 0.58074 | 0.61084 | 0.94580 | 0.27763 | 0.22532 | 0.87422 | 0.81298 | 0.79631 | 0.52515 | 0.00782 | 0.52521 | 0.99758 | N/A |
| n_hidden: 256, n_latent: 50, n_layers: 3 | Embedding | Full | 0.52215 | 0.49896 | 0.54535 | 0.94187 | 0.00000 | 0.18724 | 0.86674 | 0.68241 | 0.51693 | 0.52494 | 0.01204 | 0.53745 | 0.99830 | N/A |
| n_hidden: 256, n_latent: 50, n_layers: 3 | Embedding | HVG | 0.58618 | 0.56151 | 0.61086 | 0.94462 | 0.19731 | 0.21920 | 0.88489 | 0.81441 | 0.79847 | 0.52575 | 0.00780 | 0.52045 | 0.99826 | N/A |

**Table 7:** *Performance of LDVAE on the Zenodo 8020792 (24 PBMC Samples)*

Table 7 summarizes the LDVAE sweep on Zenodo 8020792 over $n_{\text{hidden}} \in \{128, 256\}$, $n_{\text{latent}} \in \{10, 20, 30, 40, 50\}$, $n_{\text{layers}} \in \{1, 2, 3\}$, and Features $\in \{\text{Full}, \text{HVG}\}$. The best Overall is 0.60260 at $(256, 50, 1, \text{HVG})$, while the worst is 0.49913 at $(256, 10, 2, \text{Full})$. Across matched pairs, increasing hidden size from 128 to 256 is favorable for LDVAE on this dataset. Overall improves in $19/30$ pairs (down in $11/30$), the batch-correction overall improves in $17/30$ (down in $13/30$), and the biology overall improves in $20/30$ (down in $10/30$). Thus, larger hidden units tend to raise all three composites here.

Depth has the opposite effect. Comparing $n_{\text{layers}}=3$ versus 1 at fixed $(n_{\text{hidden}}, n_{\text{latent}}, \text{Features})$, Overall rises in $6/20$ and falls in $14/20$ pairs; the batch-correction overall rises in $4/20$ and falls in $16/20$; the biology overall rises in $6/20$ and falls in $14/20$. Deeper encoders are broadly unfavorable for the aggregates in this benchmark.

Latent dimensionality shows the most systematic trend. Stepwise comparisons $10 \rightarrow 20 \rightarrow 30 \rightarrow 40 \rightarrow 50$ aggregated over fixed $(n_{\text{hidden}}, n_{\text{layers}}, \text{Features})$ yield Overall up in $31/48$ steps (down $16/48$), the batch-correction overall up in $36/48$ (down $11/48$), and the biology overall up in $24/48$ (down $23/48$). The endpoint comparison $50-10$ strengthens this picture: Overall is higher in $11/12$ matched triplets (down $1/12$), batch overall in $11/12$ (down $1/12$), and biology overall in $10/12$ (down $2/12$). Increasing the latent dimension therefore helps LDVAE's aggregate scores here, especially the batch-oriented composite.

Features strongly favor HVG for LDVAE in this dataset. In matched $(n_{\text{hidden}}, n_{\text{latent}}, n_{\text{layers}})$ pairs, HVG$-$Full for Overall with counts $30/30$ up, the batch overall $30/30$ up, and the biology overall with $29/30$ up. HVG consistently dominates Full for the composites.

Metric-wise optima align with these trends. The batch-correction metrics favor high latent and often larger hidden size: Batch ASW peaks at $(256, 50, 1, \text{Full})$, PCR at $(256, 50, 1, \text{HVG})$, while iLISI achieves its maximum at $(128, 10, 1, \text{HVG})$. Graph connectivity peaks at $(128, 50, 1, \text{HVG})$. Clustering agreement is strongest at moderate latent with higher hidden size, with NMI and ARI both maximized at $(256, 20, 1, \text{HVG})$. Label compactness (Label

ASW) peaks at $(256, 10, 3, \text{HVG})$, reflecting that deeper and lower-latent settings can tighten local label structure even as they hurt the composite scores.

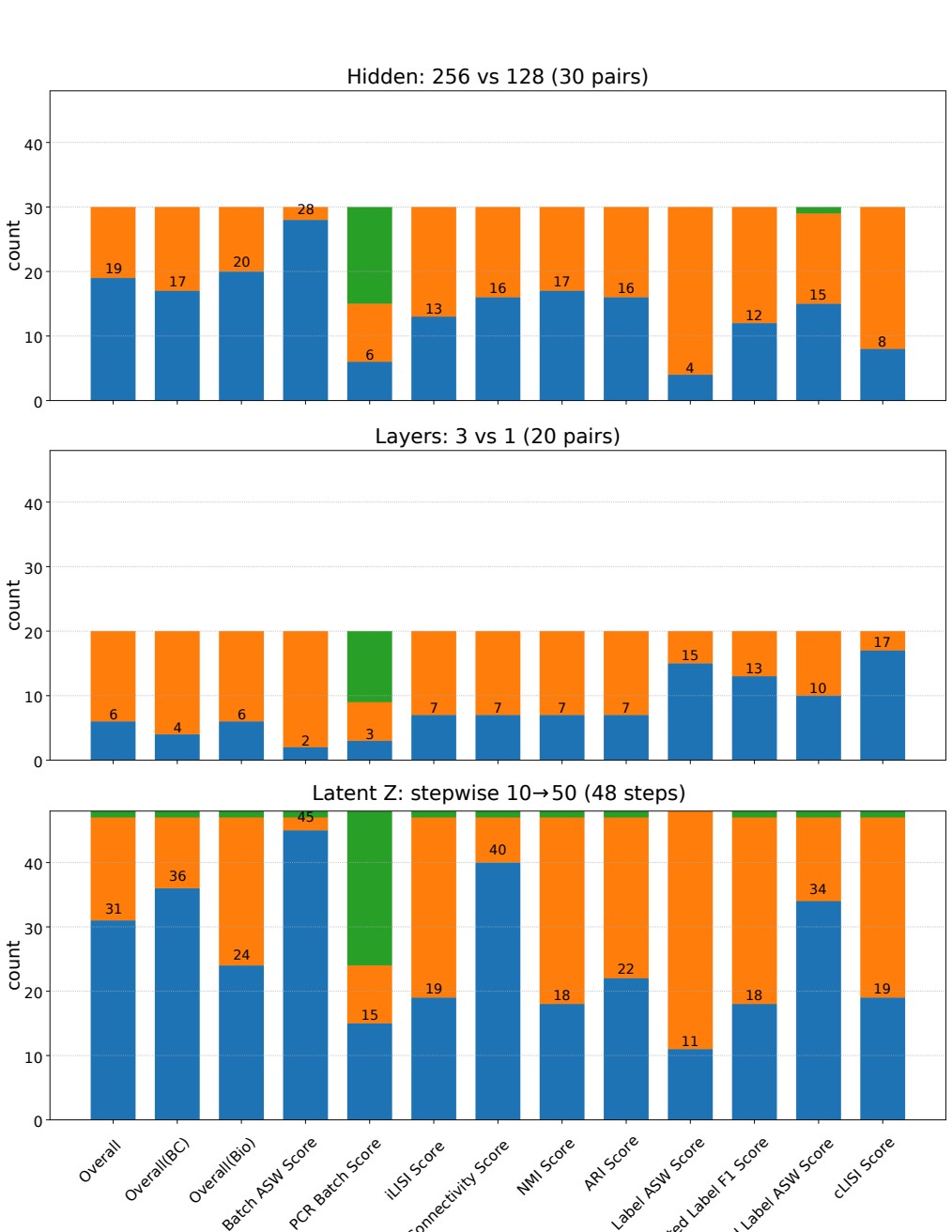

**Figure 21:** *Paired comparison of LDVAE hyperparameters on the Zenodo 8020792 (24 PBMC Samples) dataset, showing the number of metrics that improve ("up"), decline ("down"), or remain unchanged ("flat") when varying hidden units (256 vs. 128), network depth (3 vs. 1 layers), latent dimension z (stepwise 10→50) on both full and HVG feature sets.*

## K  PER METRIC PERFORMANCE ON ZENODO 11100300 (18 PBMC SAMPLES)

| Hyperparameters | Output | Features | Overall | Overall(BC) | Overall(Bio) | Batch correction | | | | Bio Conservation | | | | | | |
| | | | Overall | Overall(BC) | Overall(Bio) | ASW | PCR | iLISI | GC | NMI | ARI | ASW | IL F1 | IL ASW | cLISI | TC |
|---|---|---|---|---|---|---|---|---|---|---|---|---|---|---|---|---|
| n_hidden: 128, n_latent: 10, n_layers: 1 | Embedding | Full | 0.74357 | 0.79067 | 0.69647 | 0.93655 | 0.86584 | 0.36092 | 0.99937 | 0.71434 | 0.49695 | 0.57586 | N/A | N/A | 0.99871 | N/A |
| n_hidden: 128, n_latent: 10, n_layers: 1 | Embedding | HVG | 0.73700 | 0.78083 | 0.69317 | 0.94925 | 0.81664 | 0.35916 | 0.99827 | 0.70142 | 0.49437 | **0.57865** | N/A | N/A | 0.99825 | N/A |
| n_hidden: 128, n_latent: 10, n_layers: 2 | Embedding | Full | 0.73842 | 0.79513 | 0.68171 | 0.93177 | 0.88430 | 0.36541 | 0.99904 | 0.68888 | 0.47270 | 0.56771 | N/A | N/A | 0.99754 | N/A |
| n_hidden: 128, n_latent: 10, n_layers: 2 | Embedding | HVG | 0.73006 | 0.78232 | 0.67780 | 0.94277 | 0.82257 | 0.36570 | 0.99823 | 0.67910 | 0.46637 | 0.56817 | N/A | N/A | 0.99756 | N/A |
| n_hidden: 128, n_latent: 10, n_layers: 3 | Embedding | Full | 0.75741 | 0.79657 | 0.71826 | 0.93719 | 0.88280 | 0.36667 | 0.99961 | 0.71811 | 0.59776 | 0.56012 | N/A | N/A | 0.99704 | N/A |
| n_hidden: 128, n_latent: 10, n_layers: 3 | Embedding | HVG | 0.72656 | 0.77860 | 0.67453 | 0.94542 | 0.81243 | 0.35829 | 0.99824 | 0.67862 | 0.46732 | 0.55572 | N/A | N/A | 0.99645 | N/A |
| n_hidden: 128, n_latent: 20, n_layers: 1 | Embedding | Full | 0.73332 | 0.79921 | 0.66722 | 0.94498 | 0.89199 | 0.36015 | **0.99973** | 0.66800 | 0.45710 | 0.54662 | N/A | N/A | 0.99717 | N/A |
| n_hidden: 128, n_latent: 20, n_layers: 1 | Embedding | HVG | 0.74026 | 0.78192 | 0.69861 | 0.96380 | 0.80793 | 0.35738 | 0.99857 | 0.69219 | 0.55992 | 0.54762 | N/A | N/A | 0.99470 | N/A |
| n_hidden: 128, n_latent: 20, n_layers: 2 | Embedding | Full | 0.76084 | 0.80487 | 0.71682 | 0.95225 | 0.90314 | 0.36455 | 0.99952 | 0.72047 | 0.59984 | 0.55024 | N/A | N/A | 0.99673 | N/A |
| n_hidden: 128, n_latent: 20, n_layers: 2 | Embedding | HVG | 0.73099 | 0.79000 | 0.67197 | 0.96418 | 0.83718 | 0.36011 | 0.99853 | 0.67987 | 0.46834 | 0.54458 | N/A | N/A | 0.99510 | N/A |
| n_hidden: 128, n_latent: 20, n_layers: 3 | Embedding | Full | 0.76463 | 0.80267 | 0.72658 | 0.95060 | 0.89509 | 0.36548 | 0.99953 | 0.73494 | 0.62635 | 0.54869 | N/A | N/A | 0.99636 | N/A |
| n_hidden: 128, n_latent: 20, n_layers: 3 | Embedding | HVG | 0.74421 | 0.77823 | 0.71018 | 0.95221 | 0.80400 | 0.35862 | 0.99810 | 0.71431 | 0.57852 | 0.55226 | N/A | N/A | 0.99563 | N/A |
| n_hidden: 128, n_latent: 30, n_layers: 1 | Embedding | Full | 0.75697 | 0.80376 | 0.71018 | 0.95588 | 0.89788 | 0.36265 | 0.99861 | 0.71446 | 0.59704 | 0.53556 | N/A | N/A | 0.99452 | N/A |
| n_hidden: 128, n_latent: 30, n_layers: 1 | Embedding | HVG | 0.75391 | 0.80835 | 0.69947 | 0.95904 | 0.90942 | 0.36562 | 0.99930 | 0.70537 | 0.56287 | 0.53611 | N/A | N/A | 0.99355 | N/A |
| n_hidden: 128, n_latent: 30, n_layers: 2 | Embedding | Full | 0.75817 | 0.80245 | 0.71389 | 0.95011 | 0.89697 | 0.36330 | 0.99942 | 0.71634 | 0.59704 | 0.54592 | N/A | N/A | 0.99626 | N/A |
| n_hidden: 128, n_latent: 30, n_layers: 2 | Embedding | HVG | 0.75281 | 0.79241 | 0.71320 | 0.96590 | 0.84148 | 0.36334 | 0.99892 | 0.72315 | 0.59310 | 0.54238 | N/A | N/A | 0.99417 | N/A |
| n_hidden: 128, n_latent: 30, n_layers: 3 | Embedding | Full | 0.75163 | 0.79771 | 0.70555 | 0.94822 | 0.88044 | 0.36258 | 0.99959 | 0.70648 | 0.57037 | 0.54802 | N/A | N/A | 0.99732 | N/A |
| n_hidden: 128, n_latent: 30, n_layers: 3 | Embedding | HVG | 0.72833 | 0.77842 | 0.67825 | 0.95184 | 0.80137 | 0.36213 | 0.99834 | 0.69273 | 0.47436 | 0.55008 | N/A | N/A | 0.99582 | N/A |
| n_hidden: 128, n_latent: 40, n_layers: 1 | Embedding | Full | 0.75400 | 0.81121 | 0.69679 | 0.96324 | 0.92027 | 0.36290 | 0.99843 | 0.69793 | 0.56360 | 0.53356 | N/A | N/A | 0.99207 | N/A |
| n_hidden: 128, n_latent: 40, n_layers: 1 | Embedding | HVG | 0.74457 | 0.80000 | 0.68913 | 0.97286 | 0.86810 | 0.36010 | 0.99895 | 0.68666 | 0.55207 | 0.52825 | N/A | N/A | 0.98955 | N/A |
| n_hidden: 128, n_latent: 40, n_layers: 2 | Embedding | Full | 0.75102 | 0.80034 | 0.70169 | 0.94753 | 0.89093 | 0.36336 | 0.99954 | 0.70841 | 0.56100 | 0.54319 | N/A | N/A | 0.99416 | N/A |
| n_hidden: 128, n_latent: 40, n_layers: 2 | Embedding | HVG | 0.74592 | 0.79337 | 0.69847 | 0.96375 | 0.84618 | 0.36503 | 0.99850 | 0.69539 | 0.56119 | 0.54279 | N/A | N/A | 0.99450 | N/A |
| n_hidden: 128, n_latent: 40, n_layers: 3 | Embedding | Full | 0.76433 | 0.80190 | 0.72677 | 0.95385 | 0.88818 | 0.36594 | 0.99962 | 0.74668 | 0.61357 | 0.55009 | N/A | N/A | 0.99672 | N/A |
| n_hidden: 128, n_latent: 40, n_layers: 3 | Embedding | HVG | 0.73752 | 0.77117 | 0.70386 | 0.94733 | 0.78102 | 0.35824 | 0.99809 | 0.71969 | 0.54312 | 0.55654 | N/A | N/A | 0.99611 | N/A |
| n_hidden: 128, n_latent: 50, n_layers: 1 | Embedding | Full | 0.73829 | 0.81062 | 0.66597 | 0.96529 | 0.91620 | 0.36140 | 0.99958 | 0.65365 | 0.49030 | 0.52979 | N/A | N/A | 0.99015 | N/A |
| n_hidden: 128, n_latent: 50, n_layers: 1 | Embedding | HVG | 0.75599 | 0.80459 | 0.70738 | 0.96506 | 0.87891 | 0.36506 | 0.99855 | 0.72821 | 0.59033 | 0.52444 | N/A | N/A | 0.98654 | N/A |
| n_hidden: 128, n_latent: 50, n_layers: 2 | Embedding | Full | 0.76067 | 0.80500 | 0.71634 | 0.95229 | 0.90492 | 0.36330 | 0.99952 | 0.72066 | 0.60189 | 0.54710 | N/A | N/A | 0.99569 | N/A |
| n_hidden: 128, n_latent: 50, n_layers: 2 | Embedding | HVG | 0.74897 | 0.79320 | 0.70473 | 0.96617 | 0.84502 | 0.36276 | 0.99886 | 0.72354 | 0.56011 | 0.54192 | N/A | N/A | 0.99337 | N/A |
| n_hidden: 128, n_latent: 50, n_layers: 3 | Embedding | Full | 0.75520 | 0.80057 | 0.70983 | 0.94705 | 0.88916 | 0.36674 | 0.99935 | 0.70826 | 0.58533 | 0.54892 | N/A | N/A | 0.99680 | N/A |
| n_hidden: 128, n_latent: 50, n_layers: 3 | Embedding | HVG | 0.72946 | 0.77679 | 0.68212 | 0.95020 | 0.79778 | 0.36067 | 0.99950 | 0.67569 | 0.50605 | 0.55025 | N/A | N/A | 0.99649 | N/A |
| n_hidden: 256, n_latent: 10, n_layers: 1 | Embedding | Full | 0.73380 | 0.78465 | 0.68295 | 0.94463 | 0.83537 | 0.35935 | 0.99925 | 0.68725 | 0.47386 | 0.57157 | N/A | N/A | **0.99914** | N/A |
| n_hidden: 256, n_latent: 10, n_layers: 1 | Embedding | HVG | 0.73574 | 0.79156 | 0.67992 | 0.94852 | 0.85987 | 0.35928 | 0.99857 | 0.68321 | 0.46419 | 0.57382 | N/A | N/A | 0.99848 | N/A |
| n_hidden: 256, n_latent: 10, n_layers: 2 | Embedding | Full | 0.72833 | 0.78952 | 0.66714 | 0.93917 | 0.85669 | 0.36278 | 0.99944 | 0.65511 | 0.45673 | 0.55990 | N/A | N/A | 0.99682 | N/A |
| n_hidden: 256, n_latent: 10, n_layers: 2 | Embedding | HVG | 0.73961 | 0.77663 | 0.70259 | 0.95079 | 0.79482 | 0.36288 | 0.99802 | 0.69292 | 0.55473 | 0.56531 | N/A | N/A | 0.99739 | N/A |
| n_hidden: 256, n_latent: 10, n_layers: 3 | Embedding | Full | 0.74645 | 0.79936 | 0.69353 | 0.94676 | 0.88296 | 0.36822 | 0.99950 | 0.68980 | 0.53891 | 0.55064 | N/A | N/A | 0.99477 | N/A |
| n_hidden: 256, n_latent: 10, n_layers: 3 | Embedding | HVG | 0.72476 | 0.77882 | 0.67069 | 0.94712 | 0.80469 | 0.36494 | 0.99855 | 0.67001 | 0.46716 | 0.54984 | N/A | N/A | 0.99576 | N/A |
| n_hidden: 256, n_latent: 20, n_layers: 1 | Embedding | Full | 0.75578 | 0.79603 | 0.71553 | 0.95356 | 0.87299 | 0.35800 | 0.99957 | 0.71781 | 0.59845 | 0.54942 | N/A | N/A | 0.99642 | N/A |
| n_hidden: 256, n_latent: 20, n_layers: 1 | Embedding | HVG | 0.73043 | 0.77968 | 0.68117 | 0.96378 | 0.79843 | 0.35781 | 0.99872 | 0.69075 | 0.49358 | 0.54461 | N/A | N/A | 0.99574 | N/A |
| n_hidden: 256, n_latent: 20, n_layers: 2 | Embedding | Full | 0.73627 | 0.80415 | 0.66840 | 0.94979 | 0.90012 | 0.36719 | 0.99950 | 0.67267 | 0.46441 | 0.54290 | N/A | N/A | 0.99362 | N/A |
| n_hidden: 256, n_latent: 20, n_layers: 2 | Embedding | HVG | 0.75693 | 0.79451 | 0.71934 | 0.97087 | 0.84519 | 0.36335 | 0.99863 | 0.72407 | 0.62336 | 0.53755 | N/A | N/A | 0.99238 | N/A |
| n_hidden: 256, n_latent: 20, n_layers: 3 | Embedding | Full | 0.75147 | 0.80113 | 0.70180 | 0.95474 | 0.88167 | 0.36843 | 0.99967 | 0.70791 | 0.56761 | 0.53826 | N/A | N/A | 0.99343 | N/A |
| n_hidden: 256, n_latent: 20, n_layers: 3 | Embedding | HVG | 0.73483 | 0.78791 | 0.68174 | 0.96678 | 0.82378 | 0.36202 | 0.99907 | 0.69114 | 0.51043 | 0.53415 | N/A | N/A | 0.99123 | N/A |
| n_hidden: 256, n_latent: 30, n_layers: 1 | Embedding | Full | 0.73591 | 0.80199 | 0.66984 | 0.95662 | 0.88562 | 0.36647 | 0.99923 | 0.66984 | 0.47134 | 0.53330 | N/A | N/A | 0.99211 | N/A |
| n_hidden: 256, n_latent: 30, n_layers: 1 | Embedding | HVG | **0.78615** | 0.79398 | **0.77832** | 0.97341 | 0.84452 | 0.35920 | 0.99881 | **0.99881** | 0.59039 | 0.53232 | N/A | N/A | 0.99177 | N/A |
| n_hidden: 256, n_latent: 30, n_layers: 2 | Embedding | Full | 0.73206 | 0.81125 | 0.65287 | 0.96490 | 0.91204 | **0.37281** | 0.99916 | 0.64486 | 0.44339 | 0.53457 | N/A | N/A | 0.98866 | N/A |
| n_hidden: 256, n_latent: 30, n_layers: 2 | Embedding | HVG | 0.74810 | 0.80222 | 0.69398 | 0.97551 | 0.86988 | 0.36480 | 0.99871 | 0.71758 | 0.53823 | 0.52941 | N/A | N/A | 0.99069 | N/A |
| n_hidden: 256, n_latent: 30, n_layers: 3 | Embedding | Full | 0.74015 | 0.80267 | 0.67763 | 0.95935 | 0.88780 | 0.36461 | 0.99893 | 0.69060 | 0.48713 | 0.53933 | N/A | N/A | 0.99347 | N/A |
| n_hidden: 256, n_latent: 30, n_layers: 3 | Embedding | HVG | 0.75578 | 0.79367 | 0.71788 | 0.96751 | 0.84201 | 0.36609 | 0.99907 | 0.73627 | 0.60606 | 0.53596 | N/A | N/A | 0.99305 | N/A |
| n_hidden: 256, n_latent: 40, n_layers: 1 | Embedding | Full | 0.75422 | 0.80927 | 0.69916 | 0.96387 | 0.90714 | 0.36694 | 0.99914 | 0.71054 | 0.56767 | 0.52836 | N/A | N/A | 0.99007 | N/A |
| n_hidden: 256, n_latent: 40, n_layers: 1 | Embedding | HVG | 0.69409 | 0.80166 | 0.58653 | 0.97486 | 0.87259 | 0.35996 | 0.99923 | 0.72256 | 0.57646 | 0.52355 | N/A | N/A | 0.52355 | N/A |
| n_hidden: 256, n_latent: 40, n_layers: 2 | Embedding | Full | 0.75557 | 0.80907 | 0.70207 | 0.96346 | 0.90539 | 0.36781 | 0.99962 | 0.71322 | 0.56861 | 0.53444 | N/A | N/A | 0.99201 | N/A |
| n_hidden: 256, n_latent: 40, n_layers: 2 | Embedding | HVG | 0.76875 | 0.80719 | 0.73031 | 0.97697 | 0.88476 | 0.36832 | 0.99871 | 0.75554 | 0.65302 | 0.52601 | N/A | N/A | 0.98667 | N/A |
| n_hidden: 256, n_latent: 40, n_layers: 3 | Embedding | Full | 0.74767 | 0.80476 | 0.69058 | 0.95807 | 0.89233 | 0.36896 | 0.99967 | 0.71877 | 0.51386 | 0.53681 | N/A | N/A | 0.99286 | N/A |
| n_hidden: 256, n_latent: 40, n_layers: 3 | Embedding | HVG | 0.76265 | 0.78816 | 0.73713 | 0.96879 | 0.82256 | 0.36224 | 0.99907 | 0.74440 | **0.67602** | 0.53640 | N/A | N/A | 0.99170 | N/A |
| n_hidden: 256, n_latent: 50, n_layers: 1 | Embedding | Full | 0.74719 | **0.81558** | 0.67881 | 0.97070 | **0.92144** | 0.37150 | 0.99866 | 0.67643 | 0.53109 | 0.52449 | N/A | N/A | 0.98322 | N/A |
| n_hidden: 256, n_latent: 50, n_layers: 1 | Embedding | HVG | 0.75087 | 0.80678 | 0.69496 | **0.97824** | 0.88862 | 0.36167 | 0.99860 | 0.71970 | 0.56266 | 0.51912 | N/A | N/A | 0.97835 | N/A |
| n_hidden: 256, n_latent: 50, n_layers: 2 | Embedding | Full | 0.75673 | 0.81165 | 0.70180 | 0.96136 | 0.91403 | 0.37160 | 0.99961 | 0.71812 | 0.56323 | 0.53517 | N/A | N/A | 0.99069 | N/A |
| n_hidden: 256, n_latent: 50, n_layers: 2 | Embedding | HVG | 0.74350 | 0.80860 | 0.67841 | 0.97814 | 0.88740 | 0.36990 | 0.99911 | 0.66990 | 0.53513 | 0.52511 | N/A | N/A | 0.98348 | N/A |
| n_hidden: 256, n_latent: 50, n_layers: 3 | Embedding | Full | 0.74081 | 0.80604 | 0.67559 | 0.95842 | 0.89584 | 0.37048 | 0.99942 | 0.68939 | 0.47890 | 0.53921 | N/A | N/A | 0.99484 | N/A |
| n_hidden: 256, n_latent: 50, n_layers: 3 | Embedding | HVG | 0.72107 | 0.78626 | 0.65587 | 0.96577 | 0.81659 | 0.36387 | 0.99882 | 0.67190 | 0.42336 | 0.53537 | N/A | N/A | 0.99286 | N/A |

**Table 8:** *Performance of scVI on the Zenodo 11100300 (18 PBMC Samples)*

Table 8 reports the scVI sweep on Zenodo 11100300 over $n_{\text{hidden}} \in \{128, 256\}$, $n_{\text{latent}} \in \{10, 20, 30, 40, 50\}$, $n_{\text{layers}} \in \{1, 2, 3\}$, and Features $\in \{\text{Full}, \text{HVG}\}$. The best Overall is $0.78615$ at $(256, 30, 1, \text{HVG})$, while the worst Overall is $0.69409$ at $(256, 40, 1, \text{HVG})$. The trajectory, isolated label F1, and isolated label ASW are not applicable for this dataset and therefore reported as N/A.

Changing the hidden size from 128 to 256 consistently benefits batch correction but tends to reduce biology and the combined overall. In matched pairs that fix $(n_{\text{latent}}, n_{\text{layers}}, \text{Features})$, Overall improves in $10/30$ and declines in $20/30$ pairs, Overall(BC) improves in $20/30$ and declines in $10/30$, while Overall(Bio) improves in $10/30$ and declines in $20/30$. This indicates that larger hidden size strengthens batch mixing on this dataset but generally softens biology and the combined Overall.

Depth primarily trades batch for biology with little net effect on the combined overall. Comparing $n_{\text{layers}}=3$ against 1 at fixed $(n_{\text{hidden}}, n_{\text{latent}}, \text{Features})$, Overall rises in $9/20$ and falls in $11/20$ pairs (effectively flat). The batch-oriented Overall(BC) rises in $6/20$ and falls in $14/20$, whereas the biology-oriented Overall(Bio) rises in $10/20$ and falls in $10/20$.

Latent dimensionality shows the clearest positive signal for batch removal and a modest benefit for the combined overall. Aggregating stepwise comparisons $10 \to 20 \to 30 \to 40 \to 50$ at fixed $(n_{\text{hidden}}, n_{\text{layers}}, \text{Features})$, Overall increases in $27/48$ steps, Overall(BC) in $35/48$, and Overall(Bio) in $26/48$. The endpoint contrast $50-10$ reinforces this: Overall improves in $8/12$ matched triplets, Overall(BC) in $11/12$, and Overall(Bio) in $6/12$. Increasing $n_{\text{latent}}$ therefore clearly helps batch correction and typically nudges the primary Overall upward; the impact on biology is mixed-to-slightly-positive on average.

Feature selection favors Full on this dataset, especially for batch. In paired HVG−Full comparisons at fixed $(n_\mathrm{hidden}, n_\mathrm{latent}, n_\mathrm{layers})$, Overall is higher with HVG in $11/30$ and lower in $19/30$ pairs, Overall(BC) is higher in $2/30$ and lower in $28/30$, and Overall(Bio) is higher in $10/30$ and lower in $20/30$. Thus, Full generally outperforms HVG for the composite scores on Zenodo 11100300, particularly for batch-oriented performance. However, the single best Overall configuration in this grid is HVG: $0.78615$ at $(256, 30, 1, \mathrm{HVG})$. In practice, Full is the stronger default across matched settings, but careful tuning (e.g., shallow depth with $n_\mathrm{latent} \approx 30$) can yield an HVG peak.

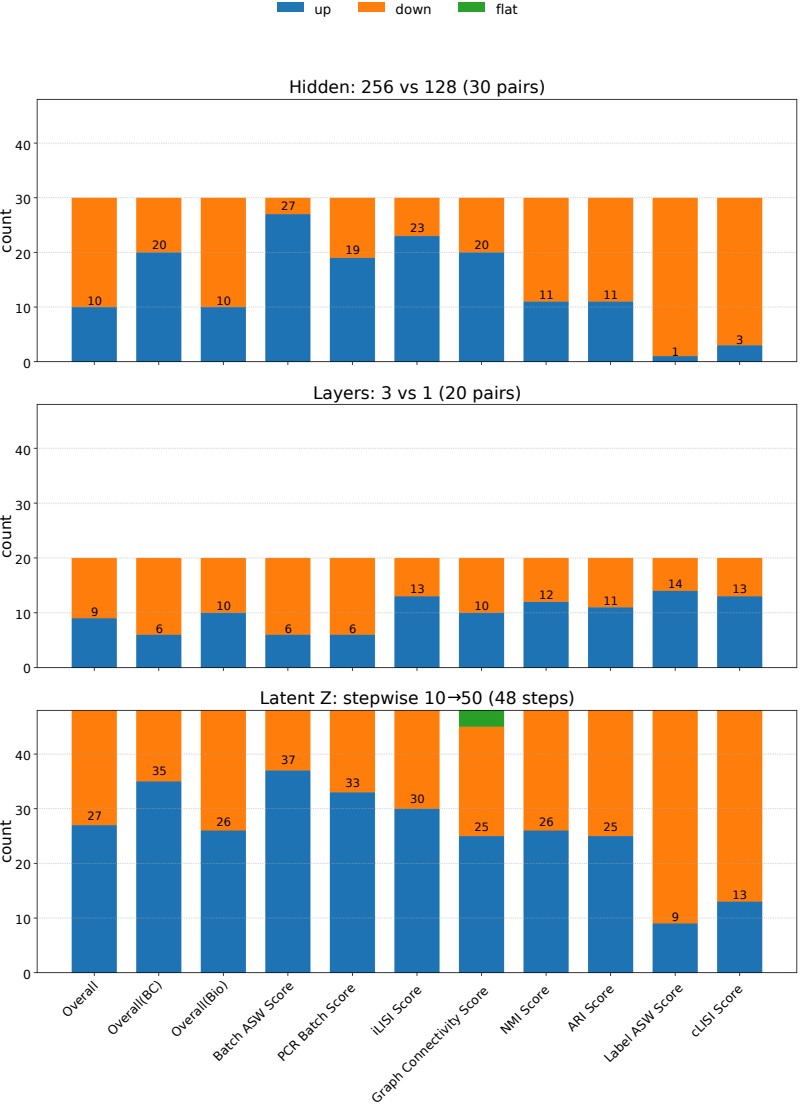

**Figure 22:** *Paired comparison of scVI hyperparameters on the Zenodo 11100300 (18 PBMC Samples) dataset, showing the number of metrics that improve ("up"), decline ("down"), or remain unchanged ("flat") when varying hidden units (256 vs. 128), network depth (3 vs. 1 layers), latent dimension z (stepwise 10→50) on both full and HVG feature sets.*

| Hyperparameters | Output | Features | Overall | Overall(BC) | Overall(Bio) | Batch correction | | | | Bio Conservation | | | | | | |
| --- | --- | --- | --- | --- | --- | --- | --- | --- | --- | --- | --- | --- | --- | --- | --- | --- |
| | | | Overall | Overall(BC) | Overall(Bio) | ASW | PCR | iLISI | GC | NMI | ARI | ASW | IL F1 | IL ASW | cLISI | TC |
| n_hidden: 128, n_latent: 10, n_layers: 1, n_latent_u: 10 | Embedding | Full | 0.72206 | 0.78466 | 0.65946 | 0.85681 | 0.91451 | 0.37134 | 0.99600 | 0.63585 | 0.36409 | 0.63863 | N/A | N/A | 0.99927 | N/A |
| n_hidden: 128, n_latent: 10, n_layers: 1, n_latent_u: 10 | Embedding | HVG | 0.71933 | 0.78719 | 0.65146 | 0.86209 | 0.92137 | 0.36917 | 0.99612 | 0.63059 | 0.33772 | 0.63823 | N/A | N/A | 0.99931 | N/A |
| n_hidden: 128, n_latent: 10, n_layers: 2, n_latent_u: 10 | Embedding | Full | 0.72062 | 0.78490 | 0.65634 | 0.86290 | 0.90837 | 0.37185 | 0.99648 | 0.63102 | 0.36052 | 0.63459 | N/A | N/A | 0.99925 | N/A |
| n_hidden: 128, n_latent: 10, n_layers: 2, n_latent_u: 10 | Embedding | HVG | 0.72032 | 0.78497 | 0.65568 | 0.86205 | 0.90850 | 0.37269 | 0.99663 | 0.63548 | 0.35389 | 0.63396 | N/A | N/A | 0.99940 | N/A |
| n_hidden: 128, n_latent: 10, n_layers: 3, n_latent_u: 10 | Embedding | Full | 0.72680 | 0.78696 | 0.66664 | 0.85763 | 0.92134 | 0.37267 | 0.99620 | 0.65308 | 0.37522 | 0.63881 | N/A | N/A | 0.99944 | N/A |
| n_hidden: 128, n_latent: 10, n_layers: 3, n_latent_u: 10 | Embedding | HVG | 0.72176 | 0.78976 | 0.65376 | 0.86159 | 0.92669 | 0.37430 | 0.99648 | 0.62965 | 0.34765 | 0.63852 | N/A | N/A | 0.99922 | N/A |
| n_hidden: 128, n_latent: 20, n_layers: 1, n_latent_u: 10 | Embedding | Full | 0.73283 | 0.79277 | 0.67289 | 0.90797 | 0.89916 | 0.36614 | 0.99781 | 0.66933 | 0.39960 | 0.62332 | N/A | N/A | 0.99930 | N/A |
| n_hidden: 128, n_latent: 20, n_layers: 1, n_latent_u: 10 | Embedding | HVG | 0.72897 | 0.79046 | 0.66748 | 0.90340 | 0.89593 | 0.36489 | 0.99761 | 0.66411 | 0.39194 | 0.61495 | N/A | N/A | 0.99891 | N/A |
| n_hidden: 128, n_latent: 20, n_layers: 2, n_latent_u: 10 | Embedding | Full | 0.72669 | 0.78754 | 0.66583 | 0.89963 | 0.88899 | 0.36388 | 0.99767 | 0.63403 | 0.41812 | 0.61247 | N/A | N/A | 0.99871 | N/A |
| n_hidden: 128, n_latent: 20, n_layers: 2, n_latent_u: 10 | Embedding | HVG | 0.72868 | 0.79094 | 0.66641 | 0.89737 | 0.90239 | 0.36634 | 0.99766 | 0.65707 | 0.39657 | 0.61272 | N/A | N/A | 0.99929 | N/A |
| n_hidden: 128, n_latent: 20, n_layers: 3, n_latent_u: 10 | Embedding | Full | 0.73540 | 0.79384 | 0.67697 | 0.88808 | 0.92371 | 0.36643 | 0.99712 | 0.67273 | 0.41822 | 0.61789 | N/A | N/A | 0.99905 | N/A |
| n_hidden: 128, n_latent: 20, n_layers: 3, n_latent_u: 10 | Embedding | HVG | 0.72664 | 0.78971 | 0.66358 | 0.89219 | 0.90128 | 0.36796 | 0.99742 | 0.65050 | 0.39256 | 0.61202 | N/A | N/A | 0.99921 | N/A |
| n_hidden: 128, n_latent: 30, n_layers: 1, n_latent_u: 10 | Embedding | Full | 0.74372 | 0.80148 | 0.68596 | 0.92463 | 0.92149 | 0.36355 | 0.99625 | 0.67899 | 0.43705 | 0.62841 | N/A | N/A | 0.99939 | N/A |
| n_hidden: 128, n_latent: 30, n_layers: 1, n_latent_u: 10 | Embedding | HVG | 0.71877 | 0.80107 | 0.63646 | 0.92244 | 0.91901 | 0.36471 | 0.99813 | 0.61191 | 0.32700 | 0.60987 | N/A | N/A | 0.99705 | N/A |
| n_hidden: 128, n_latent: 30, n_layers: 2, n_latent_u: 10 | Embedding | Full | 0.73606 | 0.79442 | 0.67769 | 0.91073 | 0.90245 | 0.36662 | 0.99788 | 0.67692 | 0.42629 | 0.60873 | N/A | N/A | 0.99883 | N/A |
| n_hidden: 128, n_latent: 30, n_layers: 2, n_latent_u: 10 | Embedding | HVG | 0.71389 | 0.78651 | 0.64127 | 0.89762 | 0.88815 | 0.36276 | 0.99752 | 0.61729 | 0.34274 | 0.60848 | N/A | N/A | 0.99659 | N/A |
| n_hidden: 128, n_latent: 30, n_layers: 3, n_latent_u: 10 | Embedding | Full | 0.75098 | 0.80219 | 0.69977 | 0.90350 | 0.94140 | 0.36657 | 0.99729 | 0.68668 | 0.49009 | 0.62387 | N/A | N/A | 0.99845 | N/A |
| n_hidden: 128, n_latent: 30, n_layers: 3, n_latent_u: 10 | Embedding | HVG | 0.72950 | 0.79728 | 0.66173 | 0.90578 | 0.91541 | 0.37061 | 0.99733 | 0.64701 | 0.38737 | 0.61439 | N/A | N/A | 0.99814 | N/A |
| n_hidden: 128, n_latent: 40, n_layers: 1, n_latent_u: 10 | Embedding | Full | 0.72395 | 0.79302 | 0.65489 | 0.92370 | 0.88932 | 0.36111 | 0.99794 | 0.62858 | 0.38956 | 0.60411 | N/A | N/A | 0.99731 | N/A |
| n_hidden: 128, n_latent: 40, n_layers: 1, n_latent_u: 10 | Embedding | HVG | 0.73113 | 0.79279 | 0.66947 | 0.91707 | 0.89075 | 0.36502 | 0.99831 | 0.64430 | 0.43850 | 0.59819 | N/A | N/A | 0.99687 | N/A |
| n_hidden: 128, n_latent: 40, n_layers: 2, n_latent_u: 10 | Embedding | Full | 0.73472 | 0.79926 | 0.67018 | 0.92368 | 0.91050 | 0.36555 | 0.99730 | 0.65273 | 0.41843 | 0.61124 | N/A | N/A | 0.99831 | N/A |
| n_hidden: 128, n_latent: 40, n_layers: 2, n_latent_u: 10 | Embedding | HVG | 0.74035 | 0.80248 | 0.67823 | 0.92141 | 0.92549 | 0.36493 | 0.99809 | 0.66445 | 0.42404 | 0.62621 | N/A | N/A | 0.99821 | N/A |
| n_hidden: 128, n_latent: 40, n_layers: 3, n_latent_u: 10 | Embedding | Full | 0.73560 | 0.79908 | 0.67213 | 0.90325 | 0.92703 | 0.36850 | 0.99752 | 0.66677 | 0.40765 | 0.61487 | N/A | N/A | 0.99922 | N/A |
| n_hidden: 128, n_latent: 40, n_layers: 3, n_latent_u: 10 | Embedding | HVG | 0.74078 | 0.80125 | 0.68032 | 0.90975 | 0.92663 | 0.37156 | 0.99704 | 0.67766 | 0.43076 | 0.61365 | N/A | N/A | 0.99922 | N/A |
| n_hidden: 128, n_latent: 50, n_layers: 1, n_latent_u: 10 | Embedding | Full | 0.74528 | 0.80529 | 0.68528 | 0.92550 | 0.93172 | 0.36640 | 0.99755 | 0.68174 | 0.43863 | 0.62144 | N/A | N/A | 0.99930 | N/A |
| n_hidden: 128, n_latent: 50, n_layers: 1, n_latent_u: 10 | Embedding | HVG | 0.74732 | 0.79971 | 0.69493 | 0.92149 | 0.91355 | 0.36578 | 0.99803 | 0.69756 | 0.47126 | 0.61263 | N/A | N/A | 0.99827 | N/A |
| n_hidden: 128, n_latent: 50, n_layers: 2, n_latent_u: 10 | Embedding | Full | 0.73173 | 0.79916 | 0.66429 | 0.92175 | 0.91528 | 0.36160 | 0.99803 | 0.65881 | 0.38401 | 0.61609 | N/A | N/A | 0.99826 | N/A |
| n_hidden: 128, n_latent: 50, n_layers: 2, n_latent_u: 10 | Embedding | HVG | 0.73662 | 0.80290 | 0.67034 | 0.91964 | 0.92953 | 0.36423 | 0.99819 | 0.65646 | 0.41452 | 0.61285 | N/A | N/A | 0.99751 | N/A |
| n_hidden: 128, n_latent: 50, n_layers: 3, n_latent_u: 10 | Embedding | Full | 0.72559 | 0.78850 | 0.66268 | 0.89428 | 0.89627 | 0.36640 | 0.99706 | 0.64475 | 0.40431 | 0.60303 | N/A | N/A | 0.99863 | N/A |
| n_hidden: 128, n_latent: 50, n_layers: 3, n_latent_u: 10 | Embedding | HVG | 0.73587 | 0.79278 | 0.67897 | 0.90159 | 0.90333 | 0.36823 | 0.99795 | 0.65093 | 0.44474 | 0.62191 | N/A | N/A | 0.99831 | N/A |
| n_hidden: 256, n_latent: 10, n_layers: 1, n_latent_u: 10 | Embedding | Full | 0.72701 | 0.79112 | 0.66290 | 0.86494 | 0.93190 | 0.37231 | 0.99533 | 0.64149 | 0.36809 | 0.64249 | N/A | N/A | 0.99955 | N/A |
| n_hidden: 256, n_latent: 10, n_layers: 1, n_latent_u: 10 | Embedding | HVG | 0.73094 | 0.79262 | 0.66925 | 0.87356 | 0.92723 | 0.37331 | 0.99638 | 0.65257 | 0.38544 | 0.63972 | N/A | N/A | 0.99928 | N/A |
| n_hidden: 256, n_latent: 10, n_layers: 2, n_latent_u: 10 | Embedding | Full | 0.71492 | 0.78855 | 0.64129 | 0.86107 | 0.92013 | 0.37664 | 0.99633 | 0.60333 | 0.33028 | 0.63256 | N/A | N/A | 0.99898 | N/A |
| n_hidden: 256, n_latent: 10, n_layers: 2, n_latent_u: 10 | Embedding | HVG | 0.71951 | 0.78053 | 0.65849 | 0.84607 | 0.90385 | 0.37594 | 0.99627 | 0.64233 | 0.36007 | 0.63218 | N/A | N/A | 0.99939 | N/A |
| n_hidden: 256, n_latent: 10, n_layers: 3, n_latent_u: 10 | Embedding | Full | 0.73138 | 0.79591 | 0.66686 | 0.86875 | 0.94433 | 0.37521 | 0.99535 | 0.65297 | 0.36789 | 0.64723 | N/A | N/A | 0.99935 | N/A |
| n_hidden: 256, n_latent: 10, n_layers: 3, n_latent_u: 10 | Embedding | HVG | 0.71654 | 0.78509 | 0.64798 | 0.86711 | 0.93839 | 0.37169 | 0.96318 | 0.62390 | 0.32862 | 0.63999 | N/A | N/A | 0.99942 | N/A |
| n_hidden: 256, n_latent: 20, n_layers: 1, n_latent_u: 10 | Embedding | Full | 0.73004 | 0.78664 | 0.67345 | 0.89887 | 0.88281 | 0.36726 | 0.99760 | 0.67384 | 0.40879 | 0.61218 | N/A | N/A | 0.99900 | N/A |
| n_hidden: 256, n_latent: 20, n_layers: 1, n_latent_u: 10 | Embedding | HVG | 0.74265 | 0.79610 | 0.68920 | 0.91248 | 0.90530 | 0.36877 | 0.99787 | 0.68934 | 0.44756 | 0.62091 | N/A | N/A | 0.99900 | N/A |
| n_hidden: 256, n_latent: 20, n_layers: 2, n_latent_u: 10 | Embedding | Full | 0.72896 | 0.78404 | 0.67388 | 0.87884 | 0.88811 | 0.37253 | 0.99668 | 0.66501 | 0.42382 | 0.60776 | N/A | N/A | 0.99895 | N/A |
| n_hidden: 256, n_latent: 20, n_layers: 2, n_latent_u: 10 | Embedding | HVG | 0.71766 | 0.78231 | 0.65300 | 0.88393 | 0.88082 | 0.36724 | 0.99725 | 0.64418 | 0.36741 | 0.60173 | N/A | N/A | 0.99870 | N/A |
| n_hidden: 256, n_latent: 20, n_layers: 3, n_latent_u: 10 | Embedding | Full | 0.70797 | 0.77866 | 0.63728 | 0.86900 | 0.90920 | 0.37524 | 0.96117 | 0.60641 | 0.32371 | 0.61975 | N/A | N/A | 0.99924 | N/A |
| n_hidden: 256, n_latent: 20, n_layers: 3, n_latent_u: 10 | Embedding | HVG | 0.72467 | 0.79756 | 0.65178 | 0.88853 | 0.93225 | 0.37579 | 0.99367 | 0.63037 | 0.35406 | 0.62412 | N/A | N/A | 0.99856 | N/A |
| n_hidden: 256, n_latent: 30, n_layers: 1, n_latent_u: 10 | Embedding | Full | 0.71804 | 0.79783 | 0.63826 | 0.91841 | 0.90738 | 0.36753 | 0.99802 | 0.61881 | 0.33626 | 0.60044 | N/A | N/A | 0.99752 | N/A |
| n_hidden: 256, n_latent: 30, n_layers: 1, n_latent_u: 10 | Embedding | HVG | 0.73216 | 0.79717 | 0.66715 | 0.92173 | 0.90478 | 0.36398 | 0.99819 | 0.66216 | 0.40066 | 0.60726 | N/A | N/A | 0.99853 | N/A |
| n_hidden: 256, n_latent: 30, n_layers: 2, n_latent_u: 10 | Embedding | Full | 0.71715 | 0.80315 | 0.63115 | 0.90927 | 0.93335 | 0.37234 | 0.99765 | 0.61023 | 0.31581 | 0.60191 | N/A | N/A | 0.99663 | N/A |
| n_hidden: 256, n_latent: 30, n_layers: 2, n_latent_u: 10 | Embedding | HVG | 0.72175 | 0.79462 | 0.64888 | 0.89439 | 0.91561 | 0.37160 | 0.99689 | 0.61133 | 0.36920 | 0.61679 | N/A | N/A | 0.99821 | N/A |
| n_hidden: 256, n_latent: 30, n_layers: 3, n_latent_u: 10 | Embedding | Full | 0.72045 | 0.79117 | 0.64973 | 0.89203 | 0.93652 | 0.37547 | 0.96064 | 0.63334 | 0.33662 | 0.62950 | N/A | N/A | 0.99947 | N/A |
| n_hidden: 256, n_latent: 30, n_layers: 3, n_latent_u: 10 | Embedding | HVG | 0.72010 | 0.79042 | 0.64979 | 0.88677 | 0.93863 | 0.37528 | 0.96099 | 0.62818 | 0.32156 | 0.64973 | N/A | N/A | **0.99967** | N/A |
| n_hidden: 256, n_latent: 40, n_layers: 1, n_latent_u: 10 | Embedding | Full | 0.75153 | 0.80173 | 0.70133 | 0.92991 | 0.91192 | 0.36706 | 0.99805 | 0.70095 | 0.49290 | 0.61273 | N/A | N/A | 0.99873 | N/A |
| n_hidden: 256, n_latent: 40, n_layers: 1, n_latent_u: 10 | Embedding | HVG | 0.74616 | 0.79745 | 0.69486 | 0.92164 | 0.90403 | 0.36625 | 0.99790 | 0.69490 | 0.47763 | 0.60903 | N/A | N/A | 0.99790 | N/A |
| n_hidden: 256, n_latent: 40, n_layers: 2, n_latent_u: 10 | Embedding | Full | 0.71765 | 0.78794 | 0.64737 | 0.88480 | 0.90158 | 0.36882 | 0.99654 | 0.62749 | 0.35104 | 0.61278 | N/A | N/A | 0.99814 | N/A |
| n_hidden: 256, n_latent: 40, n_layers: 2, n_latent_u: 10 | Embedding | HVG | 0.74237 | 0.79825 | 0.68649 | 0.90988 | 0.91796 | 0.36906 | 0.99609 | 0.67887 | 0.44168 | 0.62644 | N/A | N/A | 0.99898 | N/A |
| n_hidden: 256, n_latent: 40, n_layers: 3, n_latent_u: 10 | Embedding | Full | 0.72572 | 0.80499 | 0.64645 | 0.90436 | 0.94556 | 0.37536 | 0.99467 | 0.62498 | 0.33937 | 0.62217 | N/A | N/A | 0.99928 | N/A |
| n_hidden: 256, n_latent: 40, n_layers: 3, n_latent_u: 10 | Embedding | HVG | 0.72150 | 0.80404 | 0.63896 | 0.89545 | 0.94951 | 0.37586 | 0.99534 | 0.60538 | 0.31576 | 0.63547 | N/A | N/A | 0.99920 | N/A |
| n_hidden: 256, n_latent: 50, n_layers: 1, n_latent_u: 10 | Embedding | Full | 0.73069 | 0.79090 | 0.67047 | 0.90638 | 0.89399 | 0.36514 | 0.99809 | 0.65615 | 0.43029 | 0.59720 | N/A | N/A | 0.99825 | N/A |
| n_hidden: 256, n_latent: 50, n_layers: 1, n_latent_u: 10 | Embedding | HVG | 0.72609 | 0.78964 | 0.66255 | 0.90964 | 0.88483 | 0.36610 | 0.99798 | 0.63485 | 0.41751 | 0.60036 | N/A | N/A | 0.99748 | N/A |
| n_hidden: 256, n_latent: 50, n_layers: 2, n_latent_u: 10 | Embedding | Full | 0.74736 | 0.79605 | 0.69867 | 0.90233 | 0.91392 | 0.37163 | 0.99633 | 0.68396 | 0.49091 | 0.62091 | N/A | N/A | 0.99891 | N/A |
| n_hidden: 256, n_latent: 50, n_layers: 2, n_latent_u: 10 | Embedding | HVG | 0.72176 | 0.79043 | 0.65309 | 0.89493 | 0.89761 | 0.37157 | 0.99760 | 0.61000 | 0.40147 | 0.60374 | N/A | N/A | 0.99716 | N/A |
| n_hidden: 256, n_latent: 50, n_layers: 3, n_latent_u: 10 | Embedding | Full | 0.73704 | 0.81064 | 0.66344 | 0.90459 | **0.96658** | **0.37803** | 0.99338 | 0.65262 | 0.37965 | 0.62222 | N/A | N/A | 0.99926 | N/A |
| n_hidden: 256, n_latent: 50, n_layers: 3, n_latent_u: 10 | Embedding | HVG | 0.72772 | 0.79636 | 0.65909 | 0.89503 | 0.92057 | 0.37421 | 0.99562 | 0.63622 | 0.38203 | 0.61940 | N/A | N/A | 0.99870 | N/A |
| n_hidden: 128, n_latent: 10, n_layers: 1, n_latent_u: 20 | Embedding | Full | 0.73220 | 0.80011 | 0.66430 | 0.88973 | 0.93908 | 0.37471 | 0.99690 | 0.64693 | 0.36523 | 0.64606 | N/A | N/A | 0.99900 | N/A |
| n_hidden: 128, n_latent: 10, n_layers: 1, n_latent_u: 20 | Embedding | HVG | 0.73129 | 0.80348 | 0.65910 | 0.89869 | 0.94533 | 0.37306 | 0.99684 | 0.63952 | 0.35353 | 0.64404 | N/A | N/A | 0.99930 | N/A |
| n_hidden: 128, n_latent: 10, n_layers: 2, n_latent_u: 20 | Embedding | Full | 0.72278 | 0.79106 | 0.65449 | 0.88465 | 0.91349 | 0.36960 | 0.99650 | 0.62737 | 0.37453 | 0.61808 | N/A | N/A | 0.99799 | N/A |
| n_hidden: 128, n_latent: 10, n_layers: 2, n_latent_u: 20 | Embedding | HVG | 0.72398 | 0.79727 | 0.65068 | 0.89146 | 0.93031 | 0.37134 | 0.99596 | 0.62199 | 0.35037 | 0.63207 | N/A | N/A | 0.99830 | N/A |
| n_hidden: 128, n_latent: 10, n_layers: 3, n_latent_u: 20 | Embedding | Full | 0.72866 | 0.79324 | 0.66408 | 0.87920 | 0.92301 | 0.37416 | 0.99660 | 0.64667 | 0.38358 | 0.62748 | N/A | N/A | 0.99862 | N/A |
| n_hidden: 128, n_latent: 10, n_layers: 3, n_latent_u: 20 | Embedding | HVG | 0.72527 | 0.79463 | 0.65591 | 0.87993 | 0.93127 | 0.37086 | 0.99647 | 0.64554 | 0.35070 | 0.62808 | N/A | N/A | 0.99932 | N/A |
| n_hidden: 128, n_latent: 20, n_layers: 1, n_latent_u: 20 | Embedding | Full | 0.72348 | 0.78132 | 0.66565 | 0.89284 | 0.87121 | 0.36345 | 0.99777 | 0.66359 | 0.39090 | 0.60884 | N/A | N/A | 0.99925 | N/A |
| n_hidden: 128, n_latent: 20, n_layers: 1, n_latent_u: 20 | Embedding | HVG | 0.73580 | 0.79420 | 0.67740 | 0.90705 | 0.90820 | 0.36434 | 0.99719 | 0.65953 | 0.43059 | 0.62045 | N/A | N/A | 0.99904 | N/A |
| n_hidden: 128, n_latent: 20, n_layers: 2, n_latent_u: 20 | Embedding | Full | 0.74313 | 0.79300 | 0.69325 | 0.89858 | 0.91233 | 0.36371 | 0.99740 | 0.68963 | 0.45324 | 0.63096 | N/A | N/A | 0.99915 | N/A |
| n_hidden: 128, n_latent: 20, n_layers: 2, n_latent_u: 20 | Embedding | HVG | 0.72247 | 0.78950 | 0.65544 | 0.90137 | 0.89682 | 0.36261 | 0.99722 | 0.64860 | 0.35668 | 0.61775 | N/A | N/A | 0.99871 | N/A |
| n_hidden: 128, n_latent: 20, n_layers: 3, n_latent_u: 20 | Embedding | Full | 0.71909 | 0.78172 | 0.65646 | 0.88238 | 0.88771 | 0.36932 | 0.99748 | 0.62609 | 0.38513 | 0.61605 | N/A | N/A | 0.99858 | N/A |
| n_hidden: 128, n_latent: 20, n_layers: 3, n_latent_u: 20 | Embedding | HVG | 0.72431 | 0.79968 | 0.64893 | 0.89875 | 0.93588 | 0.36702 | 0.99708 | 0.62901 | 0.33883 | 0.62868 | N/A | N/A | 0.99922 | N/A |
| n_hidden: 128, n_latent: 30, n_layers: 1, n_latent_u: 20 | Embedding | Full | 0.73401 | 0.80104 | 0.66699 | 0.92865 | 0.91257 | 0.36458 | **0.99835** | 0.65647 | 0.39590 | 0.61716 | N/A | N/A | 0.99841 | N/A |
| n_hidden: 128, n_latent: 30, n_layers: 1, n_latent_u: 20 | Embedding | HVG | 0.74389 | 0.80519 | 0.68258 | 0.93409 | 0.92570 | 0.36304 | 0.99794 | 0.67112 | 0.43600 | 0.62424 | N/A | N/A | 0.99898 | N/A |
| n_hidden: 128, n_latent: 30, n_layers: 2, n_latent_u: 20 | Embedding | Full | 0.75120 | 0.80448 | 0.69792 | 0.92684 | 0.92819 | 0.36523 | 0.99767 | 0.69367 | 0.47295 | 0.62615 | N/A | N/A | 0.99889 | N/A |
| n_hidden: 128, n_latent: 30, n_layers: 2, n_latent_u: 20 | Embedding | HVG | 0.71992 | 0.80188 | 0.63797 | 0.92282 | 0.92340 | 0.36328 | 0.99800 | 0.61385 | 0.32452 | 0.61474 | N/A | N/A | 0.99879 | N/A |
| n_hidden: 128, n_latent: 30, n_layers: 3, n_latent_u: 20 | Embedding | Full | 0.74362 | 0.80678 | 0.68046 | 0.91828 | 0.94168 | 0.36987 | 0.99730 | 0.67009 | 0.41459 | 0.63767 | N/A | N/A | 0.99951 | N/A |
| n_hidden: 128, n_latent: 30, n_layers: 3, n_latent_u: 20 | Embedding | HVG | 0.75068 | **0.81303** | 0.68833 | 0.92837 | 0.95689 | 0.36907 | 0.99781 | 0.68201 | 0.42827 | 0.64421 | N/A | N/A | 0.99881 | N/A |
| n_hidden: 128, n_latent: 40, n_layers: 1, n_latent_u: 20 | Embedding | Full | 0.74983 | 0.81089 | 0.68878 | **0.94321** | 0.93841 | 0.36408 | 0.99786 | 0.68832 | 0.44342 | 0.62458 | N/A | N/A | 0.99878 | N/A |
| n_hidden: 128, n_latent: 40, n_layers: 1, n_latent_u: 20 | Embedding | HVG | 0.73352 | 0.80302 | 0.66401 | 0.93285 | 0.91634 | 0.36459 | 0.99829 | 0.65021 | 0.39395 | 0.61444 | N/A | N/A | 0.99745 | N/A |
| n_hidden: 128, n_latent: 40, n_layers: 2, n_latent_u: 20 | Embedding | Full | 0.71969 | 0.80400 | 0.63538 | 0.92869 | 0.92363 | 0.36550 | 0.99820 | 0.61858 | 0.31547 | 0.61073 | N/A | N/A | 0.99674 | N/A |
| n_hidden: 128, n_latent: 40, n_layers: 2, n_latent_u: 20 | Embedding | HVG | 0.73400 | 0.80652 | 0.66186 | 0.93251 | 0.92738 | 0.36652 | 0.99815 | 0.65439 | 0.37963 | 0.61512 | N/A | N/A | 0.99830 | N/A |
| n_hidden: 128, n_latent: 40, n_layers: 3, n_latent_u: 20 | Embedding | Full | 0.75341 | 0.81269 | 0.69413 | 0.93255 | 0.94995 | 0.37043 | 0.99782 | 0.69398 | 0.44968 | 0.63387 | N/A | N/A | 0.99899 | N/A |
| n_hidden: 128, n_latent: 40, n_layers: 3, n_latent_u: 20 | Embedding | HVG | 0.73505 | 0.80858 | 0.66153 | 0.92447 | 0.94385 | 0.36841 | 0.99759 | 0.62975 | 0.39424 | 0.62387 | N/A | N/A | 0.99825 | N/A |
| n_hidden: 128, n_latent: 50, n_layers: 1, n_latent_u: 20 | Embedding | Full | 0.75627 | 0.80630 | 0.70624 | 0.93634 | 0.92973 | 0.36260 | 0.99652 | 0.71005 | 0.48818 | 0.62770 | N/A | N/A | 0.99902 | N/A |
| n_hidden: 128, n_latent: 50, n_layers: 1, n_latent_u: 20 | Embedding | HVG | 0.74215 | 0.80100 | 0.68329 | 0.93488 | 0.90920 | 0.36169 | 0.99824 | 0.67316 | 0.45745 | 0.60471 | N/A | N/A | 0.99785 | N/A |
| n_hidden: 128, n_latent: 50, n_layers: 2, n_latent_u: 20 | Embedding | Full | 0.72261 | 0.79402 | 0.65120 | 0.91651 | 0.89719 | 0.36454 | 0.99786 | 0.62440 | 0.38184 | 0.60161 | N/A | N/A | 0.99695 | N/A |
| n_hidden: 128, n_latent: 50, n_layers: 2, n_latent_u: 20 | Embedding | HVG | 0.75280 | 0.80976 | 0.69585 | 0.93421 | 0.94417 | 0.36300 | 0.99767 | 0.70052 | 0.45690 | 0.62690 | N/A | N/A | 0.99906 | N/A |
| n_hidden: 128, n_latent: 50, n_layers: 3, n_latent_u: 20 | Embedding | Full | 0.73094 | 0.80509 | 0.65679 | 0.91623 | 0.93705 | 0.36911 | 0.99798 | 0.63404 | 0.37111 | 0.62465 | N/A | N/A | 0.99738 | N/A |
| n_hidden: 128, n_latent: 50, n_layers: 3, n_latent_u: 20 | Embedding | HVG | 0.72434 | 0.81117 | 0.63751 | 0.93125 | 0.94672 | 0.36951 | 0.99720 | 0.61145 | 0.32354 | 0.61762 | N/A | N/A | 0.99745 | N/A |
| n_hidden: 256, n_latent: 10, n_layers: 1, n_latent_u: 20 | Embedding | Full | 0.72352 | 0.80084 | 0.64621 | 0.89677 | 0.93658 | 0.37355 | 0.99645 | 0.61460 | 0.33954 | 0.63209 | N/A | N/A | 0.99861 | N/A |
| n_hidden: 256, n_latent: 10, n_layers: 1, n_latent_u: 20 | Embedding | HVG | 0.72999 | 0.80263 | 0.65735 | 0.89672 | 0.94714 | 0.36993 | 0.99672 | 0.63249 | 0.34715 | 0.65054 | N/A | N/A | 0.99921 | N/A |
| n_hidden: 256, n_latent: 10, n_layers: 2, n_latent_u: 20 | Embedding | Full | 0.73604 | 0.79864 | 0.67345 | 0.88572 | 0.94263 | 0.37110 | 0.99513 | 0.65458 | 0.40648 | 0.63398 | N/A | N/A | 0.99874 | N/A |
| n_hidden: 256, n_latent: 10, n_layers: 2, n_latent_u: 20 | Embedding | HVG | 0.73187 | 0.79400 | 0.66974 | 0.87909 | 0.92989 | 0.37175 | 0.99526 | 0.64776 | 0.39549 | 0.63709 | N/A | N/A | 0.99861 | N/A |
| n_hidden: 256, n_latent: 10, n_layers: 3, n_latent_u: 20 | Embedding | Full | 0.71022 | 0.79095 | 0.62948 | 0.88472 | 0.94511 | 0.37323 | 0.96076 | 0.58262 | 0.30268 | 0.63403 | N/A | N/A | 0.99858 | N/A |
| n_hidden: 256, n_latent: 10, n_layers: 3, n_latent_u: 20 | Embedding | HVG | 0.72213 | 0.79251 | 0.65176 | 0.88118 | 0.95165 | 0.37421 | 0.96300 | 0.62727 | 0.34414 | 0.63707 | N/A | N/A | 0.99857 | N/A |
| n_hidden: 256, n_latent: 20, n_layers: 1, n_latent_u: 20 | Embedding | Full | 0.73820 | 0.79578 | 0.68063 | 0.91071 | 0.91226 | 0.36423 | 0.99593 | 0.67329 | 0.42339 | 0.62660 | N/A | N/A | 0.99923 | N/A |
| n_hidden: 256, n_latent: 20, n_layers: 1, n_latent_u: 20 | Embedding | HVG | 0.72122 | 0.79362 | 0.64623 | 0.90521 | 0.91539 | 0.36599 | 0.99825 | 0.63092 | 0.33953 | 0.61620 | N/A | N/A | 0.99828 | N/A |
| n_hidden: 256, n_latent: 20, n_layers: 2, n_latent_u: 20 | Embedding | Full | 0.72552 | 0.79818 | 0.65285 | 0.89768 | 0.92800 | 0.36930 | 0.99776 | 0.63714 | 0.34865 | 0.62720 | N/A | N/A | 0.99840 | N/A |
| n_hidden: 256, n_latent: 20, n_layers: 2, n_latent_u: 20 | Embedding | HVG | 0.72009 | 0.78880 | 0.65139 | 0.88506 | 0.90238 | 0.37013 | 0.99763 | 0.62839 | 0.36003 | 0.61912 | N/A | N/A | 0.99801 | N/A |
| n_hidden: 256, n_latent: 20, n_layers: 3, n_latent_u: 20 | Embedding | Full | 0.73150 | 0.79120 | 0.67181 | 0.89385 | 0.93706 | 0.37073 | 0.96315 | 0.65891 | 0.39234 | 0.63655 | N/A | N/A | 0.99945 | N/A |
| n_hidden: 256, n_latent: 20, n_layers: 3, n_latent_u: 20 | Embedding | HVG | 0.73429 | 0.79661 | 0.67197 | 0.88763 | 0.93006 | 0.37318 | 0.99558 | 0.65665 | 0.39498 | 0.63720 | N/A | N/A | 0.99906 | N/A |
| n_hidden: 256, n_latent: 30, n_layers: 1, n_latent_u: 20 | Embedding | Full | 0.75026 | 0.81107 | 0.68945 | 0.94015 | 0.94001 | 0.36612 | 0.99802 | 0.69322 | 0.44205 | 0.62377 | N/A | N/A | 0.99876 | N/A |
| n_hidden: 256, n_latent: 30, n_layers: 1, n_latent_u: 20 | Embedding | HVG | 0.74092 | 0.80648 | 0.67536 | 0.93647 | 0.92622 | 0.36543 | 0.99781 | 0.67200 | 0.40842 | 0.62225 | N/A | N/A | 0.99875 | N/A |
| n_hidden: 256, n_latent: 30, n_layers: 2, n_latent_u: 20 | Embedding | Full | 0.73697 | 0.80313 | 0.67081 | 0.91700 | 0.92683 | 0.37155 | 0.99714 | 0.65109 | 0.41039 | 0.62310 | N/A | N/A | 0.99866 | N/A |
| n_hidden: 256, n_latent: 30, n_layers: 2, n_latent_u: 20 | Embedding | HVG | 0.73341 | 0.80439 | 0.66243 | 0.91709 | 0.93158 | 0.37138 | 0.99752 | 0.64600 | 0.37760 | 0.62782 | N/A | N/A | 0.99830 | N/A |
| n_hidden: 256, n_latent: 30, n_layers: 3, n_latent_u: 20 | Embedding | Full | 0.74845 | 0.80990 | 0.68700 | 0.91807 | 0.95210 | 0.37343 | 0.99599 | 0.67459 | 0.42837 | 0.64597 | N/A | N/A | 0.99906 | N/A |
| n_hidden: 256, n_latent: 30, n_layers: 3, n_latent_u: 20 | Embedding | HVG | 0.74979 | 0.81117 | 0.68841 | 0.91647 | 0.95774 | 0.37581 | 0.99466 | 0.67716 | 0.43492 | 0.64209 | N/A | N/A | 0.99947 | N/A |
| n_hidden: 256, n_latent: 40, n_layers: 1, n_latent_u: 20 | Embedding | Full | 0.74842 | 0.80995 | 0.68690 | 0.94027 | 0.93660 | 0.36488 | 0.99804 | 0.68109 | 0.43920 | 0.62894 | N/A | N/A | 0.99835 | N/A |
| n_hidden: 256, n_latent: 40, n_layers: 1, n_latent_u: 20 | Embedding | HVG | 0.75021 | 0.81234 | 0.68808 | 0.94106 | 0.94142 | 0.36911 | 0.99777 | 0.68905 | 0.44708 | 0.61744 | N/A | N/A | 0.99876 | N/A |
| n_hidden: 256, n_latent: 40, n_layers: 2, n_latent_u: 20 | Embedding | Full | 0.74259 | 0.80163 | 0.68354 | 0.91308 | 0.92580 | 0.37027 | 0.99738 | 0.67992 | 0.44268 | 0.61334 | N/A | N/A | 0.99822 | N/A |
| n_hidden: 256, n_latent: 40, n_layers: 2, n_latent_u: 20 | Embedding | HVG | 0.75400 | 0.81097 | 0.69703 | 0.92129 | 0.95549 | 0.37008 | 0.99701 | 0.70034 | 0.45714 | 0.63165 | N/A | N/A | 0.99917 | N/A |
| n_hidden: 256, n_latent: 40, n_layers: 3, n_latent_u: 20 | Embedding | Full | 0.74326 | 0.80663 | 0.67988 | 0.90799 | 0.94797 | 0.37502 | 0.99556 | 0.66553 | 0.40160 | 0.65296 | N/A | N/A | 0.99942 | N/A |
| n_hidden: 256, n_latent: 40, n_layers: 3, n_latent_u: 20 | Embedding | HVG | 0.73677 | 0.80933 | 0.66421 | 0.90938 | 0.95837 | 0.37523 | 0.99433 | 0.64333 | 0.35109 | **0.66279** | N/A | N/A | 0.99962 | N/A |
| n_hidden: 256, n_latent: 50, n_layers: 1, n_latent_u: 20 | Embedding | Full | **0.75718** | 0.80533 | 0.70904 | 0.93433 | 0.92187 | 0.36719 | 0.99791 | **0.71083** | 0.50708 | 0.61950 | N/A | N/A | 0.99875 | N/A |
| n_hidden: 256, n_latent: 50, n_layers: 1, n_latent_u: 20 | Embedding | HVG | 0.75026 | 0.80163 | **0.71090** | 0.93268 | 0.91194 | 0.36364 | 0.99825 | 0.70339 | **0.52213** | 0.61953 | N/A | N/A | 0.99853 | N/A |
| n_hidden: 256, n_latent: 50, n_layers: 2, n_latent_u: 20 | Embedding | Full | 0.74676 | 0.81098 | 0.68255 | 0.92705 | 0.94965 | 0.37014 | 0.99707 | 0.66760 | 0.44937 | 0.61661 | N/A | N/A | 0.99662 | N/A |
| n_hidden: 256, n_latent: 50, n_layers: 2, n_latent_u: 20 | Embedding | HVG | 0.74279 | 0.80984 | 0.67574 | 0.92246 | 0.94616 | 0.37309 | 0.99766 | 0.66760 | 0.40787 | 0.62266 | N/A | N/A | 0.99836 | N/A |
| n_hidden: 256, n_latent: 50, n_layers: 3, n_latent_u: 20 | Embedding | Full | 0.72299 | 0.79756 | 0.64842 | 0.90462 | 0.94866 | 0.37575 | 0.96121 | 0.60890 | 0.34624 | 0.63946 | N/A | N/A | 0.99908 | N/A |
| n_hidden: 256, n_latent: 50, n_layers: 3, n_latent_u: 20 | Embedding | HVG | 0.74121 | 0.81154 | 0.67087 | 0.92012 | 0.95594 | 0.37432 | 0.99581 | 0.65290 | 0.39022 | 0.64138 | N/A | N/A | 0.99897 | N/A |

**Table 9:** *Performance of MrVI on the Zenodo 11100300 (18 PBMC Samples) dataset*

Table 9 reports the MrVI sweep on Zenodo 11100300 over $n_{\text{hidden}} \in \{128, 256\}$, $n_{\text{latent}} \in \{10, 20, 30, 40, 50\}$, $n_{\text{layers}} \in \{1, 2, 3\}$, and Features $\in \{\text{Full}, \text{HVG}\}$. The best Overall is $0.75718$ at $(256, 50, 1, \text{Full})$, while the worst Overall is $0.70797$ at $(256, 20, 3, \text{Full})$.

Changing hidden size from $128$ to $256$ yields small but consistent gains across the composites. In matched pairs that fix $(n_{\text{latent}}, n_{\text{layers}}, \text{Features})$, Overall improves in $17/30$ and declines in $13/30$ pairs, Overall(BC) improves in $19/30$ and declines in $11/30$, and Overall(Bio) improves in $17/30$ and declines in $13/30$. Thus, larger hidden size modestly lifts both batch and biology composites and nudges the primary Overall upward.

Increasing depth from $n_{\text{layers}}{=}1$ to $3$ is broadly unfavorable for the aggregate scores. At fixed $(n_{\text{hidden}}, n_{\text{latent}}, \text{Features})$, Overall rises in $7/20$ and falls in $13/20$ pairs, Overall(BC) is essentially flat with $10/20$ up and $10/20$ down, and Overall(Bio) rises in $7/20$ and falls in $13/20$. Deeper encoders, therefore, tend to reduce biological conservation and the primary Overall, while not improving the batch composite on average.

Latent dimensionality shows a clear preference toward higher capacity in this grid. Although the table does not contain enough perfectly matched stepwise chains to report full $10 \rightarrow 20 \rightarrow 30 \rightarrow 40 \rightarrow 50$ counts at fixed $(n_{\text{hidden}}, n_{\text{layers}}, \text{Features})$, the strongest configurations concentrate at larger $n_{\text{latent}}$: the best Overall occurs at $n_{\text{latent}}{=}50$ with shallow depth, and several batch- and clustering-oriented peaks are also at higher latent values.

Stepping $n_{\text{latent\_u}}$ from $10$ to $20$ yields consistent gains on this dataset. In paired comparisons at fixed $(n_{\text{hidden}}, n_{\text{latent}}, n_{\text{layers}}, \text{Features})$, the primary Overall increases in $40/60$ pairs, the batch composite Overall(BC) in $51/60$, and the biology composite Overall(Bio) in $32/60$. Batch metrics benefit strongly: Batch ASW rises in $54/60$ and PCR in $50/60$; iLISI shifts only slightly on average (up $24/60$). Clustering agreement improves as well: NMI up $31/60$ and ARI up $34/60$. Label compactness also trends upward (Label ASW up $42/60$), while graph connectivity is nearly unchanged on average (up $32/60$, 1 flat) and cLISI is essentially flat (up $29/60$).

By feature set, the $u$ increase helps both Full and HVG, with a stronger batch gain under HVG. For Full: Overall up $18/30$, Overall(BC) up $23/30$, Overall(Bio) up $15/30$. For HVG: Overall up $22/30$, Overall(BC) up $28/30$, Overall(Bio) up $17/30$. In short, enlarging $n_{\text{latent\_u}}$ from $10$ to $20$ reliably improves batch removal and usually lifts the primary Overall, while also providing modest gains in clustering agreement and biology; neighborhood-level batch mixing (iLISI) moves only slightly on average.

Metric-wise optima align with these trends. Overall peaks at $(256, 50, 1, \text{Full})$ with $0.75718$; the batch composite Overall(BC) peaks at $(128, 30, 3, \text{HVG})$ with $0.81304$; the biology composite Overall(Bio) peaks at $(256, 50, 1, \text{HVG})$ with $0.71090$. Among individual metrics: Batch ASW is highest at $(128, 40, 1, \text{Full})$ with $0.94321$; PCR and iLISI peak at $(256, 50, 3, \text{Full})$ with $0.96658$ and $0.37803$, respectively; graph connectivity is highest at $(128, 30, 1, \text{Full})$ with $0.99835$; clustering agreement is strongest at high latent with shallow depth, with NMI peaking at $(256, 50, 1, \text{Full})$ with $0.71083$ and ARI at $(256, 50, 1, \text{HVG})$ with $0.52213$; Label ASW peaks at $(256, 40, 3, \text{HVG})$ with $0.66279$; cLISI peaks at $(256, 30, 3, \text{HVG})$ with $0.99967$.

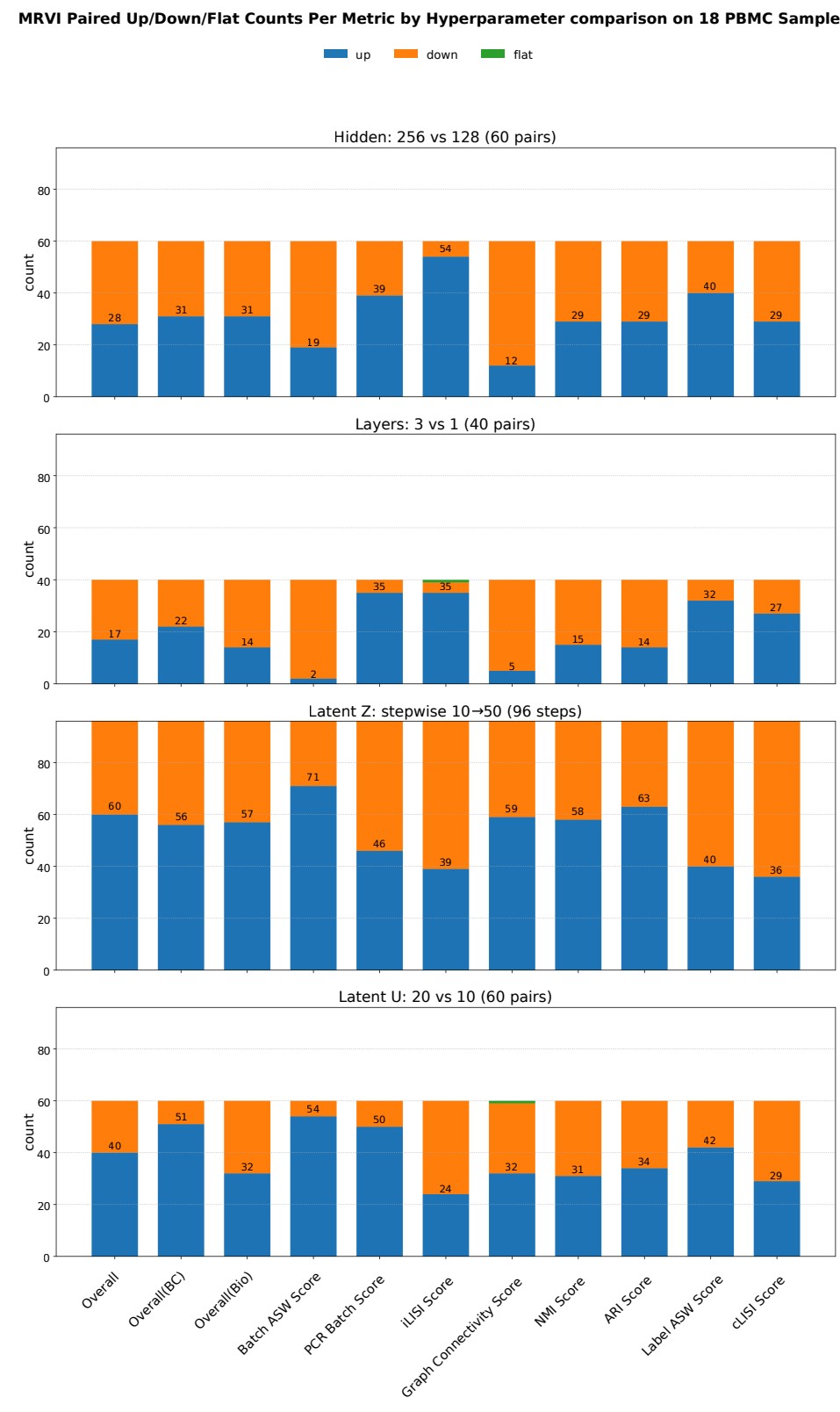

**Figure 23:** *Paired comparison of MrVI hyperparameters on the Zenodo 11100300 (18 PBMC Samples) dataset, showing the number of metrics that improve ("up"), decline ("down"), or remain unchanged ("flat") when varying hidden units (256 vs. 128), network depth (3 vs. 1 layers), latent dimension z (stepwise 10→50), and latent dimension u evaluated on both full and HVG feature sets.*

| Hyperparameters | Output | Features | Overall | Overall(BC) | Overall(Bio) | Batch correction | | | | Bio Conservation | | | | | | |
| | | | Overall | Overall(BC) | Overall(Bio) | ASW | PCR | iLISI | GC | NMI | ARI | ASW | IL F1 | IL ASW | cLISI | TC |
| n_hidden: 128, n_latent: 10, n_layers: 1 | Embedding | Full | 0.66922 | 0.68485 | 0.65359 | 0.89983 | 0.51210 | 0.32851 | 0.99894 | 0.64332 | 0.38295 | 0.58919 | N/A | N/A | 0.99890 | N/A |
| n_hidden: 128, n_latent: 10, n_layers: 1 | Embedding | HVG | 0.69900 | 0.70558 | 0.69243 | 0.93282 | 0.55843 | 0.33332 | 0.99774 | 0.68997 | 0.45604 | 0.62465 | N/A | N/A | 0.99905 | N/A |
| n_hidden: 128, n_latent: 10, n_layers: 2 | Embedding | Full | 0.67226 | 0.68766 | 0.65686 | 0.89205 | 0.52919 | 0.33041 | 0.99900 | 0.66042 | 0.38010 | 0.58774 | N/A | N/A | 0.99916 | N/A |
| n_hidden: 128, n_latent: 10, n_layers: 2 | Embedding | HVG | 0.70885 | 0.72790 | 0.68980 | 0.92680 | 0.64955 | 0.33761 | 0.99763 | 0.68709 | 0.45304 | 0.62009 | N/A | N/A | 0.99898 | N/A |
| n_hidden: 128, n_latent: 10, n_layers: 3 | Embedding | Full | 0.68873 | 0.71742 | 0.66005 | 0.89809 | 0.64061 | 0.33202 | 0.99896 | 0.66387 | 0.38156 | 0.59516 | N/A | N/A | 0.99961 | N/A |
| n_hidden: 128, n_latent: 10, n_layers: 3 | Embedding | HVG | 0.69579 | 0.71869 | 0.67290 | 0.92532 | 0.61756 | 0.33452 | 0.99733 | 0.66307 | 0.40421 | **0.62502** | N/A | N/A | 0.99930 | N/A |
| n_hidden: 128, n_latent: 20, n_layers: 1 | Embedding | Full | 0.61313 | 0.56658 | 0.65967 | 0.95915 | 0.00000 | 0.30768 | 0.99950 | 0.65994 | 0.40886 | 0.57070 | N/A | N/A | 0.99918 | N/A |
| n_hidden: 128, n_latent: 20, n_layers: 1 | Embedding | HVG | 0.70715 | 0.72204 | 0.69226 | 0.95870 | 0.59276 | 0.33829 | 0.99839 | 0.70365 | 0.49034 | 0.57689 | N/A | N/A | 0.99816 | N/A |
| n_hidden: 128, n_latent: 20, n_layers: 2 | Embedding | Full | 0.63171 | 0.60380 | 0.65963 | 0.95415 | 0.15074 | 0.31077 | 0.99953 | 0.66210 | 0.40311 | 0.57406 | N/A | N/A | 0.99925 | N/A |
| n_hidden: 128, n_latent: 20, n_layers: 2 | Embedding | HVG | 0.70848 | 0.73149 | 0.68547 | 0.95561 | 0.63471 | 0.33741 | 0.99825 | 0.68254 | 0.47434 | 0.58661 | N/A | N/A | 0.99840 | N/A |
| n_hidden: 128, n_latent: 20, n_layers: 3 | Embedding | Full | 0.64390 | 0.61499 | 0.67282 | 0.95028 | 0.19762 | 0.31278 | 0.99928 | 0.68321 | 0.43191 | 0.57659 | N/A | N/A | 0.99956 | N/A |
| n_hidden: 128, n_latent: 20, n_layers: 3 | Embedding | HVG | 0.70999 | 0.74944 | 0.67053 | 0.94880 | 0.71010 | 0.34067 | 0.99819 | 0.66540 | 0.42680 | 0.59109 | N/A | N/A | 0.99884 | N/A |
| n_hidden: 128, n_latent: 30, n_layers: 1 | Embedding | Full | 0.63633 | 0.60215 | 0.67051 | 0.96330 | 0.13281 | 0.31289 | **0.99960** | 0.67682 | 0.44288 | 0.56346 | N/A | N/A | 0.99889 | N/A |
| n_hidden: 128, n_latent: 30, n_layers: 1 | Embedding | HVG | 0.70546 | 0.72589 | 0.68503 | 0.96659 | 0.59804 | 0.34019 | 0.99875 | 0.69742 | 0.48514 | 0.56006 | N/A | N/A | 0.99750 | N/A |
| n_hidden: 128, n_latent: 30, n_layers: 2 | Embedding | Full | 0.64692 | 0.63221 | 0.66163 | 0.96235 | 0.24953 | 0.31774 | 0.99923 | 0.66341 | 0.41367 | 0.56978 | N/A | N/A | 0.99967 | N/A |
| n_hidden: 128, n_latent: 30, n_layers: 2 | Embedding | HVG | 0.71049 | 0.74963 | 0.67135 | 0.96334 | 0.69679 | 0.34030 | 0.99806 | 0.67562 | 0.44076 | 0.57074 | N/A | N/A | 0.99828 | N/A |
| n_hidden: 128, n_latent: 30, n_layers: 3 | Embedding | Full | 0.63434 | 0.61503 | 0.65365 | 0.96285 | 0.18332 | 0.31447 | 0.99949 | 0.65318 | 0.38983 | 0.57158 | N/A | N/A | **1.00000** | N/A |
| n_hidden: 128, n_latent: 30, n_layers: 3 | Embedding | HVG | 0.71103 | 0.75603 | 0.66604 | 0.96007 | **0.72364** | 0.34199 | 0.99841 | 0.66345 | 0.42711 | 0.57505 | N/A | N/A | 0.99855 | N/A |
| n_hidden: 128, n_latent: 40, n_layers: 1 | Embedding | Full | 0.63766 | 0.60095 | 0.67438 | 0.96982 | 0.12054 | 0.31390 | 0.99954 | 0.69063 | 0.45829 | 0.55013 | N/A | N/A | 0.99845 | N/A |
| n_hidden: 128, n_latent: 40, n_layers: 1 | Embedding | HVG | 0.71253 | 0.74455 | 0.68052 | 0.97159 | 0.66565 | 0.34217 | 0.99878 | 0.69499 | 0.48060 | 0.54996 | N/A | N/A | 0.99652 | N/A |
| n_hidden: 128, n_latent: 40, n_layers: 2 | Embedding | Full | 0.64718 | 0.61952 | 0.67483 | 0.96673 | 0.19443 | 0.31691 | 0.99950 | 0.68321 | 0.46455 | 0.55228 | N/A | N/A | 0.99926 | N/A |
| n_hidden: 128, n_latent: 40, n_layers: 2 | Embedding | HVG | 0.71840 | 0.72964 | 0.70717 | 0.96619 | 0.61405 | 0.33987 | 0.99847 | 0.71069 | 0.56770 | 0.55258 | N/A | N/A | 0.99769 | N/A |
| n_hidden: 128, n_latent: 40, n_layers: 3 | Embedding | Full | 0.62854 | 0.61290 | 0.64418 | 0.96547 | 0.17010 | 0.31669 | 0.99936 | 0.65823 | 0.36317 | 0.55578 | N/A | N/A | 0.99955 | N/A |
| n_hidden: 128, n_latent: 40, n_layers: 3 | Embedding | HVG | 0.70254 | 0.74067 | 0.66441 | 0.96305 | 0.66255 | 0.33900 | 0.99806 | 0.66393 | 0.42747 | 0.56776 | N/A | N/A | 0.99847 | N/A |
| n_hidden: 128, n_latent: 50, n_layers: 1 | Embedding | Full | 0.65101 | 0.63033 | 0.67168 | 0.97123 | 0.23177 | 0.31883 | 0.99949 | 0.68602 | 0.45757 | 0.54507 | N/A | N/A | 0.99806 | N/A |
| n_hidden: 128, n_latent: 50, n_layers: 1 | Embedding | HVG | 0.72147 | 0.74746 | 0.69549 | 0.97587 | 0.67587 | 0.34287 | 0.99855 | 0.71950 | 0.52053 | 0.54600 | N/A | N/A | 0.99593 | N/A |
| n_hidden: 128, n_latent: 50, n_layers: 2 | Embedding | Full | 0.64379 | 0.62776 | 0.65982 | 0.96814 | 0.22649 | 0.31682 | 0.99958 | 0.67170 | 0.41838 | 0.55000 | N/A | N/A | 0.99919 | N/A |
| n_hidden: 128, n_latent: 50, n_layers: 2 | Embedding | HVG | 0.72941 | 0.74135 | 0.71746 | 0.96833 | 0.65883 | 0.33937 | 0.99887 | 0.72210 | 0.59529 | 0.55454 | N/A | N/A | 0.99793 | N/A |
| n_hidden: 128, n_latent: 50, n_layers: 3 | Embedding | Full | 0.62294 | 0.60653 | 0.63934 | 0.97032 | 0.14141 | 0.31507 | 0.99930 | 0.64999 | 0.35428 | 0.55361 | N/A | N/A | 0.99949 | N/A |
| n_hidden: 128, n_latent: 50, n_layers: 3 | Embedding | HVG | 0.72420 | 0.75199 | 0.69651 | 0.96222 | 0.70696 | 0.33977 | 0.99862 | 0.68980 | 0.52873 | 0.56881 | N/A | N/A | 0.99870 | N/A |
| n_hidden: 256, n_latent: 10, n_layers: 1 | Embedding | Full | 0.68101 | 0.70862 | 0.65340 | 0.90189 | 0.60293 | 0.33074 | 0.99892 | 0.65243 | 0.37347 | 0.58840 | N/A | N/A | 0.99930 | N/A |
| n_hidden: 256, n_latent: 10, n_layers: 1 | Embedding | HVG | 0.70327 | 0.70459 | 0.70196 | 0.90854 | 0.54774 | 0.33591 | 0.99739 | 0.70159 | 0.48754 | 0.61940 | N/A | N/A | 0.99930 | N/A |
| n_hidden: 256, n_latent: 10, n_layers: 2 | Embedding | Full | 0.67382 | 0.68452 | 0.66312 | 0.90854 | 0.50046 | 0.32998 | 0.99911 | 0.65912 | 0.40110 | 0.59269 | N/A | N/A | 0.99958 | N/A |
| n_hidden: 256, n_latent: 10, n_layers: 2 | Embedding | HVG | 0.69601 | 0.71431 | 0.67770 | 0.93357 | 0.58962 | 0.33673 | 0.99734 | 0.67147 | 0.41974 | 0.62062 | N/A | N/A | 0.99897 | N/A |
| n_hidden: 256, n_latent: 10, n_layers: 3 | Embedding | Full | 0.67723 | 0.68634 | 0.66812 | 0.90834 | 0.50758 | 0.33094 | 0.99849 | 0.67443 | 0.41019 | 0.58848 | N/A | N/A | 0.99938 | N/A |
| n_hidden: 256, n_latent: 10, n_layers: 3 | Embedding | HVG | 0.69707 | 0.71346 | 0.68069 | 0.92903 | 0.59068 | 0.33699 | 0.99716 | 0.67125 | 0.43346 | 0.61908 | N/A | N/A | 0.99896 | N/A |
| n_hidden: 256, n_latent: 20, n_layers: 1 | Embedding | Full | 0.62700 | 0.58520 | 0.66881 | 0.96007 | 0.07103 | 0.31017 | 0.99954 | 0.68082 | 0.42169 | 0.57338 | N/A | N/A | 0.99933 | N/A |
| n_hidden: 256, n_latent: 20, n_layers: 1 | Embedding | HVG | 0.71483 | 0.73610 | 0.69355 | 0.95630 | 0.65078 | 0.33897 | 0.99834 | 0.70543 | 0.49129 | 0.57918 | N/A | N/A | 0.99831 | N/A |
| n_hidden: 256, n_latent: 20, n_layers: 2 | Embedding | Full | 0.64195 | 0.61763 | 0.66627 | 0.94961 | 0.20773 | 0.31371 | 0.99947 | 0.66770 | 0.42324 | 0.57489 | N/A | N/A | 0.99924 | N/A |
| n_hidden: 256, n_latent: 20, n_layers: 2 | Embedding | HVG | 0.69957 | 0.72969 | 0.66946 | 0.95716 | 0.62667 | 0.33721 | 0.99770 | 0.66985 | 0.43037 | 0.57958 | N/A | N/A | 0.99803 | N/A |
| n_hidden: 256, n_latent: 20, n_layers: 3 | Embedding | Full | 0.60572 | 0.56740 | 0.64404 | 0.95804 | 0.00376 | 0.30856 | 0.99926 | 0.65183 | 0.39470 | 0.57448 | N/A | N/A | 0.99935 | N/A |
| n_hidden: 256, n_latent: 20, n_layers: 3 | Embedding | HVG | 0.69670 | 0.70928 | 0.68412 | 0.95738 | 0.54704 | 0.33453 | 0.99818 | 0.68266 | 0.47468 | 0.58086 | N/A | N/A | 0.99828 | N/A |
| n_hidden: 256, n_latent: 30, n_layers: 1 | Embedding | Full | 0.64997 | 0.63624 | 0.66371 | 0.96638 | 0.26638 | 0.31840 | 0.99960 | 0.67410 | 0.42422 | 0.55739 | N/A | N/A | 0.99912 | N/A |
| n_hidden: 256, n_latent: 30, n_layers: 1 | Embedding | HVG | 0.72052 | 0.74827 | 0.69276 | 0.96901 | 0.68418 | 0.34147 | 0.99842 | 0.70099 | 0.51371 | 0.55950 | N/A | N/A | 0.99685 | N/A |
| n_hidden: 256, n_latent: 30, n_layers: 2, | Embedding | Full | 0.63691 | 0.60311 | 0.67071 | 0.96494 | 0.13512 | 0.31283 | 0.99955 | 0.68363 | 0.43328 | 0.56647 | N/A | N/A | 0.99947 | N/A |
| n_hidden: 256, n_latent: 30, n_layers: 2, | Embedding | Full | 0.70498 | 0.73418 | 0.67578 | 0.96778 | 0.63214 | 0.33857 | 0.99823 | 0.68771 | 0.45385 | 0.56397 | N/A | N/A | 0.99759 | N/A |
| n_hidden: 256, n_latent: 30, n_layers: 3 | Embedding | Full | 0.63773 | 0.60856 | 0.66691 | 0.96530 | 0.15305 | 0.31629 | 0.99958 | 0.67815 | 0.42802 | 0.56220 | N/A | N/A | 0.99926 | N/A |
| n_hidden: 256, n_latent: 30, n_layers: 3 | Embedding | HVG | 0.70053 | 0.72777 | 0.67329 | 0.96734 | 0.60678 | 0.33870 | 0.99827 | 0.68378 | 0.45102 | 0.56108 | N/A | N/A | 0.99728 | N/A |
| n_hidden: 256, n_latent: 40, n_layers: 1 | Embedding | Full | 0.63521 | 0.60489 | 0.66552 | 0.97242 | 0.13242 | 0.31795 | 0.99948 | 0.67587 | 0.43682 | 0.55096 | N/A | N/A | 0.99845 | N/A |
| n_hidden: 256, n_latent: 40, n_layers: 1 | Embedding | HVG | 0.71049 | 0.73680 | 0.68417 | 0.97484 | 0.63097 | 0.34325 | 0.99815 | 0.69811 | 0.49464 | 0.54731 | N/A | N/A | 0.99662 | N/A |
| n_hidden: 256, n_latent: 40, n_layers: 2 | Embedding | Full | 0.64448 | 0.62069 | 0.66828 | 0.96645 | 0.20168 | 0.31523 | 0.99940 | 0.68030 | 0.44205 | 0.55159 | N/A | N/A | 0.99917 | N/A |
| n_hidden: 256, n_latent: 40, n_layers: 2 | Embedding | HVG | 0.71339 | 0.73423 | 0.69256 | 0.97423 | 0.62281 | 0.34156 | 0.99831 | 0.69093 | 0.53241 | 0.55036 | N/A | N/A | 0.99655 | N/A |
| n_hidden: 256, n_latent: 40, n_layers: 3 | Embedding | Full | 0.64538 | 0.62161 | 0.66915 | 0.96556 | 0.20558 | 0.31577 | 0.99953 | 0.68046 | 0.43960 | 0.55715 | N/A | N/A | 0.99938 | N/A |
| n_hidden: 256, n_latent: 40, n_layers: 3 | Embedding | HVG | 0.70052 | 0.73334 | 0.66771 | 0.97249 | 0.62162 | 0.34065 | 0.99860 | 0.68068 | 0.43939 | 0.55315 | N/A | N/A | 0.99761 | N/A |
| n_hidden: 256, n_latent: 50, n_layers: 1 | Embedding | Full | 0.65889 | 0.63772 | 0.68006 | 0.97166 | 0.26112 | 0.31895 | 0.99918 | 0.69559 | 0.48283 | 0.54378 | N/A | N/A | 0.99804 | N/A |
| n_hidden: 256, n_latent: 50, n_layers: 1 | Embedding | HVG | **0.74219** | **0.75603** | **0.72835** | 0.97573 | 0.70291 | **0.34685** | 0.99863 | **0.74726** | **0.63073** | 0.54145 | N/A | N/A | 0.99397 | N/A |
| n_hidden: 256, n_latent: 50, n_layers: 2 | Embedding | Full | 0.64468 | 0.62555 | 0.66382 | 0.96726 | 0.21867 | 0.31688 | 0.99938 | 0.67895 | 0.43176 | 0.54604 | N/A | N/A | 0.99851 | N/A |
| n_hidden: 256, n_latent: 50, n_layers: 2 | Embedding | HVG | 0.73305 | 0.74983 | 0.71627 | 0.96725 | 0.68357 | 0.34237 | 0.99861 | 0.72899 | 0.59677 | 0.54367 | N/A | N/A | 0.99564 | N/A |
| n_hidden: 256, n_latent: 50, n_layers: 3 | Embedding | Full | 0.66128 | 0.62774 | 0.69482 | 0.97240 | 0.22309 | 0.31653 | 0.99896 | 0.69834 | 0.53510 | 0.54665 | N/A | N/A | 0.99916 | N/A |
| n_hidden: 256, n_latent: 50, n_layers: 3 | Embedding | HVG | 0.72043 | 0.74848 | 0.69239 | 0.97486 | 0.67910 | 0.34140 | 0.99858 | 0.69172 | 0.53446 | 0.54748 | N/A | N/A | 0.99588 | N/A |

**Table 10:** *Performance of LDVAE on the Zenodo 11100300 (18 PBMC Samples)*

Table 10 reports the LDVAE sweep on Zenodo 11100300 over $n_{\text{hidden}} \in \{128, 256\}$, $n_{\text{latent}} \in \{10, 20, 30, 40, 50\}$, $n_{\text{layers}} \in \{1, 2, 3\}$, and Features $\in \{\text{Full}, \text{HVG}\}$. The best Overall is $0.74219$ at $(256, 50, 1, \text{HVG})$, while the worst Overall is $0.60572$ at $(256, 20, 3, \text{Full})$.

Changing hidden size from $128$ to $256$ yields a small net gain in the primary aggregate and a clearer lift in biology, with a slight reduction in the batch composite. In paired comparisons that fix $(n_{\text{latent}}, n_{\text{layers}}, \text{Features})$, Overall rises in $16/30$ and falls in $14/30$ pairs, Overall(BC) rises in $14/30$ and falls in $16/30$, and Overall(Bio) rises in $20/30$ and falls in $10/30$.

Depth is broadly unfavorable for the main and biology composites. From $n_{\text{layers}}=2$ versus $1$, Overall is up $8/20$ and down $12/20$, Overall(BC) is essentially flat with $10/20$ up, and Overall(Bio) is up $5/20$ and down $15/20$. From $n_{\text{layers}}=3$ versus $1$, Overall is up $7/20$ and down $13/20$, Overall(BC) is up $11/20$, and Overall(Bio) is up $1/20$ and down $19/20$. Deeper encoders, therefore, tend to reduce biological conservation and slightly depress the primary. Overall, the batch composite is flat to modestly positive only at three layers.

Latent dimensionality shows the clearest positive signal for batch removal and a modest lift for the primary aggregate. Stepwise across $10 \to 20 \to 30 \to 40 \to 50$ at fixed $(n_{\text{hidden}}, n_{\text{layers}}, \text{Features})$, Overall increases in $31/48$ steps, Overall(BC) in $40/48$, while Overall(Bio) increases in $18/48$. The endpoint contrast $50-10$ reinforces this: Overall improves in $8/12$ matched triplets, Overall(BC) in $10/12$, and Overall(Bio) in $6/12$. Increasing $n_{\text{latent}}$ thus reliably helps batch correction and typically nudges the primary Overall upward; the biology composite often softens stepwise but is roughly balanced end-to-end.

Feature selection favors HVG on this dataset. In paired HVG$-$Full comparisons at fixed $(n_{\text{hidden}}, n_{\text{latent}}, n_{\text{layers}})$, Overall is higher with HVG in $16/30$ and lower in $14/30$, Overall(BC) is higher in $16/30$, and Overall(Bio) is higher in $18/30$. HVG is therefore modestly better for the primary, batch, and biology composites in this grid.

Metric-wise optima are consistent with these trends. Overall peaks at $(256, 50, 1, \mathrm{HVG})$ with $0.74219$; the batch composite Overall(BC) peaks at $(128, 30, 3, \mathrm{HVG})$ with $0.75603$; the biology composite Overall(Bio) peaks at $(256, 50, 1, \mathrm{HVG})$ with $0.72835$. Among individual metrics: Batch ASW is highest at $(256, 50, 1, \mathrm{HVG})$ with $0.97573$; PCR peaks at $(128, 30, 3, \mathrm{HVG})$ with $0.72364$; iLISI peaks at $(256, 50, 1, \mathrm{HVG})$ with $0.34685$; graph connectivity is highest at $(128, 30, 1, \mathrm{Full})$ with $0.99960$; clustering agreement is strongest at high latent with shallow depth, with NMI and ARI both peaking at $(256, 50, 1, \mathrm{HVG})$ with $0.74726$ and $0.63073$, respectively; Label ASW peaks at $(128, 10, 3, \mathrm{HVG})$ with $0.62502$; cLISI peaks at $(128, 30, 3, \mathrm{Full})$ with $1.00000$.

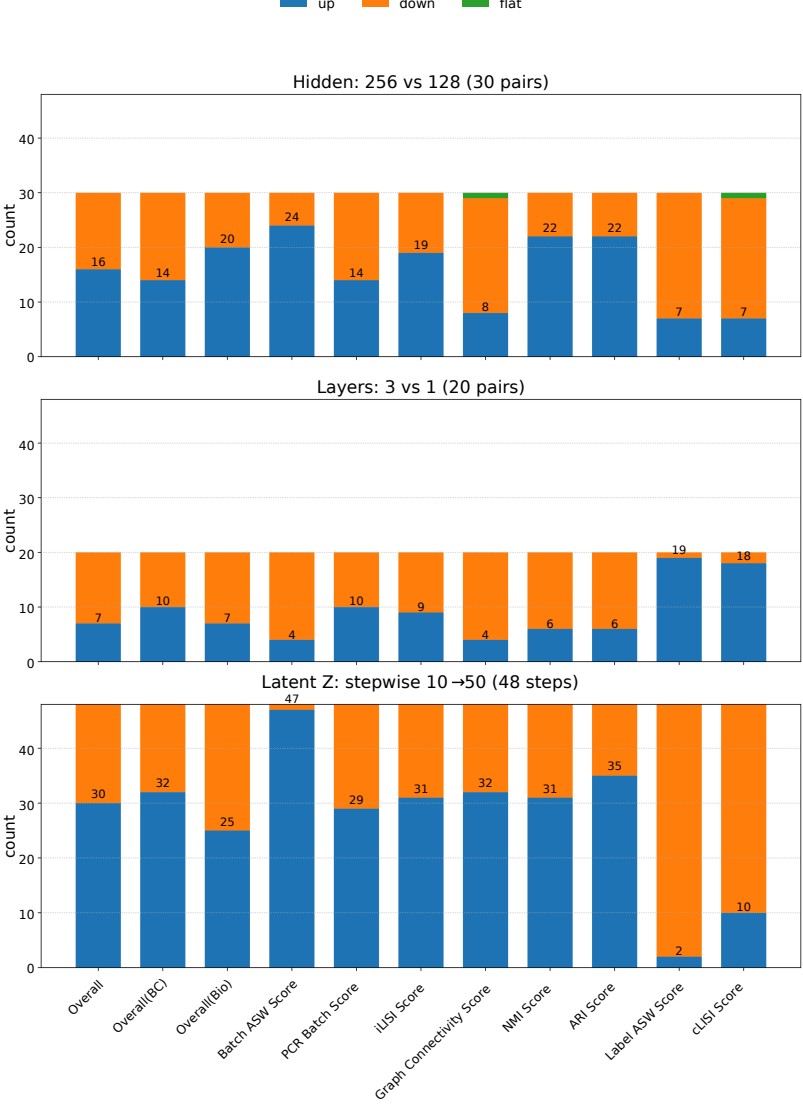

**Figure 24:** *Paired comparison of LDVAE hyperparameters on the Zenodo 11100300 (18 PBMC Samples) dataset, showing the number of metrics that improve ("up"), decline ("down"), or remain unchanged ("flat") when varying hidden units (256 vs. 128), network depth (3 vs. 1 layers), latent dimension z (stepwise 10→50) on both full and HVG feature sets.*

