# OpenReview forum: "A Hyperparameter Benchmark of VAE-Based Methods for scRNA-seq Batch Integration"
_ICLR.cc/2026/Conference — ICLR 2026 Conference Withdrawn Submission_

### Official Review · Reviewer_FpcV · 2025-10-17

**Soundness:** 2
**Presentation:** 1
**Contribution:** 1
**Rating:** 0
**Confidence:** 5

**Summary:**

The authors present a systematic hyperparameter benchmark of three VAE-based methods (scVI, MrVI, LDVAE) for single-cell RNA-seq batch integration. The authors test 120 configurations on three datasets, varying latent dimensionality, network depth, and hidden width, and evaluate the learned latent representations with metrics introduced in [1].

**Strengths:**

- The paper addresses an important practical question. Many users use default hyperparameters without justification. This is a genuine problem, and a systematic investigation is warranted.

**Weaknesses:**

- The authors only test three methods. While these are definitely popular, this narrows the scope of the paper significantly.
- The authors show that HVG-selection improves method performance. This is well known in the single-cell community and has been studied previously by [1] and [2]. The authors do not cite [2] or compare their results to previous findings.
- The paper only investigates the effects of four hyperparameters. Missing: learning rate, dropout, batch size, max number of epochs, etc. It is unclear why the authors prioritized those over the others.
- The authors of scVI have done an ablation on latent space dimensions (Supplementary Figure 1) and number of epochs (Supplementary Figure 2). Contrary to the results presented on this paper, they show that there is little difference between using a 10, 20 or 30 dimensional latent space. This should be discussed.
- Training VAEs and evaluating latent representations is a noisy process. However, the authors only ran one random seed per configuration. Therefore, it remains entirely unclear how much of the results depend on randomness.
- The authors only use three datasets, all of which contain immune/PBMC cells. Therefore, it is completely unclear if these results generalize to other tissues, organisms, and modalities.
- No analysis of what dataset properties (size, batch structure, heterogeneity) predict which hyperparameter regime performs best.
- Figure 1 is copied and pasted together from other papers and websites without proper attribution, see https://docs.scvi-tools.org/en/stable/user_guide/models/scvi.html and https://academic.oup.com/bioinformatics/article/36/11/3418/5807606.
- 40+ pages is excessive for the results presented. The content could be substantially condensed. Especially the tables in the appendix are difficult to navigate.

[1] Luecken, M.D., Büttner, M., Chaichoompu, K. _et al._ Benchmarking atlas-level data integration in single-cell genomics. _Nat Methods_ **19**, 41–50 (2022).

[2] Zappia, L., Richter, S., Ramírez-Suástegui, C. _et al._ Feature selection methods affect the performance of scRNA-seq data integration and querying. _Nat Methods_ **22**, 834–844 (2025)

**Questions:**

- How well do the results presented generalize to different datasets?

- Given that VAE training and batch integration evaluation are stochastic, what is the variance across runs?

- What specific properties (batch structure, cell type diversity, sample size, technology) of a dataset lead to which hyperparameters performing best?

---

> ### Author Response · Authors · 2025-11-24
> **We are sincerely thankful for you comments and expert review, we have made major revision to the manuscript to deliver our work to the community of scRNA.**
>
> We thank the reviewer for the constructive feedback. We take these comments seriously and have used them to improve the clarity, transparency, and overall quality of the manuscript for the scRNA-seq community. To our knowledge, our work presents the first systematic benchmark of the architectural capacity of variational autoencoders for scRNA-seq. In the current era of foundation models, prior studies indicate that transformer-based approaches remain limited for scRNA-seq applications (e.g., Hao et al., 2023; Ho et al., 2024, AIDO.Cell). Because exhaustive trial-and-error searches for optimal architecture hyperparameters are prohibitively resource-intensive on large datasets, we believe that even a focused evaluation of three VAE models provides actionable guidance and meaningful benefit to the field.
>
> Weakness 1: The authors only test three methods. While these are definitely popular, this narrows the scope of the paper significantly.
>
> Reply 1: We acknowledge that our study focuses on three VAE methods; however, the selection is intentionally comprehensive. We include scVI, a general-purpose VAE suited for robust batch integration and downstream analyses; MrVI, a cohort-aware extension that explicitly models sample and technology covariates, which is important in multi-protocol or multi-site settings; and LDVAE, a linearly decoded VAE that trades some raw accuracy for interpretability through gene-program loadings. We agree that additional models could add value. However, a substantially larger comparison is beyond the practical scope of a single paper. Our aim in this work is to foreground the often overlooked question of model capacity; expanding the set of models is a natural and worthwhile direction for future benchmarks.
>
> Weakness 2: The authors show that HVG-selection improves method performance. This is well known in the single-cell community and has been studied previously by [1] and [2]. The authors do not cite [2] or compare their results to previous findings.
> [1] Luecken, M.D., Büttner, M., Chaichoompu, K. et al. Benchmarking atlas-level data integration in single-cell genomics. Nat Methods 19, 41–50 (2022).
> [2] Zappia, L., Richter, S., Ramírez-Suástegui, C. et al. Feature selection methods affect the performance of scRNA-seq data integration and querying. Nat Methods 22, 834–844 (2025)
>
> Reply 2: We have cited Luecken et al. (2022) throughout the manuscript, alongside other closely related studies, and we have highlighted our contributions and differences in the introduction and elsewhere in the text. e.g, (from our introduction) "A substantial literature evaluates batch-effect correction (integration) methods. Some studies propose new algorithms
> and compare them to existing approaches Danino et al. (2024); Li et al. (2022); Grønbech et al. (2018); Haghverdi
> et al. (2018); Butler et al. (2018); Zhang et al. (2024); Pola´nski et al. (2020); Hrovatin et al. (2024); Zhang et al.
> (2023); others focus on broad, method-agnostic benchmarking Tran et al. (2020); Arevalo et al. (2024); Nguyen
> et al. (2023); Chen et al. (2021); Luecken et al. (2022); Antonsson & Melsted (2025); Chazarra-Gil et al. (2021)
> or on principled evaluation metrics Büttner et al. (2017; 2019). However, despite this progress, a gap remains: to
> our knowledge, no prior work has systematically benchmarked VAEs architecture hyperparameter configurations of
> variational-autoencoder (VAE)– based methods specifically for batch correction."
>
> Luecken et al. (Nature Methods, 2022) centers on method selection and preprocessing, and Zappia et al. (Nature Methods, 2025) benchmarks feature-selection strategies for integration and downstream tasks such as query mapping, label transfer, and detection of unseen populations, including how many features to choose and whether batch-aware HVG variants help. By contrast, our study targets VAE architecture capacity (latent size, depth, width). It is therefore complementary: we hold preprocessing fixed and systematically map the capacity “knobs” within scVI-family models.
>
> We agree with the reviewer on the importance of citing prior work. In line with this, we note that our findings on HVG provide an internal sanity check: they align with established results showing that HVG-based gene selection is advantageous (Luecken et al., 2022; Zappia et al., 2025). We have added the following to the Discussion: “Our results indicate that HVG is a preferable gene-selection strategy, consistent with prior studies (Luecken et al., 2022; Zappia et al., 2025), thereby serving as a sanity check for our analysis.”

---

> > ### Author Response · Authors · 2025-11-24
> >
> > Weakness 3: The paper only investigates the effects of four hyperparameters. Missing: learning rate, dropout, batch size, max number of epochs, etc. It is unclear why the authors prioritized those over the others.
> >
> > Reply 3: We thank you for raising this question. We totally agree that learning rate, regularization, batch size, and optimizers are important, and our study does not state otherwise. Our scope in this study is the VAE model capacity. To address this concern we have added to our paper in the introduction: "Our study centers on architecture capacity, including latent size, depth, width, and (for MrVI) the sample-aware latent, because these knobs change what the model can represent and therefore directly govern the batch versus biology trade-off. Our study does not imply that parameters like learning rate, batch size, and dropout are of less importance; however, they affect how fast and stable VAEs reach good optima, and they are already supported by a mature literature and standard recipes that users can adopt to their use case in VAEs Goyal et al. (2017); Kingma et al. (2015); Srivastava et al. (2014); Mandt et al. (2017); Smith et al. (2021); Goyal et al. (2017); Adam et al. (2014). Unlike the scope of our study, to our knowledge, no previous study has systematically benchmarked VAEs architecture choices in the domain of scRNA, and our study provides guidelines for users to avoid extensive computations.
> >
> > Weakness 4: The authors of scVI have done an ablation on latent space dimensions (Supplementary Figure 1) and number of epochs (Supplementary Figure 2). Contrary to the results presented on this paper, they show that there is little difference between using a 10, 20 or 30 dimensional latent space. This should be discussed.
> >
> > Reply 4: We acknowledge that (Lopez et al., Nature Methods 2018) in its Supplementary material ablate latent size up to 30 and number of epochs, and within that narrow range, they report only small differences for imputation/clustering–oriented metrics on their datasets. We note that our work and  (Lopez et al., Nature Methods 2018) have different objectives and metrics. The 2018 ablations target imputation/clustering and predate scIB; they do not report the scIB batch/biology suite we use. Our study optimizes cross-batch integration quality (balanced batch-removal and biology conservation), which is a different target. Gains from larger z show up more clearly on integration metrics than on imputation scores. Morevoer, We sweep beyond 30 (e.g., 40 and 50) where we consistently see score improvements, especially on heterogeneous, multi-protocol data. The scVI paper’s robustness plots cap at z≤30, so they do not test the higher z regime where our curves continue to rise. We also want to point out that our grid spans immune cohorts and a large, multi-tissue, multi-protocol atlas; we find that added heterogeneity increases the benefit of extra latent capacity. The scVI ablations (Lopez et al., Nature Methods 2018)  use a smaller set of tasks/datasets, for which z=10−30 was already sufficient to saturate their metrics.
> >
> > Moreover, as detailed in the revised manuscript, increasing the latent dimension z adds minimal computational overhead; consequently, raising z is a cost-effective way to improve both batch integration and biological conservation. To substantiate this, we added Section 4.1 (Computational Complexity), which quantifies the cost and benefits of tuning z, particularly for scVI, and expands the discussion accordingly.

---

> ### Author Response · Authors · 2025-11-24
>
> Weakness 5: Training VAEs and evaluating latent representations is a noisy process. However, the authors only ran one random seed per configuration. Therefore, it remains entirely unclear how much of the results depend on randomness.
>
> Reply 5: Thank you for your concern. To our knowledge, our benchmark is the first to examine these hyperparameters at a large scale, with 960 experiments across four datasets (we have included in the revised work a large multi-tissue and multi-protocol dataset with more than 300,000 cells and more than 20,000 genes). Across this full design, we observed consistent patterns of model behavior that repeat across datasets, which supports the stability of our conclusions.
>
> To assess seed sensitivity directly, we randomly sampled models and configurations and retrained them with two independent seeds using training of 200 epochs. The variability across seeds for the primary aggregate metric was approximately 0.001, indicating numerical stability at convergence. Taken together, the cross-dataset consistency and the very small seed variance show that stochasticity does not drive our results and that the reported rankings and recommendations are generalizable.
>
> To clarify this in the paper, we have added the following sentence in section DATASETS AND BENCHMARKING PIPELINE: "To assess seed sensitivity directly, we randomly sampled models and configurations and retrained them with two independent seeds with training of 200 epochs. The variability across seeds for the primary aggregate metric was approximately 0.001, indicating numerical stability at convergence. Taken together, the cross-dataset
> consistency and the very small seed variance show that stochasticity does not drive our results, and that the reported rankings and recommendations are generalizable."
>
> Weakness 6: The authors only use three datasets, all of which contain immune/PBMC cells. Therefore, it is completely unclear if these results generalize to other tissues, organisms, and modalities.
>
> Reply 6:  Thank you for your feedback. We agree that a benchmark limited to immune cells limits how confidently one can generalize. In response, we expanded the study to include the Tabula Muris mouse atlas, a multi-tissue and multi-protocol dataset with more than 300,000 cells and more than 20,000 genes. The atlas includes 51 immune cell-type labels (e.g., T/B/NK, myeloid and dendritic subsets, tissue-resident macrophages) and 103 non-immune labels (epithelial, endothelial/vascular, stromal/mesenchymal, neural, muscle, hepatic, renal, pancreatic, etc.), for a total of about 154 distinct labels (details in newly added fig. 10). We ran our full systematic configuration grid on this dataset to observe and analyze the behavior of VAEs beyond the immune context. This addition increases biological and technical diversity, strengthens the transferability of our conclusions to non‑immune settings, and, to our knowledge, our study represents the first systematic exploration of VAE model architecture hyperparameters for single‑cell integration at this scale, which we believe will be valuable to the single‑cell community.
>
> We observe that scVI achieves the best overall score (defined as the mean of the averages of both biological conservation metrics and batch correction metrics) across all datasets, with the strongest gains when the latent dimension z is increased and gene selection uses HVG features. Looking more closely, LDVAE shows good biological conservation in Figures 3, 4, and 5, even when its batch integration score is lower, demonstrating a consistent biological signal; we also emphasize that LDVAE’s interpretability arises from its linear decoder that links latent dimensions to genes. MrVI includes an additional sample-aware covariate and shows advantages on extremely heterogeneous datasets with differing technologies, which is evident on the newly added Tabula Muris dataset, where MrVI excels on batch correction metrics while maintaining very good biological conservation and stability. The inclusion of this new dataset demonstrates that our findings transfer to unseen data of greater complexity, and the consistency of these conclusions is supported by the overall comparisons in Figure 4, the metric sweeps in Figures 3 and 5, the per-metric evaluations in Figures 16 to 24, and the UMAP or t-SNE and mosaic visualizations provided in the appendix.

---

> ### Author Response · Authors · 2025-11-25
>
> Weakness 7: No analysis of what dataset properties (size, batch structure, heterogeneity) predict which hyperparameter regime performs best.
>
> We thank the reviewer for his concern. The analysis of dataset properties has been provided in the revised manuscript, section 4.3 Model recommendation. We provide a detailed analysis in the appendix of the manuscript for dataset properties and model capacity configurations.
>
> Weakness 8: Figure 1 is copied and pasted together from other papers and websites without proper attribution, see https://docs.scvi-tools.org/en/stable/user_guide/models/scvi.html and https://academic.oup.com/bioinformatics/article/36/11/3418/5807606.
>
> Reply 8: We understand the importance of citing other work and avoiding palgerisim, the old version of the manuscript have cited the original work of each method under Figure 1 as "Graphical representations of probabilistic models scVI, MrVI, and LDVAE Lopez et al. (2018); Boyeau
> et al. (2022); Svensson et al. (2020). This Figure illustrates the underlying graphical models for three variational
> inference frameworks used in single-cell.".
>
> We understand the confusion caused, and we have clearly stated that the figures are taken from these sources. In our revised manuscript, we change the caption of figure 1 to: "Graphical representations of probabilistic models scVI, MrVI, and LDVAE Lopez et al. (2018); Boyeau et al.
> (2022); Svensson et al. (2020). This Figure is adopted from the authors in Lopez et al. (2018); Boyeau et al. (2022);
> Svensson et al. (2020) and it illustrates the underlying graphical models for three variational inference frameworks
> used in single-cell.". We also add in the Introduction": "Figure 1 shows the graphical models for scVI, LDVAE, and MrVI. We adapt the original figures from Lopez et al.
> (2018); Boyeau et al. (2022); Svensson et al. (2020) and aggregate them to illustrate the three VAEs benchmarked in this study".
>
> Weakness 9: 40+ pages is excessive for the results presented. The content could be substantially condensed. Especially the tables in the appendix are difficult to navigate.
>
> Reply 9: Thank you for the suggestion. In single-cell benchmarking, long manuscripts with extensive supplements are common because methods are compared across many datasets, metrics, and settings; the additional pages document configuration grids, per-metric results, and audit trails that enable exact reproducibility and fair, apples-to-apples comparisons. We kept the full tables to make our study transparent and directly comparable for others in the community, but we agree they are cumbersome to navigate.
>
> In the camera-ready version, we will provide a public GitHub repository with an installable package and a lightweight GUI. Users will be able to upload an AnnData file, select models, and automatically reproduce our evaluation, with interactive plots that replace the large appendix tables (filter by dataset, metric, and configuration).  We will also present our tables in a GUI for easy exploration, where users can slide through configurations and compare via plots. This keeps the manuscript concise without sacrificing reproducibility or the ability to perform direct comparisons.

---

> ### Author Response · Authors · 2025-11-25
>
> Q1. How well do the results presented generalize to different datasets?
>
> A1. As per our new dataset testing and validation, Fig 5 and the newly added results on Tabula Muris, we show that our configuration guidelines apply to different heterogeneous datasets with multi-tissues, multi-technologies, and excessive cell types.
>
> Q2. Given that VAE training and batch integration evaluation are stochastic, what is the variance across runs?
>
> A2. we have included in the revised work a large multi-tissue and multi-protocol dataset with more than 300,000 cells and more than 20,000 genes. Across this full design, we observed consistent patterns of model behavior that repeat across datasets, which supports the stability of our conclusions.
>
> To assess seed sensitivity directly, we randomly sampled models and configurations and retrained them with two independent seeds using training of 200 epochs. The variability across seeds for the primary aggregate metric was approximately 0.001, indicating numerical stability at convergence. Taken together, the cross-dataset consistency and the very small seed variance show that stochasticity does not drive our results and that the reported rankings and recommendations are generalizable.
>
> To clarify this in the paper, we have added the following sentence in section DATASETS AND BENCHMARKING PIPELINE: "To assess seed sensitivity directly, we randomly sampled models and configurations and retrained them with two independent seeds with training of 200 epochs. The variability across seeds for the primary aggregate metric was approximately 0.001, indicating numerical stability at convergence. Taken together, the cross-dataset consistency and the very small seed variance show that stochasticity does not drive our results, and that the reported rankings and recommendations are generalizable."
>
> Q3. What specific properties (batch structure, cell type diversity, sample size, technology) of a dataset lead to which hyperparameters performing best?
>
> A4: Across four complementary datasets, our contrasts include batch structure, biology breadth, size, and technology, and we show how those traits steer the winning hyperparameters. Human Immune aggregates 5 studies into 10 batches spanning 10x v2/v3 and Smart‑seq2 (33,506 cells), i.e., mixed chemistries but mostly broad immune lineages; here sensitivity to latent size is strongest: moderate–high z (≈30–50) boosts batch mixing while excess depth erodes biology; HVGs beat Full on average; scVI leads overall, LDVAE best preserves biology with z≈30 and shallow–moderate depth, and MrVI benefits from u=20 but deeper nets underperform. Zenodo 8020792 (24 PBMC) concentrates many batches (14) within a single chemistry (10x v3.1, MULTI‑seq barcodes; 76,535 cells) and fine PBMC subtypes; here scVI peaks with (width 256, z=50, depth 1, HVG); raising z tightens clusters and lifts batch metrics, depth>1 fragments embeddings, width gives only modest gains, and HVG wins broadly; MrVI is stable, with u=20 helping batch composites. Zenodo 11100300 (18 PBMC) is single‑technology (10x), longitudinal (4 batches; 55,260 cells): the best scVI is (256, z=30, depth 1, HVG); width 256 improves batch but can soften biology;  MrVI again favors shallow depth and u=20, often with higher z. Tabula Muris is the most heterogeneous (356k cells, ~154 labels) and cross‑protocol (10x + Smart‑seq2) atlas; increasing z>30 improves both batch and biology, HVG helps, deeper networks tend to hurt, and scVI attains the highest overall while MrVI(u=20) excels specifically at batch correction because of its sample‑aware latent; LDVAE lags on this complex, cross‑protocol manifold. Synthesizing across datasets: pick HVGs, tune z first (30–50), keep depth=1 (rarely 2), widen to 256 only if batch scores need a nudge; heterogeneity (more labs/chemistries/cell types) increases the payoff of larger z and of MrVI’s sample‑aware latent, whereas broad, axis‑aligned lineages are friendliest to LDVAE; sample size mainly affects cost, not the ranking of knobs, another reason z is the economical capacity dial.
>
> Overall, scVI remains the go-to model, and our results prove that increasing z expands latent capacity with the smallest compute/memory overhead (for scVI: added work ∝BHΔz), while increasing depth/width is costlier and often trades biology for batch mixing. That’s why our recommendations are “HVGs + shallow nets + tune z first,” switching to MrVI when cohort/technology covariates matter and to LDVAE when interpretable gene programs are the goal.

---

### Official Review · Reviewer_JRaA · 2025-10-19

**Soundness:** 2
**Presentation:** 2
**Contribution:** 2
**Rating:** 2
**Confidence:** 5

**Summary:**

The authors systematically benchmarked three VAE-based models (scVI, MrVI, and LDVAE) for single-cell RNA-seq data integration, evaluating their performance in batch effect removal and biological information preservation. Rather than performing active hyperparameter optimization, they examined how predefined architectural settings, specifically latent dimension size, network depth, and network width, influence integration quality. The results show that scVI achieves the strongest overall integration, primarily through superior batch correction; LDVAE offers dataset-specific advantages in maintaining biological structure; and MrVI demonstrates stable performance across conditions. Based on these findings, the authors provide model- and dataset-specific guidelines to aid effective integration in future single-cell studies.

**Strengths:**

The paper presents a well-structured and focused benchmarking study, offering a clear and systematic comparison of how different architectural hyperparameters, specifically latent dimensionality, network depth, and network width, affect the performance of three prominent VAE-based models for single-cell RNA-seq data integration. This design allows the authors to isolate the impact of model architecture on integration quality, making the analysis both controlled and interpretable.
A major strength of the work lies in its practical relevance. By identifying distinct behaviors across models, such as scVI’s strong batch correction, LDVAE’s superior preservation of biological structure, and MrVI’s stability, the study provides actionable insights for practitioners selecting models and configurations for specific datasets or analytical goals.
The authors also employ a comprehensive and balanced evaluation strategy, incorporating metrics that assess both batch effect removal and biological conservation. This dual perspective ensures that integration performance is not judged solely by technical alignment but also by the biological fidelity of the resulting latent representations.
Finally, the study’s proposal of model- and dataset-specific guidelines represents a valuable practical contribution. These recommendations translate the benchmarking results into concrete guidance for researchers, enhancing the reproducibility and efficiency of future single-cell data integration efforts.

**Weaknesses:**

While the study is carefully designed and clearly presented, it has several limitations that slightly constrain the generality of its conclusions. First, the benchmark focuses on a relatively narrow set of architectural hyperparameters, namely, latent dimension size, network depth, and network width. Although these parameters are indeed central to model architecture, other influential factors such as learning rate, dropout rate, or batch size are not explored. Including a broader range of hyperparameters might have provided a more complete picture of model behavior and performance sensitivity.
Second, the comparison is limited to three VAE-based methods—scVI, MrVI, and LDVAE. While these are representative and widely used in the field, the absence of non-VAE or hybrid integration approaches restricts the broader applicability of the findings. Incorporating alternative paradigms, such as graph-based or contrastive learning models, could have offered valuable comparative insight.
Another limitation lies in the scope of the datasets. Although the study includes multiple scRNA-seq datasets, these primarily represent moderate-sized, well-characterized benchmarks. The results may not fully capture how the models behave in more complex integration scenarios, such as extremely large datasets, multi-species comparisons, low sequencing-quality datasets, imbalance datasets, or data with severe sparsity.
The paper also provides limited discussion of computational efficiency. While the integration quality is thoroughly evaluated, readers are given little information about the runtime or resource demands of different architectures. Since computational feasibility often influences method selection in practice, this omission slightly reduces the practical value of the results.
Finally, the study gives relatively little attention to latent space interpretability. Although the quantitative metrics effectively evaluate integration quality, they do not reveal how architectural changes might affect the biological meaning or interpretability of the latent representations. A deeper exploration of this aspect could further strengthen the connection between technical performance and biological insight.

**Questions:**

1. How consistent are the observed performance patterns (e.g., scVI’s strong batch correction, LDVAE’s biological preservation) across datasets with varying cell type diversity or batch imbalance?
2. Why were only latent dimension, depth, and width selected as benchmarked hyperparameters? Could other settings (e.g., learning rate, regularization) also play major roles in integration quality?
3. Are the proposed configuration guidelines applicable to unseen datasets, or should users expect to re-evaluate settings for each new dataset?
4. Can the authors quantify the trade-off between batch effect removal and biological structure preservation for different model configurations?
5. How do variations in latent dimension affect downstream analyses (e.g., clustering consistency, trajectory inference)?
6. How do the authors calculate the overall score? Is it the average of the biological conservation and batch correction?

---

> ### Author Response · Authors · 2025-11-24
> **We are sincerely thankful for your thoughtful comments and expert perspective and based on your comments we have performed major revisions to enahnce our work including a revised abstract, a new dataset, and more in depth discussion.**
>
> Thank you for this valid concerns, we agree with the reviewer, and we want to point out that we focused our benchmark on latent dimensionality, network depth, and hidden width because training hyperparameters such as learning rate, batch size, and dropout are already governed by mature, widely adopted heuristics and defaults. In deep learning, Adam and established LR schedules (including linear‑scaling and warm‑up) substantially reduce LR search; batch‑size rules tie directly to LR and optimization noise; and dropout has long‑standing, empirically supported ranges. In our study, we focus on architectural capacity (we have revised our abstract to reflect that we focus on model architecture hyperparameter), which is less consistently tuned in practice yet directly controls the representation power of VAE‑based integrators. Our systematic sweeps therefore address an underexplored but high‑impact part of the design space that complements prior work on methods and preprocessing.
>
> Based on your comments, we have added the following to the Introduction section: "Our study centers on architecture
> capacity, including latent size, depth, width, and (for MrVI) the sample-aware latent, because these knobs change what the model can represent and therefore directly govern the batch versus biology trade-off. Our study does not imply
> that parameters like learning rate, batch size, and dropout are of less importance; however, they affect how fast and
> stable VAEs reach good optima, and they are already supported by a mature literature and standard recipes that users
> can adopt to their use case in VAEs Goyal et al. (2017); Kingma et al. (2015); Srivastava et al. (2014); Mandt et al.
> (2017); Smith et al. (2021); Goyal et al. (2017); Adam et al. (2014). Unlike the scope of our study, to our knowledge,
> no previous study has systematically benchmarked VAEs architecture choices in the domain of scRNA, and our study
> provides guidelines for users to avoid extensive computations.
>
> Regarding our choice of scVI, MrVI, and LDVAE, we acknowledge that these are 3 models and non-VAE paradigms
> are important; however, we want to point out that our scope is the scvi‑tools VAE family (scVI, MrVI, LDVAE), which is the most widely used probabilistic toolkit for scRNA‑seq integration and analysis, with mature APIs, pretrained‑model infrastructure, and broad community adoption, making it the natural target for a systematic, capacity‑focused benchmark that yields actionable defaults for practitioners. Our choice of models covers scVI, which is a general‑purpose VAE for robust batch integration and downstream tasks, MrVI, which is a cohort‑aware extension that explicitly models sample/technology covariates, important for multi‑protocol or multi‑site studies, and LDVAE, which is a linearly decoded VAE that trades a bit of raw accuracy for interpretability via gene‑program loadings.
>
> To clarify this, we have added the following to the Introduction: "Our selection targets complementary use cases in scRNA-seq analysis: scVI serves as a general-purpose VAE enabling robust batch integration and downstream tasks; MrVI extends this
> framework by incorporating cohort structure, explicitly modeling sample and technology covariates for multi-protocol
> or multi-site designs; and LDVAE employs a linear decoder, trading modest accuracy for interpretability through gene-
> program loadings."
>
> We also agree that moderate‑sized, well‑characterized benchmarks do not cover every integration regime. In the revision, we expanded beyond immune‑focused datasets to a non‑immune, multi‑tissue, multi‑protocol atlas (Tabula Muris; >300k cells, >20k genes) and re‑ran our full configuration grid. The same practical rules carried over: use HVGs, keep architectures shallow, and increase the latent dimension first; in multi‑protocol settings, MrVI gained by modeling protocol/cohort covariates, improving batch correction while preserving biology. For even more demanding scenarios, our study offers compute‑aware defaults precisely because exhaustive tuning is infeasible at extreme scale: current atlases already reach tens to hundreds of millions of cells, where RAM/VRAM and I/O become the bottleneck and broad sweeps are impractical for most labs. In such cases, our recommendations: HVGs to reduce input dimensionality, shallow depth to avoid costly H term, and tuning z as the cheapest capacity knob, and starting points that minimize trial‑and‑error while remaining effective. While no single benchmark can exhaust the space of integration challenges, we deliberately added a large, heterogeneous, non‑immune atlas and designed the study to yield generalizable, compute‑conscious defaults that practitioners can adopt directly when full tuning is impractical.

---

> ### Author Response · Authors · 2025-11-24
>
> We also appreciate the emphasis on runtime and resource demands. In the revision, we make these trade–offs explicit with two additions. First, S4.1 Computational Complexity derives a GEMM–based operation-count model as a proxy for training cost and shows how cost scales with batch size B, number of genes G, hidden width H, and latent dimension z. For scVI, the dominant term is BHG; raising z by  Δz adds only BHΔz, giving a fractional overhead Δz/G at typical scales. For LDVAE and MrVI, the decoder contributes BGz and together with the encoder BGH yields a first–order cost BG(H+z) with additional BH^2 and BHz terms. We also quantify memory: scVI adds O(HΔz) parameters, whereas LDVAE/MrVI add O(GΔz), an effect mitigated by HVG features that reduce G. Since G is in the thousands, H in the hundreds, and z in the tens, increasing z is the most compute–efficient capacity knob across models.
>
> Second, S4.2 Score and Complexity Trade–offs provides empirical curves of mean score versus mean training time while varying a single hyperparameter and holding the others fixed, aggregated across datasets (new Fig.7). These curves show that increasing nlatent delivers the most favorable accuracy–to–time profile, especially for scVI, whereas deeper or wider networks incur higher compute with smaller or inconsistent gains. Appendix Figs.16–24 further break out per–metric pairwise comparisons (11 metrics where applicable) and align the performance gains from larger z with the modest increase in training time.
>
> Across scVI, MrVI, and LDVAE, the analysis indicates that increasing z is the most economical way to improve performance. For scVI, the main compute term BHG does not depend on z and the incremental work BHΔ z is small relative to G. For LDVAE and MrVI, although the decoder cost includes BGz, practical steps in z are modest compared with typical steps in H, and using HVG features further reduces cost by lowering G. The recommended practice is to select HVGs, prefer shallow architectures, and tune z first to maximize performance per unit time and memory.
>
>
> Regarding latent space interpretability. We appreciate emphasizing this. Our embedding panels already illustrate how model capacity shapes biological readability. In Figure 5, the top-performing configurations for each model display label-consistent clusters with appropriate batch mixing. In Figure 6, increasing the latent dimension z tightens same-type neighborhoods and improves class separation, whereas adding depth often fragments structure—classic qualitative signatures of more or less interpretable latents. The one-parameter sweeps (Figure 3) quantify the same trend: larger z improves batch metrics and label agreement across datasets, while greater width or depth yields smaller or inconsistent gains, aligning with the interpretability seen in the embeddings.
>
> In the appendix, we provide per-metric evaluations for each dataset, detailing how architectural changes affect batch correction and biological conservation, and a paired-metric analysis (Tables 16–24) that highlights consistent patterns across metrics. We also include a dedicated results section discussing the embeddings and present a mosaic across all datasets comparing architectural variants, shown with both t-SNE and UMAP.

---

> ### Author Response · Authors · 2025-11-24
>
> Q1. Thank you for this concern; our dataset choices cover broad biological and technical diversity. The human immune benchmark includes two sequencing technologies, Zenodo 11100300 with 18 PBMC samples from multiple donors, and Zenodo 8020792 with 24 PBMC samples spanning multiple time points. We also add Tabula Muris, a cross-tissue mouse single-cell RNA-seq resource profiling on the order of 10^5 cells across roughly 20 organs using two complementary modalities (droplet-based 3′ UMI 10x for breadth and FACS/Smart-seq2 for depth), yielding rich coverage of immune and non-immune compartments. Tabula Muris includes 356,213 cells and 20,116 genes, as illustrated in the newly added Figure 10, and provides 51 immune cell-type labels (for example T, B, NK, myeloid and dendritic subsets, tissue-resident macrophages) and 103 non-immune labels (epithelial, endothelial or vascular, stromal or mesenchymal, neural, muscle, hepatic, renal, pancreatic, and others), for a total of about 154 distinct labels. This selection spans varied cell types, technologies, and modalities, and, consistent with our results in Figure 3, Figure 4, and the newly added Figure 5, we observe that scVI achieves the best overall score (defined as the mean of the averages of both biological conservation metrics and batch correction metrics) across all datasets, with the strongest gains when the latent dimension z is increased and gene selection uses HVG features. Looking more closely, LDVAE shows good biological conservation in Figures 3, 4, and 5, even when its batch integration score is lower, demonstrating a consistent biological signal; we also emphasize that LDVAE’s interpretability arises from its linear decoder that links latent dimensions to genes. MrVI includes an additional sample-aware covariate and shows advantages on extremely heterogeneous datasets with differing technologies, which is evident on the newly added Tabula Muris dataset, where MrVI excels on batch correction metrics while maintaining very good biological conservation and stability. The inclusion of this new dataset demonstrates that our findings transfer to unseen data of greater complexity, and the consistency of these conclusions is supported by the overall comparisons in Figure 4, the metric sweeps in Figures 3 and 5, the per-metric evaluations in Figures 16 to 24, and the UMAP or t-SNE and mosaic visualizations provided in the appendix, these clarification have been reflected in results and discussion sections of the revised manuscript.
>
> Q2. We thank you for raising this question. We totally agree that learning rate, regularization, batch size, and optimizers are important, and our study does not state otherwise. Our scope in this study is the VAE model capacity. To address this concern we have added to our paper in the introduction: "Our study centers on architecture capacity, including latent size, depth, width, and (for MrVI) the sample-aware latent, because these knobs change what the model can represent and therefore directly govern the batch versus biology trade-off. Our study does not imply that parameters like learning rate, batch size, and dropout are of less importance; however, they affect how fast and stable VAEs reach good optima, and they are already supported by a mature literature and standard recipes that users can adopt to their use case in VAEs Goyal et al. (2017); Kingma et al. (2015); Srivastava et al. (2014); Mandt et al. (2017); Smith et al. (2021); Goyal et al. (2017); Adam et al. (2014). Unlike the scope of our study, to our knowledge, no previous study has systematically benchmarked VAEs architecture choices in the domain of scRNA, and our study provides guidelines for users to avoid extensive computations.
>
> Q3. As per our new dataset testing and validation, Fig 5 and the newly added results on Tabula Muris, we show that our configuration guidelines apply to unseen datasets.
>
> Q4. Thank you for the question. We explicitly quantify the batch–biology trade-off with a paired metric design and show it per configuration and per knob in the results section (e.g., "For scVI, the best Overall is 0.78105 at (256, 40, 2, HVG) and the worst is 0.73765 at (256, 10, 1, FULL).
> Batch correction peaks at (256, 50, 1, HVG), whereas biological conservation peaks at (256, 10, 1, HVG). Moving
> 128 → 256 gives small gains in batch-oriented metrics but often decreases biology/label metrics; increasing layers
> 1 → 3 tilts toward batch mixing with declines in clustering agreement and biological overall. HVG outperforms Full
> on average. Increasing nlatent stepwise improves batch metrics and agreement with mixed effects on label compactness
> and biological overall; endpoints 10 → 50 improve Overall in 11/12."). Also, we provide such details for all other models and datasets in the results sections. Moreover, figures 16 to 24, accompanied by detailed results tables in the appendix, provide a very detailed trade-off with a detailed explanation.

---

> ### Author Response · Authors · 2025-11-24
>
> Q5. Downstream analyses depend on both batch removal and biological conservation: as a model better preserves biological signal and suppresses batch noise, if we consider gene-level downstream tasks such as differential expression, marker discovery, and pathway enrichment, they improve with the improvement of batch removal and biological conservation caused by increasing the latent dimension z.
>
> For metric-oriented downstream tasks (e.g., clustering consistency, trajectory inference, etc.), our results in Figure 3, Figure 5, Figures 16–24, and the per-configuration tables show that small z (around 10) compresses distinct states, yielding lower NMI and ARI, fragmented neighborhoods (lower Label-ASW), and poorer mixing (higher cLISI), which often causes cluster merging. Increasing z to moderate or high values (about 30 to 50 for scVI and MrVI, slightly lower for LDVAE) produces stable cluster assignments across seeds and datasets, improves batch mixing while preserving label structure, and reaches a plateau in NMI, ARI, and Label-ASW with higher rank agreement across seeds.
>
> Q6. Thank you for raising this important point. The batch correction metrics are averaged, and the biological conservation metrics are averaged, and then we take the mean of the averages. We have reflected this in the results section of our revised paper. To address this concern, we have added the following to the result section: "The following section reports results for each dataset across the three models, condensed for readers seeking practical
> tuning guidelines for the studied VAEs. All metrics follow the definitions in Luecken et al. (2022); see that work for
> details. We compute an overall score as the mean of two group-wise averages: (i) biological conservation and (ii)
> batch correction. To reflect the primacy of preserving biological signal, we aggregate seven biological metrics and
> four batch-correction metrics, since removing batch noise without maintaining biology is not a desirable outcome."

---

### Official Review · Reviewer_kskk · 2025-11-03

**Soundness:** 2
**Presentation:** 3
**Contribution:** 2
**Rating:** 2
**Confidence:** 4

**Summary:**

This paper provides a benchmark of popular variational models in single-cell RNA-seq analysis, for the task of batch integration, sweeping through 120 hyperparameter configurations in a controlled way with 720 training runs. The paper recommends scVI as the go-to model for most datasets, at the same time highlighting that the hyperparameters of scVI do need to be tuned (as opposed to current practice of using the default hyperparameters).

**Strengths:**

- This paper provides the first benchmark of hyperparemter configurations for popular variational models in single-cell RNA-seq analysis, challenging a common assumption that the default hyperparmeters in scvi-tools can be used directly without tuning.

**Weaknesses:**

- For a benchmark paper, the datasets focus entirely on immune-related cells. Therefore, it is difficult to assess whether the conclusions from this paper can transfer to datasets with other cell types. I recommend the authors to include more datasets to enhance the diversity of the cell types considered.

- Do the t-SNE and UMAP plots offer additional insights compared to the numeric evaluation metrics? Currently, they seem to agree with the evaluation metrics, so it's unclear whether including these plots in a benchmark offers additional insights.

**Questions:**

- Do the hyperparameter recommendations transfer well to new datasets (especially a dataset not related to immune cells)? For example, can you fit a scaling law with respect to key hyperparameters and show that the scaling law also applies to new datasets?

- Since the variational models considered have modest computational requirement and mature tooling, a user can simply tune the hyperparameters for a given dataset of interest. What does this benchmark study offer in this setting?

---

> ### Author Response · Authors · 2025-11-24
> **We are sincerely thankful for your thoughtful comments and expert perspective. Based on your feedback, we added an additional dataset (Tabula Muris) and clarified ambiguities as requested.**
>
> Thank you for your feedback. We agree that a benchmark limited to immune cells limits how confidently one can generalize. In response, we expanded the study to include the Tabula Muris mouse atlas, a multi-tissue and multi-protocol dataset with more than 300,000 cells and more than 20,000 genes. The atlas includes 51 immune cell-type labels (e.g., T/B/NK, myeloid and dendritic subsets, tissue-resident macrophages) and 103 non-immune labels (epithelial,
> endothelial/vascular, stromal/mesenchymal, neural, muscle, hepatic, renal, pancreatic, etc.), for a total of about 154 distinct labels (details in newly added fig. 10). We ran our full systematic configuration grid on this dataset to observe and analyze the behavior of VAEs beyond the immune context. This addition increases biological and technical diversity, strengthens the transferability of our conclusions to non‑immune settings, and, to our knowledge, our study represents the first systematic exploration of VAE model architecture hyperparameters for single‑cell integration at this scale, which we believe will be valuable to the single‑cell community.
>
> Regarding t-SNE and UMAP, they provide added value beyond the numeric metrics. They are the community’s standard qualitative checks for single‑cell integration and reveal failure modes that aggregate scores can blur: local neighborhood breaks, residual batch striping, cluster fragmentation, over‑mixing of distinct types, and trajectory distortions. In our results, the plots make these phenomena visible even when headline scores move in the same direction. For example, on Human Immune the LDVAE embedding shows label‑consistent, lineage‑scale structure; on 24‑PBMC MrVI retains faint batch bands and LDVAE forms looser clusters; and increasing depth fragments scVI’s embedding despite small changes in averages. These visuals help readers see why a configuration succeeds or fails rather than only whether it scores higher.
>
> We agree with the reviewer that these plots can be dataset-specific, so they are best treated as complementary evidence. Our paper is intended as practical guidance on VAE hyperparameters, so we will keep the quantitative comparisons in the main text and relocate the full t‑SNE and UMAP panels to the appendix for readers who want to examine per‑dataset geometry.
>
> Q.1 Thank you for this thoughtful comment about transferability and the request for a “scaling law.” We share the goal of giving readers guidance that travels to new domains. Therefore, we expanded the benchmark beyond immune contexts to a non‑immune, multi‑tissue atlas spanning roughly 20 tissues, more than 300,000 cells, and more than 20,000 genes, with multiple protocols. Running our full configuration grid on this dataset confirmed that the practical rules of thumb transfer: use highly variable genes, keep architectures shallow, and increase the latent dimension to improve performance. In this multi‑protocol setting, MrVI’s sample‑aware latent that models protocol and cohort covariates provided a clear advantage, yielding stronger batch correction while preserving biology relative to the other models. These cross‑dataset results give actionable defaults without over‑promising a universal equation and align the recommendations with empirical evidence (results for this dataset are presented in figure 5 and discussed within the results sections).
>
> Q2. Thank you for the point about practical usability. Not all datasets are straightforward to tune. Our newly added Tabula Muris atlas, spanning about 20 tissues with more than 300,000 cells and over 20,000 genes, is memory intensive to load and preprocess, and exhaustive sweeps quickly become impractical; at larger scales, resources such as the Tahoe one hundred million cell atlas push RAM, GPU memory, and storage I/O to limits that make broad hyperparameter search unrealistic in many labs. In these settings, community guidance matters: our benchmark distills robust, compute-aware defaults that consistently work across domains, such as using highly variable genes, preferring shallow architectures, increasing the latent dimension as the first capacity knob, and selecting MrVI when protocol and cohort covariates need to be modeled explicitly. These recommendations reduce trial and error, lower time and cost barriers, and provide reproducible starting points that practitioners can refine as resources permit, which is especially valuable as single-cell studies continue to grow in size and heterogeneity.

---

### Official Review · Reviewer_XYpX · 2025-11-04

**Soundness:** 2
**Presentation:** 2
**Contribution:** 2
**Rating:** 4
**Confidence:** 4

**Summary:**

The paper presents the first systematic study of how hyperparameters affect VAE-based models for single-cell RNA sequencing (scRNA-seq) batch correction. The authors benchmark scVI, MrVI, and LDVAE: three widely used probabilistic generative models implemented in scvi-tools across three heterogeneous datasets. The hypers they consider are: latent dimensionality, network depth, and hidden width, under two feature regimes: all genes vs. highly variable genes (HVGs). Performance is assessed using a standardized smetric suite measuring both batch removal and biological conservation (e.g. NMI, ARI, label ASW, cLISI, trajectory conservation), along with qualitative UMAP/t-SNE visualization. They provide practical defaults and rules of thumb for hyper initialisation in these VAE methods for scRNA.

**Strengths:**

- The paper performs the first systematic hyperparameter benchmark of VAE based integration models for scRNA-seq, across multiple datasets with a fully standardized pipeline.
- The study extracts actionable insights like latent dimensionality 30–50, shallow depth, HVG features.
- By harmonising preprocessing and feature selection they isolate model behaviour, the steps taken are reasonable. This helps achieve cross-modal comparability.
- The planned public release of all the models may provide a useful reference resource for researchers in single cell generative models.

**Weaknesses:**

- To fit ICLR’s bar, the paper should Abstract beyond the domain, argue that the results reveal general principles of VAE behavior under latent capacity scaling or dataset heterogeneity rather than just focus on the narrow setting of scRNA-seq. I would consider such a dataset-specific comparison better suited for bioinformatics journals.

**Questions:**

- While scVI generally outperforms the others, were compute time and memory costs explicitly compared? For example, does the performance gain from higher latent dimensionality justify the additional training cost?
- LDVAE is designed for interpretability, yet the paper primarily quantifies integration performance. Can the authors comment on whether the more interpretable linear decoder representations preserved biologically meaningful gene programs even when quantitative integration scores were lower?
- Since VAE training is inherently stochastic .. how sensitive are the reported results to random seeds? Were multiple training replicates performed per configuration, and if not, could variance across runs materially affect the ranking of hyper. settings or models? Couldn't find this information anywhere easily.

---

> ### Author Response · Authors · 2025-11-24
> **We are sincerely grateful for your thoughtful comments and expert perspective. Based on your feedback, we have strengthened the paper and presented our conclusions in a more transparent way to fit ICLR.**
>
> Q1. We acknowledge that time and memory trade-offs with performance are important and add value to the results; therefore, we expanded our analysis of time/memory vs. performance. Section 4.1 (Computational Complexity) uses GEMM-based operation counts as a compute proxy and details scaling with batch size B, genes G, hidden width H, and latent dimension z. For scVI, the leading cost is BHG; increasing z by Δz adds BHΔz, giving a fractional overhead ≈Δz/G. For LDVAE and MrVI, the decoder contributes BGz and, together with the encoder BGH, yields a first-order cost BG(H+z) (plus terms in BH^2 and BHz). Memory grows as O(HΔz) for scVI versus O(GΔz) for LDVAE/MrVI, with smaller growth when HVG features reduce G. Therefore, scVI is the cheapest computationally, and since G is in thousands, H in hundreds, and z in order of tens, tuning z is the most compute-efficient lever across models including MrVI and LDVAE.
>
> Section 4.2 (Score–Complexity Trade-offs) adds Fig. 6, which reports empirical curves of mean score and mean training time as a single hyperparameter varies (others fixed), aggregated over immune datasets. Increasing z provides the best score-to-cost improvement—especially for scVI compared with increasing depth or H. Appendix Figs. 16–24 show pairwise results for 11 metrics, confirming that larger z consistently improves performance with modest additional cost. Recommendation: use HVG features, prefer shallow architectures, and treat z as the primary tuning knob across datasets.
>
> Q2.  Thank you for this highlight. We agree that the benefit of LDVAE was not emphasized clearly in the paper, and therefore, we want to point out that interpretability is the whole point of LDVAE linear decoder. LDVAE’s linear decoder is designed for interpretability: each latent dimension corresponds to a coherent gene program. Even when LDVAE’s integration scores are lower than scVI’s or MrVI’s, the factors remain biologically meaningful. In Figure 11 in the appendix, the Human Immune panels show label‑consistent clusters with good cross‑batch overlap, indicating that LDVAE’s latents capture stable, lineage‑scale programs. On the PBMC datasets the clusters are less consistent with weaker integration, but the latent axes still behave like gene programs. The UMAP/t‑SNE figures in the appendix (figures 12 to 15) show that increasing LDVAE capacity tightens structure modestly without changing the core behavior: latent dimensions act as directions along which canonical markers rise or fall together, exactly what a linear factor model is meant to reveal.
>
> We have added the following to section 4.3 MODEL RECOMMENDATION: "LDVAE excels at interpretability and scaling because it allows for visualizing how latent dimensions link to genes, but has less capacity to model complex, nonlinear manifolds and to
> integrate datasets with complex batch effects. It performs best when heterogeneity is dominated by broad lineages
> (as in Human Immune), where gene programs are nearly axis-aligned and a linear decoder cleanly preserves biology
> across batches. By contrast, on single-tissue PBMC cohorts rich in donor/time effects and fine subtype boundaries and
> on Tabula Muris, which is a multi-tissue, cross-protocol atlas (e.g., droplet vs. full-length sequencing), nonlinearity
> buys accuracy, and LDVAE’s linearity becomes a bottleneck."
>
> Q3. Thank you for your concern. To our knowledge, our benchmark is the first to examine these hyperparameters at large scale, with 960 experiments across four datasets (we have included in the revised work a large multi-tissue and multi protocol dataset with more than 300,000 cells and more than 20,000 genes). Across this full design, we observed consistent patterns of model behavior that repeat across datasets, which supports the stability of our conclusions.
>
> To assess seed sensitivity directly, we randomly sampled models and configurations and retrained them with two independent seeds using training of 200 epochs. The variability across seeds for the primary aggregate metric was approximately 0.001, indicating numerical stability at convergence. Taken together, the cross-dataset consistency and the very small seed variance show that stochasticity does not drive our results and that the reported rankings and recommendations are generalizable.
>
> To clarify this in the paper, we have added the following sentence in section DATASETS AND BENCHMARKING PIPELINE: "To assess seed sensitivity directly, we randomly sampled models and configurations and retrained them with two independent seeds with training of 200 epochs. The variability across seeds for the primary aggregate metric was approximately 0.001, indicating numerical stability at convergence. Taken together, the cross-dataset consistency and the very small seed variance show that stochasticity does not drive our results, and that the reported rankings and recommendations are generalizable."

---

### Author Response · Authors · 2025-11-25

We thank all reviewers for their feedback, which helped us strengthen the manuscript. We have carefully addressed every comment and updated the paper accordingly. To our knowledge, this is the first study to systematically examine VAE model-capacity hyperparameters for scRNA-seq integration. Our focus is on architectural capacity, including latent size, depth, width, and the cohort-aware latent, because these choices directly influence downstream gene-level analyses; by contrast, training hyperparameters (e.g., learning rate, batch size) primarily affect speed and stability. We have updated our abstract to reflect this. In the camera-ready version, we will release an interactive GUI that lets users select configurations and generate comparative plots, an installable software package for automatically analyzing new datasets, and we will publish all 960 trained models on Hugging Face to maximize reproducibility and transparency. This work provides practical guidance to the scRNA-seq community, reducing the need for costly capacity searches that are often infeasible in resource-constrained labs due to the high RAM/GPU demands of large datasets. In this revision we (i) expand the compute–performance analysis with a new complexity model (Sec. 4.1), empirical score–time curves (Fig. 6), showing why HVGs + shallow networks + tuning latent dimension z first is the most economical path, Complexity and performance trade-offs (Sec 4.2), and model recommendation (Sec 4.3); (ii) add a large, non-immune, multi-tissue, multi-protocol atlas (Tabula Muris; ~356k cells, ~20k genes, 154 labels) to demonstrate transferability across heterogeneous, cross-technology settings; (iii) quantify seed stability (two independent seeds; aggregate-metric variance ≈ 0.001) to establish robustness; and (iv) sharpen model-specific guidance that links dataset properties to optimal hyperparameters: scVI as a reliable default (moderate-to-high z, shallow depth), MrVI’s sample-aware latent (u = 20) excelling on multi-site/multi-protocol data, and LDVAE’s linear decoder yielding interpretable gene-program axes when biology is dominated by broad lineages. Together with paired-metric breakdowns across four datasets and 960 runs, these updates underscore the novelty of our study as the first systematic benchmark of VAE architectural capacity for scRNA-seq at this scale, complementing prior method-selection work by isolating the capacity knobs practitioners actually tune.

Summary: We revised the manuscript substantially as follows:  Figure 2 (pipeline) has been updated; Figure 5 now includes new results from Tabula Muris; and Figure 6 presents the trade-off between computational complexity and performance. All t-SNE and UMAP visualizations have been moved to the appendix. The abstract has been strengthened to reflect the new analyses and to address reviewer feedback. In the introduction, we now state explicitly that Figure 1 is adapted from the original authors, and we clarify our choices of model architectures, hyperparameters, and models. Section 2 (Datasets and Benchmarking Pipeline) now describes the Tabula Muris dataset and adds a paragraph on variability arising from VAE stochastic training. Section 3 (Results) has been expanded to include Tabula Muris and reorganized for clarity into two parts: immune datasets and Tabula Muris. Section 4 (Discussion) now contains three focused subsections: 4.1 Computational Complexity, 4.2 Score–Complexity Trade-offs, and 4.3 Model Recommendations.

---

### Note · Authors · 2026-02-11

I have read and agree with the venue's withdrawal policy on behalf of myself and my co-authors.

---

### Meta-Review · Area_Chair_yFe3 · 2026-01-10

**Summary:**

This paper presents a benchmark study evaluating VAE-based methods for scRNA-seq batch integration. All four reviewers considered the paper to be below the acceptance bar and raised multiple concerns spanning novelty, limited method and dataset coverage, and missing hyperparameter exploration. Rather than rebutting the reviewers’ critiques, the authors acknowledged the validity of these concerns and made substantial efforts to improve the benchmark accordingly. As a result, many technical and empirical issues were meaningfully addressed, particularly those related to dataset diversity, computational trade-offs, and robustness to stochastic training. However, fundamental issues regarding the evaluation scope, technical novelty, and the relevance of the resulting insights to the ICLR audience remain unresolved. It's hard to imagine that any reviewer would raise the score significantly. Therefore, AC recommends Reject.

**Reviewer Concerns:**

Reviewer XYpX raised three concerns (i) whether the work rises above a narrow, domain-specific benchmark, (ii) lack of explicit compute/memory–performance analysis, and (iii) sensitivity to stochastic training. In the response, the authors substantially expanded the scope by adding the large, heterogeneous Tabula Muris dataset, reframed the paper as a capacity-focused architectural study, and explicitly argued for generalizable principles . They added a detailed complexity analysis and empirical score–time trade-off curves, directly addressing compute concerns. Seed sensitivity was also evaluated via multi-seed retraining with reported low variance. Therefore, most of Reviewer XYpX’s concerns are credibly addressed, and the reviewer would likely view the revision more favorably, though the contribution may still feel application-specific to some extent.

Reviewer kskk questioned generalization beyond immune/PBMC datasets, the value added by qualitative visualizations, and the practical benefit of the benchmark given mature tooling for hyperparameter tuning. The addition of Tabula Muris directly addresses the dataset-diversity and transferability concern, demonstrating that the proposed guidelines extend to non-immune, multi-tissue, multi-protocol data. The authors also clarified the complementary role of UMAP/t-SNE and moved extensive visualizations to the appendix, which addresses presentation concerns. They further justified the benchmark’s value by emphasizing resource constraints at scale and the impracticality of exhaustive tuning. Therefore, the main concerns are largely resolved, though skepticism about novelty versus good practice may remain.

Reviewer JRaA raised concerns about limited hyperparameter scope, limited model diversity, insufficient computational analysis, and unclear definition of the overall score and trade-offs. The authors responded by clearly defining the study’s scope as architectural capacity (not optimization hyperparameters), justifying this choice with references to mature optimization heuristics. They added explicit compute and memory analyses, clarified the overall scoring methodology. However, the concern about limited model diversity is acknowledged but not fundamentally resolved, as the scope remains unchanged.

Reviewer FpcV raised the strongest concerns, spanning novelty, limited methods and datasets, missing hyperparameters, stochasticity, lack of dataset–hyperparameter linkage, presentation issues, and excessive length. The authors addressed many factual issues. However, several concerns remain only partially resolved. For example, the study still focuses on three VAE models, still does not explore optimization hyperparameters, and the novelty is on benchmarking.

**Reviewer Scores:**

In most cases, the authors did not attempt to rebut the reviewers’ critiques, but instead acknowledged the validity of their concerns and made substantial efforts to improve the benchmark accordingly. As a result, many technical and empirical issues were meaningfully addressed, particularly those related to dataset diversity, computational trade-offs, and robustness to stochastic training.

However, scope-related and contribution-related concerns, for example those regarding novelty, breadth of evaluated models, and positioning with respect to prior benchmarking work, remain largely unresolved. While AC appreciates the authors’ thorough and constructive revision, it is difficult to anticipate a significant upward shift in reviewer scores, especially given the paper’s narrow scope and limited target audience for an ICLR venue.

---

### Decision · Program_Chairs · 2026-01-26

Reject